evolution/palaeontology

birds, growth and functional maturity, ontogeny, precocial–altricial development, quantitative bone histology, radial porosity profile

**Author for correspondence:**
Edina Prondvai
e-mails: edina.prondvai@gmail.com,
e.prondvai@bham.ac.uk

# Radial porosity profiles: a new bone histological method for comparative developmental analysis of diametric limb bone growth

Edina Prondvai[1,2], Adam T. Kocsis[3], Anick Abourachid[4], Dominique Adriaens[5], Pascal Godefroit[6], Dong-Yu Hu[7,8] and Richard J. Butler[1]

[1]School of Geography, Earth and Environmental Sciences, University of Birmingham, Edgbaston, Birmingham, UK
[2]MTA-MTM-ELTE Research Group for Paleontology, Budapest, Hungary
[3]Department of Palaeobiology, Friedrich-Alexander-University of Erlangen-Nürnberg, Erlangen, Germany
[4]Département Adaptations du Vivant, UMR 7179 Muséum National d'Histoire Naturelle – CNRS, Paris, France
[5]Department of Biology, Evolutionary Morphology of Vertebrates, Ghent University, Ghent, Belgium
[6]Directorate Earth and History of Life, Royal Belgian Institute of Natural Sciences, Brussels, Belgium
[7]Key Laboratory for Evolution of Past Life in Northeast Asia, Ministry of Land and Resources, Paleontological Institute of Shenyang Normal University, Shenyang, People's Republic of China
[8]Paleontological Museum of Liaoning, Shenyang, People's Republic of China

EP, 0000-0002-1284-8311; ATK, 0000-0002-9028-665X; DA, 0000-0003-3610-2773; RJB, 0000-0003-2136-7541

In fossil tetrapods, limb bone histology is considered the most reliable tool not only for inferring skeletal maturity—a crucial assessment in palaeobiological and evolutionary studies—but also for evaluating the growth dynamics within the ontogenetic window represented by the primary bone cortex. Due to its complex relationship with bone growth and functional maturation, primary cortical vascularity is an indispensable osteohistological character for reconstructing growth dynamics, especially in the context of various developmental strategies along the precocial–altricial spectrum. Using this concept as our working hypothesis, we developed a new quantitative osteohistological parameter, radial porosity profile (RPP), that captures relative cortical porosity changes in limb bones as trajectories. We built a proof-of-concept RPP dataset on extant birds, then added fossil paravian dinosaurs and performed a set of trajectory-grouping analyses to identify potential RPP

categories and evaluate them in the context of our ontogeny—developmental strategy working hypothesis. We found that RPPs, indeed, reflect important developmental features within and across elements, specimens and taxa, supporting their analytical power. Our RPPs also revealed unexpected potential osteohistological correlates of growth and functional development of limb bones. The diverse potential applications of RPPs open up new research directions in the evolution of locomotor ontogeny.

# 1. Background

Assessing ontogenetic stage, growth and development in long-extinct fossil vertebrates is of crucial importance in evolutionary and palaeobiological studies, such as evolutionary developmental biology, life-history analysis, comparative functional morphology, paleoecology, behavioural inferences, and taxonomic and phylogenetic assessments (see reviews in [1–13]). Tracing the ontogenetic status of fossil vertebrates mainly relies on the identification of important developmental events and/or periods in their mineralized skeletons that characterize different stages of the ontogenetic trajectory. In amniotes, these may include the late-embryonic period (when ossification is extensive enough for skeletal preservation), hatching/birth, weaning, the fastest growth period, the phase of growth saturation, reproductive maturation, skeletal maturation and senescence.

Extensive literature exists on the diverse size-independent methods used for assessing the skeletal maturity in fossils [13]. These methods are based on osteological and other morphological maturity indices, such as the closure sequence of sutures, degree of fusion, robusticity and surface texture of elements, and allometric changes of different skeletal parts (e.g. [14–20]), on osteohistological indicators of growth dynamics and bone maturation (e.g. [2,8,13,21–34]), or some combinations of these (see review in [13]). However, some osteological and other morphological correlates of skeletal maturity may be unreliable. For instance, Bailleul *et al.* [35] argued against using the degree of suture closure as indicator of skeletal maturity level by demonstrating that a generalized progressive fusion pattern throughout growth in non-avian diapsids and mammals is disputable. Skeletal robusticity is also well known to vary between sexes (e.g. [36,37]) and hence is unreliable for ontogenetic inferences in fossils. Finally, methods requiring preservation of associated elements prevent ontogenetic staging of fragmentary fossils and isolated bones, and interspecific variation in growth strategies may also render surface texture analysis less efficient [38]. Postcranial osteohistology, on the other hand, can provide useful ontogenetic information even for fragmentary and isolated mineralized elements, as long as the primary growth record is not obliterated by extensive secondary remodelling and the outermost cortex is preserved. Most importantly, and in contrast with other ontogenetic characters, primary bone tissues do not merely record the skeletal maturity status but also part of the preceding ontogenetic history of the animal, providing details of growth dynamics in the preserved ontogenetic window.

Developmental sequences and the course of developmental trajectories up to skeletal maturity and maximum body size can vary considerably, and not only among different taxa but also within the same species (e.g. [17,39–47]). Such phenotypic plasticity is now widely accepted as one of the most important sources of variability that selection acts upon, and thus as key to adaptive evolution (e.g. [48] and references therein). With the exception of birds, most extant and fossil members of the group Archosauromorpha (including birds, crocodilians and extinct species more closely related to them than to lepidosaurs), which have been comparatively extensively studied, apparently reveal high variability in the ontogenetic development of skeletal traits [18,19,29–32,45,49]. By contrast, birds show a uniquely low intraspecific phenotypic plasticity in their growth and development among modern vertebrates [18,50–53].

This intraspecific developmental variability has been documented in growth series of the early theropod dinosaurs *Coelophysis bauri* and *Megapnosaurus rhodesiensis* by applying ontogenetic sequence analysis (OSA) [18–20]. The OSA approach evaluates various, supposedly irreversible ontogenetic traits—in the case of *Coelophysis* and *Megapnosaurus* osteological maturity indices—with a parsimony-based method to map hierarchic developmental sequences that should lead from the least to the most mature phenotype [44]. However, due to the uncertainties related to embryological polarity and to the assumption of irreversibility and discrete states of characters [44], analysis of fossil taxa remains problematic, even if they are represented by a large number of specimens in different ontogenetic stages.

In addition to this intraspecific developmental variability, allometric development of different body parts within an individual poses further difficulties in ontogenetic studies of fossils. The thoroughly characterized precocial–altricial developmental spectrum, which describes low–high postnatal growth rates coupled with

high–low degree of functional maturity in the developing animal [54–57], is intimately linked to such allometric developmental processes. For instance, some developmental strategies in birds are associated with disparate growth and functional maturation of the fore- and hind-limbs [58–62]. Because these fine aspects of allometric growth and development are rarely known or studied in extinct amniotes, analysing fossil bones that may be taxonomically diagnostic but are preserved in isolation to the rest of the skeleton can lead to under- or overestimation of the actual ontogenetic stage of the individual in question (e.g. [30,33,34,45,63,64]). Since the uniformly weight-bearing elements in terrestrial vertebrates, such as the femur, are expected to stop growing when body mass stops accumulating (i.e. final body size is reached), investigating hind-limb bones might circumvent the ontogenetic staging problem related to allometric development-induced intraskeletal differences. Nevertheless, beyond the pure femur-based ontogenetic staging in fossils, for studies focusing on whole-body functional development through ontogeny and evolution, it remains important to map allometric developmental processes among multiple functional units in the body within and across taxa.

Based on these methodological comparisons, we consider that analysing bone tissue structure is the most reliable and efficient approach for skeletal growth-related ontogenetic assignment in fossil vertebrates and may also be informative of locomotor developmental strategies. In this context, primary cortical vascularity—hereafter referred to as primary porosity—is an essential osteohistological character due to its complex relationship with bone growth and functional maturation. Primary porosity averaged over the entire cortex thickness can be used to assess and compare the ontogenetic stages of conspecifics based on homologous bones, the relative growth rates of different bones within the skeleton or even different taxa if their relative ontogenetic stages are similar. However, this averaging approach loses information of the fine-scale diametric growth dynamics of the studied element. By contrast, changes in primary porosity in the radial growth direction reflect growth dynamics within the given ontogenetic window represented by the cortex. Such fine resolution of bone development can be crucial for inferring the functional development of locomotor modules within the skeleton, and eventually the ontogenetic locomotor strategies adopted by the animal.

We propose here a new quantitative histological method for long bones, referred to as radial porosity profiles (RPPs), that builds upon the general principles of diametric bone growth and development in the primary cortex of terrestrial tetrapod long bones. Our method abstracts growth and developmental dynamics from the sampled element on the basis of changes in relative primary porosity measured along radial trajectories (i.e. in the direction of diametric cortical growth) across the preserved posthatching cortex. The course of the RPPs—short trajectories capturing the radial porosity changes—is not only informative of the specific ontogenetic status of the specimen at the time of death, but also documents the growth dynamics of the element as preserved in the primary cortex in the given ontogenetic window. Thus, RPPs may give important insights into the developmental strategy of the studied bones in the context of overall growth trajectory and ontogenetic functional aspects of the individual's skeleton.

In this study, we introduce this RPP method with data collected from ontogenetic series of Rouen ducks (*Anas platyrhynchos domesticus*) of known ages, juvenile hoatzin specimens (*Opisthocomus hoazin*) with approximate age estimates, adult birds of other taxa and extinct paravian dinosaurs. We demonstrate the RPPs' analytical potential by evaluating these datasets with three different grouping approaches. We also provide a preliminary interpretation of the results in a growth- and function-related ontogenetic context, bearing in mind that accumulating such data in diverse extant tetrapods throughout their ontogeny is necessary to establish a reliable and wide-ranging comparative basis for applying this method successfully in fossils.

# 2. Material and methods

## 2.1. Materials

### 2.1.1. Ducks (*Anas platyrhynchos domesticus*, *Anas* sp.)

Representing an avian group with a pronounced modularity in fore-limb–hind-limb development [59,61,62], anatids are excellent model animals to study the osteohistology of disparate limb growth and function in the precocial–altricial developmental context [65]. Altogether we gathered 14 Rouen ducks from two different sources for this study.

Thirteen ethanol-preserved duck specimens were provided for bone histological sampling from the wet collection of the Museum National D'Histoire Naturelle (MNHN), Paris, France. These included

three from each of the age categories—4, 8, 15 and 30 days posthatching age (dph)—giving a growth series represented by three specimens in each cohort, and a single individual of 50 dph age. These ducks, along with another 85 specimens, were part of a previous muscle developmental study [66] in which hatchlings purchased from a commercial market at 1 dph were raised under the same temperature and light conditions with ad libitum food source in an enclosure measuring 200 × 600 cm$^2$ without swimming facilities at the French National Research Institute for Agriculture, Food and Environment (INRAE, UE0089 Unite Experimentale Palmipèdes a Foie Gras). Cohorts representing early (4 and 8 dph), mid- (15 dph) and late juveniles (30 dph) and close to fledging (50 dph) ontogenetic stages were euthanized [66]. Euthanized specimens that were not dissected in that study, including specimens donated for our investigation, were kept in 70% ethanol at MNHN.

In addition to these specimens, two associated hind-limb bones, a femur and a tibiotarsus, of an adult duck (*Anas* sp.) of unknown age were collected post-mortem and donated for our investigation by a private individual.

### 2.1.2. Hoatzins (*Opisthocomus hoazin*)

Hoatzins are highly specialized South American tropical birds. About two weeks after hatching, their juveniles have the unique ability to climb about in trees with the aid of two pairs of claws on their wings, and they can drop into water and swim to escape nest predators (e.g. [67,68]). These climbing and swimming activities frequently involve a remarkable alternating fore-limb movement that contrasts with the usual synchronous wing flapping observed in other modern nestling birds [69]. This peculiar precocity of the wings related to arboreal and aquatic locomotion but not to flight in hoatzin chicks is unparalleled among extant birds and may therefore be associated with distinct osteohistology. The limb bone histology of juvenile hoatzins was explored for the first time within the framework of this study.

Three juvenile specimens of wild hoatzins collected in Venezuela used in a locomotor performance analysis and euthanized for the study of Abourachid *et al*. [69] were provided for osteohistological sampling (see further details on the origin of specimens, experimental design and ethical statements in [69]). The exact age of these specimens was not known; however, their age could be estimated based on their body mass using the body mass–age relationship described for hoatzins by Dominguez-Bello *et al*. [68] as

$$y = 9.61 + 5.63x, \tag{2.1}$$

where $y$ = body mass in grams and $x$ = posthatching age in days (dph). This linear equation gave estimated ages of 6, 9 and 1 dph for the three investigated hoatzin chicks. This interval represents the early- to mid-juvenile stages of their 45–55 days growth trajectory from hatching to fledging [70].

### 2.1.3. Adult birds of other taxa

We sampled the limb bones of three additional bird specimens that we considered here as adults based on their functionally mature plumage and flight capability at the time of their death. These adult birds, a common kestrel (*Falco tinnunculus*), a Eurasian woodcock (*Scolopax rusticola*) and a rook (*Corvus frugilegus*), were all found on public land or as roadkill in Hungary and collected post-mortem by private individuals who offered the carcasses for scientific purposes to the Eötvös Loránd University (ELTE), Budapest, Hungary. The carcasses were prepared by maceration and their skeletal remains are housed at the Department of Paleontology at ELTE. Based on the known growth strategy and sexually mature plumage characteristics of these species (e.g. [54,71,72]), these specimens were estimated to have been as old as or older than 90 dph.

### 2.1.4. Fossil paravian dinosaurs

Multiple skeletal elements of five different Mesozoic paravian dinosaur taxa, *Anchiornis huxleyi*, *Aurornis xui*, *Eosinopteryx brevipenna*, *Serikornis sungei* and the avialan *Jeholornis curvipes*, were sampled and histologically analysed by Prondvai *et al*. [33]. These sections were re-analysed with the method introduced in this study. For information on the origin and current curation of the sampled specimens and their sections, see Prondvai *et al*. [33].

Electronic supplementary material, figure S1, shows the phylogenetic interrelationships among the studied taxa. Table 1 summarizes and gives further details on the specimens and the sampled elements used in this study.

**Table 1.** Sampled taxa, specimens and elements used in this study. Estimated minimum ages were based on the size and the incipient OCL in the elements of *Anas* sp., on the fresh body weight in hoatzin chicks, and on the plumage maturity and flight capability in the context of known growth strategy and lifestyle in *Falco*, *Scolopax* and *Corvus*. Abbreviations: Al, alula; Cmc, carpometacarpus; Fe, femur, Hu, humerus; Ra, radius; Ti, tibiotarsus; Tmt, tarsometatarsus; Ul, ulna.

| taxon | collection ID | specimen ID | age (dph) | sampled bones |
|---|---|---|---|---|
| *Anas platyrhynchos domesticus* | MNHN.ZMO 2014 - 215 | Anas no. 215 | 4 | Hu; Fe; Ti; Tmt |
| | MNHN.ZMO 2014 - 230 | Anas no. 230 | 4 | Hu; Fe; Ti; Tmt |
| | MNHN.ZMO 2014 - 290 | Anas no. 290 | 4 | Hu; Fe; Ti; Tmt |
| | MNHN.ZMO 2014 - 227 | Anas no. 227 | 8 | Hu; Ul; Cmc; Fe; Ti; Tmt |
| | MNHN.ZMO 2014 - 218 | Anas no. 218 | 8 | Hu; Ul; Fe; Ti; Tmt |
| | MNHN.ZMO 2014 - 246 | Anas no. 246 | 8 | Hu; Ul; Fe; Ti; Tmt |
| | MNHN.ZMO 2014 - 241 | Anas no. 241 | 15 | Hu; Ul; Cmc; Fe; Ti; Tmt |
| | MNHN.ZMO 2014 - 254 | Anas no. 254 | 15 | Hu; Ul; Cmc; Fe; Ti; Tmt |
| | MNHN.ZMO 2014 - 255 | Anas no. 255 | 15 | Hu; Ul; Cmc; Fe; Ti; Tmt |
| | MNHN.ZMO 2014 - 208 | Anas no. 208 | 30 | Hu; Ul; Cmc; Fe; Ti; Tmt |
| | MNHN.ZMO 2014 - 221 | Anas no. 221 | 30 | Hu; Ul; Cmc; Fe; Ti; Tmt |
| | MNHN.ZMO 2014 - 298 | Anas no. 298 | 30 | Hu; Ul; Cmc; Fe; Ti; Tmt |
| | MNHN.ZMO 2014 - 236 | Anas no. 236 | 50 | Hu; Ul; Cmc; Fe; Ti; Tmt |
| | NA | Anas no. 6 | ≥90 | Fe; Ti |
| *Opisthocomus hoatzin* | NA | Opis P4 | 6 | Hu; Ul; Cmc; Fe; Ti; Tmt |
| | NA | Opis P1 | 9 | Hu; Ul; Cmc; Fe; Ti; Tmt |
| | NA | Opis P3 | 16 | Hu; Ul; Cmc; Fe; Ti; Tmt |
| *Falco tinnunculus* | MDE A.5 | Falco | ≥90 | Hu; Ul; Cmc; Fe; Ti |
| *Scolopax rusticola* | MDE A.20 | Scol | ≥90 | Hu; Ul; Cmc; Fe; Ti; Tmt |
| *Corvus frugilegus* | MDE A.47 | Corv | ≥90 | Hu; Ul; Cmc; Fe; Ti; Tmt |
| *Anchiornis huxleyi* | YFGP – T5199 | Anch | NA | Hu; Ra; Ul; Cmc; Al; Fe |
| *Aurornis xui* | YFGP – T5198 | Aur | NA | Hu; Ra; Ul; Cmc; Al; Fe |
| *Eosinopteryx brevipenna* | YFGP – T519716 | Eos | NA | Hu; Ra; Ul; Cmc; Al; Fe |
| *Jeholornis curvipes* | YFGP – yb21 | Jeh | NA | Hu; Ra; Ul; Fe |
| *Serikornis sungei* | PMOL-AB002004 | Ser | NA | Hu; Ra; Ul; Cmc; Al; Fe |

## 2.2. Methods

### 2.2.1. Theoretical background of radial porosity profiles

From taking an overview of the main principles of diaphyseal bone growth and the processes involved in it in tetrapods representing size and metabolic ranges where vascularity appears in compact bone (see electronic supplementary material, Information S1; figure 1), four important and universally valid points emerge related to primary porosity patterns in the cortex through ontogeny: (i) primary porosity in the developing bone cortex reflects the rate at which bone volume is expanding; (ii) where osteons are present, there is a gradient of primary osteonal compaction being generally greater in the older (i.e. deeper) cortex than in the younger (i.e. more superficial) cortex resulting in a radial transition of lower to higher porosity that is most pronounced in the fastest growth period; (iii) medullary cavity expansion can secondarily increase porosity within the deep cortex due to resorption processes that invade the perimedullary regions and counteract the osteonal compaction; and (iv) following the fast juvenile growth period and approaching skeletal maturity, porosity levels decrease eventually to near-zero or zero in the outermost cortex (figure 1). Whether growth is overall slow or fast, cyclical or continuous, isometric or allometric, precocial or altricial, porosity patterns are influenced by the cumulative effects of all these developmental processes and thus represent an information-rich abstraction of the ontogenetic trajectories and potentially element-specific functional aspects of tetrapod skeletons.

Thus, we predict that a characteristic porosity profile along the radial axis of the bone cortex—that is, a pattern of primary porosity changes in the direction of radial growth from the inner to the outer cortex surface—will appear in each limb bone through ontogeny. We hypothesize that these porosity profiles can be linked to the ontogenetic stage and developmental strategy of the individual. Developmental strategy includes growth- and function-related aspects that may also involve modularity in limb development. Hence, RPPs of limb bones are expected to reflect the element's history of growth and function within the ontogenetic window preserved in the primary cortex. By sampling multiple elements of any skeleton, RPPs are therefore predicted to inform about intraskeletal growth dynamics, including iso- or allometric growth and function of different elements. RPPs are expected to be comparable among conspecifics or specimens of other taxa to decipher similarities and differences in relative growth phases and developmental strategies.

### 2.2.2. Selection of the data-collecting technique

Our ontogenetic and developmental approach is based on the osteohistological evaluation of the RPPs in limb bone diaphyses. Limb bone shafts are the most frequently analysed elements in palaeohistological studies because the mid-diaphyseal region usually has the highest primary bone content providing insight into individual growth trajectories [73]. Long bone histology is also shaped by diverse locomotor demands [74–83] and therefore is expected to reflect the functional development of the limbs, which is intimately associated with precocial–altricial ontogenetic strategies.

The following fore- and hind-limb elements were selected for analysis in most extant birds (table 1): humerus, ulna, carpometacarpus, femur, tibiotarsus and tarsometatarsus. The only exceptions were some duckling specimens of the early ontogenetic cohorts (4 and 8 dph), in which the ulna and carpometacarpus had not yet developed enough posthatching cortex for a reasonable radial measurement, the adult *Anas* sp. for which only the femur and the tibiotarsus were available for sampling, and the adult kestrel in which the tarsometatarsal was lost during preparation. Since of the hind-limb bones only the femur was sampled in the paravian dinosaurs [33], we also evaluated sections of the radius and the first phalanx of digit I. (alula) to have enough between-element comparative data for each fossil specimen. The non-homologous nature of elements was taken into account in the interspecific comparisons.

Our input variable, the mid-diaphyseal posthatching cortical porosity data, can be collected by diverse means (e.g. bone sections, μCT and synchrotron). For this demonstration, we used two-dimensional data from undemineralized transverse ground sections, the most accessible and widely used technique for fossil osteohistological investigations, to provide a common basis for direct comparisons between extant and extinct taxa.

### 2.2.3. Dissection and osteohistological ground section preparation

Duck and hoatzin specimens acquired from MNHN Paris were dissected in the histological laboratory of the Department of Biology, Ghent University (Ghent, Belgium).

**Figure 1.** (Caption overleaf.)

Up until the point of resin embedding, we followed the same dissection and bone sample preparation procedure for the ducks as described in Prondvai *et al.* [65]. After embedding the bone halves in resin, the transversely cut mid-diaphyses were exposed by cutting off surplus resin using a

**Figure 1.** (*Overleaf.*) Patterns of development of primary cortical porosity in the growing limb bones of ducks (*a*–*g*) and an adult rook (*h*,*i*) shown in mid-shaft transverse ground sections. (*a*,*b*) Overview of the sections of humerus and femur of duckling '218' at 8 dph following altricial and precocial development, respectively. Black dashed arrows indicate direction of radial cortex growth from medullary cavity (mc) to periosteal surface (ps). Note the more compact inner bony ring representing the prehatching cortex (prc) and the highly porous early stage posthatching cortex (poc) separated by a distinct hatching line (hl). (*c*,*d*) Humerus and femur of duckling '241' with different degrees of osteonal compaction of vascular spaces from the inner to the outer cortex at 15 dph. Higher osteonal compaction is evident in the inner cortex of the precocially developing femur. (*e*) Radius of '241' at 15 dph showing a partial layer of early developing endosteal lamellae (el) rimming the medullary cavity and large resorption cavities in between the pre- and posthatching cortices. A thick layer of the soft periosteum (po) is also preserved in this section. (*f*) Tibia of duckling '221' at 30 dph reveals active resorption opening up large cavities between a prominent layer of endosteal lamellae and the innermost primary bone cortex with some primary bone remnants still visible inside the resorption cavities. Enlarged vascular spaces (svs) in this area also indicate inner cortical resorption associated with medullary cavity expansion. Note (also in *e*) that in contrast with the general assumption (see main text and electronic supplementary material, Information), the EL is present despite the still ongoing radial cortical growth and dynamically changing and expanding medullary cavity. Well-developed primary osteons (pro) characterize most of the cortex at this stage; however, the outer cortical region clearly shows less compacted vascular spaces, i.e. higher porosity, indicating still ongoing diametric bone growth. (*g*) Femur of an adult duck of unknown age. Note the abrupt decrease in vascularity from the inner to the outer cortex. However, only incipient OCL and no EL can be observed. (*h*,*i*) Femur of an adult rook (*Corvus frugilegus*) showing tripartite cortex with well-developed EL (el), vascularized but highly compacted primary cortex, and a thick OCL. Note in (*i*) that the diameters of highly compacted vascular canals (versus) are almost within the size range of the osteocytic lacunae. Scale bars: 300 µm in (*a*,*b*,*f*); 120 µm in (*c*–*e*), (*g*,*h*); 50 µm in (*i*).

GEOFORM combined cutting–grinding machine (Metkon GEOFORM thin sectioning system), then manually ground on SiC powder of 400 grit size and mounted on custom-made plastic slides. Using the cutting module of GEOFORM, the resin blocks were cut off leaving an approximately 150 µm thick section of embedded bone sample on the slides. These were ground down further to approximately 70–100 µm thickness on the 65 µm diamond cup grinding wheel module of GEOFORM. Section surfaces were then manually smoothened on SiC powder of 800 grit size and protected with glass coverslips. These sections were all prepared in the histological laboratory of the Department of Biology, Ghent University.

For the hoatzins, kestrel, woodcock and rook, all ground section preparation steps were performed in the palaeohistology laboratory of the Department of Physical and Applied Geology, ELTE (Budapest, Hungary), following the procedures and using the same equipment as in Prondvai *et al.* [65]. Paravian dinosaur bones were already sampled and prepared as ground sections for the study of Prondvai *et al.* [33].

### 2.2.4. Microscopic investigation, visualization and measurement tools

Transverse mid-diaphyseal ground sections were examined under polarized light microscopes (Olympus CX22 and Nikon ECLIPSE LV100 POL), photographed with digital microscope cameras (Olympus Microscope U-TVO.5XC and QImaging MP5.0), and the images were digitally processed (ANALYSIS 5.0 and Image Pro Insight 8.0 software). All subsequent histomorphometric measurements were taken from the ground section images with ImageJ 1.53c [84] using the freehand and polygon selection options, and area measurements. Figures were assembled in Inkscape 0.92.1 [85].

### 2.2.5. Radial porosity profile measurement

In each element, three measurement sectors (*S*) were selected, wherever possible in homologous regions of the cortex, while trying to keep a sufficient distance among sectors to adequately represent the intracortical histovariability (figure 2*a*). However, selection of cortical areas and spacing of sectors were also influenced by the need to minimize areas of extensive secondary remodelling which obscure primary porosity information, thereby decreasing sample size (electronic supplementary material, figure S2A–D). In addition to these criteria, the procedure for selection of the measurement areas had to be the most flexible and opportunistic in the generally fragmentary and incomplete sections of paravian dinosaurs (electronic supplementary material, figure S2D).

The sectors were drawn on the cortex with two borders following the curvature of the inner and outer surface of the posthatching cortex, and the other two running roughly radially but adjusted to local relative cortical thickness and histology (figure 2*a*). Because of the radial orientation of the

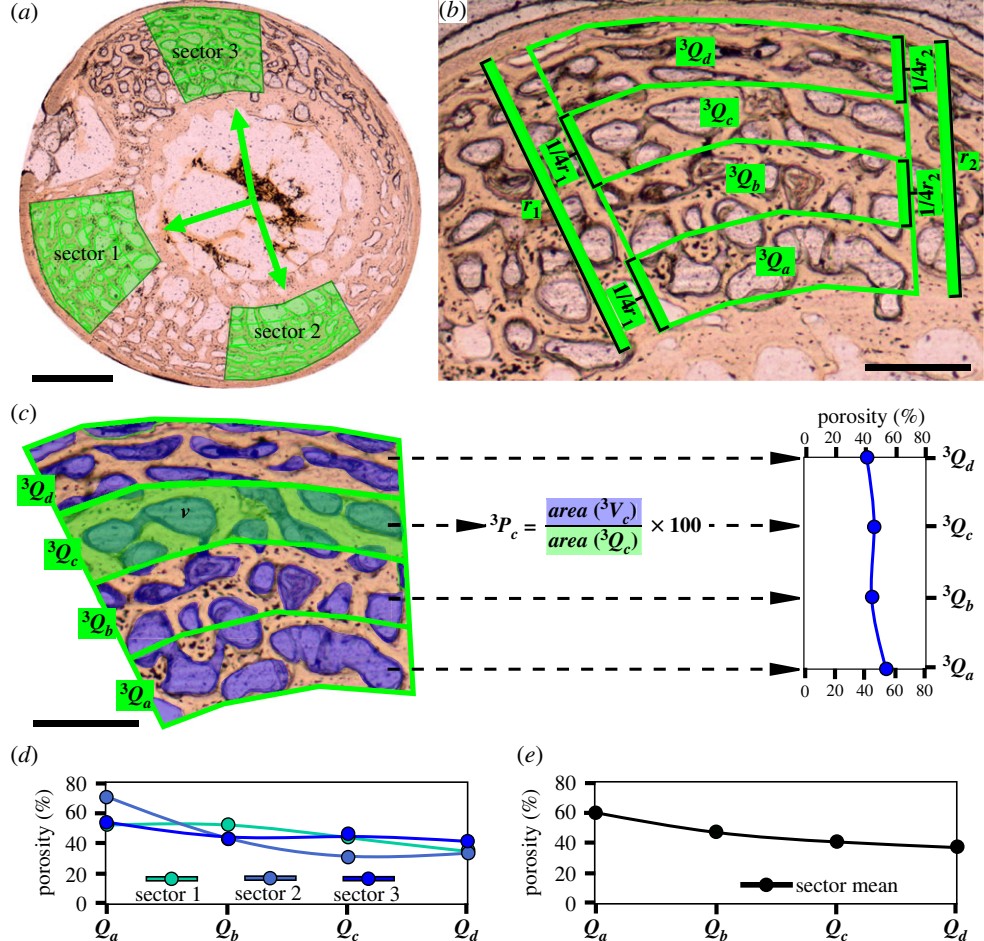

**Figure 2.** Measurement method for defining RPP shown on the example of the humeral cross-section of the duckling '218' at 8 dph. (*a*) The positions of the three sectors (green shaded polygons) are assigned based on set homologous anatomical directions and/or to cover representative and roughly equally distributed sampling areas in the posthatching cortex. Note that the shapes and dimensions of sectors 1, 2 and 3 are not set the same but are rather adjusted to local posthatching cortex qualities, including cortex thickness, radial growth direction, arch shape of inner and outer posthatching cortex surface, and a representative inclusion of primary cortical porosity in each quadrant ($Q$). (*b*) The outline and subdivision of the sectors into four quadrants ($^3Q_{a-d}$), using sector 3 as an example. The radial extent of each quadrant is determined by dividing the lengths of the two radii ($r_1$ and $r_2$) of the polygon into four equal sections. $^3Q_a$ designates the innermost (i.e. oldest) and $^3Q_d$ the outermost (i.e. youngest) posthatching cortical quadrant. (*c*) Areas of vascular spaces (v, blue shaded areas) are measured in each quadrant separately, and porosity ($P$) is determined as the percentage of the summed area of all vascular spaces (V) in a quadrant to total quadrant area. The porosity values of the four quadrants give the RPP of sector 3 taken from the inner ($^3Q_a$) to the outer ($^3Q_d$) cortex. (*d*) RPPs taken of the three sectors demonstrated in a single graph reflecting RPP diversity throughout the cortex. (*e*) Mean RPP of the humerus of '218' calculated from the mean porosity values of matching quadrants over the three sectors. Scale bars: 300 µm in (*a*); 120 µm in (*b,c*).

sides of each sector, the encompassed inner cortical areas are generally smaller (figure 2*b*), and the areal difference between the inner and outer cortical areas depends on the size of the medullary cavity relative to the cortex thickness. Because the radial borders of the sectors encompass smaller areas in the inner than the outer cortex (figure 2*b*), and this areal difference depends on the size of the medullary cavity relative to the cortex thickness, the dimensions of the sectors could not be standardized. Instead, they were adjusted to the regional relative cortex thickness in such a way as to include a histologically representative measurement area throughout the cortex thickness. In practice this means a large enough area in which repetitive patterns of the large-scale vascular architecture can be observed (figure 2*a*; electronic supplementary material, figure S2E,F). Based on the standard deviations of proportional porosity values measured from the inner to the outer cortex (see below), areal differences among and within sectors did not introduce any measurement bias.

Each of the three sectors was divided up to four quadrants ($Q$) along the radial axis from the inner to the outer cortex ($Q_a$–$Q_d$). Quadrants were delineated by dividing each radial side of the sector into four sections of equal length and connecting the corresponding sections of the two radial sides following the section circumference (figure 2$b$). These four quadrants represented the sampling areas for the four points of measurement defining the RPP for each sector as follows. (i) The area of each quadrant was measured, along with the area of each primary vascular space ($V_x$) within the quadrant. (ii) The porosity percentage in each quadrant ($P_x$) was calculated as

$$P_x = \frac{\text{area}(V_x)}{\text{area}(Q_x)} \times 100. \tag{2.2}$$

(iii) RPP within a sector was given as a trajectory defined by the four $P_x$ (i.e. $P_a$–$P_d$) values of quadrants starting from the innermost (i.e. the oldest, $Q_a$) to the outermost (i.e. the youngest, $Q_d$) cortical layer (figure 2$c$). Thus, this RPP trajectory, which is a four-point vector defined here by $P_a$–$P_d$, reflects element-specific developmental dynamics within the ontogenetic window of the specimen. First, the intrasectional variability was assessed based on the RPPs of the three sectors (figure 2$d$). Thereafter, a mean RPP was extracted for each bone by calculating the mean values of $P_x$ of the corresponding quadrants ($P_a$–$P_d$) over the three sectors ($S = 3$) (figure 2$e$). This calculation for four quadrants can be described by the following generalized formula:

$$\text{RPP} = \left( \sum_{i=1}^{S} Pa_i \frac{1}{S}, \sum_{i=1}^{S} Pb_i \frac{1}{S}, \sum_{i=1}^{S} Pc_i \frac{1}{S}, \sum_{i=1}^{S} Pd_i \frac{1}{S} \right). \tag{2.3}$$

The courses of these RPPs (electronic supplementary material, figure S3) were numerically analysed by diverse trajectory modelling and clustering methods in various comparative contexts (see §2.2.6).

Porosity measurements were only taken in primary cortical areas, including secondarily enlarged primary vascular channels where no sign of secondary bone deposition was present. Wherever secondary remodelling was present (i.e. resorption followed by secondary re-deposition of bone tissue, such as secondary osteons or compacted coarse cancellous bone (CCCB)), their areas were measured within the quadrants and subtracted from the sampled areas (electronic supplementary material, figure S2B,D). In those bones where secondary remodelling affected extensive cortical regions obliterating most or all of the primary cortex in a given quadrant (most frequently $Q_{a,b}$), the porosity values of the affected quadrants were considered missing (NAs) (electronic supplementary material, figure S2B,C). Only those RPPs were analysed in which a minimum of two of the four points defining the sector-averaged trajectory could be calculated at least in one sector. The extent of secondary remodelling was also considered in the interpretation of the results.

The relative thicknesses of the outer circumferential layer (OCL) and the endosteal lamellar layer (EL), rimming the medullary cavity in most bones of the paravian dinosaurs and some bones of the extant bird specimens (figure 1$h$), were quantified in relation to overall cortex thickness as the area percentage it occupies of the total cortex area in the measured sectors. Similar to secondarily remodelled areas, OCL and EL were not included in the numeric analyses in this study but were considered during the interpretation of the results.

## 2.2.6. Numeric analysis: a selection of methods

In the next step, we aimed to demonstrate the diverse analytical potential of RPPs by exploring whether different types of RPPs can be identified in our dataset and if yes, whether and how those types reflect the ontogenetic stage of the animal and the developmental strategy of the sampled bone. Here, we predict that RPPs of those bones which follow similar developmental strategies and come from individuals of comparable ontogenetic stages will be grouped together. Accordingly, we also expect a certain level of skeletal dissociation (i.e. bones coming from a single individual but following different developmental strategies will be separated), especially in ducks that show strong disparity in their fore- and hind-limb development. For this, we selected three different grouping approaches: (i) K-means clustering; (ii) group-based trajectory modelling (GBTM) [86]; and (iii) three types of trajectory similarity measures (dynamic time warping (DTW), 'edit distance' (ED) and longest common subsequence (LCSS)) [87]. Our reason for selecting these specific methods was to see how applying different grouping algorithms on the same dataset influences the outcome and hence our interpretation of the results. Furthermore, these exploratory approaches are applicable to a broad

range of trajectory-type datasets as they do not have strict assumptions about the independence of observations, data structure and distribution or other conditions usually set by parametric tests.

*K*-means clustering is a widely used method applied in a diverse range of fields for classifying multivariate observations into groups. *K*-means algorithm partitions observations into a pre-defined number of clusters, in which each observation is assigned to the nearest cluster mean (or centroid) parametrized by vectors. Cluster centroids are searched for by iteration algorithms to find local optima (e.g. [88]). GBTM is a statistical approach that was developed to analyse individual-level developmental trajectories—a progression of any phenomenon described by longitudinal data—and to identify distinct groups of trajectories within a population [86,89]. Using multi-nomial modelling strategy, GBTM assigns group membership probabilities to individual trajectories and compares the analytical support for all models to find the optimal (best-supported) number of groups [86,89,90]. The third main approach, trajectory similarity measures, comes from the field of computational movement analysis in which trajectories are defined as a 'sequence of time-stamped locations' [91] and similarity is calculated by various distance measures between trajectory pairs [87]. In the DTW method, the sum of the distances at each point along the trajectories is calculated as the smallest warping path. In ED, the minimum number of edits is calculated for two trajectories to be considered equivalent. The last trajectory similarity measure considered here, LCSS, works with the translation of the trajectories in all possible dimensions (in our case two-dimensional) to find the maximum number of equivalent points between two trajectories [87,91,92]. After calculating distances among all trajectory pairs by these three measures, we organized the pairs of similarity values into a distance matrix for each method and analysed potential trajectory groups by agglomerative hierarchical cluster analysis using the complete linkage method. The latter clustering technique builds a binary tree that starts out with each individual observation being in its own cluster, then it merges pairs of the closest clusters at each step until it reaches the root containing all observations in one cluster (e.g. [93]). Hereafter, we refer to these similarity measures–hierarchical cluster analysis combinations as DTW-hclust, ED-hclust and LCSS-hclust.

In all analysis types, we first ran the tests separately on the RPPs of the duck ontogenetic series (electronic supplementary material, figure S3A,B), then on those of the extinct paravian dinosaurs (electronic supplementary material, figure S3C,D), and finally on the complete dataset including the ducks, the extinct paravians, the juvenile hoatzins and the other adult birds (electronic supplementary material, figure S3E,F). Using these separate analyses, we could adjust the resolution of the results to broaden the context of our interpretations and elaborate on the finer details. In the similarity measure methods which, in contrast with GBTM and *K*-means, can work with trajectories containing missing values, we also analysed the duck and complete datasets with and without incomplete RPPs to see their effect on the group compositions. The level of skeletal dissociation of specimens—that is, the number of different groups an individual's bones were assigned to—was assessed in each analysis. Furthermore, we also ran tests on centred RPPs calculated for each quadrant as the difference from the mean sector porosity within a given trajectory (electronic supplementary material, figure S3B,D,E). In these tests, only the shape of the trajectories determines their group memberships. Finally, in the duck dataset, for which we had the most complete ontogenetic series with multiple specimens in some cohorts and exact information about ages and functional development, these expected potential predictors of group memberships could be tested by logistic regressions in the best-supported RPP-clustering models.

All analyses were performed in RStudio (1.4.1) integrated development environment (RStudio, PBC) for the free programming language and statistical computing software R [94]. *K*-means clustering was called in from the basic 'stats' function *kmeans()*, GBTM was performed using the function *crimCV()* in package 'crimCV' [90,95], and the three different trajectory similarity measures were run with the functions *DTW()*, *EditDist()* and *LCSS()* in package 'SimilarityMeasures' [87], after which an agglomerative hierarchical clustering was performed on their distance matrix outputs using the function *hclust()* in 'stats'. For all analyses, we used the available ontogenetic, age and developmental strategy data of the specimens, and the graphical distribution patterns of individual RPPs, to select the number of groups to be tested. *K*-means uses *a priori* defined number of clusters, and we tested three–five clusters in the different analyses with 100 iterations in each to find local optima for cluster means. GBTM, on the other hand, works with a pre-set maximum number of groups, which we set to five, so that it analysed all models with group numbers 1–5 to select the number of groups that best fits our RPP data. Because GBTM also has a stochastic component, we ran 100 iterations in each analysis to find a globally best-supported model using three model selection methods (Bayesian information criterion (BIC), Akaike information criterion (AIC) and cross-validation error (CVE)) with

the function *crimCV()* in which CVE was developed for testing GBTM itself [95]. Finally, the dichotomous dendrograms of the distance matrix-based hierarchical clustering resulting from DTW, ED and LCSS were also cut at 3–5 clusters.

We tested specimen, age, bone and developmental strategy of elements as potential predictor variables of group memberships in the duck dataset by running multi-nomial logistic regressions with the functions *multinom()* in package 'nnet' [96] using membership outputs of the four grouping methods as response variables. As a model selection procedure, we first used the function *step()* available in {stats} [94]. This function compares the AIC values of each model that results from dropping predictors from the original, multi-predictor model in a stepwise manner and identifies the model that has the lowest AIC value. We then calculated the second-order AIC (AICc) values for these models with the function *AICc()* in package 'MuMIn' [97] to check AIC overfitting in our comparatively small dataset [98]. Finally, we used the function *fitted()* in {stats} [94] to extract memberships for each bone predicted by the most supported models and compared their performance based on the number of matches between the actual and predicted memberships. See complete datasets, detailed R scripts with analysis set-ups for each case, and results of all analyses in the electronic supplementary material, table S1 and Information S2–S6.

Results were interpreted in various contexts, such as the developmental strategies of different elements within an individual, individual variation within a cohort of a species, ontogenetic patterns of development of homologous elements and the whole skeleton, and broader within-clade and interspecific comparisons.

# 3. Results

## 3.1. Qualitative evaluation of radial porosity profiles

Inspection of the simple graphical representation of RPPs reveals interesting patterns concerning the developmental trajectories of limb elements (figure 3; electronic supplementary material, figures S4–S18). In this qualitative description, we mostly focus on the ducks where the ontogenetic spectrum and the presence of three individuals in each age group (hereafter cohort) up to 30 dph allowed more detailed interpretations (electronic supplementary material, figures S4–S12). Other taxa in our sample (electronic supplementary material, figures S13–S18) are considered in the context of the findings in ducks.

### 3.1.1. Ducks

Although sample size is obviously inadequate to properly assess intracohort variance in RPPs, the cohorts containing three specimens in the duck growth series seem to show certain ontogenetic trends in the course of the trajectories and their variance that may be related to the disparate locomotor functions of the fore- and hind-limbs.

At 4 dph, neither the ulna nor the carpometacarpus has developed enough posthatching periosteal cortex to be evaluated. All the other analysed limb bones of the ducklings show high overall porosity levels throughout the cortex (approx. 20–70%) with erratic-looking RPPs. The intrasectional variance of RPPs (i.e. among the three sectors) lies within the same range as the intracohort variance among homologous bones, and no separation appears between altricial fore-limb and precocial hind-limb elements either in absolute porosity levels or in the course of RPPs (electronic supplementary material, figures S4, S9 and S10). By 8 dph, a trend appears in the hind-limb elements with more channelized RPP courses, that is, with lower intrasectional and intracohort variance in a well-defined course and radially increasing porosity, especially in the femur and tibiotarsus. On the other hand, wing elements show no such trend in their RPPs and continue to be largely irregular (electronic supplementary material, figures S5, S9 and S10). At 15 dph, the general pattern stays the same as at 8 dph, except that the humerus starts showing similar channelization, and porosity may increase in the inner cortex of some elements due to the formation of large resorption cavities related to medullary cavity expansion (figure 3; electronic supplementary material, figures S6, S9 and S10). Furthermore, overall porosity seems to be higher in the altricial wing bones, although the precocial tarsometatarsus remains in the same porosity range as the wing bones at this stage. By 30 dph, RPP channelization occurs in every element with overall higher porosity levels in the altricial wing bones (approx. 20–50%) than in the precocial leg bones (approx. 5–30%) (electronic supplementary material,

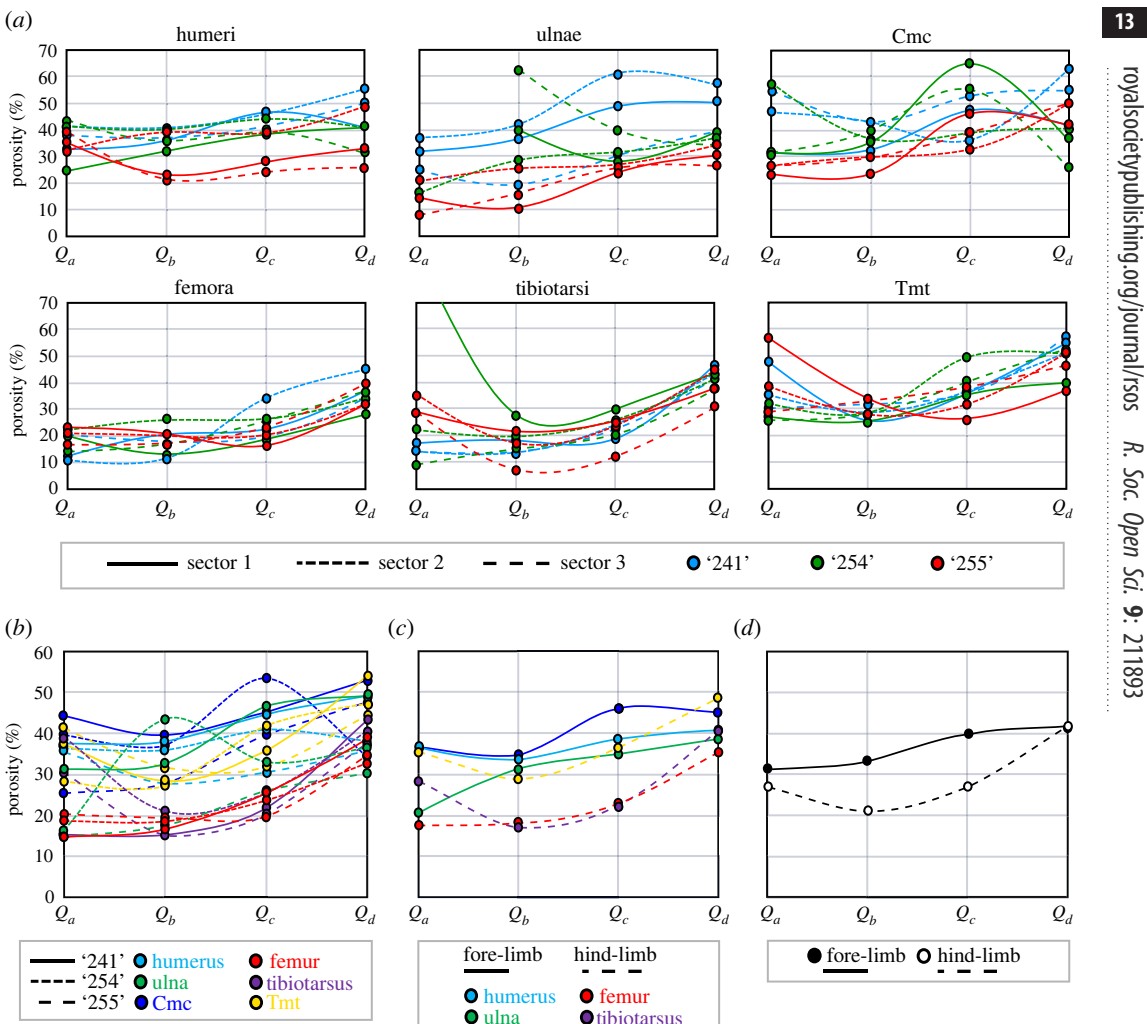

**Figure 3.** Various visual RPP assessments using graphical representations exemplified on the RPPs of the 15 dph cohort of ducklings. (a) RPPs of homologous bones in ducklings '241', '254' and '255' showing both intrasectional (among sectors) and intracohort (among specimens) variance at 15 dph. Note that the porosity value of $Q_a$ in a single tibial RPP is off the scale which is due to its $Q_a$ being occupied by a large resorption cavity. Line types and colours code the sectors and specimens, respectively, as indicated in the panel legend. (b) Intracohort variance among homologous bones narrowed down to mean RPPs averaged over all sectors in each element. Line types and colours code the specimens and elements, respectively, as indicated in the panel legend. (c) Mean RPPs averaged over homologous bones of pooled individuals within the cohort. Line types and colours code fore-limb versus hind-limb and elements, respectively, as indicated in the panel legend. (d) Mean RPPs averaged over all fore-limb versus all hind-limb elements of pooled duck individuals within the cohort. Line types code fore-limb versus hind-limb, as indicated in the panel legend. Abbreviations: cmc, carpometacarpus; tmt, tarsometatarsus.

figures S7, S9 and S10), although pooled bones of the fore-limb start to show higher overall porosity than hind-limb bones already from 8 dph (electronic supplementary material, figure S11). Porosity is generally the highest in the outer cortex (i.e. $P_d$ in $Q_d$) in all elements at 30 dph (electronic supplementary material, figures S7, S9 and S10). Nevertheless, extensive secondary remodelling, mostly in the form of CCCB, appears in one tibiotarsus and two tarsometatarsi, limiting primary porosity measurements to the outer half of the cortex in these elements. At 50 dph, overall porosity decreases further in every element compared to that at 30 dph (electronic supplementary material, figures S6–S8). Again, CCCB occupies a large part of the inner cortex in the tibiotarsus and tarsometatarsus, although there remain areas in the section where primary bone is present in the entire cortex thickness, and hence could be measured for RPP. Wing bones still show higher overall porosity with an increasing trend towards the outer cortex, whereas trajectories of hind-limb bones seem to be more levelled off and reach considerably lower $P_d$ (approx. 8–18%) than those of wing bones (approx. 20–40%) (electronic

supplementary material, figures S8–S10). Finally, in the two hind-limb elements of the adult *Anas* sp. (greater than or equal to 90 dph), overall primary porosity levels are the lowest measured in the ontogenetic series (approx. 2–12%) with an incipient OCL starting to form in the outermost cortex and exhibiting the least vascularity (electronic supplementary material, figures S8–S10). EL is only present in the tibiotarsus and even there only regionally. These features imply that had this animal lived longer it could still have developed a thick OCL, while adjusting relative cortex thickness via endosteal resorption and/or deposition of an EL. Furthermore, both elements reveal secondarily remodelled regions; it is low and patchy in the femur and more extensive in the inner cortical half of the tibiotarsus where only a single sector of primary bone remained to assess its RPP.

The same dynamically changing pattern through ontogeny is observed when fore- and hind-limb elements of all individuals are pooled within each cohort to generate mean RPPs of altricial wing versus precocial leg bones (electronic supplementary material, figure S11). Fore-limb RPPs generally run above hind-limb RPPs in all cohorts except for 4 dph. The shape of hind-limb RPPs with an overall increase in porosity towards $Q_d$ seems consistent through ontogeny from 8 dph to 50 dph, while fore-limb RPPs adopt a similar shape only from 30 dph onwards. As expected, the hind-limb RPP in the sole adult (greater than or equal to 90 dph) showed the lowest $P_d$ which is consistent with the presence of an incipient OCL. Unfortunately, no wing bones were available of this specimen for comparison.

Concerning RPP patterns among homologous elements through ontogeny of ducks, in all series there is a gradual decrease in overall cortical porosity (electronic supplementary material, figure S12). However, whereas homologous wing elements in the growth series tend to converge on a similar $P_d$ at least up to 30 dph, homologous leg bones instead show a stepwise decrease in their respective $P_d$ from 8 dph onwards. Nonetheless, the tarsometatarsus at 15 dph also seems to retain similar $P_d$ to those of earlier ontogenetic stages.

Even though not part of RPP evaluation itself, it is important to note that partial or complete EL was observed in several elements from early ontogeny on, apparently with a dynamically changing pattern of presence/absence, thickness and position as growth and maturation progress (see figure 1*e,f*). These observations, along with earlier reports (e.g. [65,99]), disprove the general and widespread assumption that EL is a hallmark of the cessation of medullary cavity expansion and radial bone growth (e.g. [22,26]).

### 3.1.2. Other taxa

RPPs of wing and leg bones of the juvenile hoatzins and the adult kestrel, woodcock and rook also reveal patterns that fit well within the working hypothesis of avian ontogeny and developmental strategy, as discussed in ducks.

RPPs of the two younger (approx. 6 and 9 dph) and the older (approx. 16 dph) juvenile hoatzins are roughly within the porosity range of duckling RPPs at 15 dph (approx. 20–60%) and 30 dph (approx. 10–50%), respectively (electronic supplementary material, figure S13). This is also in line with their estimated growth stages with regard to the length of their respective growth period (ducklings at 15 and 30 dph have covered about the sixth and third of their total approximately 90 days growth period; hoatzins at approximately 6, approximately 9 and approximately 16 dph are roughly at the eighth, fifth and third of their total approximately 50 days growth period). However, the course of the trajectories of the hoatzin wing bones deviates from that in the duckling wing bones at roughly corresponding porosity ranges and relative ontogenetic stages. Whereas RPPs of wing bones in ducklings show consistently higher porosity levels than RPPs of femora and tibiotarsi at 15 and 30 dph (electronic supplementary material, figure S8), this fore- and hind-limb divergence in RPPs cannot be observed in the hoatzins, in which wing and leg bone RPPs stay in about the same porosity range in all three specimens (electronic supplementary material, figure S11). In fact, when fore- and hind-limb elements in each specimen are pooled, respectively, the hind-limb RPP runs slightly above the fore-limb RPP in the approximately 6 and approximately 9 dph chicks, whereas the fore- and hind-limb RPPs intertwine in the oldest, approximately 16 dph, chick (electronic supplementary material, figure S14). It has to be emphasized that among all elements, porosity is outstandingly the highest in the tarsometatarsi in the two younger hoatzins. A similar trend of higher porosity of tarsometatarsi as compared to femora and tibiotarsi was also observed in ducks up to 15 dph, albeit tarsometatarsal porosity did not exceed that of the wing bones in these ducklings (electronic supplementary material, figures S9 and S10), unlike the pattern found in hoatzins. This high tarsometatarsal porosity can explain why hind-limb RPPs run slightly above fore-limb RPPs in the two younger hoatzin chicks; a pattern that disappears in the

oldest chick (electronic supplementary material, figure S14). Finally, RPP channelization characterizes all elements in these hoatzin specimens with wing and leg bones showing not only similar porosity ranges but also similar shapes of RPPs in each specimen, respectively (electronic supplementary material, figure S13).

The adult kestrel, woodcock and rook consistently reveal low porosity with little variance across the cortex, and hence largely flat RPPs plateaued at approximately 5–10% porosity which eventually drops down to almost 0% in $Q_d$ (electronic supplementary material, figure S15). The only exception seems to be the kestrel in which porosity roughly stays at 10% with its tibia even showing greater than 15% porosity in $Q_a$. The latter is due to large resorption cavities in some areas of the innermost cortex where the bone seems to be partially remodelled, skewing the RPPs. When compared to the other adult birds, the overall histology of this kestrel specimen also reveals a relatively thin OCL in its leg bones and only incipient or missing OCL in its wing bones, while an EL is absent in all of its bones. Along with the active bone resorption from the endosteal surface of the medullary cavity, these features indicate that this kestrel was not yet skeletally fully mature, and it would have continued to thicken OCL as a residual growth in bone diameter and would have developed an EL, had it lived longer. Although wing bones have slightly higher $P_d$ values than leg bones, which reflects the lack of a definite OCL in these elements, the difference is not as pronounced as to be evident in the RPPs at this developmental stage (electronic supplementary material, figures S15 and S16). Nevertheless, the fact that there are still detectable differences in the overall osteohistological maturity between the wing and leg bones in this kestrel suggests a certain level of disparity in the growth trajectories of fore- and hind-limb bones in earlier stages of development.

Limb elements of the adult woodcock and rook show the same general RPP characteristics as the hind-limb elements in the adult duck (electronic supplementary material, figures S15 and S16). Here, no disparity between wing and leg bones is evident in overall histology with all having a well-defined OCL and EL (electronic supplementary material, figure S16). In general, OCL makes up most of $Q_d$ in the bones of these birds, and, accordingly, RPPs converge on very low $P_d$ levels (0–5%) in the outermost cortex (electronic supplementary material, figure S15). Furthermore, tarsometatarsi in both specimens are heavily remodelled to the extent that only the OCL remains as primary cortex. Unfortunately, tarsometatarsi were unavailable for comparison in the kestrel and the adult duck. However, tarsometatarsi showed extensive remodelling in the inner half of the cortex already at 30 dph in all three ducklings, and a less extensive but sensible remodelling in the sole duck specimen available at 50 dph. The high degrees of secondary remodelling in the inner cortical half of tarsometatarsi deprive us of an informed consideration of their RPPs in these later ontogenetic stages.

As for the extinct paravian dinosaurs, the ontogenetic range of the sample is incomparably smaller than that available for the extant birds. Four of the five individuals were described as subadults–adults, and only *Eosinopteryx* as a (late) juvenile, based on their comparative overall osteohistological maturity [33]. Mean RPPs of these paravians averaged over all sectors in each element (electronic supplementary material, figure S17) stayed within the range of 0–9% porosity along their full length, with the humerus of *Eosinopteryx* representing the highest porosity range (7–9%). Furthermore, $P_d$ levels in all other elements of *Eosinopteryx* also cover the highest range (approx. 2–4%), compared to those of the other specimens. These data correspond with the earlier observation that *Eosinopteryx* is the least mature 'dinobird' in the sample and its humerus is relatively the most altricially developing element in its skeleton [33]. *Serikornis* elements reveal the second-highest $P_d$ range (1.2–3.5%), again supporting its identification as the second 'youngest' (late juvenile or subadult) specimen in the sample [33]. RPPs of all other specimens converge on less than 3% $P_d$ levels (electronic supplementary material, figure S17).

Interestingly, and in contrast with the general pattern seen in our extant bird sample, an EL is present in all sampled elements of these fossil paravians, while a definite OCL is missing in all *Eosinopteryx* bones, and in some bones of the other, more mature specimens. Even though only femora could be sampled of the hind-limbs and compared to fore-limb bones in these extinct paravians, femoral RPPs seem to be neatly aligned with RPPs of the fore-limb elements showing no evident deviation at these ontogenetic stages (electronic supplementary material, figures S17 and S18).

## 3.2. Numeric analyses of radial porosity profiles

In essence, all analyses of the raw data by K-means, GBTM and similarity measures (DTW and ED combined with K-means) gave similar results for each dataset, that is, for the ducks, extinct paravians and the complete dataset (figures 4–6; electronic supplementary material, figures S19–S21 and

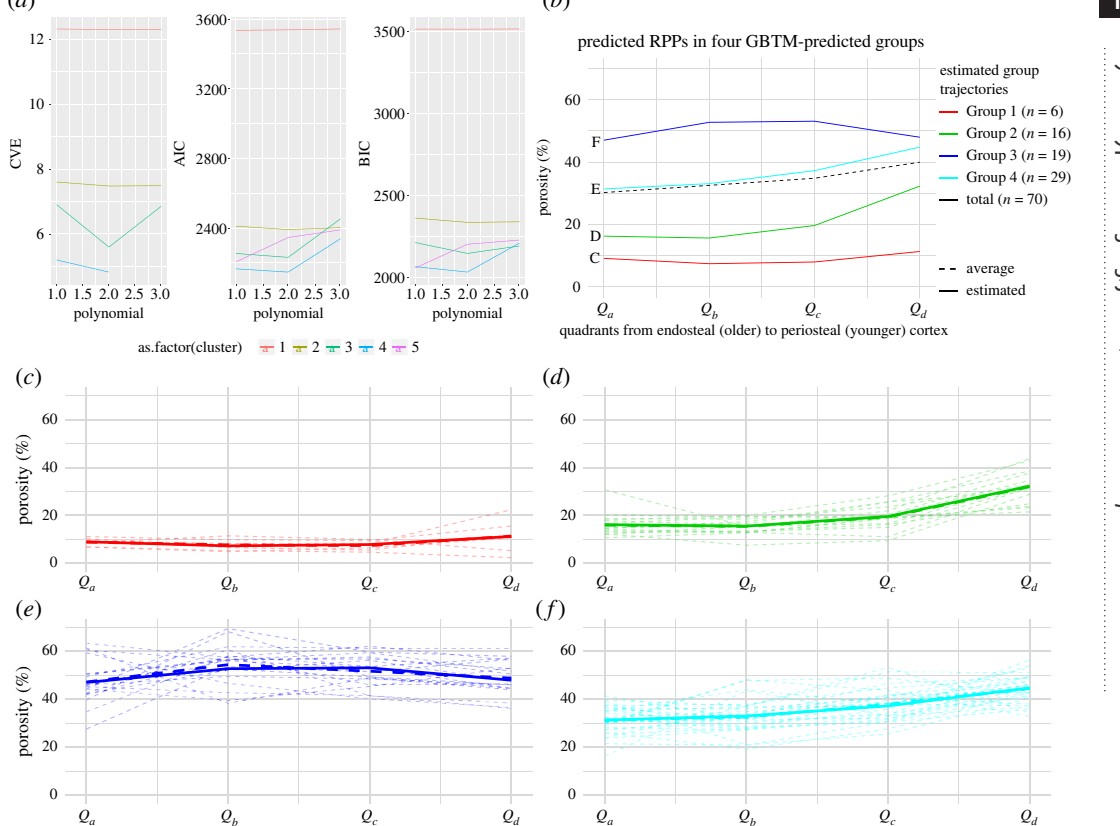

**Figure 4.** Summary of results of the GBTM analysis of the duck RPP dataset. (*a*) Model selection output of CVE, AIC and BIC showing that the overall best-supported group number is four with a polynomial degree of two. (*b*) Predicted trajectories of the four groups of RPPs indicated by different colours and the mean RPP averaged over the dataset shown with a dashed line. (*c*–*f*) RPP distribution pattern in each group overlain by their respective estimated trajectories.

Information S4–S6). On the other hand, centred data and LCSS gave quite erratic memberships in each case which apparently did not relate to any studied parameters (electronic supplementary material, figures S22 and S23). Therefore, only the raw data-based results are shown here, whereas those of centred data and LCSS are considered further only in the Discussion.

### 3.2.1. Ducks

When the duck growth series are analysed with 3, 4 and 5 groups for describing trajectory types from 4 dph up to adulthood, the 4-group set-up consistently represents the best model across different analysis methods (electronic supplementary material, figure S19). Numeric support is provided by GBTM iterations in which the 4-group model shows the most frequently, as well as overall, the lowest AIC, BIC and CVE values (figure 4). RPP group compositions, excluding two bones with incomplete RPPs, are fairly consistent among different analyses with 61% of the elements clustered the same way by all four methods, and another 27% by three of the four (table 2). The most inconsistency in membership assignments of bones across methods occurred in the 15 dph cohort, followed by 30 dph. Group memberships appear to largely reflect the combination of the individual's ontogenetic stage and the developmental strategy of the element (figure 5). Accordingly, these RPP groups (figure 5*a*) tend to 'dissociate' an animal's skeleton along its differently developing limb modules. For instance, in both *K*-means and GBTM, the precocially developing hind-limb bones of the subadult (50 dph) duck are grouped together with bones of adult animals, while its altricially developing wing bones are assigned to a separate group composed mostly of bones representing earlier ontogenetic stages, such as mid- and late juveniles. Similarly, fore-limb versus hind-limb bones of late juveniles (30 dph) tend to be grouped together with ontogenetically and/or functionally less versus more mature elements, respectively (figure 5*b* and table 2).

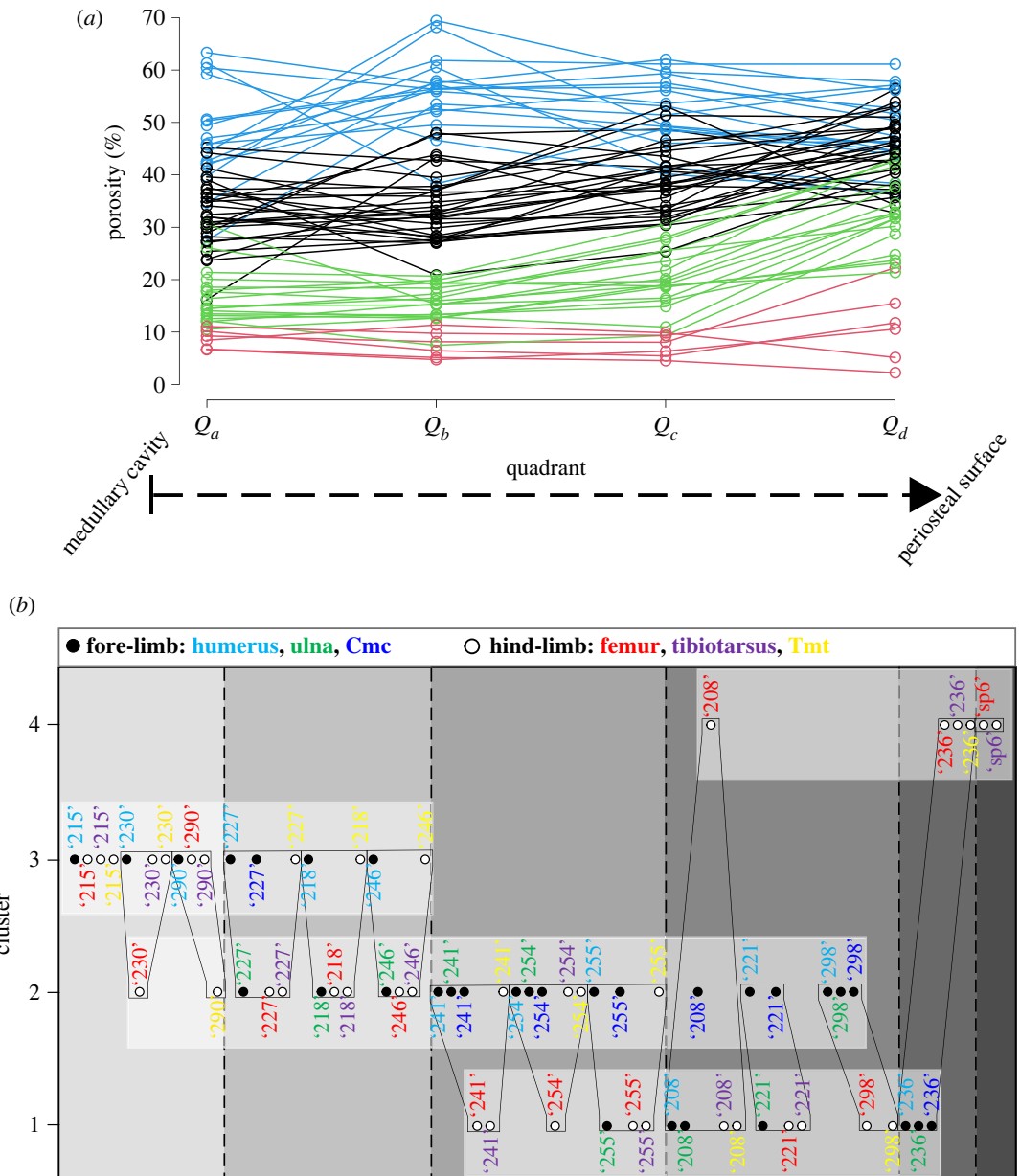

**Figure 5.** 4-group *K*-means clustering of the duck RPP dataset—excluding two hind-limb elements with missing data points—demonstrated by different graphical representations. (*a*) Raw RPPs with their cluster memberships indicated by the four different colours. (*b*) Cluster compositions with each element identifiable within their respective groups. Elements are sorted along the horizontal axis by the age of the ducks. Black polygons connect bones belonging to the same individual. Note how cluster memberships within cohorts reflect the characteristics of skeletal dissociation (i.e. assignment of an individual's bones into different clusters) as ontogeny progresses, showing a trend of separation of certain wing and leg elements. Colours indicate homologous elements as shown in the legend. Element IDs as given in the electronic supplementary material, table S1.

At this analytical resolution encompassing the entire ontogenetic series up to adulthood, group memberships also reveal that the predictive power of precocial versus altricial development in separation of element RPPs changes throughout ontogeny. The overall pattern in the dynamics of RPP disparity can be traced in group compositions along the age axis combined with the level of skeletal dissociation of specimens (figure 5*b* and table 3). In the analyses with four groups, skeletal dissociation is not strong at 4 dph (i.e. all bones within an individual tend to be clustered in a single group) but starts to appear more consistently at 8 dph. At this age, the wing and leg bones of an individual tend to be separated into two groups, although in most analyses the ulnae and

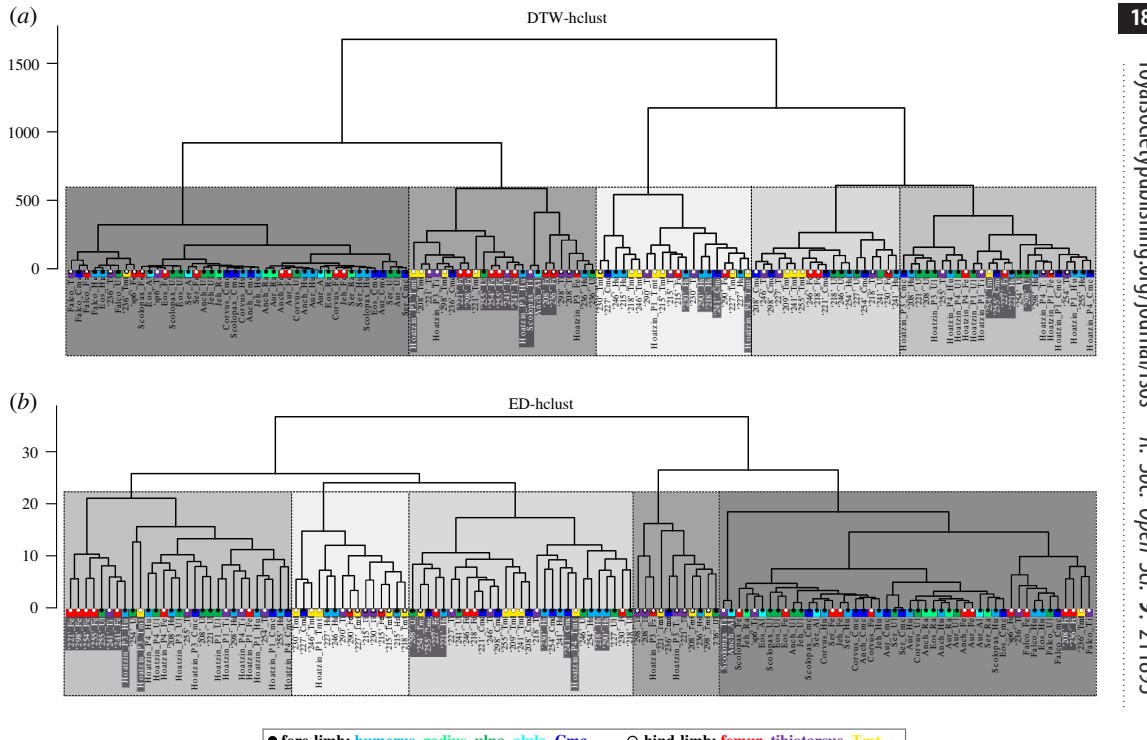

**Figure 6.** Clustering of the complete RPP dataset—including RPPs with missing points—by DTW-hclust (*a*) and ED-hclust (*b*) with cut-off points at five groups demonstrated on dendrograms. Clusters are indicated by five different shades of grey with each shade representing a corresponding cluster-pair between (*a*) DTW-hclust and (*b*) ED-hclust dendrograms. Corresponding cluster-pairs were assessed based on the high congruency in cluster composition. Elements that have differing cluster assignments in DTW-hclust and ED-hclust are highlighted by white font against a dark grey background. Overall, 82% of the elements are clustered identically. Colours indicate homologous bones, black and white circles indicate fore- and hind-limb elements, respectively, as shown in the legend. Element IDs as given in the electronic supplementary material, table S1.

tarsometatarsi are grouped the opposite way: the ulnae with the femora and tibiotarsi, and the tarsometatarsi with the humeri and the single sampled carpometacarpal (table 2). The level of skeletal dissociation increases further at 15 and 30 dph where the scattering of an individual's bones across three different clusters regularly occurs and reaches its maximum in the 15 dph cohort with one specimen dissociated into four clusters by ED-hclust (table 3). At 50 dph, skeletal dissociation returns to a bipartite division with perfect separation of wing and leg bones by GBTM and *K*-means (figure 5*b* and table 2). Figure 5*b*, which shows the sorting of bones by *K*-means, further illustrates these dynamics in clusters no. 1 and no. 2 which contain all elements of 15 dph and all but one of 30 dph. The distribution pattern in these clusters is a transition from the younger ages (left-hand side) being mostly represented by precocial hind-limb bones (white circles) to the older ages (right-hand side) containing predominantly altricial fore-limb bones (black circles). The mid-age range in each cluster shows a more balanced mixture of fore- and hind-limb bones in the respective groups. More specifically, in cluster no. 2, the number of fore- versus hind-limb elements belonging to early juveniles (4 and 8 dph) is 3 versus 8, in mid-juveniles (15 dph) 8 versus 4 and in late juveniles (30 dph) 6 versus 0. In cluster no. 1, the same numbers are 1 versus 5 at 15 dph, 3 versus 6 at 30 dph and 3 versus 0 at 50 dph. Thus, modularity in fore- and hind-limb development results in groups spanning three *ad hoc* maturity degrees (from early to late juveniles in no. 2 and from mid-juveniles to subadults in no. 1) and clustering together elements that originate from very differently aged animals (figure 5*b* and table 2).

To further unfold how this complex RPP pattern might be related to the disparate growth and development of altricial versus precocial elements, we separately analysed by GBTM and *K*-means the subsets of those ontogenetic stages where we had at least three animals in each category, that is, for early (4–8 dph), mid- (15 dph) and late juveniles (30 dph), respectively (electronic supplementary material, table S2). Here, we set the maximum number of groups to 2 to see whether and how precocial versus altricial strategy is reflected in the RPP group compositions through ontogeny. These higher resolution analyses showed a similar trend to that recovered in the analysis spanning the entire

**19**

**Table 2.** Group memberships of duck elements, excluding bones with incomplete RPPs, and their congruence across the four methods (GBTM, DTW-hclust, ED-hclust and *K*-means) analysed in the 4-group set-up. Colours indicate corresponding group memberships within and across methods. Shades of grey indicate group membership congruence of each element across methods. Inter-method congruence refers to the percentage of cases in which the given method-pair grouped elements the same way. Overall congruence of the methods is calculated as the sum of paired congruence percentages with each of the other methods. Note the changing pattern in group memberships related to the individual's ontogenetic stage and the developmental strategy of the element. Element and specimen IDs and abbreviations as in the electronic supplementary material, table S1.

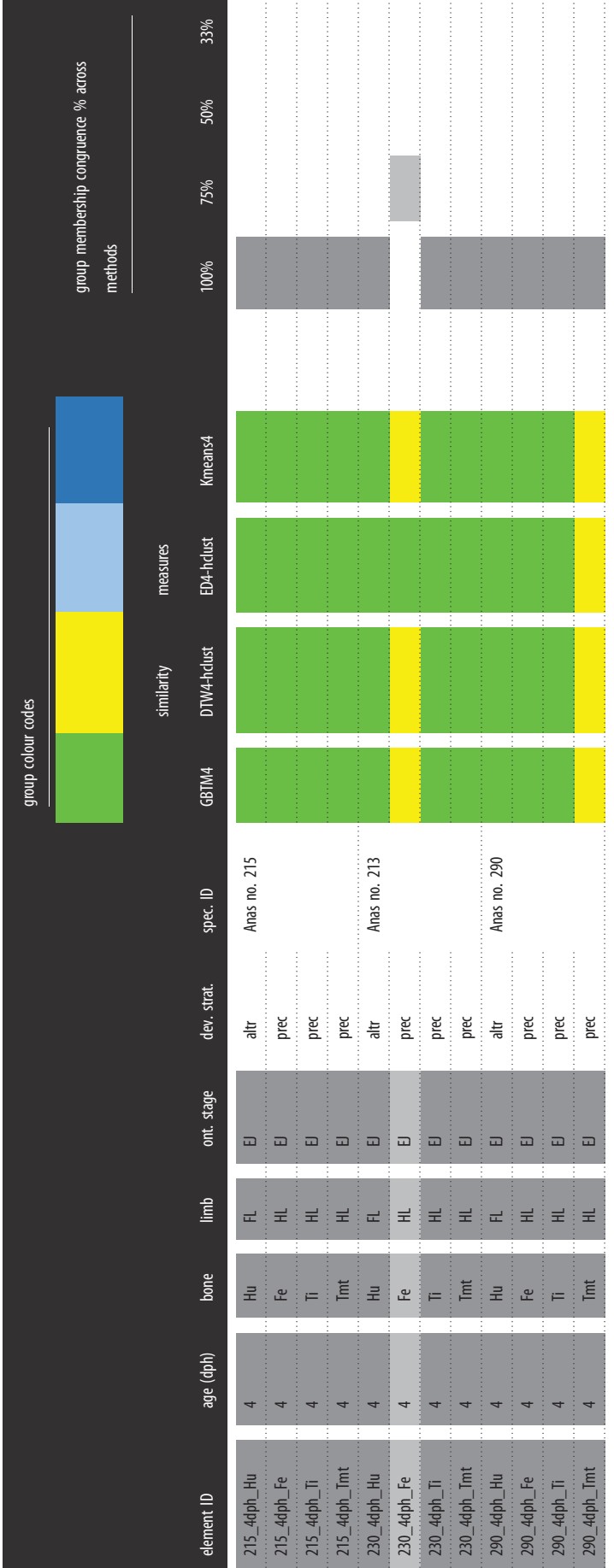

| element ID | age (dph) | bone | limb | ont. stage | dev. strat. | spec. ID | GBTM4 | DTW4-hclust | ED4-hclust | Kmeans4 | | 100% | 75% | 50% | 33% |
|---|---|---|---|---|---|---|---|---|---|---|---|---|---|---|---|
| 215_4dph_Hu | 4 | Hu | FL | EJ | altr | Anas no. 215 | | | | | | | | | |
| 215_4dph_Fe | 4 | Fe | HL | EJ | prec | | | | | | | | | | |
| 215_4dph_Ti | 4 | Ti | HL | EJ | prec | | | | | | | | | | |
| 215_4dph_Tmt | 4 | Tmt | HL | EJ | prec | | | | | | | | | | |
| 230_4dph_Hu | 4 | Hu | FL | EJ | altr | Anas no. 213 | | | | | | | | | |
| 230_4dph_Fe | 4 | Fe | HL | EJ | prec | | | | | | | | | | |
| 230_4dph_Ti | 4 | Ti | HL | EJ | prec | | | | | | | | | | |
| 230_4dph_Tmt | 4 | Tmt | HL | EJ | prec | | | | | | | | | | |
| 290_4dph_Hu | 4 | Hu | FL | EJ | altr | Anas no. 290 | | | | | | | | | |
| 290_4dph_Fe | 4 | Fe | HL | EJ | prec | | | | | | | | | | |
| 290_4dph_Ti | 4 | Ti | HL | EJ | prec | | | | | | | | | | |
| 290_4dph_Tmt | 4 | Tmt | HL | EJ | prec | | | | | | | | | | |

group colour codes

similarity        measures

group membership congruence % across methods

(*Continued.*)

**Table 2.** (*Continued.*)

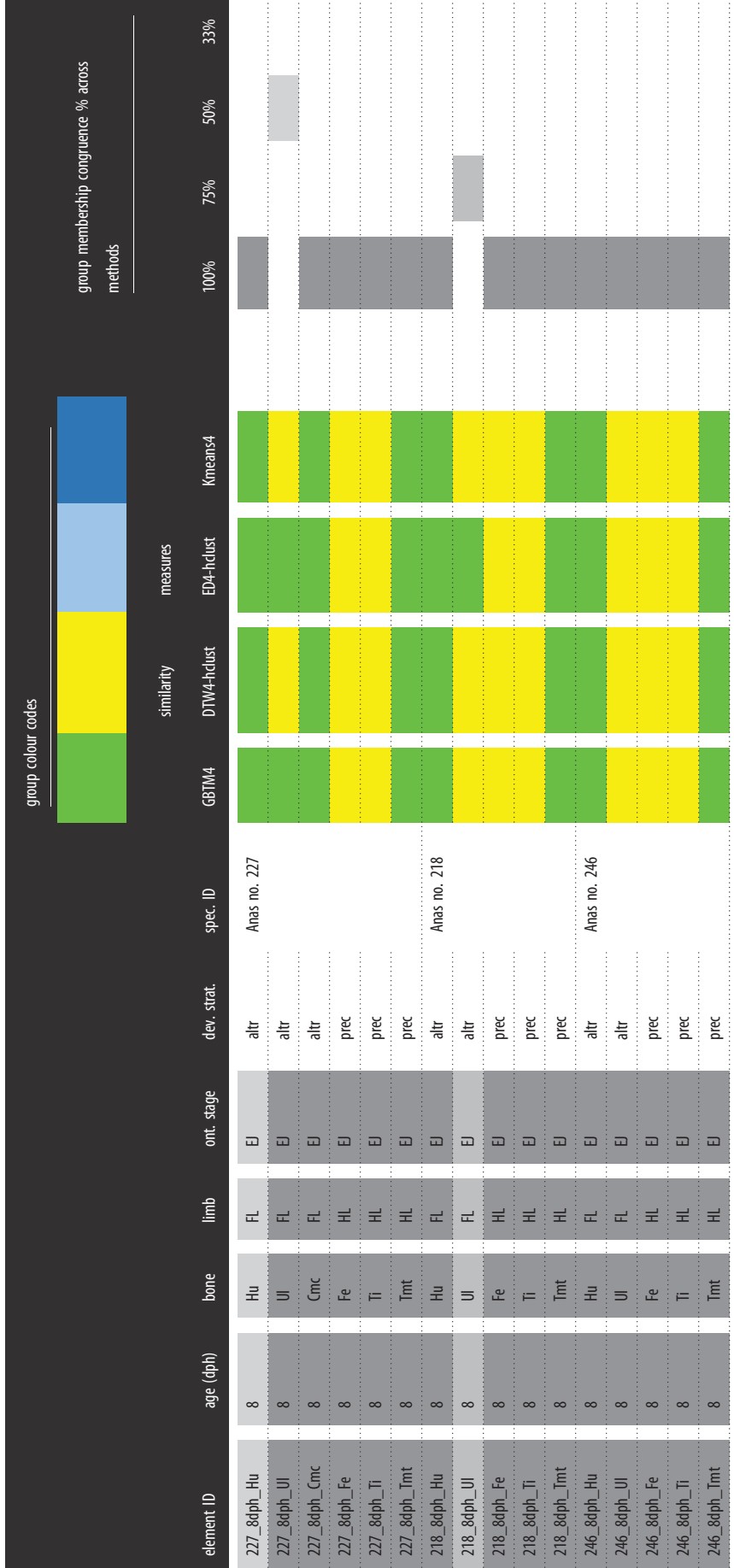

| element ID | age (dph) | bone | limb | ont. stage | dev. strat. | spec. ID | group colour codes | | | | group membership congruence % across methods | | | |
|---|---|---|---|---|---|---|---|---|---|---|---|---|---|---|
| | | | | | | | GBTM4 | DTW4-hclust | ED4-hclust | Kmeans4 | 100% | 75% | 50% | 33% |
| 227_8dph_Hu | 8 | Hu | FL | EJ | altr | Anas no. 227 | | | | | | | | |
| 227_8dph_Ul | 8 | Ul | FL | EJ | altr | | | | | | | | | |
| 227_8dph_Cmc | 8 | Cmc | FL | EJ | altr | | | | | | | | | |
| 227_8dph_Fe | 8 | Fe | HL | EJ | prec | | | | | | | | | |
| 227_8dph_Ti | 8 | Ti | HL | EJ | prec | | | | | | | | | |
| 227_8dph_Tmt | 8 | Tmt | HL | EJ | prec | | | | | | | | | |
| 218_8dph_Hu | 8 | Hu | FL | EJ | altr | Anas no. 218 | | | | | | | | |
| 218_8dph_Ul | 8 | Ul | FL | EJ | altr | | | | | | | | | |
| 218_8dph_Fe | 8 | Fe | HL | EJ | prec | | | | | | | | | |
| 218_8dph_Ti | 8 | Ti | HL | EJ | prec | | | | | | | | | |
| 218_8dph_Tmt | 8 | Tmt | HL | EJ | prec | | | | | | | | | |
| 246_8dph_Hu | 8 | Hu | FL | EJ | altr | Anas no. 246 | | | | | | | | |
| 246_8dph_Ul | 8 | Ul | FL | EJ | altr | | | | | | | | | |
| 246_8dph_Fe | 8 | Fe | HL | EJ | prec | | | | | | | | | |
| 246_8dph_Ti | 8 | Ti | HL | EJ | prec | | | | | | | | | |
| 246_8dph_Tmt | 8 | Tmt | HL | EJ | prec | | | | | | | | | |

Colour code legend: GBTM4, DTW4-hclust (similarity), ED4-hclust (measures), Kmeans4.

(*Continued.*)

**Table 2.** (*Continued.*)

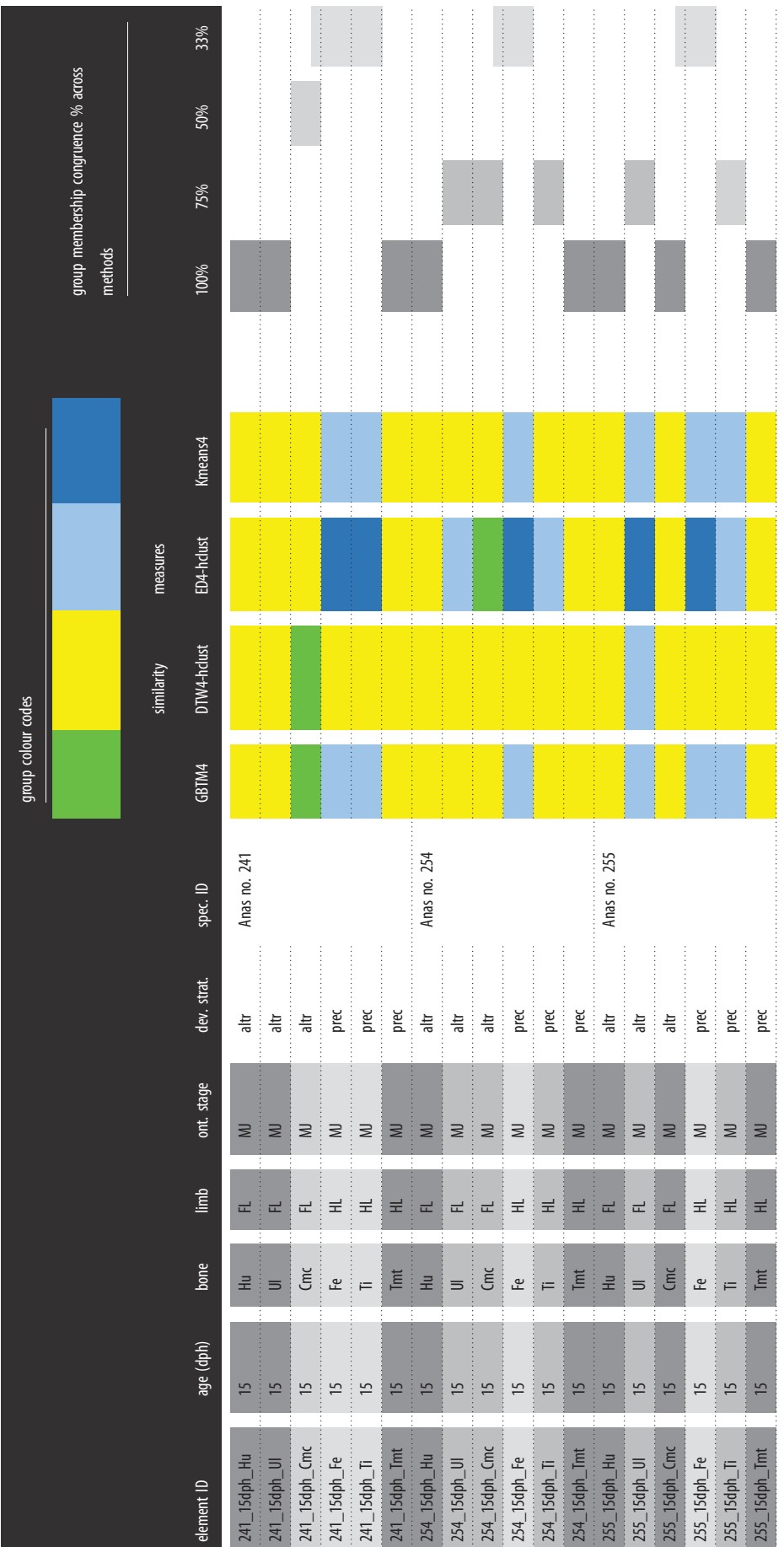

| element ID | age (dph) | bone | limb | ont. stage | dev. strat. | spec. ID | GBTM4 | DTW4-hclust | ED4-hclust | Kmeans4 | 100% | 75% | 50% | 33% |
|---|---|---|---|---|---|---|---|---|---|---|---|---|---|---|
| 241_15dph_Hu | 15 | Hu | FL | MJ | altr | Anas no. 241 | | | | | | | | |
| 241_15dph_Ul | 15 | Ul | FL | MJ | altr | | | | | | | | | |
| 241_15dph_Cmc | 15 | Cmc | FL | MJ | altr | | | | | | | | | |
| 241_15dph_Fe | 15 | Fe | HL | MJ | prec | | | | | | | | | |
| 241_15dph_Ti | 15 | Ti | HL | MJ | prec | | | | | | | | | |
| 241_15dph_Tmt | 15 | Tmt | HL | MJ | prec | | | | | | | | | |
| 254_15dph_Hu | 15 | Hu | FL | MJ | altr | Anas no. 254 | | | | | | | | |
| 254_15dph_Ul | 15 | Ul | FL | MJ | altr | | | | | | | | | |
| 254_15dph_Cmc | 15 | Cmc | FL | MJ | altr | | | | | | | | | |
| 254_15dph_Fe | 15 | Fe | HL | MJ | prec | | | | | | | | | |
| 254_15dph_Ti | 15 | Ti | HL | MJ | prec | | | | | | | | | |
| 254_15dph_Tmt | 15 | Tmt | HL | MJ | prec | | | | | | | | | |
| 255_15dph_Hu | 15 | Hu | FL | MJ | altr | Anas no. 255 | | | | | | | | |
| 255_15dph_Ul | 15 | Ul | FL | MJ | altr | | | | | | | | | |
| 255_15dph_Cmc | 15 | Cmc | FL | MJ | altr | | | | | | | | | |
| 255_15dph_Fe | 15 | Fe | HL | MJ | prec | | | | | | | | | |
| 255_15dph_Ti | 15 | Ti | HL | MJ | prec | | | | | | | | | |
| 255_15dph_Tmt | 15 | Tmt | HL | MJ | prec | | | | | | | | | |

group colour codes

similarity    measures

group membership congruence % across methods

(*Continued.*)

**Table 2.** (Continued.)

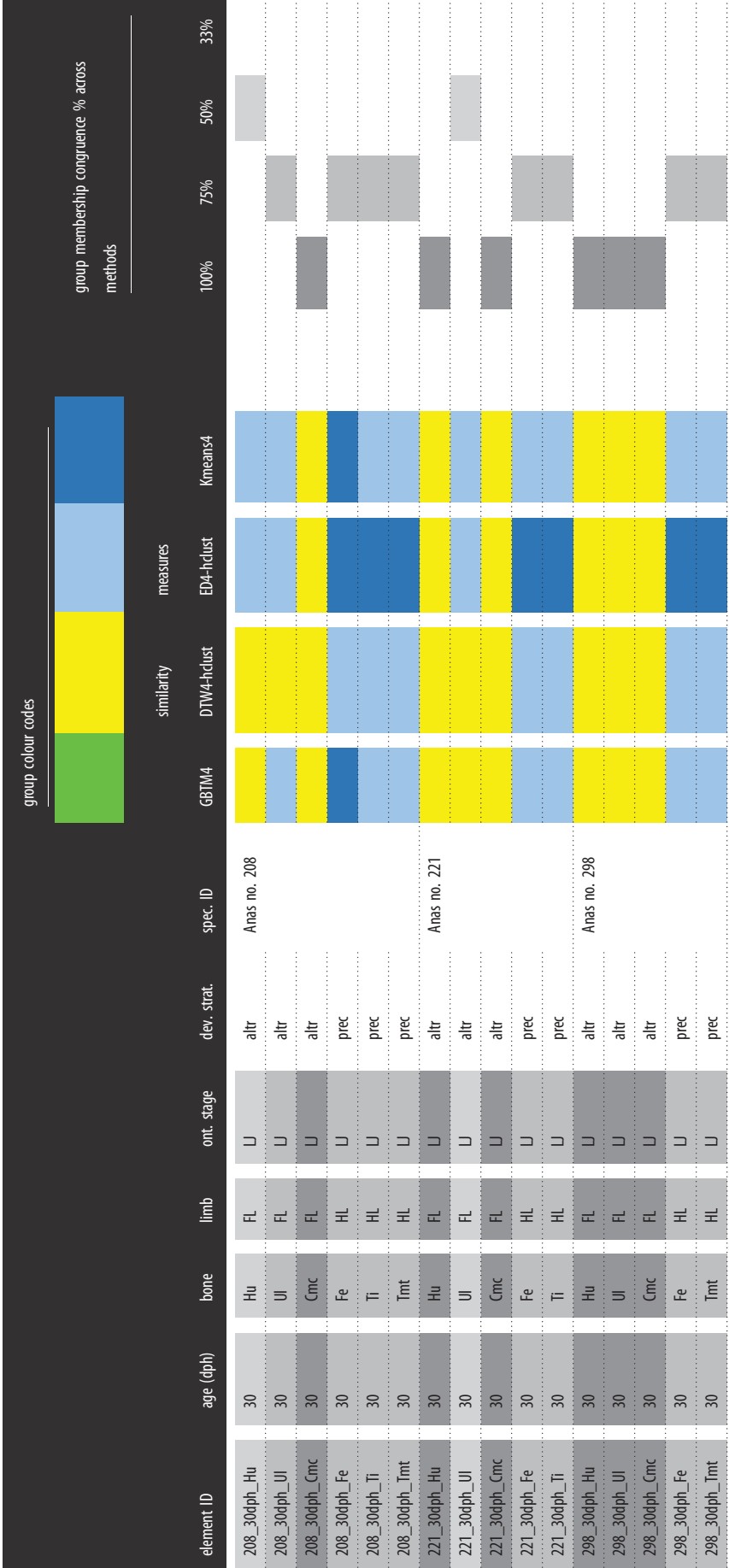

| element ID | age (dph) | bone | limb | ont. stage | dev. strat. | spec. ID | group colour codes GBTM4 | DTW4-hclust | ED4-hclust | Kmeans4 | group membership congruence % across methods |
|---|---|---|---|---|---|---|---|---|---|---|---|
| 208_30dph_Hu | 30 | Hu | FL | LJ | altr | Anas no. 208 | | | | | |
| 208_30dph_Ul | 30 | Ul | FL | LJ | altr | | | | | | |
| 208_30dph_Cmc | 30 | Cmc | FL | LJ | altr | | | | | | |
| 208_30dph_Fe | 30 | Fe | HL | LJ | prec | | | | | | |
| 208_30dph_Ti | 30 | Ti | HL | LJ | prec | | | | | | |
| 208_30dph_Tmt | 30 | Tmt | HL | LJ | prec | | | | | | |
| 221_30dph_Hu | 30 | Hu | FL | LJ | altr | Anas no. 221 | | | | | |
| 221_30dph_Ul | 30 | Ul | FL | LJ | altr | | | | | | |
| 221_30dph_Cmc | 30 | Cmc | FL | LJ | altr | | | | | | |
| 221_30dph_Fe | 30 | Fe | HL | LJ | prec | | | | | | |
| 221_30dph_Ti | 30 | Ti | HL | LJ | prec | | | | | | |
| 298_30dph_Hu | 30 | Hu | FL | LJ | altr | Anas no. 298 | | | | | |
| 298_30dph_Ul | 30 | Ul | FL | LJ | altr | | | | | | |
| 298_30dph_Cmc | 30 | Cmc | FL | LJ | altr | | | | | | |
| 298_30dph_Fe | 30 | Fe | HL | LJ | prec | | | | | | |
| 298_30dph_Tmt | 30 | Tmt | HL | LJ | prec | | | | | | |

Group membership congruence % across methods: 100% | 75% | 50% | 33%

Measures: similarity, GBTM4, DTW4-hclust; ED4-hclust, Kmeans4

**Table 2.** (*Continued.*)

| element ID | age (dph) | bone | limb | ont. stage | dev. strat. | spec. ID | GBTM4 | DTW4-hclust | ED4-hclust | Kmeans4 |
|---|---|---|---|---|---|---|---|---|---|---|
| 236_50dph_Hu | 50 | Hu | FL | SA | altr | Anas no. 236 | | | | |
| 236_50dph_Ul | 50 | Ul | FL | SA | altr | | | | | |
| 236_50dph_Cmc | 50 | Cmc | FL | SA | altr | | | | | |
| 236_50dph_Fe | 50 | Fe | HL | SA | prec | | | | | |
| 236_50dph_Ti | 50 | Ti | HL | SA | prec | | | | | |
| 236_50dph_Tmt | 50 | Tmt | HL | SA | prec | | | | | |
| Anas_sp6_Fe | ≥90 | Fe | HL | A | prec | Anas no. 6 | | | | |
| Anas_sp6_Ti | ≥90 | Ti | HL | A | prec | | | | | |

group colour codes

similarity | measures

group membership congruence % across methods

| | 100% | 75% | 50% | 33% |
|---|---|---|---|---|

| | no. 43 | no. 19 | no. 4 | no. 4 |
|---|---|---|---|---|
| no. bones | | | | |
| % bones | 61% | 27% | 6% | 6% |

inter-method congruence (%)

| | GBTM4 | DTW4 | ED4 | Kmeans4 |
|---|---|---|---|---|
| GBTM4 | 100% | | | |
| DTW4 | 87% | 100% | | |
| ED4 | 69% | 61% | 100% | |
| Kmeans4 | 94% | 84% | 71% | 100% |

overall congruence

| | |
|---|---|
| GBTM4 | 87 + 69 + 94 = 250 |
| DTW4 | 87 + 61 + 84 = 232 |
| ED4 | 69 + 61 + 71 = 201 |
| Kmeans4 | 94 + 84 + 71 = 249 |

ontogenetic series but provided a much clearer pattern and stronger signals related to the developmental strategies of individual bones.

The major trend is still the progressively clearer separation of the RPPs of altricially versus precocially developing fore- versus hind-limb elements as ontogeny progresses. GBTM of early juveniles (4–8 dph) showed no separation of bones at 4 dph, whereas RPPs of differently developing elements start to differ more evidently at 8 dph with all of the altricial wing bones being grouped together with bones of 4 dph, while all femora and tibiotarsi of the precocial leg bones are grouped separately (electronic supplementary material, table S2). Nevertheless, tarsometatarsi at 8 dph are clustered with the wing bones, and the distinctness of the groups is still weak, which is also reflected in the closeness of support for both number of groups (1 and 2) provided by all GBTM model selection methods (electronic supplementary material, Information S4). $K$-means gave very similar results, although the ulnae at 8 dph were consistently assigned to the same cluster as the femora and tibiotarsi. On the other hand, in late juveniles (30 dph), both GBTM and $K$-means were able to identify and separate precocially and altricially developing bones with 100% accuracy based on their RPPs. Finally, the strength of separation of RPPs of precocial and altricial elements in mid-juveniles (15 dph) is intermediate with a less distinct group composition than in late juveniles but higher support for the two groups than in early juveniles by all GBTM model selection methods (electronic supplementary material, table S2, Information S4).

As for individual elements, the altricial ulna and the precocial tarsometatarsus show the most unexpected group memberships in all analyses concerning their development. On the one hand, from 8 dph (the earliest age of a measurable ulnar cortex in our sample) up to 30 dph, the ulnae tend to group together with their contemporary precocial femora and tibiae. On the other hand, the tarsometatarsus usually shows a highly porous cortex as compared with the other two precocial hind-limb elements, the femur and the tibiotarsus, and hence is frequently grouped together with the altricial wing elements within the same skeleton.

Based on our working hypothesis and the observed patterns in group compositions in the entire duck dataset and its cohort subsets, age, bone type and developmental strategy of bones were selected to be tested as predictors of group memberships resulting from the different grouping methods with four groups. Even though bones belonging to the same specimen are clearly not independent, the changing level of skeletal dissociation through ontogeny (table 3) implies that it is not a main constraint in group memberships. Nevertheless, we first tested 'specimen' as the only predictor in a null model, after which we built a complex multi-nomial logistic regression model including specimen as well as all other predictors in linear combination and ran an AIC-based stepwise model selection.

In each case, dropping 'specimen' from predictors resulted in the largest drop in AIC values (electronic supplementary material, Information S4). This selection procedure supported the linear combination of the predictors' age and developmental strategy of the bone in GBTM, and age, type and developmental strategy of the bone in $K$-means, DTW-hclust and ED-hclust. On the other hand, AICc values were the lowest for the simpler 'age + developmental strategy' model in $K$-means, DTW-hclust and ED-hclust too. However, when the actual group memberships were compared with the memberships predicted by these models, the simpler, two-predictor model underperformed the three-predictor models in each case. While memberships predicted by the 'age + bone + developmental strategy' models matched greater than or equal to 89% of the actual group memberships of elements, this percentage value was less than or equal to 74% for the 'age + developmental strategy' models. In GBTM, in which only the simpler two-predictor model was supported, 76% of the bones matched their original GBTM group membership (table 4). Based on the stepwise model selection and the $p$-values calculated for the predictors, all multi-nomial logistic regressions showed that age is overall the strongest predictor of group memberships, followed by developmental strategy, and then the type of bone. For further details see electronic supplementary material, Information S4.

### 3.2.2. Fossil paravian dinosaurs

Different types of analyses of the five specimens of fossil paravian dinosaurs gave moderately variable results. Based on a previous ontogenetic study of the specimens [33] and on the visual inspection of their RPPs, we maximized the number of groups to three. In GBTM, all model selection methods supported the model where RPPs of all studied elements belong to a single group (electronic supplementary material, figure S20; Information S5). In the 2-group set-up of DTW-hclust and ED-hclust analyses, one group contains a single element, namely the humerus of the late juvenile *Eosinopteryx*, while all other elements are clustered together in the other group. $K$-means, on the other

**Table 3.** Level of skeletal dissociation of ducks in the different grouping analyses in the 4-group set-up. Dissociation level is zero if all bones of an individual were assigned to the same group, while the series of hyphen-separated numbers indicate, on the one hand, the number of groups and, on the other hand, the number of bones assigned to each group (e.g. '1-3' means the bones of the individual were separated into two groups with one bone in one group and three bones in the other group). The maximum of four groups appears with ED-hclust in a single 15 dph duck, but skeletal dissociation into three groups occurs at 15 and 30 dph with all methods but DTW-hclust. Note how skeletal dissociation level increases up to 15 dph, and then seems to drop again at more advanced ages.

| age (dph) | specimen ID | no. sampled bones | skeletal dissociation level/specimen | | | |
| --- | --- | --- | --- | --- | --- | --- |
| | | | GBTM4 | DTW-hclust4 | ED-hclust4 | Kmeans4 |
| 4 | Anas no. 215 | 4 | 0 | 0 | 0 | 0 |
| | Anas no. 230 | 4 | 1-3 | 1-3 | 0 | 1-3 |
| | Anas no. 290 | 4 | 1-3 | 1-3 | 1-3 | 1-3 |
| 8 | Anas no. 227 | 6 | 2-4 | 3-3 | 2-4 | 3-3 |
| | Anas no. 218 | 5 | 2-3 | 2-3 | 2-3 | 2-3 |
| | Anas no. 246 | 5 | 2-3 | 2-3 | 2-3 | 2-3 |
| 15 | Anas no. 241 | 6 | 1-2-3 | 1-5 | 2-4 | 2-4 |
| | Anas no. 254 | 6 | 1-5 | 0 | 1-1-2-2 | 1-5 |
| | Anas no. 255 | 6 | 3-3 | 1-5 | 1-2-3 | 3-3 |
| 30 | Anas no. 208 | 6 | 1-2-3 | 3-3 | 1-2-3 | 1-1-4 |
| | Anas no. 221 | 5 | 2-3 | 2-3 | 2-3 | 2-3 |
| | Anas no. 298 | 5 | 2-3 | 2-3 | 2-3 | 2-3 |
| 50 | Anas no. 236 | 6 | 3-3 | 2-4 | 0 | 3-3 |
| ≥90 | Anas no. 6 | 2 | 0 | 0 | 0 | 0 |

**Table 4.** Summary of results of the multi-nomial logistic regression analyses to test potential predictor variables of group memberships resulting from GBTM, K-means, DTW-hclust and ED-hclust with four groups in ducks. For each grouping method, a stepwise model selection was run with the original model including the linear combination of the following predictors: specimen, age, type of bone (Bone) and developmental strategy of bone (Dev_strat). This table only shows the best-supported models with the lowest AIC values and/or the lowest AICc values. Note that the simpler models (with lower AICc) have a considerably lower prediction power indicated by the lower percentage of matches between actual and predicted group memberships (PM%). For further details, see electronic supplementary material, Information S4.

| method | best-supported model for four groups | AIC | AICc | PM% |
| --- | --- | --- | --- | --- |
| GBTM | GBTM4 ~ Age + Dev_strat | 95.884 | NA | 76 |
| K-means | Kmeans4 ~ Age + Dev_strat | 90.447 | 93.447 | 74 |
| | Kmeans4 ~ Age + Dev_strat + Bone | 87.517 | 106.767 | 90 |
| DTW-hclust | DTW4-hclust ~ Age + Dev_strat | 104.88 | 107.783 | 68 |
| | DTW4-hclust ~ Age + Dev_strat + Bone | 89.632 | 108.112 | 90 |
| ED-hclust | ED4-hclust ~ Age + Dev_strat | 93.942 | 96.846 | 62.5 |
| | ED4-hclust ~ Age + Dev_strat + Bone | 93.264 | 111.744 | 89 |

hand, assigns four of the six elements of *Eosinopteryx* along with two elements of *Serikornis* in a separate group from all the rest of the bones. The 3-group setting provided more diverse group memberships with different analyses; however, all methods resulted in one of the groups containing solely the *Eosinopterx* humerus (electronic supplementary material, figure S20). This recurring separation of *Eosinopterx* elements by different RPP analysis methods is in accordance with the previously assigned juvenile status of this specimen [33]. However, the rest of the 'dinobird' elements were clustered in variable ways. Only DTW-hclust sorted three more *Eosinopteryx* elements into a separate group and all other

elements in the third group, while *K*-means and ED-hclust gave less interpretable element sorting (electronic supplementary material, Information S5).

Group membership probabilities of RPPs are best explained when precocity ranks of elements (a qualitative order of relative osteohistological maturity of elements within the skeleton [33]) are also considered in combination with the assigned ontogenetic stage of the specimen (deduction based on overall osteohistological maturity [33]). This is shown by the prevalent separation of the humerus of *Eosinopteryx* from all other elements in both the 2-group and 3-group set-ups. On the other hand, only the 2-group *K*-means memberships seem to reflect the immature ontogenetic status of *Serikornis* [33], while this signal is absent in all other analyses.

### 3.2.3. Complete dataset

When tested for 3–5 groups (figure 6; electronic supplementary material, figure S21 and Information S6), model selection methods performed in the GBTM analyses supported 5 groups as the best model to cluster RPPs in the complete dataset. However, the four different analysis types gave much less congruent results for the composition of the 5 groups (electronic supplementary material, figure S21) than they did for the duck-only dataset with 4 groups. Here, excluding four elements with incomplete RPPs, only 24% of the elements clustered identically by all four methods, and 46% by three of the four methods (table 5). This is largely due to the GBTM group assignments in the adult bird and dinobird segment of the dataset which deviated from those of the other clustering methods in all but a single case. Nevertheless, this still gives a total of 70% of the elements being grouped with a minimum of 75% consistency across different methods. Skeletal dissociation only occurs at juvenile stages in birds and its maximum level is three (table 6). Once again, group memberships tend to reflect the ontogeny-dependent dynamics of the elements' developmental strategies (figure 6 and table 5; electronic supplementary material, Information S6). Overall, no phylogenetic coherence appears in any of the RPP-clustering patterns.

Within the duck subgroup in the complete dataset, only 40% of the bones were identically clustered by all four methods, while 26% and 31% were clustered by three and two of the four methods, respectively (table 5). Furthermore, when cluster assignments of duck elements in this pooled dataset with 5 groups were compared with that of the duck-only dataset with 4 groups, 41% of the elements were still clustered the same way by all four methods as in the duck-only analyses, despite the difference in the number of groups. Another 16% of the bones were grouped identically by three out of the four methods altogether giving a greater than or equal to 75% clustering congruence for 56% of the duck elements across these analyses. The highest level of skeletal dissociation is three which is the most frequent in the 15 dph and 30 dph cohorts.

The subgroup of juvenile hoatzin bones shows less overall consistency revealing that only two elements were clustered identically by all four and another two by three out of four methods. On the other hand, 100% and 89% identical clustering of the hoatzin bones occurred between GBTM and ED-hclust, and *K*-means and DTW-hclust, respectively (table 5). Furthermore, each analysis scatters the hoatzin bones across three clusters. In the two younger specimens, i.e. the approximately 6 dph and approximately 9 dph hoatzin chicks, only the tarsometatarsus was separated from the rest of the skeleton by all four methods and in each case was put in the cluster representing the least mature stages (table 5). The level of skeletal dissociation within the hoatzin subgroup is fairly consistent across different methods and is represented by two clusters in each specimen with a single case of level 3 skeletal dissociation in ED-hclust (table 6). By and large, hoatzin bones cover the same range of clusters in the respective analysis as do duckling bones of 8 to 30 dph with the following recurring pattern across the four grouping methods: (i) most bones of approximately 6 dph and approximately 9 dph hoatzin chicks are grouped together with most bones of 15 dph and the wing bones of 30 dph ducklings; (ii) the tarsometatarsals of approximately 6 dph and approximately 9 dph hoatzin chicks are assigned to the same group as most 4 and 8 dph duckling bones; and (iii) most bones of the approximately 16 dph hoatzin chick are clustered with the leg bones of 15 and 30 dph ducklings and the wing bones of 50 dph ducklings. These trends in bone distribution seem to reflect not only the estimated ages/ontogenetic stages of these hoatzin juveniles, but also the semiprecocial nature of their developing limb bones in the context of the duckling bone composition in the respective clusters (figure 6 and table 5; electronic supplementary material, Information S6).

With one exception, bones of extant adult birds were grouped together with those of extinct paravian dinosaurs by all four methods (table 5). GBTM picked out a single paravian element—the carpometacarpus of *Aurornis*—for its fifth cluster, while filling up the fourth cluster with all the other

**Table 5.** Group memberships of elements in the complete dataset, excluding bones with incomplete RPPs and their congruence across the four methods (GBTM, DTW-hclust and ED-hdust, K-means) analysed in the 5-group set-up. Colours, shades of grey, inter-method and overall congruence are interpreted as in table 2. Note how membership congruence as well as inter-method congruence decreased as compared to the duck-only dataset (table 2). Element and specimen IDs and abbreviations as in the electronic supplementary material, table S1.

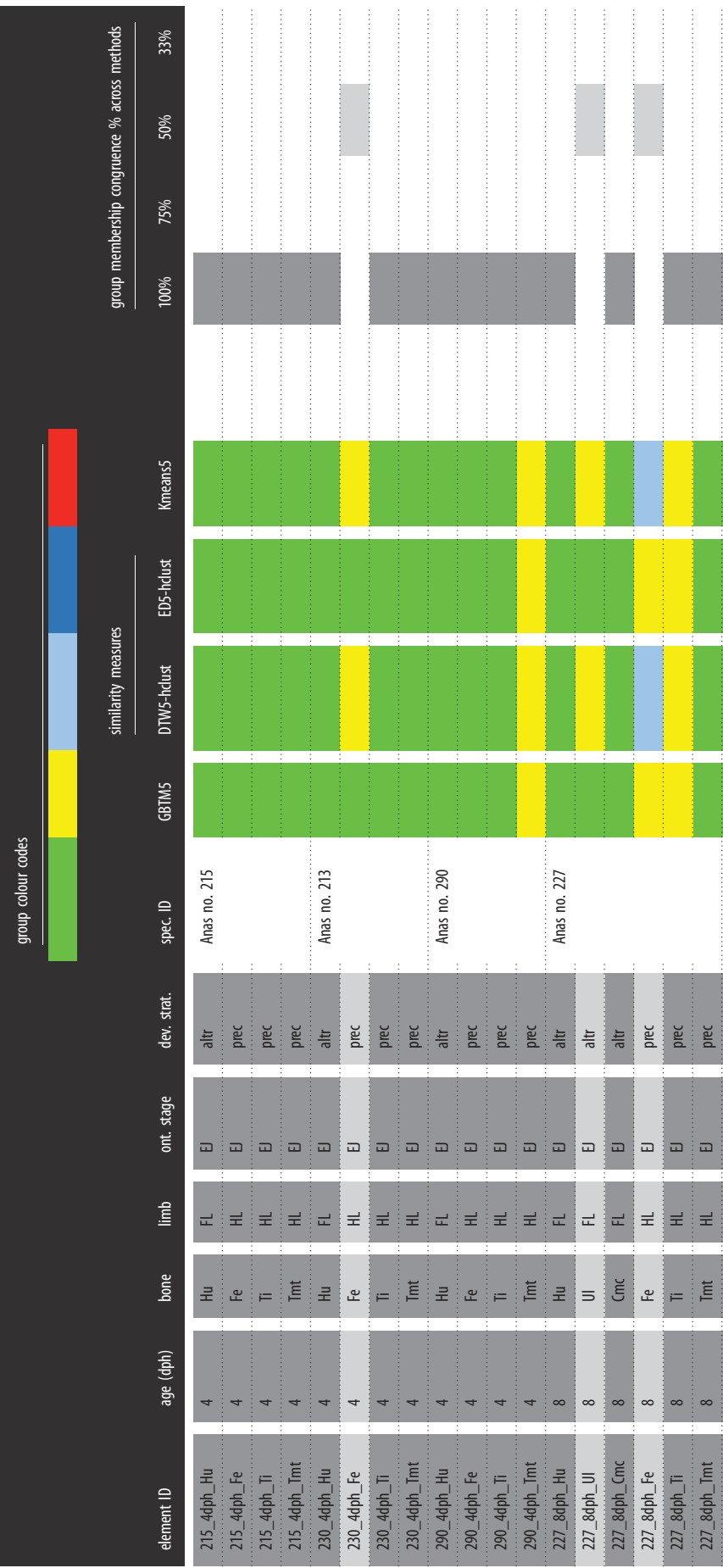

(Continued.)

**Table 5.** (*Continued.*)

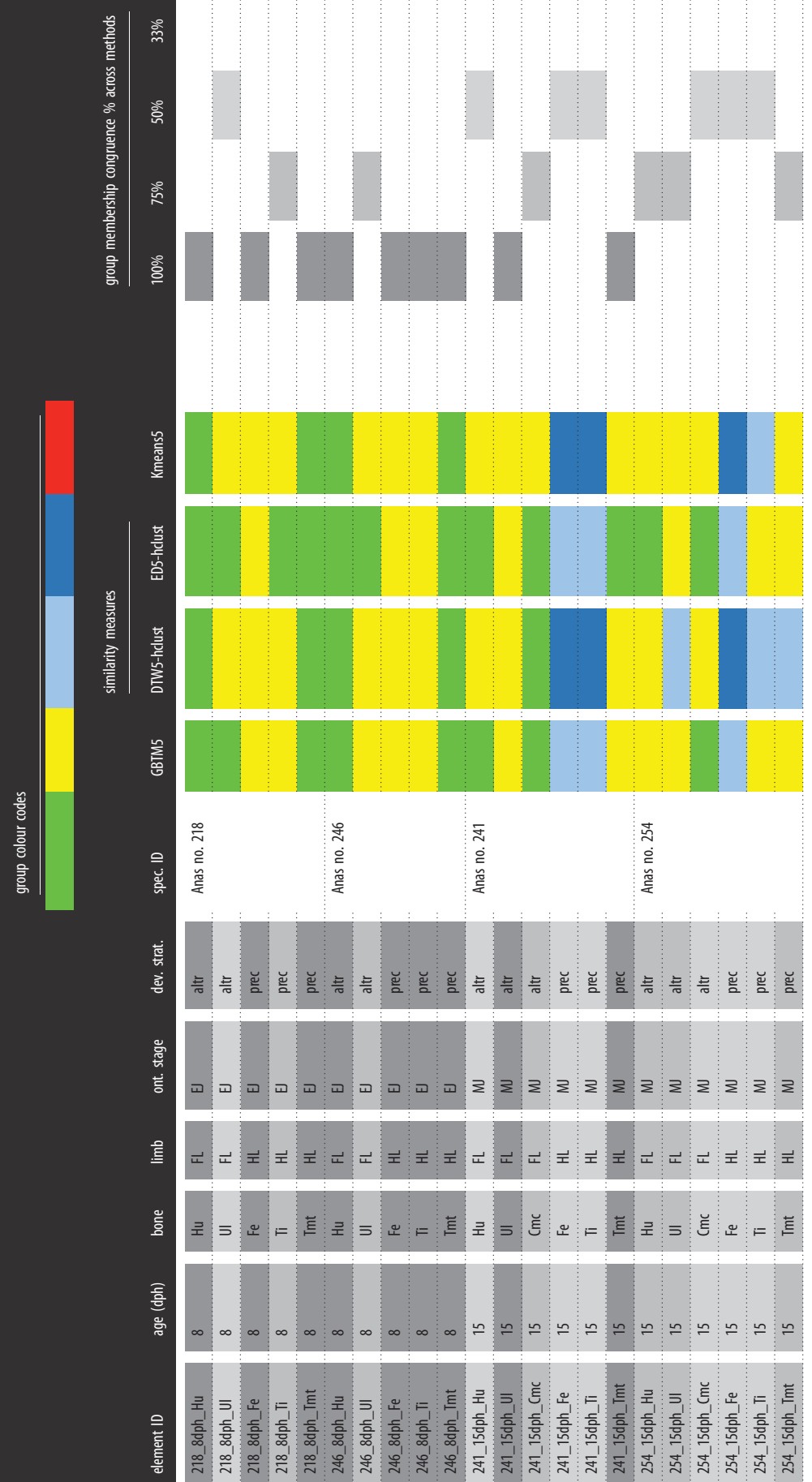

| element ID | age (dph) | bone | limb | ont. stage | dev. strat. | spec. ID | GBTM5 | DTW5-hclust | ED5-hclust | Kmeans5 | 100% | 75% | 50% | 33% |
|---|---|---|---|---|---|---|---|---|---|---|---|---|---|---|
| 218_8dph_Hu | 8 | Hu | FL | EJ | altr | Anas no. 218 | | | | | ■ | | | |
| 218_8dph_Ul | 8 | Ul | FL | EJ | altr | | | | | | | | ■ | |
| 218_8dph_Fe | 8 | Fe | HL | EJ | prec | | | | | | ■ | | | |
| 218_8dph_Ti | 8 | Ti | HL | EJ | prec | | | | | | | ■ | | |
| 218_8dph_Tmt | 8 | Tmt | HL | EJ | prec | | | | | | ■ | | | |
| 246_8dph_Hu | 8 | Hu | FL | EJ | altr | Anas no. 246 | | | | | | ■ | | |
| 246_8dph_Ul | 8 | Ul | FL | EJ | altr | | | | | | ■ | | | |
| 246_8dph_Fe | 8 | Fe | HL | EJ | prec | | | | | | ■ | | | |
| 246_8dph_Ti | 8 | Ti | HL | EJ | prec | | | | | | ■ | | | |
| 246_8dph_Tmt | 8 | Tmt | HL | EJ | prec | | | | | | ■ | | | |
| 241_15dph_Hu | 15 | Hu | FL | MJ | altr | Anas no. 241 | | | | | ■ | | | |
| 241_15dph_Ul | 15 | Ul | FL | MJ | altr | | | | | | | ■ | | |
| 241_15dph_Cmc | 15 | Cmc | FL | MJ | altr | | | | | | | | ■ | |
| 241_15dph_Fe | 15 | Fe | HL | MJ | prec | | | | | | ■ | | | |
| 241_15dph_Ti | 15 | Ti | HL | MJ | prec | | | | | | ■ | | | |
| 241_15dph_Tmt | 15 | Tmt | HL | MJ | prec | | | | | | ■ | | | |
| 254_15dph_Hu | 15 | Hu | FL | MJ | altr | Anas no. 254 | | | | | | ■ | | |
| 254_15dph_Ul | 15 | Ul | FL | MJ | altr | | | | | | ■ | | | |
| 254_15dph_Cmc | 15 | Cmc | FL | MJ | altr | | | | | | | | ■ | |
| 254_15dph_Fe | 15 | Fe | HL | MJ | prec | | | | | | | | ■ | |
| 254_15dph_Ti | 15 | Ti | HL | MJ | prec | | | | | | | ■ | | |
| 254_15dph_Tmt | 15 | Tmt | HL | MJ | prec | | | | | | | | ■ | |

group colour codes

similarity measures

group membership congruence % across methods

**29**

**Table 5.** (*Continued.*)

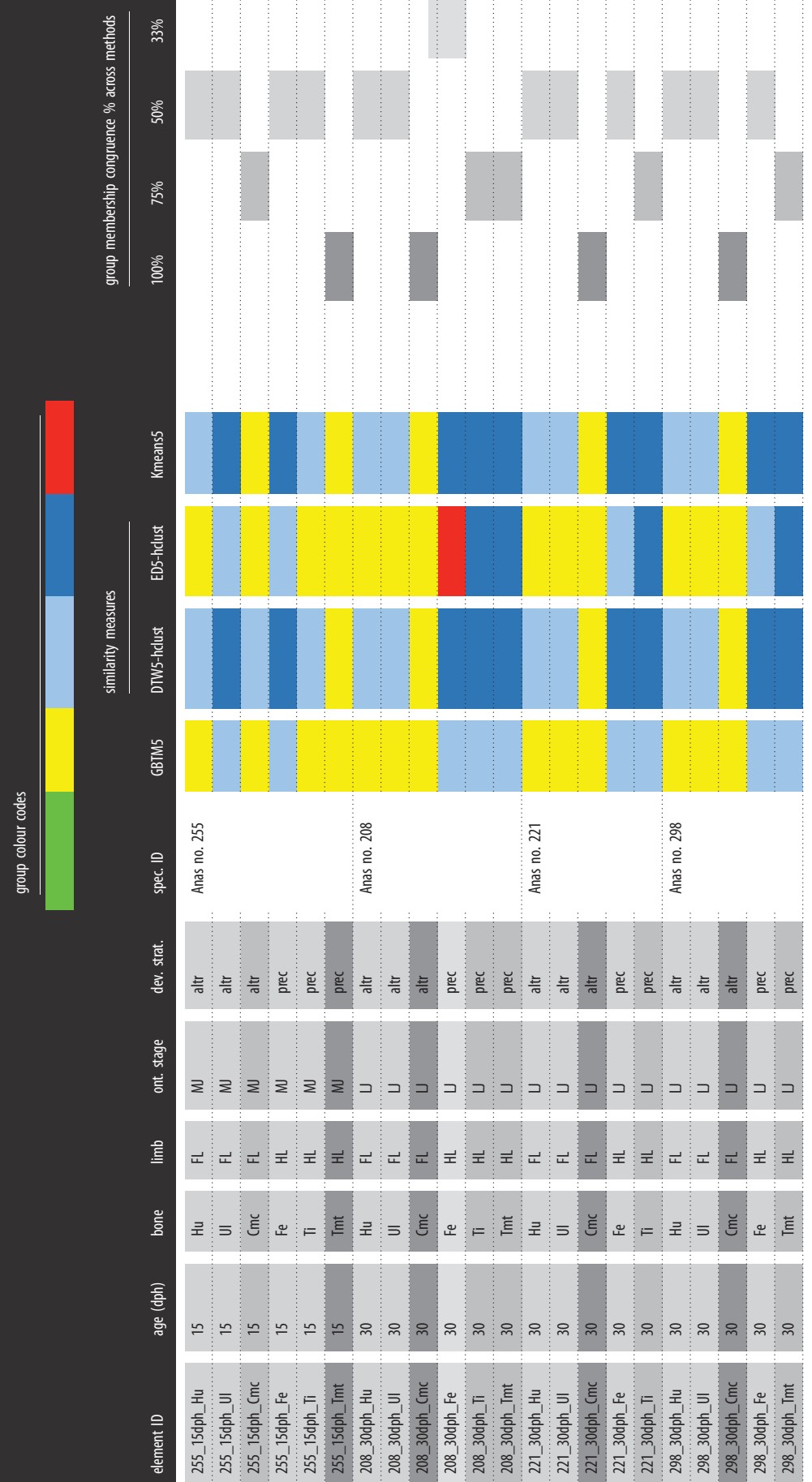

| element ID | age (dph) | bone | limb | ont. stage | dev. strat. | spec. ID | similarity measures | | | | group membership congruence % across methods | | | |
|---|---|---|---|---|---|---|---|---|---|---|---|---|---|---|
| | | | | | | | GBTM5 | DTW5-hclust | ED5-hclust | Kmeans5 | 100% | 75% | 50% | 33% |
| 255_15dph_Hu | 15 | Hu | FL | MJ | altr | Anas no. 255 | | | | | | | | |
| 255_15dph_Ul | 15 | Ul | FL | MJ | altr | | | | | | | | | |
| 255_15dph_Cmc | 15 | Cmc | FL | MJ | altr | | | | | | | | | |
| 255_15dph_Fe | 15 | Fe | HL | MJ | prec | | | | | | | | | |
| 255_15dph_Ti | 15 | Ti | HL | MJ | prec | | | | | | | | | |
| 255_15dph_Tmt | 15 | Tmt | HL | MJ | prec | | | | | | | | | |
| 208_30dph_Hu | 30 | Hu | FL | LJ | altr | Anas no. 208 | | | | | | | | |
| 208_30dph_Ul | 30 | Ul | FL | LJ | altr | | | | | | | | | |
| 208_30dph_Cmc | 30 | Cmc | FL | LJ | altr | | | | | | | | | |
| 208_30dph_Fe | 30 | Fe | HL | LJ | prec | | | | | | | | | |
| 208_30dph_Ti | 30 | Ti | HL | LJ | prec | | | | | | | | | |
| 208_30dph_Tmt | 30 | Tmt | HL | LJ | prec | | | | | | | | | |
| 221_30dph_Hu | 30 | Hu | FL | LJ | altr | Anas no. 221 | | | | | | | | |
| 221_30dph_Ul | 30 | Ul | FL | LJ | altr | | | | | | | | | |
| 221_30dph_Cmc | 30 | Cmc | FL | LJ | altr | | | | | | | | | |
| 221_30dph_Fe | 30 | Fe | HL | LJ | prec | | | | | | | | | |
| 221_30dph_Ti | 30 | Ti | HL | LJ | prec | | | | | | | | | |
| 298_30dph_Hu | 30 | Hu | FL | LJ | altr | Anas no. 298 | | | | | | | | |
| 298_30dph_Ul | 30 | Ul | FL | LJ | altr | | | | | | | | | |
| 298_30dph_Cmc | 30 | Cmc | FL | LJ | altr | | | | | | | | | |
| 298_30dph_Fe | 30 | Fe | HL | LJ | prec | | | | | | | | | |
| 298_30dph_Tmt | 30 | Tmt | HL | LJ | prec | | | | | | | | | |

group colour codes

(*Continued.*)

**Table 5.** (*Continued.*)

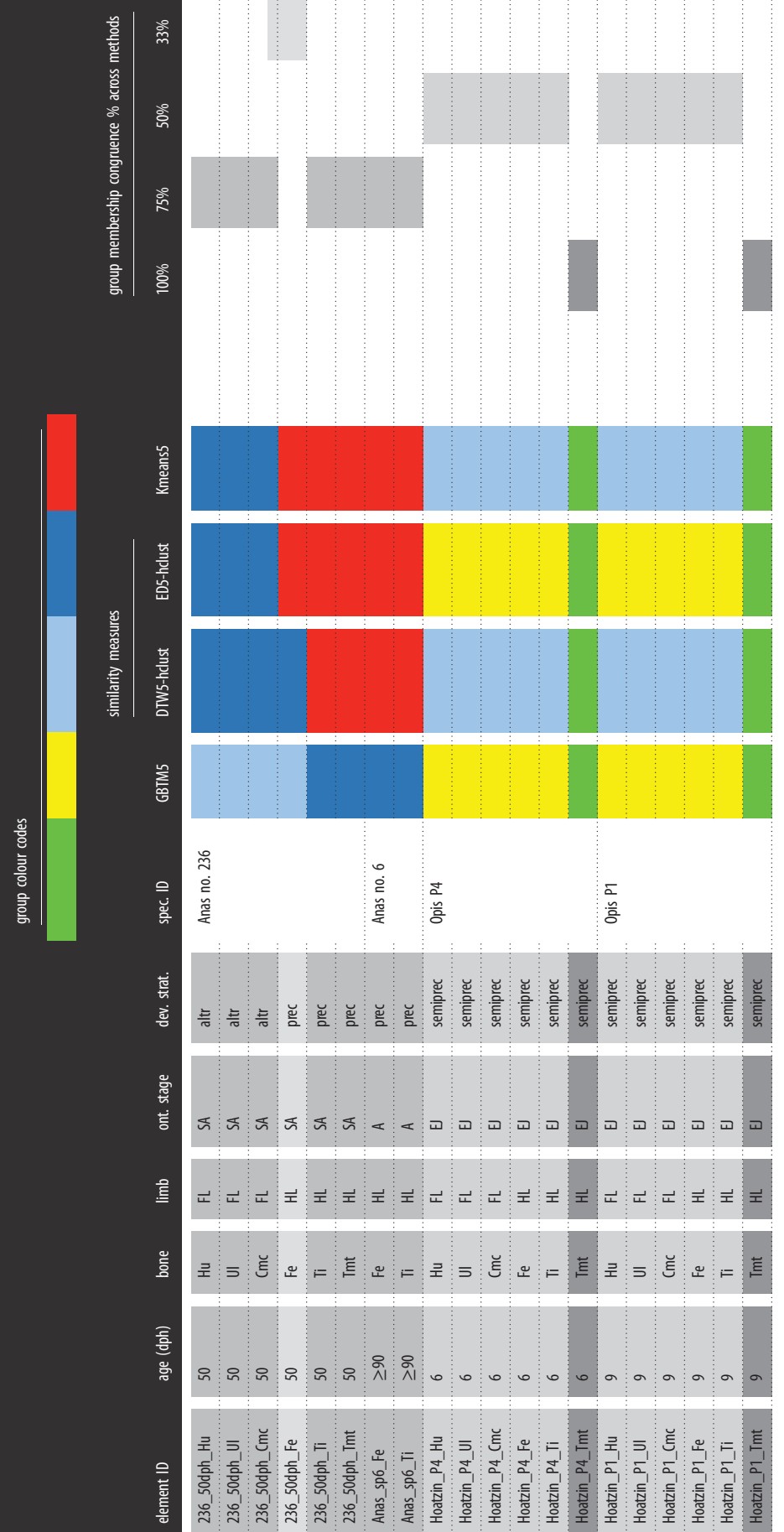

| element ID | age (dph) | bone | limb | ont. stage | dev. strat. | spec. ID | similarity measures | | | | group membership congruence % across methods | | | |
|---|---|---|---|---|---|---|---|---|---|---|---|---|---|---|
| | | | | | | | GBTM5 | DTW5-hclust | ED5-hclust | Kmeans5 | 100% | 75% | 50% | 33% |
| 236_50dph_Hu | 50 | Hu | FL | SA | altr | Anas no. 236 | | | | | | | | |
| 236_50dph_Ul | 50 | Ul | FL | SA | altr | | | | | | | | | |
| 236_50dph_Cmc | 50 | Cmc | FL | SA | altr | | | | | | | | | |
| 236_50dph_Fe | 50 | Fe | HL | SA | prec | | | | | | | | | |
| 236_50dph_Ti | 50 | Ti | HL | SA | prec | | | | | | | | | |
| 236_50dph_Tmt | 50 | Tmt | HL | SA | prec | | | | | | | | | |
| Anas_sp6_Fe | ≥90 | Fe | HL | A | prec | Anas no. 6 | | | | | | | | |
| Anas_sp6_Ti | ≥90 | Ti | HL | A | prec | | | | | | | | | |
| Hoatzin_P4_Hu | 6 | Hu | FL | EJ | semiprec | Opis P4 | | | | | | | | |
| Hoatzin_P4_Ul | 6 | Ul | FL | EJ | semiprec | | | | | | | | | |
| Hoatzin_P4_Cmc | 6 | Cmc | FL | EJ | semiprec | | | | | | | | | |
| Hoatzin_P4_Fe | 6 | Fe | HL | EJ | semiprec | | | | | | | | | |
| Hoatzin_P4_Ti | 6 | Ti | HL | EJ | semiprec | | | | | | | | | |
| Hoatzin_P4_Tmt | 6 | Tmt | HL | EJ | semiprec | | | | | | | | | |
| Hoatzin_P1_Hu | 9 | Hu | FL | EJ | semiprec | Opis P1 | | | | | | | | |
| Hoatzin_P1_Ul | 9 | Ul | FL | EJ | semiprec | | | | | | | | | |
| Hoatzin_P1_Cmc | 9 | Cmc | FL | EJ | semiprec | | | | | | | | | |
| Hoatzin_P1_Fe | 9 | Fe | HL | EJ | semiprec | | | | | | | | | |
| Hoatzin_P1_Ti | 9 | Ti | HL | EJ | semiprec | | | | | | | | | |
| Hoatzin_P1_Tmt | 9 | Tmt | HL | EJ | semiprec | | | | | | | | | |

(*Continued.*)

**Table 5.** (*Continued.*)

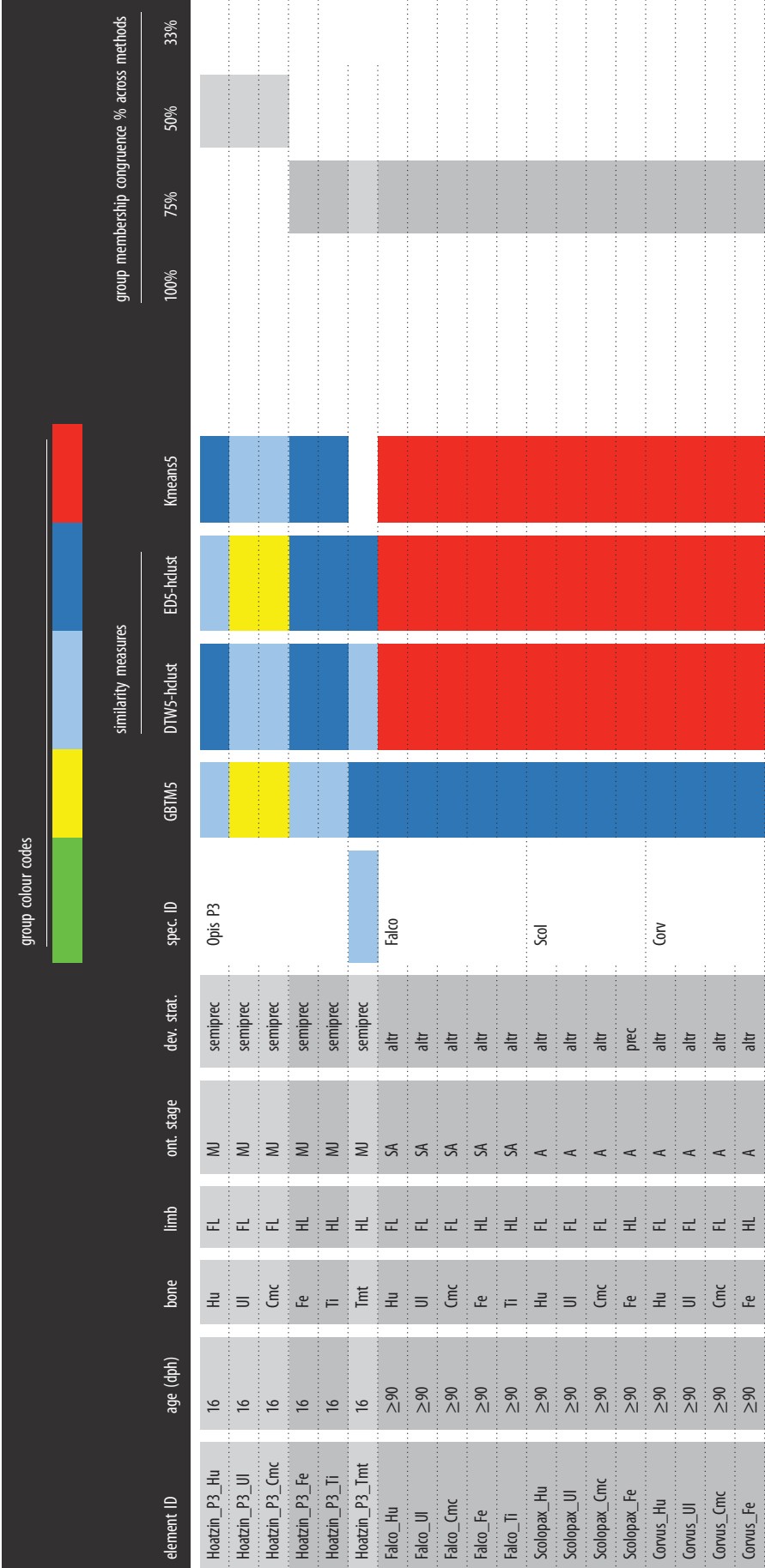

| element ID | age (dph) | bone | limb | ont. stage | dev. strat. | spec. ID | similarity measures | | | | group membership congruence % across methods | | | |
|---|---|---|---|---|---|---|---|---|---|---|---|---|---|---|
| | | | | | | | GBTM5 | DTW5-hclust | ED5-hclust | Kmeans5 | 100% | 75% | 50% | 33% |
| Hoatzin_P3_Hu | 16 | Hu | FL | MJ | semiprec | Opis P3 | | | | | | | | |
| Hoatzin_P3_Ul | 16 | Ul | FL | MJ | semiprec | | | | | | | | | |
| Hoatzin_P3_Cmc | 16 | Cmc | FL | MJ | semiprec | | | | | | | | | |
| Hoatzin_P3_Fe | 16 | Fe | HL | MJ | semiprec | | | | | | | | | |
| Hoatzin_P3_Ti | 16 | Ti | HL | MJ | semiprec | | | | | | | | | |
| Hoatzin_P3_Tmt | 16 | Tmt | HL | MJ | semiprec | | | | | | | | | |
| Falco_Hu | ≥90 | Hu | FL | SA | altr | Falco | | | | | | | | |
| Falco_Ul | ≥90 | Ul | FL | SA | altr | | | | | | | | | |
| Falco_Cmc | ≥90 | Cmc | FL | SA | altr | | | | | | | | | |
| Falco_Fe | ≥90 | Fe | HL | SA | altr | | | | | | | | | |
| Falco_Ti | ≥90 | Ti | HL | SA | altr | | | | | | | | | |
| Scolopax_Hu | ≥90 | Hu | FL | A | altr | Scol | | | | | | | | |
| Scolopax_Ul | ≥90 | Ul | FL | A | altr | | | | | | | | | |
| Scolopax_Cmc | ≥90 | Cmc | FL | A | altr | | | | | | | | | |
| Scolopax_Fe | ≥90 | Fe | HL | A | prec | | | | | | | | | |
| Corvus_Hu | ≥90 | Hu | FL | A | altr | Corv | | | | | | | | |
| Corvus_Ul | ≥90 | Ul | FL | A | altr | | | | | | | | | |
| Corvus_Cmc | ≥90 | Cmc | FL | A | altr | | | | | | | | | |
| Corvus_Fe | ≥90 | Fe | HL | A | altr | | | | | | | | | |

group colour codes

(*Continued.*)

**Table 5.** (*Continued.*)

group colour codes

similarity measures

group membership congruence % across methods

| element ID | age (dph) | bone | limb | ont. stage | dev. strat. | spec. ID | GBTM5 | DTW5-hclust | ED5-hclust | Kmeans5 | 100% | 75% | 50% | 33% |
|---|---|---|---|---|---|---|---|---|---|---|---|---|---|---|
| Anch_Hu | NA | Hu | FL | A | NA | Anch | | | | | | | | |
| Anch_Ra | NA | Ra | FL | A | NA | | | | | | | | | |
| Anch_Ul | NA | Ul | FL | A | NA | | | | | | | | | |
| Anch_Cmc | NA | Cmc | FL | A | NA | | | | | | | | | |
| Anch_Fe | NA | Fe | HL | A | NA | | | | | | | | | |
| Aur_Hu | NA | Hu | FL | A | NA | Aur | | | | | | | | |
| Aur_Ra | NA | Ra | FL | A | NA | | | | | | | | | |
| Aur_Ul | NA | Ul | FL | A | NA | | | | | | | | | |
| Aur_Cmc | NA | Cmc | FL | A | NA | | | | | | | | | |
| Aur_Al | NA | Al | FL | A | NA | | | | | | | | | |
| Aur_Fe | NA | Fe | HL | A | NA | | | | | | | | | |
| Eos_Hu | NA | Hu | FL | U | NA | Eos | | | | | | | | |
| Eos_Ra | NA | Ra | FL | U | NA | | | | | | | | | |
| Eos_Ul | NA | Ul | FL | U | NA | | | | | | | | | |
| Eos_Cmc | NA | Cmc | FL | U | NA | | | | | | | | | |
| Eos_Al | NA | Al | FL | U | NA | | | | | | | | | |
| Eos_Fe | NA | Fe | HL | U | NA | | | | | | | | | |
| Jeh_Hu | NA | Hu | FL | A | NA | Jel | | | | | | | | |
| Jeh_Ra | NA | Ra | FL | A | NA | | | | | | | | | |
| Jeh_Ul | NA | Ul | FL | A | NA | | | | | | | | | |
| Jeh_Fe | NA | Fe | HL | A | NA | | | | | | | | | |

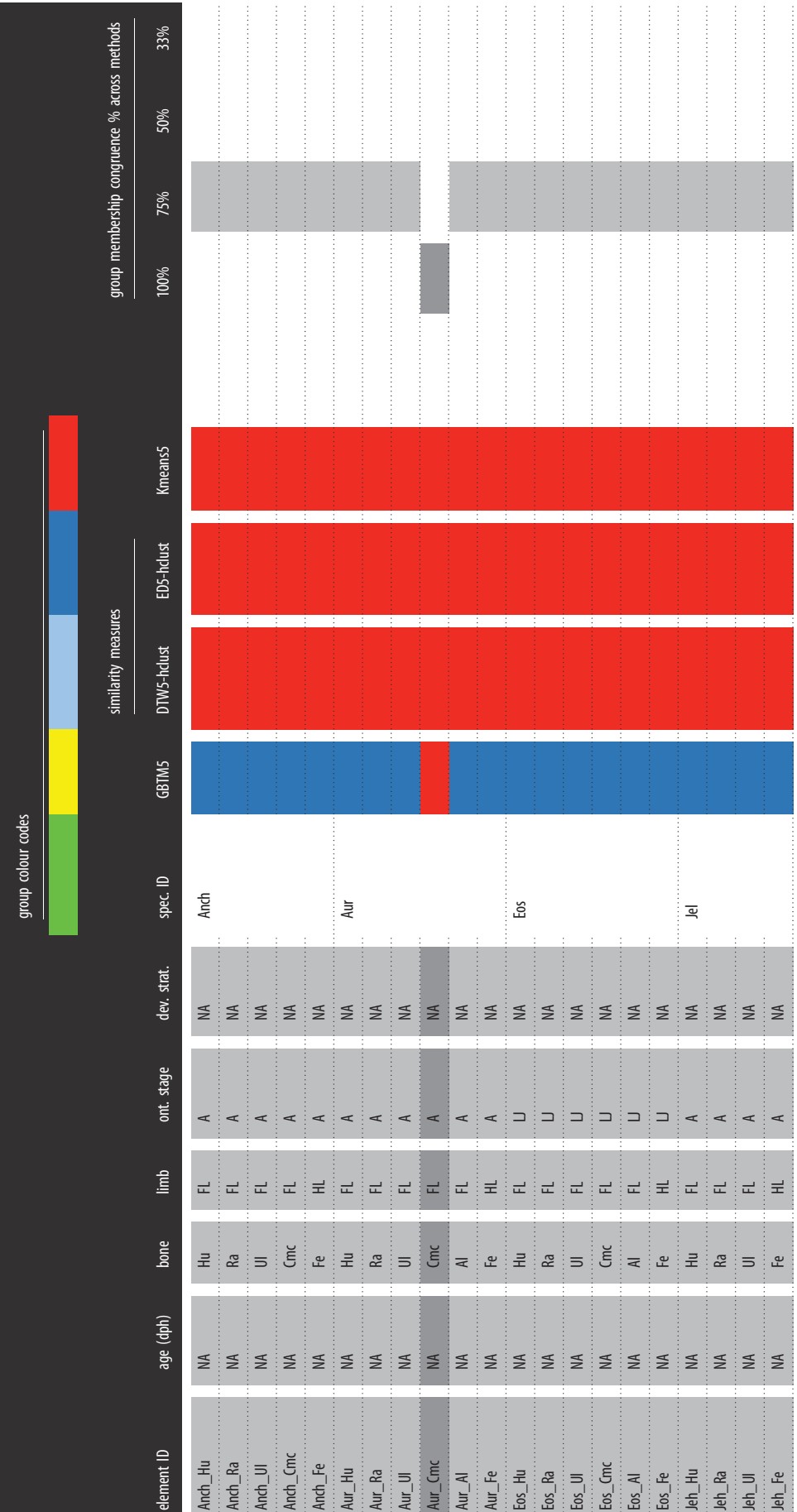

| element ID | age (dph) | bone | limb | ont. stage | dev. strat. | spec. ID | GBTM5 | DTW5-hclust | EDS-hclust | Kmeans5 |
|---|---|---|---|---|---|---|---|---|---|---|
| Ser_Hu | NA | Hu | FL | SA | NA | Ser | | | | |
| Ser_Ra | NA | Ra | FL | SA | NA | | | | | |
| Ser_Ul | NA | Ul | FL | SA | NA | | | | | |
| Ser_Cmc | NA | Cmc | FL | SA | NA | | | | | |
| Ser_Al | NA | Al | FL | SA | NA | | | | | |
| Ser_Fe | NA | Fe | HL | SA | NA | | | | | |

group colour codes

similarity measures

**group membership congruence % across methods**

| | 100% | 75% | 50% | 33% |
|---|---|---|---|---|
| no. bones | no. 31 | no. 59 | no. 36 | no. 2 |
| % bones | 24% | 46% | 28% | 1.5% |

inter-method congruence (%)

| | GBTM5 | DTW5 | EDS | Kmeans5 |
|---|---|---|---|---|
| GBTM5 | 100% | | | |
| DTW5 | 27% | 100% | | |
| EDS | 55% | 66% | 100% | |
| Kmeans5 | 29% | 96% | 68% | 100% |

overall congruency

| | |
|---|---|
| GBTM5 | 27 + 55 + 29 = 111 |
| DTW5 | 27 + 66 + 96 = 189 |
| EDS | 55 + 66 + 68 = 189 |
| Kmeans5 | 29 + 96 + 68 = 193 |

**Table 6.** Level of skeletal dissociation of specimens in the different grouping analyses of the complete dataset in the 5-group set-up. Interpretation of dissociation level as in table 3. The maximum skeletal dissociation into three groups is frequent at 15 and 30 dph in ducks with all four methods. Note that skeletal dissociation only characterizes juvenile stages in birds. Specimen IDs as in the electronic supplementary material, table S1.

| age (dph) | specimen ID | no. sampled bones | skeletal dissociation level/specimen | | | |
| | | | GBTM5 | DTW-hclust5 | ED-hclust5 | Kmeans5 |
|---|---|---|---|---|---|---|
| 4 | Anas no. 215 | 4 | 0 | 0 | 0 | 0 |
| | Anas no. 230 | 4 | 0 | 1-3 | 0 | 1-3 |
| | Anas no. 290 | 4 | 1-3 | 1-3 | 1-3 | 1-3 |
| 8 | Anas no. 227 | 6 | 2-4 | 1-2-3 | 2-4 | 1-2-3 |
| | Anas no. 218 | 5 | 2-3 | 2-3 | 1-4 | 2-3 |
| | Anas no. 246 | 5 | 2-3 | 2-3 | 2-3 | 2-3 |
| 15 | Anas no. 241 | 6 | 2-2-2 | 1-2-3 | 2-2-2 | 2-4 |
| | Anas no. 254 | 6 | 1-1-4 | 1-2-3 | 1-2-3 | 1-1-4 |
| | Anas no. 255 | 6 | 2-4 | 1-2-3 | 2-4 | 2-2-2 |
| 30 | Anas no. 208 | 6 | 3-3 | 1-2-3 | 1-2-3 | 1-2-3 |
| | Anas no. 221 | 5 | 2-3 | 1-2-2 | 1-1-3 | 1-2-2 |
| | Anas no. 298 | 5 | 2-3 | 1-2-2 | 1-1-3 | 1-2-2 |
| 50 | Anas no. 236 | 6 | 2-4 | 2-4 | 3-3 | 3-3 |
| ≥90 | Anas no. 6 | 2 | 0 | 0 | 0 | 0 |
| 6 | hoatzin_P4 | 6 | 1-5 | 1-5 | 1-5 | 1-5 |
| 9 | hoatzin_P1 | 6 | 1-5 | 1-5 | 1-5 | 1-5 |
| 16 | hoatzin_P3 | 6 | 2-4 | 2-4 | 2-2-2 | 2-4 |
| ≥90 | Falco | 5 | 0 | 0 | 0 | 0 |
| ≥90 | Scolopax | 5 | 0 | 0 | 0 | 0 |
| ≥90 | Corvus | 4 | 0 | 0 | 0 | 0 |
| NA | Anch | 6 | 1-5 | 0 | 0 | 0 |
| NA | Aur | 6 | 0 | 0 | 0 | 0 |
| NA | Eos | 6 | 0 | 0 | 0 | 0 |
| NA | Jeh | 4 | 0 | 0 | 0 | 0 |
| NA | Ser | 6 | 0 | 0 | 0 | 0 |

bones of extinct paravians, adult birds and two hind-limb elements of the 50 dph duck (table 5). Thus, the recurring separation of the *Eosinopteryx* humerus in the 2-group 'dinobird'-only analyses disappears from the 'dinobird' subgroup of the complete dataset in all analyses.

### 3.2.4. Performance of and congruency among different methods

Besides the comparisons of clustering congruency of elements presented above, performance and consistency of each method across the analyses can also be compared. While GBTM and *K*-means can only work with complete RPPs that have all four trajectory-defining points, DTW-hclust and ED-hclust tolerate incomplete RPPs with up to two missing values. However, their sensitivity to missing values was not consistent between the analyses of the duck-only and the complete dataset. Whereas only 66% of the elements were sorted the same way by DTW-hclust in the analyses of the duck-only dataset with and without incomplete RPPs, ED-hclust performed better with 93% (electronic supplementary material, table S3). On the other hand, the same percentage values for the complete dataset were 100% and 75% in DTW-hclust and ED-hclust, respectively (electronic supplementary

material, table S4), meaning that here, ED-hclust was more sensitive to missing values in RPPs than DTW-hclust.

Congruency in element sorting was examined between method pairs as well. When we removed all elements with incomplete RPPs, the duck and the complete dataset included 70 and 128 elements, respectively. In this reduced duck dataset, the highest congruency in membership assignments was found between GBTM and K-means resulting in 94% of the elements clustered the same way. The lowest congruence was surprisingly detected between DTW-hclust and ED-hclust with only 61% congruent memberships. Among the four methods, GBTM showed the highest, while ED-hclust the lowest overall congruence calculated as the sum of their paired congruence percentages with each of the other methods (table 2).

However, in the complete dataset encompassing 128 elements spanning the same ontogenetic range but broadening the included taxa and developmental strategies, the pattern is quite different. Here, congruency levels of GBTM with respect to all other methods dropped considerably and its overall level is also the lowest. By contrast, the DTW-hclust–K-means method pair shows the highest congruence with 96% of identical membership assignments of the bones, and K-means has the highest overall congruence with all other methods (table 5).

Finally, when DTW-hclust and ED-hclust were run on datasets including incomplete RPPs, the two methods showed 89% and 82% congruence in element sorting in the duck-only and complete dataset, respectively (electronic supplementary material, tables S3 and S4).

# 4. Discussion

The RPP method was built on the working hypothesis that RPPs in limb bone shafts reflect the combination of growth and functional maturity. Using mostly extant birds to create a proof-of-concept dataset, we applied different analytical methods to identify potential RPP categories and then cross-tested these RPP groups to see if and how they are related to the ontogeny–developmental strategy predictors of our working hypothesis. As laid out in the sections below, the implications of the results of this study are extremely diverse and pave the way for future research in many different directions.

## 4.1. Proof of concept: predictions and findings

Despite the limited sample size and the taxonomically and ontogenetically unbalanced composition of our osteohistological dataset of extant and extinct paravians, our case study demonstrates how this simple quantitative parameter, RPP, offers diverse analytical possibilities and can deliver information about complex growth- and locomotor ontogeny-related aspects of bone development. Ducks clearly represented the most extensive source of information in every aspect, including (i) element-specific ontogenetic patterns revealed by the almost complete growth series with known ages (but lacking adult wing elements and senescent specimens); (ii) a glimpse of intraspecific variability provided by cohorts containing three specimens at each stage up to 30 dph; and (iii) the osteohistological characteristics of modular fore- and hind-limb development in the context of locomotor ontogenetic strategy.

The results of the analyses of the duck subset showed where RPPs meet our predictions, as defined by bone growth and functional maturation principles (see electronic supplementary material, Information S1), and where they seem to deviate from them. For instance, we assumed that precocially developing leg bones grow more slowly and are functionally more mature with more advanced osteonal development—and hence lower overall porosity—than altricially developing wing bones. We therefore expected to find hind-limb element RPPs mostly running below fore-limb element RPPs within an individual from hatching on until about fledging. This seemed to be the dominant pattern supported not only by visual RPP inspection (electronic supplementary material, figure S11) but also by the most frequent analytical separation of wing and leg RPPs of individuals (i.e. the dissociation of skeletons) and the significant prediction power of developmental disparity in sorting the elements (tables 2–4). Most importantly, 4-group models successfully show how altricial and precocial bone development within an individual can assign an individual's elements to groups containing bones of younger and older individuals, respectively (table 2). These results suggest that RPP groups indeed reflect different osteohistological maturity levels related to ontogenetic functional development. However, the presence and strength of this functional signal are not uniform through ontogeny and across elements.

In particular, the erratic course of RPPs of all studied bones at 4 dph did not show any such trend (electronic supplementary material, figures S4, S9 and S10), although it has to be noted that of the wing bones only humeri could be profiled because the ulnae and carpometacarpals—having a delayed, altricial growth—had extremely underdeveloped posthatching cortices at this early stage. Even though the expected trend started to appear at 8 dph (electronic supplementary material, figures S5, S9 and S10), the signal was not as evident as predicted based on the fact that ducklings can run and swim with their hind-limbs right after hatching, while they do not use their minute wings for any sensible function at this age. A further peculiarity was that up to 15 dph, the RPPs of tarsometatarsi were in the range of—and grouped together by quantitative analyses with—the RPPs of wing elements in their respective cohorts revealing an unexpectedly high porosity for a precocial hind-limb element. On the other hand, ulnar RPPs sometimes clustered with elements of higher functional maturity (e.g. with hind-limb bones of the same cohort or with elements of more advanced ontogenetic stages) (table 2; electronic supplementary material, figures S9 and S10). These deviations from the expected pattern may have many different reasons, including small sample size, inadequate number of quadrants, more subtle differences in the functional development of individual bones and/or other underlying factors influencing RPPs that are unknown or unaccounted for in our study. For instance, the high level of skeletal dissociation up into three groups per individual that occurred in some specimens at 15 dph and 30 dph may suggest a period of accelerated developmental partitioning of individual bones in mid- and late juvenile phases of growth. Along with raising new questions, ducks provided a valuable basis for highlighting the application potential of RPPs and for drawing our preliminary, but promising, inferences on how RPPs can also reflect developmental dynamics of limb bone growth in other taxa.

For instance, in the light of disparate RPP patterns in the altricially developing fore-limbs and precocially developing hind-limbs of the ducklings, the RPPs in juvenile hoatzin limbs are intriguing. Even though sample size is too small to test whether it is a real trend, the unique wing-assisted locomotor activity of hoatzin chicks at this age seems to correspond with the findings of their wing bone RPPs being in the same range as and following similar trajectories to their leg bone RPPs (table 5; electronic supplementary material, figures S13 and S14). On the other hand, the strangely high porosity levels of tarsometatarsi in both ducklings and hoatzin chicks remain unexplained in this functional context. However, the extensive remodelling present in many of the more mature avian tarsometatarsi in our sample may well be related to this highly porous and hence possibly fragile nature early in ontogeny.

## 4.2. Unexpected new findings

Qualitative inspection of RPPs also revealed hitherto unknown patterns that are promising candidates for indicating relative functional maturity levels of limb bones.

First, a simple plotting of RPPs of homologous bones through ontogeny in ducks (electronic supplementary material, figure S12) showed an apparent difference between wing and leg bones in how porosity in the outermost cortical quadrant, which includes the actively growing periosteal surface, changes as ontogeny progresses. Whereas homologous altricial fore-limb elements converge on similar $Q_d$ porosity values up to 30 dph, there is a stepwise decrease in $Q_d$ porosity in homologous precocial hind-limb bones from 8 dph through ontogeny (electronic supplementary material, figure S12). This pattern may suggest that osteonal development, and hence functional maturity, stays higher in precocial elements even in the actively growing outer cortical region. Thus, any potential lag between growth and functional maturation is likely tighter in precocial bones than in altricial bones.

The second pattern becomes evident when the course of RPPs of multiple elements is jointly investigated within an individual and/or within a cohort where all points defining the RPPs are, as much as possible, homologous (i.e. representing the same ontogenetic window) (figure 3; electronic supplementary material, figures S9, S10 and S13). The pattern itself could be described as a progressive channelization of RPPs that, as observed in the ducks, appears to correlate with the process of functional maturation. Low functional maturity seems to be associated with RPPs that apparently run independently from one another showing erratic-looking courses and a comparatively high intrasectional (electronic supplementary material, figures S4–S8) as well as intracohort (electronic supplementary material, figure S9) variability among homologous bones. By contrast, elements that perform locomotor functional tasks reveal RPPs that become aligned and converge on a more definite course and porosity level, depending on the ontogenetic window captured. Accordingly, in ducks, this progressive channelization could be observed the earliest in the precocial leg bones, while wing bones

started to show alignment of RPPs much later in development. Furthermore, aligned RPPs of wing elements ran at higher porosity levels than did hind-limb elements up to fledging (electronic supplementary material, figures S4–S10). Our sample of juvenile hoatzins also seems to reflect this relationship. In hoatzin chicks, the RPPs of wing and leg elements are all channelized and run in the same porosity range (electronic supplementary material, figures S13 and S14), a finding congruent with the equal levels of locomotor involvement of the wing and leg in climbing and swimming at this stage of development. Supporting this qualitative observation, we find that such channelized RPPs are also identified and largely grouped together by the different analytical methods (tables 2 and 5).

However, a fine detail does not seem to line up with this hypothesis and hence warrants further discussion. If locomotor loading induces RPP channelization, we would expect channelized RPPs in the hind-limb elements of ducklings already at 4 dph. However, a strong signal of channelization in the leg bones starts to appear only at 8 dph (electronic supplementary material, figure S5). Even though this may appear to represent a counter-evidence to the hypothesis of a locomotor-induced RPP channelization, the bone formation processes involved in the fast growth of the avian limb bone cortex can resolve this apparent contradiction. Posthatching radial growth of the cortex in duckling hind-limb bones at 4 dph involves the fast formation of an extensive trabecular network of woven bone that quickly encloses large, randomly shaped, soft tissue-filled areas. Osteonal compaction of these areas can only start after the first trabecular woven bone scaffolds have formed, and hence it lags behind the rapid volumetric expansion of the growing cortex (see electronic supplementary material, Information S1). Even if locomotor loading induces rapid osteonal compaction, this backlog can result in a delayed RPP channelization, which may explain the pattern observed in the ducks. The same bone formation constraint can also account for the perplexingly similar porosity ranges and erratic courses of RPPs of wing and leg bones at 4 dph, as discussed above. Potential structural weakness of the highly porous and poorly organized bone tissues in the precocial hind-limb elements of early juveniles might be compensated at higher hierarchical levels of organization (e.g. robustness of the elements) which could be included in the RPP analyses in future studies.

Finally, even though our direct information of the RPP pattern in adult ducks is quite limited, the data available in the adults of other birds in our sample provide very important clues as to what the expected RPP pattern would be in fully grown duck limb bones. The kestrel, woodcock and rook all showed very similar RPP patterns with low overall porosity and strongly channelized RPPs converging on similar porosity values in $Q_d$ in all of their sampled bones, despite the fact that these species follow different developmental strategies in the precocial–altricial range. For example, the woodcock has precocial chicks with less disparate wing and leg development and considerably earlier fledging than seen in ducklings, while the rook has fully altricial chicks staying in the nest until fledging. Their virtually identical RPP patterns strongly suggest that these profound differences in the locomotor ontogenetic strategy of their juveniles cannot be detected in the limb bone RPPs of their adults. Thus, differences detected between the RPPs of wing and leg bones in ducklings are expected to disappear by the fully grown stage in ducks as well. This scenario is also supported by the gradual channelization and convergence of RPPs of wing and leg bones detected in the duck growth series as described above.

## 4.3. Performance of analytical methods

Even though our chosen methods for analysing RPPs showed largely convergent results with raw data, differences in their sensitivity to differing sample size, composition and cluster numbers have also become evident with the diverse analytical set-ups. Choosing an adequate number of clusters is critical in all analyses, as it should reflect prior expectations based on a well-defined underlying hypothesis related to our data structure (including sample size and composition). Based on the results we conclude that, for the duck RPP subset, GBTM and $K$-means with four clusters performed by and large the best in assigning memberships that reflect the combination of the specimen's ontogenetic stage and the developmental strategy of the elements (table 2). However, analysis of the complete dataset implies that for GBTM to give sensible results, there might be an upper limit to the number of clusters that is apparently unrelated to data structure. This constraining factor may well be the resolution of trajectories, that is, the number of measurement points (quadrants) defining an RPP, which in our case study was set to four. RPP sorting performance of DTW-hclust and ED-hclust was below that of GBTM and $K$-means for ducks but way above GBTM and close to $K$-means for the complete dataset (table 5). Furthermore, the inclusion of incomplete RPPs (i.e. where one or two measurement points were missing due to remodelled inner cortex) into the analyses is only possible with DTW-hclust and ED-hclust which sorted the elements with relatively high congruency (greater

than or equal to 66%) whether or not incomplete RPPs were included in the analyses. In sum, DTW-hclust and ED-hclust appear to be the least sensitive to the low resolution of trajectories but give noisier results than GBTM and *K*-means whenever the number of clusters and measurement points of RPPs seems optimal in relation to the data structure. However, as GBTM and *K*-means require complete RPPs with equal numbers of measurement points, elements that are prone to remodelling and have no cortical sector in which the majority of each quadrant is formed of primary bone cannot be analysed with these methods.

We also ran trials on centred RPPs with all four methods as well as on raw data with LCSS which translates trajectories along both axes to find maximum match between RPPs. Our aim here was to test whether disregarding absolute porosity values and only focusing on the shape of the trajectories can be valuable for detecting corresponding relative ontogenetic stages and bone functional maturity among different taxa. While *K*-means analyses of centred data and LCSS-hclust on raw data clearly identified and grouped together channelized RPPs that followed a similar course, for a large proportion of our dataset they gave quite erratic memberships in each analysis. For instance, some of the RPPs of the youngest ducks show similar shapes to those of the adult paravians despite having four–five times higher porosity values than the latter and hence were grouped together. These results suggest that, in interspecific comparisons, the shapes of the trajectories on their own may only be useful indicators of relative ontogenetic stages of individuals and/or functional maturity stages of elements if enough prior knowledge exists of the species' general developmental characteristics. However, it has to be noted that if the RPPs are based on as few as four data points (quadrants), finer details that could be decisive in the trajectories' shape may be entirely masked. Our dataset only contained paravian long bones the cortex of which is known to be relatively thinner than that of other terrestrial tetrapods limiting the sensible number of quadrants where porosity can be assessed. By contrast, thicker bone walls can record longer periods of individual/elemental growth if primary tissue dominates the cortex. This could provide more complete RPPs with finer resolution covering a longer phase of the individual growth trajectory and increasing the comparative power of these shape analyses approaches which our dataset could not demonstrate. Furthermore, neither the phylogenetic composition nor the taxon-specific ontogenetic range of our dataset was ideal to test the real potential of these methods in interspecific comparisons. Finally, LCSS may eventually prove to be a more useful tool for intraspecific ontogenetic and intraskeletal exploration of RPPs. As LCSS translates the trajectories along the horizontal axis as well, which in the case of RPPs actually represents the ontogenetic and/or maturation axis, it has the potential to identify similar sections in RPPs even in differently aged individuals and/or differently maturing elements. This means that LCSS might be able to find the overlapping ontogenetic and/or developmental maturation stages recorded in the cortex of an ontogenetic series of specimens (electronic supplementary material, figure S24) and/or the relative timing of maturation steps in different elements within a skeleton. For this capacity to emerge, LCSS probably needs RPPs of finer resolution with more trajectory-defining points (quadrants) than included in our analyses. This potential of LCSS in RPP analysis remains to be explored.

## 4.4. Limitations and general requirements

For all its versatility, the RPP method also has some limitations. These are related to the size, ontogenetic structure and phylogenetic composition of the dataset, and the general cortical features constraining measurement areas, number of quadrants and thus evaluable data points for generating RPPs. For instance, our 'dinobird' sample only contained specimens that were clearly beyond their fastest growth period and mostly closer to or within their adult range. However, the slight differences in osteohistological maturity among the specimens [33] were not evidently detected by the numeric analyses of RPPs. Furthermore, cortical and histological preservation of the sampled bones were mostly incomplete which made unbiased selection of sectors and quadrant areas almost impossible. Finally, some cortical areas with well-preserved histology which were too small for profiling frequently showed high regional variation in tissue composition suggesting that our RPPs do not capture this diversity and hence can provide only limited information in these dinobirds. Hence, qualitative inspection of the sections along with visual and quantitative evaluations of their RPPs is equally important steps for the best possible interpretation of the RPP patterns.

RPPs need to be interpreted in the context of the overall histological features of the cortex. Therefore, using this method correctly requires explicit knowledge of osteohistology: ability to correctly identify bone tissues in order to select proper measurement areas in the cortex (i.e. avoiding

secondarily remodelled areas as much as possible), to understand potential biases in RPPs (such as resorption-induced increase in porosity; potentially eroded surfaces in fossil bones; sections taken from other than mid-diaphyseal regions) and to consider them accordingly. The methods to further analyse RPPs also have to be carefully selected and the interpretation of their results requires a good general understanding of bone growth and development. Knowledge of species-specific features that may be related to or even influence bone growth and development can be a crucial addition in the context of RPP evaluation, as shown here in the hoatzin chicks.

## 4.5. Individual adjustments and possible future extensions

RPPs can be considered as a standardized osteohistological method to infer growth and functional development-related attributes in living and extinct tetrapods. At the same time, the possibility of case-specific individual adjustments in almost every step of RPP measurements and analyses is a great advantage for any future osteohistological studies wishing to apply RPP as a core approach. These adjustments, if properly performed, do not introduce subjectivity into the analyses but rather allow the taxonomic and phylogenetic expansion of the application field and refine the search for biologically relevant patterns.

From the above discussion, it is clear that the number of RPP-defining measurement points act as a limiting factor in both the qualitative and quantitative evaluation of RPPs. Hence, increasing the resolution of trajectories (the number of quadrants in each sector) will likely result in an increase of detection power and sorting performance of all RPP-analysing methods. However, besides the lowest number of measurement points for any quantitative analysis to make sense, there is a sensible upper limit to the number of quadrants into which a sector can or should be divided. This upper limit largely depends on the absolute and relative cortex thickness as well as the overall porosity features and vascular architecture of the cortex. In this study, we set the number of quadrants to four because we had to adjust our measurement areas to the minute but extremely porous posthatching cortex present at the earliest evaluable ontogenetic stages in our sample combined with the relatively low cortex thickness characteristic of paravian limb bones. A sample of a different taxonomic composition could allow or even require very different number of quadrants.

For instance, multi-element RPP study of a single, or multiple species of slow-growing, medium-sized tetrapods with only sparse bone vascularization even in earlier ontogeny may not even need as many as four quadrants in a sector to provide qualitative insight into allometric growth patterns, ontogenetic structure of a monospecific assemblage, or interspecific differences in growth and development. On the other hand, a large-bodied, faster growing tetrapod possessing thick limb bone cortices with extensive primary growth record allows and justifies a larger number of quadrants to be taken for defining RPPs. The number of quadrants (as well as sectors) can be set to capture as much of the radial histodiversity as deemed to be informative in the context of the ontogenetic range of the sample, cortex thickness and completeness of the primary growth record. A dynamically changing pattern in bone histology may require a higher number of quadrants to reflect these changes in the most adequate resolution, while there may not be much merit in a high number of quadrants in a histologically more uniform and slowly changing cortex. In addition, unduly increasing the number of quadrants in relation to the number of sampled elements can lead to 'overparametrization' in models, such as K-means, used for RPP analyses. *Sensu* Formann [100], the minimum sample size for such multi-variate analysis should be greater than or equal to $2^d$, where $d$ is the number of parameters. In our case, this means that RPPs of four or five quadrants (parameters) would require the inclusion of at least 32 or 64 elements, respectively, in an RPP analysis. Secondary remodelling can pose further constraints on picking the ideal number of quadrants. Since remodelled areas are not quantifiable as points of RPPs, the extent of remodelling should first be mapped throughout the sample to make sure there will still be enough RPP datapoints in partially remodelled elements, if they are to be included in the quantitative analysis.

Apart from remodelling issues, the number of quadrants should preferably be the same in all sampled bones to give matching points of RPPs at any given ontogenetic window and an equal resolution of all RPPs within the sample to analyse. This means that the number of quadrants should be adjusted to the maximum that the thinnest analysed cortex in the sample allows. A thin cortex can still be counteracted by selecting a larger sector area to include enough cortex in each quadrant for a sound analysis. Even though DTW-hlcust and ED-hclust can handle differing numbers of trajectory-defining points, their sensitivity to these differences is not conclusive (see the comparison of memberships with and without incomplete RPPs in the duck and complete dataset). Since only a few

specimens had less than four RPP-defining measurement points in our paravian sample, we cannot properly assess how the sorting performance of these methods would decrease with increasing variance in the number of RPP-defining points within a sample. Nevertheless, this is another point where cautiousness is required in future studies using RPP-analysing methods.

The number and size of sectors, i.e. selected measurement areas in the cortex, also can and should be adjusted in a way as to account for the intrasectional histodiversity. If a section shows regionally different bone histological features, the number and placement of the sectors will largely determine how much of this histodiversity will be captured in RPPs. Therefore, it is recommended to pre-select those sections in the sample which show the most variable intrasectional histology and define the number of sectors to take based on these sections. This way, the degree of histodiversity can also be analysed and compared among different elements within a skeleton or among homologous elements in a growth series.

Sector selection can also be driven by other purposes, such as to test whether there is any regional histodiversity in RPPs adhering to certain anatomical directions (such as lateral, medial, cranial and caudal section portions) or loading regimes (based on biomechanical measurements). In these cases, the research question itself will determine the number and relative placement of the sectors within any given section, although the size of each sector remains freely adjustable to other constraints posed by cortex thickness or histological composition.

Numerous further tests and refinements are possible in the analyses themselves by adjusting the arguments used in the models, such as the maximum distance between corresponding RPP points to be considered equivalent in ED, the degree of polynomials used to model trajectories in GBTM (as defined by [90,92] in R packages and functions) or the number of clusters to be tested in any of these methods. How these adjustments can improve RPP clustering performance largely depends on the sample size and the taxonomic and ontogenetic composition of the dataset: the more distinct ontogenetic stages and/or developmental strategies are represented in the sample, the better separation of RPP groups can be expected. In addition, resolution can be increased by focusing on a certain subset of a larger dataset, for instance a cohort composed of multiple individuals, as evidenced here by the clearer ontogenetic pattern of separation of wing and leg bones within duckling cohorts. Graphical demonstrations of the results can be very simple and straightforward (as shown in figures 3 and 5a; electronic supplementary material, figures S4–S22) or more elaborate including multiple parameters, while still remaining easy to interpret (e.g. figures 4, 5b and 6).

Representing versatile input variables, RPPs can be qualitatively explored, numerically analysed and the results visualized in many different ways and certainly by several other methods not considered in this study. In fact, collecting data for RPP generation does not even have to be confined to two-dimensional ground sections, but can also be performed in three-dimensional, µCT- or synchrotron-generated virtual sections by both areal and volumetric measurements. Image processing algorithms can also be defined for the automatic measurement of RPPs. Provided a large enough training dataset is collected on extant animals with well-known growth and developmental traits, supervised machine learning approaches (e.g. deep neural networks) might be used to characterize these trait combinations with RPP trajectories, and then to predict the likelihood of functional traits and developmental stages of fossil tetrapods based on their osteohistology. Nevertheless, the easy and inexpensive way of RPP generation in ground sections, as shown in this study, is an important value of the method because this measurement does not require any costly, high-end equipment or elaborate software to be performed.

Finally, RPPs can be used to extend the current queries and explore other avenues of locomotor ontogenetic development. First of all, with RPP, we can probe deeper than ever into the ontogenetic context of the evolution of powered flight in the three volant vertebrate groups (pterosaurs, birds and bats) which offers a new, exciting field in osteohistological studies. This includes hotly debated questions about the proposed 'superprecocity' of pterosaurs, the first vertebrates ever to evolve powered flight (e.g. [101,102]), as well as the evolution of ontogeny of flight in maniraptoran dinosaurs [33,65]. Another interesting field to explore concerns ontogenetic shifts in posture. Ontogenetic postural changes have long been in the focus of a number of studies in dinosaurs (e.g. [103–107]), with only a single study attempting to qualitatively relate osteohistological features with a proposed shift from quadrupedal to bipedal stance [105]. Even though the number of modern taxa showing the proposed postural shifts from quadrupedal to bipedal or vice versa is extremely limited (but see e.g. hoatzins), with enough baseline RPP data collected in extant amniotes showing various allometric limb bone growth and function, RPPs can provide the first quantitative osteohistological means to construct and support models of locomotor ontogenetic strategies that may not have any direct modern analogues.

## 4.6. Added value to existing osteohistological methods

Recently, Griffin *et al.* [13] gave an extensive review of the various methods used in assessing ontogenetic maturity in fossil saurian reptiles, including their benefits and limitations. They concluded that, even with its potential caveats, long bone histology is indeed the 'overall reasonable method' to use in most fossil clades. However, their review also demonstrates that most osteohistological methods are useful only for qualitative ontogenetic staging and studying a narrow aspect of growth (fast versus slow, cyclical versus continuous), or are entirely reliant on the skeletochronological assessment of absolute age (e.g. for life-history reconstructions) for analysing patterns quantitatively. Besides all the caveats associated with such skeletochronological assessments [13,34], absolute ages can be uninformative or even confounding in comparative studies of animals that have, for instance, the same overall type of growth dynamics but on very different time scales due to differences in their absolute body size (e.g. small- versus large-bodied birds). Finally, most of these osteohistological methods largely disregard the important functional developmental aspects of skeletal growth.

By contrast, RPPs can be assessed without accurate knowledge of absolute body sizes, ages and lifespans, as long as vascular primary bone dominates at least a measurable region of the cortex. Furthermore, this method was specifically developed to capture and extract information of the intimately linked growth and functional development of individual bones and skeletons in the relative time-frame of the animals' own growth trajectory. RPPs are a standardized measure of developmental dynamics of long bones providing the first quantitative method to analyse these parameters in an intra- and interspecific comparative context. Modelling dynamics of diametric bone growth that are influenced by so many factors can be very challenging in fossils, and RPPs may be key input variables to gain insights into skeletal growth dynamics and their associated factors in an unprecedented resolution.

# 5. Conclusion

In our study, we have laid down the foundations of RPP, a new quantitative osteohistological approach that gives analytical insights into posthatching limb bone development. Using a very simple measurement technique, RPPs were generated for a sample of paravian limb bone sections with variable taxonomic and ontogenetic composition and analysed with diverse quantitative methods to compare and cluster these trajectories. Using extant bird species as 'proof-of-concept' cases, we demonstrated the versatile application and analytical potential of RPPs for exploring (i) intrasectional, interelemental and intraspecific histovariability through ontogeny; (ii) intra- and interspecific variation within a given ontogenetic window; and (iii) how all these features may relate to the known growth and developmental strategies. We showed that RPPs indeed reflect the intimately linked growth and functional maturation of limb bones and, as such, they are an extremely valuable quantitative osteohistological source of developmental information in extant and extinct tetrapods. These findings were mainly derived from RPPs in the most information-rich duck dataset and can be summarized as follows:

(1) Modular limb development largely results in differing RPP courses within an individual with altricial wing bones showing higher porosity levels and more erratically running trajectories than precocial leg bones.
(2) Accordingly, wing and leg bone RPPs tend to be separated by trajectory-grouping methods resulting in the analytical dissociation of the individual's skeleton with its altricial and precocial bones grouped together with elements of earlier and later ontogenetic stages, respectively.
(3) The level of skeletal dissociation changes through ontogeny reflecting age-dependent developmental dynamics, and in ducks, it reaches its maximum in mid- and late juveniles (15 and 30 dph).
(4) In ducks, porosity values in the outermost cortex ($Q_d$) appear to stay constant in homologous wing bones up to 30 dph, whereas they decrease in a stepwise manner in homologous leg bones, as ontogeny progresses.
(5) Within a skeleton, the courses of RPPs tend to become aligned and occupy or converge on similar porosity ranges in bones that show advanced functional maturity; a phenomenon we refer to here as RPP channelization. This phenomenon was observed in the RPPs of maturing ducklings as well as in the equally functional limbs in juvenile hoatzins and is supported by the fact that any fore-limb–hind-limb disparity in RPPs disappears by the adult stage in the sampled paravians.

(6) As for different bones, the ulnae of ducklings and the tarsometatarsi of ducklings and juvenile hoatzins show unusually low and high porosity for wing and leg elements, respectively, which renders their grouping counterintuitive in our working hypothesis. This bone-specific pattern is yet to be explained.

We also revealed the caveats of this approach along with some special cases in which extra caution needs to be taken concerning measurements and analyses of RPPs. Finally, we provided an overview of possible methodological adjustments and extensions that could be applied in future studies using RPPs as the core variable.

Our major goal was to create a quantitative osteohistological parameter that is informative of bone tissue development and could be standardized across different taxa. Whether and how bone histology reflects the complex allocational interplay between growth and functional maturation is just emerging from these preliminary results. It appears that the most promising standardizable comparative osteohistological correlates of bone growth and functional development are the porosity ranges along RPPs combined with the level of RPP channelization (course alignment and occupation of and/or convergence on similar porosity ranges). The unexpected emergence of these RPP-defined osteohistological correlates also highlights the potential of this method in discovering new patterns based on extant taxa that will allow diverse aspects of locomotor ontogeny in fossil taxa to be explored at an unprecedented level.

The emergence of such intriguing trends despite the small sample size and unbalanced ontogenetic and taxic composition of our dataset illustrates the analytical power of the RPP method. Although further baseline data collection on extant tetrapod limb bones is essential, RPP has major potential to be used as a quantitative osteohistological indicator of growth strategies across taxa and among elements within the skeleton. As the growth dynamics of elements are also strongly associated with their functional development along the precocial–altricial spectrum, these analyses can be put in the evolutionary context of diverse developmental strategies with the integration of phylogenetic data. As such, RPPs may also hold a key to elucidating the evolution of locomotor developmental strategies in different fossil groups, such as in the ontogenetic aspects of avian powered flight or in the ontogenetic postural shifts suggested to have occurred in some other archosaurian taxa.

To sum up, the following advantages of the RPP method are identified in this study:

(i) RPP is an easily quantifiable parameter by many different approaches (two- or three-dimensional measurements using thin sections or μCT data) in extant and extinct tetrapods.
(ii) Quantification strategy (number and size of sectors and quadrants) is adjustable to different types of bone samples (e.g. to relative cortex thickness, degree of remodelling and preservation quality).
(iii) Qualitative inspection and quantitative analyses of RPPs are equally valuable for inferences on developmental strategy (growth and functional maturation) of bones, individuals and taxa.
(iv) A broad range of quantitative analyses of RPPs is possible without exact knowledge of the age of the specimen.
(v) The RPP method represents the first potentially standardizable osteohistological parameter for intra- and interspecific developmental comparisons.

For the successful application of the RPP method, the following important conditions are to be met:

(i) The studied limb bone has to preserve an adequately sized region of the primary vascular cortex to be profiled.
(ii) Selection of sampling areas, and the size and number of sectors and quadrants need to be optimized for the research question, the nature of the bone samples and the chosen numeric analysis types.
(iii) A good knowledge of bone growth and maturation principles is needed for correct RPP interpretations.

Data accessibility. The data are provided in the electronic supplementary material [108].

Authors' contributions. E.P.: conceptualization, data curation, formal analysis, funding acquisition, investigation, methodology, project administration, visualization, writing—original draft and writing—review and editing; A.T.K.: formal analysis, methodology, software and visualization; A.A.: data curation and resources; D.A.: resources; P.G.: data curation and resources; D.-Y.H.: data curation and resources; R.J.B.: funding acquisition and resources.

All authors gave final approval for publication and agreed to be held accountable for the work performed therein.

Conflict of interest declaration. The authors have no conflict of interest to declare.

Funding. This work was supported by Bijzonders Onderzoeksfonds (BOF) – Universiteit Gent (grant no. 01P12815), Fonds Wetenschapelijk Onderzoeks (FWO) – Vlaanderen (grant no. 1504218N), MTA-MTM-ELTE Paleo Contribution (no. 364), by the European Union's Horizon 2020 Research and Innovation Programme under the Marie Sklodowska-Curie Grant Agreement (grant no. 882758) to E.P., by ATM MNHN and PEPS CNRS to A.A., and by NSFC (grant nos. 41688103 and 42072030) to D.-Y.H.

Acknowledgement. We are grateful to Lucas Legendre and Andrew Lee for their constructive criticism and insightful comments on earlier versions of the manuscript which helped us greatly improve the standards of our study.

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
