## [Peer Review File · Royal Society Open Science]

Review History

RSOS-211150.R0 (Original submission)

Review form: Reviewer 1 (Lucas Legendre)

Is the manuscript scientifically sound in its present form?

Yes

Are the interpretations and conclusions justified by the results?

Yes

Is the language acceptable?

Yes

Do you have any ethical concerns with this paper?

No

Have you any concerns about statistical analyses in this paper?

Yes

Recommendation?

Accept with minor revision (please list in comments)

Comments to the Author(s)

Dear Editor, Dear Authors,

Please find enclosed my review for the manuscript " Radial Porosity Profiles: a new bone histological method for comparative developmental analysis of diametric limb bone growth" (RSOS-211150), submitted to Royal Society Open Science by Prondvai and colleagues, along with an annotated PDF (see Appendix A) with additional comments on specific sections of the paper.

This paper presents a new osteohistological method, radial porosity profiles (RPP), applied to a sample of birds and non-avian dinosaurs. This method is designed to provide a simple and efficient quantitative estimator of porosity that can be used as a graphical tool and in clustering procedures to group individual bones and/or specimens by ontogenetic strategy, which allows to make inference on their position on the precociality/altriciality spectrum. Such an assessment of ontogenetic strategy and growth pattern is usually done qualitatively in histological studies; the authors took the time to write a short supplementary review to explain the qualitative processes they aim to quantify. Such an initiative can only be lauded, as standardized methods for this purpose are long overdue in the field of quantitative histology, and should really become common practice in functional studies comparing growth patterns in vertebrates – especially when including fossil species.

The paper is long and technical – as it should be, since there are many functional aspects of growth and potential caveats to consider. In particular, the lack of information on some elements that could not be sampled properly due to either lack of material or extensive remodeling, as well as a small sample size in the context of both birds and non-avian paravians, render any generalization of the results difficult. Many additional factors remained to be explored, i.e. other traits that could have an influence on the growth strategies described here, additional clustering procedures to test, uncertainties regarding some of the parameters such as the number of quadrants per sector, etc. However, the authors make it very clear that the purpose of the paper is to provide a framework in which those issues and technical aspects could be explored in future research. An extensive introduction provides the context necessary to understand why such a method is needed in a comparative framework; the discussion reviews in details all potential sources of error that could undermine these preliminary results and should be explored in future studies. This transparency makes the paper a good introduction to many aspects of quantitative bone histology, and a gateway into major future challenges to be undertaken by the community. For this reason, this paper, albeit preliminary, fills a real gap in the literature and is a significant contribution to the field.

There are a few issues that should be resolved before acceptance. The main one is a potential issue in the way some parts of the clustering procedure are performed: outputs of the trajectory similarity measure analyses are treated as distance matrices for k-means clustering, which is inadequate to treat such data format (see annotated PDF); the use of e.g. k-medoids would be more appropriate in this context, and likely will not change the results much (especially since only a fraction of the results and discussion focuses on these analyses).

The paper could also be improved in the way it sells its main point – the novelty and efficiency of the method – as something that would be easily replicable by other bone histologists. As we all know, there are too many papers and too little time, and readers will inevitably skim through this

study looking for one main way to implement RPPs, without necessarily meandering through its caveats as much as they should. In this context, it is important to make that information available in a clear and accessible format, so that even the hastiest reader understands the main pros and cons of the approach.

One way to do this would be further illustrate the results and discussion sections, which are well written but very lengthy and technical, and currently only cite supplementary figures without referring to data visualizations actually present in the main paper (or barely). These long sections would greatly benefit from summary figures, with compilations of e.g. 1 RPP for each specimen of duck or species in the sample (I realize that Figs 3 and 4 have that function already, but they are barely mentioned in the discussion). Similarly, a comparative figure with a panel for each clustering procedure for a given sample, showing the difference in output for each method and displaying the percentage of congruence between them, would really help the reader understand the value of the use of all these different procedures to assess congruence, and show that such a visual check can be used to interpret results efficiently (this is in part what Table S2-4 do). Finally, as also stated in the annotated PDF, I think the conclusion would be much clearer if it listed a few numbered bullet points (3 to 5) as the main takeaways from the paper, both in terms of what this method brings to the field and what more should be done/accounted for in the near future to make it fully operational for any sample of birds (and other amniotes too).

Since these changes do not require major reworks of the structure and analyses of this paper, I recommend minor revisions (one more round of review would probably be useful to check any addition in methods and/or figures). I look forward to seeing this manuscript published in Royal Society Open Science.

Sincerely,
Lucas Legendre

Review form: Reviewer 2

Is the manuscript scientifically sound in its present form?

Yes

Are the interpretations and conclusions justified by the results?

Yes

Is the language acceptable?

Yes

Do you have any ethical concerns with this paper?

No

Have you any concerns about statistical analyses in this paper?

Yes

Recommendation?

Major revision is needed (please make suggestions in comments)

Comments to the Author(s)

Dr. Prondvai and colleagues introduce a method of measuring bone (vascular) porosity as a new way to assess the relative ontogenetic stage of a bone specimen. This method was tested on a

variety of samples including a growth series of known-age domestic ducks, hoatzin chicks of unknown age, and a previously published dataset of fossil paravian dinosaurs. For each bone sample (i.e., transverse section at midshaft), the areal proportion of porosity was measured at four regular intervals along several radial tracts. These radial porosity profiles (RPPs) were compared using a set of clustering algorithms. Major findings include: (1) decreased variability of RPPs with maturity; (2) the least and most mature bone specimens placed into distinct groups with additional groups reflecting intermediate stages of development; and (3) disassociation of bones from a single individual into several groups reflecting staggered development of limb modules. Overall, the methods seem reasonable, the results are consistent with expectations, and the discussion covers limitations and potential applications. I do want to raise a few points that I think can be addressed with revisions.

First, I think that the focus on radial porosity profiles needs more explanation. For instance, the introduction is not clear why porosity needs to be measured regular intervals along radial tracts instead of a single averaged value (either the average of the anatomical side of bone or the whole bone). If a single average value of porosity is sufficient to infer relative ontogenetic stage, then why go through the complex RPP methodology?

Second, many readers may be unfamiliar with the clustering techniques, and I think that the current text would benefit from additional details regarding the assumptions and setup for each clustering method. Here are few questions that I had after reading the manuscript a few times: Do the clusters represent relative ontogenetic stages – so a 3-cluster grouping could represent young, intermediate, and old clusters? Does a cluster method assume that the data points are independently sampled (the data points in each RPP and RPPs from bones of a single individual are not independent because they represent repeated measures)? Do you need to adjust the degrees of freedom to reflect the number of individuals rather than the number of bones? If the clustering method involves model selection (3 vs. 5 groups), what exactly is the model specification?

Third, the analysis involving GLM and logistic regression needs adjustment. The regression models assume that data points are independent of each other. However, multiple bones are sampled from a single individual. This repeated sampling, if not properly accounted for, will artificially inflate the degrees of freedom and make p-values too small. I suggest adding a random-effect to the model with specimen as a factor. In addition, you need to include a null model to test whether group membership is predicted by age and/or developmental strategy. If the dataset and relationship is strong, then model selection will exclude the null model. But that test still needs to be performed even if the additive and interaction models have significant coefficients.

Fourth, I'm not so sure about the identity of the endosteal lamellae mentioned on lines 516-518 and shown in Figure 1. Could that tissue be remnants of outermost periosteal bone from an early stage? For example, the 15-d radius in Figure 1 has what is called endosteal lamellae. But does that tissue match the outer periosteal bone of an 8-d radius? I've attached an image from my collection of mallards that shows how the outer periosteal bone of a 8-d radius overlaps the tissue that you called endosteal lamellae. In the absence of data from in vivo bone labeling (i.e., label 8-d chicks and see if those labels have an endosteal-most position in 15-d chicks), I suggest that you temper the statement that the "observations clearly disprove the general and widespread assumption that EL is a hallmark of cessation of medullary cavity expansion and radial bone growth." You should at least present the alternative hypothesis and acknowledge that in vivo testing is needed.

Finally, Figures 5-7 could be clearer. In particular, the overlapping labels within each cluster are illegible. I suggest placing labels in boxes off to the sides. That way readers can clearly see group membership.

I've prepared an annotated version of the manuscript with the complete list of my comments and suggestions (see Appendix B).

Decision letter (RSOS-211150.R0)

Dear Dr Prondvai

The Editors assigned to your paper RSOS-211150 "Radial Porosity Profiles: a new bone histological method for comparative developmental analysis of diametric limb bone growth" have made a decision based on their reading of the paper and any comments received from reviewers.

Per the comment of the Subject Editor, while the Editors and reviewers are broadly favourable in their view of your work, the additional analyses recommended may take longer than the 3 weeks our standard revision would permit. Thus, the Editors have recommended a resubmission, which will permit you up to 6 months to complete the tasks asked by the Editors and reviewers.

Both reviewers have provided attachments with their feedback - unfortunately, ScholarOne has a relatively low limit for file sizes for attachments. In the event that you do not receive a pair of PDFs and a single JPG file with this decision, please contact the editorial office who will be glad to resend the file(s) as needed.

We invite you to respond to the comments supplied below and prepare a resubmission of your manuscript. Below the referees' and Editors' comments (where applicable) we provide additional requirements. We provide guidance below to help you prepare your revision.

Please note that resubmitting your manuscript does not guarantee eventual acceptance, and we do not generally allow multiple rounds of revision and resubmission, so we urge you to make every effort to fully address all of the comments at this stage. If deemed necessary by the Editors, your manuscript will be sent back to one or more of the original reviewers for assessment. If the original reviewers are not available, we may invite new reviewers.

Please resubmit your revised manuscript and required files (see below) no later than 13-Mar-2022. Note: the ScholarOne system will 'lock' if resubmission is attempted on or after this deadline. If you do not think you will be able to meet this deadline, please contact the editorial office immediately.

Please note article processing charges apply to papers accepted for publication in Royal Society Open Science (<https://royalsocietypublishing.org/rsos/charges>). Charges will also apply to papers transferred to the journal from other Royal Society Publishing journals, as well as papers submitted as part of our collaboration with the Royal Society of Chemistry (<https://royalsocietypublishing.org/rsos/chemistry>). Fee waivers are available but must be

requested when you submit your manuscript
(<https://royalsocietypublishing.org/rsos/waivers>).

Thank you for submitting your manuscript to Royal Society Open Science and we look forward to receiving your resubmission. If you have any questions at all, please do not hesitate to get in touch.

on behalf of Dr Jennifer Botha (Associate Editor) and Kevin Padian (Subject Editor)
openscience@royalsociety.org

Associate Editor Comments to Author (Dr Jennifer Botha):

Associate Editor: 1

Comments to the Author:

Both reviewers were highly positive about your paper, however, both require some revision, particularly regarding the statistics. Both reviewers provide an annotated copy of the manuscript. Reviewer 2's questions appear particularly important in clarifying various points for the reader.

Subject Editor comments:

The reviewers and AE are clearly enthusiastic about this manuscript, and I agree. Its complexities have brought out a range of comments and suggestions from our reviewers, both of whom provided detailed and thoughtful assessments. I am logging a "reject/resub" decision not on the basis of the quality but because our "major revision" timeline is perhaps too short to do justice to the (very complex) analyses that the reviewers suggest, and you should not be constrained by deadlines. We look forward to your resubmission and your responses to their comments.

Reviewer comments to Author:

Reviewer: 1

Comments to the Author(s)

Dear Editor, Dear Authors,

Please find enclosed my review for the manuscript "Radial Porosity Profiles: a new bone histological method for comparative developmental analysis of diametric limb bone growth" (RSOS-211150), submitted to Royal Society Open Science by Prondvai and colleagues, along with an annotated PDF with additional comments on specific sections of the paper.

This paper presents a new osteohistological method, radial porosity profiles (RPP), applied to a sample of birds and non-avian dinosaurs. This method is designed to provide a simple and efficient quantitative estimator of porosity that can be used as a graphical tool and in clustering procedures to group individual bones and/or specimens by ontogenetic strategy, which allows to make inference on their position on the precociality/altriciality spectrum. Such an assessment of ontogenetic strategy and growth pattern is usually done qualitatively in histological studies; the authors took the time to write a short supplementary review to explain the qualitative processes they aim to quantify. Such an initiative can only be lauded, as standardized methods for this purpose are long overdue in the field of quantitative histology, and should really become common practice in functional studies comparing growth patterns in vertebrates – especially when including fossil species.

The paper is long and technical – as it should be, since there are many functional aspects of growth and potential caveats to consider. In particular, the lack of information on some elements that could not be sampled properly due to either lack of material or extensive remodeling, as well as a small sample size in the context of both birds and non-avian paravians, render any generalization of the results difficult. Many additional factors remained to be explored, i.e. other traits that could have an influence on the growth strategies described here, additional clustering procedures to test, uncertainties regarding some of the parameters such as the number of quadrants per sector, etc. However, the authors make it very clear that the purpose of the paper is to provide a framework in which those issues and technical aspects could be explored in future research. An extensive introduction provides the context necessary to understand why such a method is needed in a comparative framework; the discussion reviews in details all potential sources of error that could undermine these preliminary results and should be explored in future studies. This transparency makes the paper a good introduction to many aspects of quantitative bone histology, and a gateway into major future challenges to be undertaken by the community. For this reason, this paper, albeit preliminary, fills a real gap in the literature and is a significant contribution to the field.

There are a few issues that should be resolved before acceptance. The main one is a potential issue in the way some parts of the clustering procedure are performed: outputs of the trajectory similarity measure analyses are treated as distance matrices for k-means clustering, which is inadequate to treat such data format (see annotated PDF); the use of e.g. k-medoids would be more appropriate in this context, and likely will not change the results much (especially since only a fraction of the results and discussion focuses on these analyses).

The paper could also be improved in the way it sells its main point – the novelty and efficiency of the method – as something that would be easily replicable by other bone histologists. As we all know, there are too many papers and too little time, and readers will inevitably skim through this study looking for one main way to implement RPPs, without necessarily meandering through its caveats as much as they should. In this context, it is important to make that information available in a clear and accessible format, so that even the hastiest reader understands the main pros and cons of the approach.

One way to do this would be further illustrate the results and discussion sections, which are well written but very lengthy and technical, and currently only cite supplementary figures without referring to data visualizations actually present in the main paper (or barely). These long sections would greatly benefit from summary figures, with compilations of e.g. 1 RPP for each specimen of duck or species in the sample (I realize that Figs 3 and 4 have that function already, but they are barely mentioned in the discussion). Similarly, a comparative figure with a panel for each clustering procedure for a given sample, showing the difference in output for each method and displaying the percentage of congruence between them, would really help the reader understand the value of the use of all these different procedures to assess congruence, and show that such a visual check can be used to interpret results efficiently (this is in part what Table S2–4 do). Finally, as also stated in the annotated PDF, I think the conclusion would be much clearer if it listed a few numbered bullet points (3 to 5) as the main takeaways from the paper, both in terms of what this method brings to the field and what more should be done/accounted for in the near future to make it fully operational for any sample of birds (and other amniotes too).

Since these changes do not require major reworks of the structure and analyses of this paper, I recommend minor revisions (one more round of review would probably be useful to check any addition in methods and/or figures). I look forward to seeing this manuscript published in Royal Society Open Science.

Sincerely,
Lucas Legendre

Reviewer: 2

Comments to the Author(s)

Dr. Prondvai and colleagues introduce a method of measuring bone (vascular) porosity as a new way to assess the relative ontogenetic stage of a bone specimen. This method was tested on a variety of samples including a growth series of known-age domestic ducks, hoatzin chicks of unknown age, and a previously published dataset of fossil paravian dinosaurs. For each bone sample (i.e., transverse section at midshaft), the areal proportion of porosity was measured at four regular intervals along several radial tracts. These radial porosity profiles (RPPs) were compared using a set of clustering algorithms. Major findings include: (1) decreased variability of RPPs with maturity; (2) the least and most mature bone specimens placed into distinct groups with additional groups reflecting intermediate stages of development; and (3) disassociation of bones from a single individual into several groups reflecting staggered development of limb modules. Overall, the methods seem reasonable, the results are consistent with expectations, and the discussion covers limitations and potential applications. I do want to raise a few points that I think can be addressed with revisions.

First, I think that the focus on radial porosity profiles needs more explanation. For instance, the introduction is not clear why porosity needs to be measured regular intervals along radial tracts instead of a single averaged value (either the average of the anatomical side of bone or the whole bone). If a single average value of porosity is sufficient to infer relative ontogenetic stage, then why go through the complex RPP methodology?

Second, many readers may be unfamiliar with the clustering techniques, and I think that the current text would benefit from additional details regarding the assumptions and setup for each clustering method. Here are few questions that I had after reading the manuscript a few times: Do the clusters represent relative ontogenetic stages – so a 3-cluster grouping could represent young, intermediate, and old clusters? Does a cluster method assume that the data points are independently sampled (the data points in each RPP and RPPs from bones of a single individual are not independent because they represent repeated measures)? Do you need to adjust the degrees of freedom to reflect the number of individuals rather than the number of bones? If the clustering method involves model selection (3 vs. 5 groups), what exactly is the model specification?

Third, the analysis involving GLM and logistic regression needs adjustment. The regression models assume that data points are independent of each other. However, multiple bones are sampled from a single individual. This repeated sampling, if not properly accounted for, will artificially inflate the degrees of freedom and make p-values too small. I suggest adding a random-effect to the model with specimen as a factor. In addition, you need to include a null model to test whether group membership is predicted by age and/or developmental strategy. If the dataset and relationship is strong, then model selection will exclude the null model. But that test still needs to be performed even if the additive and interaction models have significant coefficients.

Fourth, I'm not so sure about the identity of the endosteal lamellae mentioned on lines 516-518 and shown in Figure 1. Could that tissue be remnants of outermost periosteal bone from an early stage? For example, the 15-d radius in Figure 1 has what is called endosteal lamellae. But does that tissue match the outer periosteal bone of an 8-d radius? I've attached an image from my collection of mallards that shows how the outer periosteal bone of a 8-d radius overlaps the tissue that you called endosteal lamellae. In the absence of data from in vivo bone labeling (i.e., label 8-d chicks and see if those labels have an endosteal-most position in 15-d chicks), I suggest that you

temper the statement that the “observations clearly disprove the general and widespread assumption that EL is a hallmark of cessation of medullary cavity expansion and radial bone growth.” You should at least present the alternative hypothesis and acknowledge that in vivo testing is needed.

Finally, Figures 5-7 could be clearer. In particular, the overlapping labels within each cluster are illegible. I suggest placing labels in boxes off to the sides. That way readers can clearly see group membership.

I've prepared an annotated version of the manuscript with the complete list of my comments and suggestions.

===PREPARING YOUR MANUSCRIPT===

===PREPARING YOUR REVISION IN SCHOLARONE===

Author's Response to Decision Letter for (RSOS-211150.R0)

See Appendices C - F.

RSOS-211893.R0

Review form: Reviewer 1 (Lucas Legendre)

Is the manuscript scientifically sound in its present form?

Yes

Are the interpretations and conclusions justified by the results?

Yes

Is the language acceptable?

Yes

Do you have any ethical concerns with this paper?

No

Have you any concerns about statistical analyses in this paper?

No

Recommendation?

Accept with minor revision (please list in comments)

Comments to the Author(s)

Dear Editor, Dear Authors,

Please find enclosed my second round of review for the manuscript " Radial Porosity Profiles: a new bone histological method for comparative developmental analysis of diametric limb bone growth" (RSOS-211150), submitted to Royal Society Open Science by Prondvai and colleagues, along with an annotated PDF with additional comments on specific sections of the paper.

The authors reworked their manuscript, figures, and supplementary information extensively to respond to all comments made by both reviewers, and this new version is much clearer and easier to follow than the first draft. In particular, detailed descriptions of the methodology in the Material & Methods section, and clarifications of some of the patterns observed for RPPs in the Results section, allow the reader to go through the whole RPP methodology and understand the main goal of the paper very quickly. This helps conveying how useful the methodology might be for future histological studies, and how ontogenetic patterns could be compared in a standardized format on larger samples in this context. Similarly, the addition of bullet points in the conclusion clarify the main selling points of RPPs as a method, as well as their requirements to be used in a standardized comparative context.

I am also glad that some of the previously supplementary tables have now been included in the main text, which allow to compare the level of congruence of the different procedures used to cluster RPPs much more easily. The discussion on skeletal dissociation, a concept that might not always be very intuitive in light of the many distinct ontogenetic trajectories of the bones investigated in this study, is now much more developed, and replaced in the context of precociality vs. altriciality of different skeletal subunits. This structure is more cohesive and consistent with the main results of the paper, especially when it comes to interspecific comparisons and hypotheses regarding growth strategies of ducks vs. other extant birds. The authors remain cautious in their interpretations of growth patterns and the generalization of the method to other taxa depending on e.g. bone remodeling, optimal number of sectors and

quadrants, etc. which is essential when discussing such a new procedure tested on a relatively small dataset.

I only have two minor comments on the manuscript as it is:

A) l. 485–486: "The level of skeletal dissociation of specimens was assessed in each analysis." I assume you mean how many clusters were bones separated into for a given specimen (e.g. depending on whether the RPP comes from a wing or leg bone), but is there a specific metric you used to estimate skeletal dissociation? Please clarify.

B) What agglomeration method did you use in 'hclust' (e.g. Ward, complete)? This is important to ensure full replication of these results.

The edits performed in this new version are overall highly satisfactory, and I recommend acceptance of the paper after the two above minor remarks have been answered.

Sincerely,
Lucas Legendre

Review form: Reviewer 2

Is the manuscript scientifically sound in its present form?

Yes

Are the interpretations and conclusions justified by the results?

Yes

Is the language acceptable?

Yes

Do you have any ethical concerns with this paper?

No

Have you any concerns about statistical analyses in this paper?

No

Recommendation?

Accept as is

Comments to the Author(s)

In a comprehensive revision of their previous submission, Dr. Prondvai and colleagues addressed comments and questions regarding a new methodological framework to assess the ontogenetic stage of limb bones. The framework is based on measures of cortical porosity along comparable radial tracts in transverse midshaft sections of bone. The authors tested the method on datasets representing the spectrum of specimens typically studied by paleobiologists ranging from extant taxa with known ontogenetic ages to fossils with unknown ages. The radial tracts of cortical porosity (RPPs) were compared using a variety of clustering methods. Major findings are generally consistent with expectations and demonstrate the broad utility of the framework for neontological and paleontological applications.

The author's responses to reviewer comments and questions are well-reasoned and supported by sufficient evidence. Clarifications requested by the reviews were incorporated into the text, figures, and supporting R code. The revised text, although quite dense, is greatly improved. It presents the reader with the motivation for the research question, necessary background to understand the methods, and interpretation of biological meaning of the results. It also clearly states the limitations of the work and presents the results as a starting point for additional data collection and methodological refinement. Other than a couple of minor typos in the Conclusions (e.g., L 1278: "information-rick"; L 1333: "tertapods"), I think that this revision is now publication worthy.

Decision letter (RSOS-211893.R0)

Dear Dr Prondvai

On behalf of the Editors, we are pleased to inform you that your Manuscript RSOS-211893 "Radial Porosity Profiles: a new bone histological method for comparative developmental analysis of diametric limb bone growth" has been accepted for publication in Royal Society Open Science subject to minor revision in accordance with the referees' reports. Please find the referees' comments along with any feedback from the Editors below my signature.

Please submit your revised manuscript and required files (see below) no later than 7 days from today's (ie 07-Apr-2022) date. Note: the ScholarOne system will 'lock' if submission of the revision is attempted 7 or more days after the deadline. If you do not think you will be able to meet this deadline please contact the editorial office immediately.

on behalf of Dr Jennifer Botha (Associate Editor) and Kevin Padian (Subject Editor)
openscience@royalsociety.org

Associate Editor Comments to Author (Dr Jennifer Botha):

The authors have answered/corrected all the reviewers' comments satisfactorily. Reviewer 1 has just a few minor comments, but otherwise it is ready to be accepted for publication

Reviewer comments to Author:

Reviewer: 1

Comments to the Author(s)

Dear Editor, Dear Authors,

Please find enclosed my second round of review for the manuscript " Radial Porosity Profiles: a new bone histological method for comparative developmental analysis of diametric limb bone growth" (RSOS-211150), submitted to Royal Society Open Science by Prondvai and colleagues, along with an annotated PDF with additional comments on specific sections of the paper.

The authors reworked their manuscript, figures, and supplementary information extensively to respond to all comments made by both reviewers, and this new version is much clearer and easier to follow than the first draft. In particular, detailed descriptions of the methodology in the Material & Methods section, and clarifications of some of the patterns observed for RPPs in the Results section, allow the reader to go through the whole RPP methodology and understand the main goal of the paper very quickly. This helps conveying how useful the methodology might be for future histological studies, and how ontogenetic patterns could be compared in a standardized format on larger samples in this context. Similarly, the addition of bullet points in the conclusion clarify the main selling points of RPPs as a method, as well as their requirements to be used in a standardized comparative context.

I am also glad that some of the previously supplementary tables have now been included in the main text, which allow to compare the level of congruence of the different procedures used to cluster RPPs much more easily. The discussion on skeletal dissociation, a concept that might not always be very intuitive in light of the many distinct ontogenetic trajectories of the bones investigated in this study, is now much more developed, and replaced in the context of precociality vs. altriciality of different skeletal subunits. This structure is more cohesive and consistent with the main results of the paper, especially when it comes to interspecific comparisons and hypotheses regarding growth strategies of ducks vs. other extant birds. The authors remain cautious in their interpretations of growth patterns and the generalization of the method to other taxa depending on e.g. bone remodeling, optimal number of sectors and quadrants, etc. which is essential when discussing such a new procedure tested on a relatively small dataset.

I only have two minor comments on the manuscript as it is:

A) l. 485–486: "The level of skeletal dissociation of specimens was assessed in each analysis."

I assume you mean how many clusters were bones separated into for a given specimen (e.g. depending on whether the RPP comes from a wing or leg bone), but is there a specific metric you used to estimate skeletal dissociation? Please clarify.

B) What agglomeration method did you use in 'hclust' (e.g. Ward, complete)? This is important to ensure full replication of these results.

The edits performed in this new version are overall highly satisfactory, and I recommend acceptance of the paper after the two above minor remarks have been answered.

Sincerely,

Lucas Legendre

Reviewer: 2

Comments to the Author(s)

In a comprehensive revision of their previous submission, Dr. Prondvai and colleagues addressed comments and questions regarding a new methodological framework to assess the ontogenetic stage of limb bones. The framework is based on measures of cortical porosity along comparable radial tracts in transverse midshaft sections of bone. The authors tested the method on datasets representing the spectrum of specimens typically studied by paleobiologists ranging from extant taxa with known ontogenetic ages to fossils with unknown ages. The radial tracts of cortical porosity (RPPs) were compared using a variety of clustering methods. Major findings are generally consistent with expectations and demonstrate the broad utility of the framework for neontological and paleontological applications.

The author's responses to reviewer comments and questions are well-reasoned and supported by sufficient evidence. Clarifications requested by the reviews were incorporated into the text, figures, and supporting R code. The revised text, although quite dense, is greatly improved. It presents the reader with the motivation for the research question, necessary background to understand the methods, and interpretation of biological meaning of the results. It also clearly states the limitations of the work and presents the results as a starting point for additional data collection and methodological refinement. Other than a couple of minor typos in the Conclusions (e.g., L 1278: "information-rick"; L 1333: "tertapods"), I think that this revision is now publication worthy.

===PREPARING YOUR MANUSCRIPT===

one version should clearly identify all the changes that have been made (for instance, in coloured highlight, in bold text, or tracked changes);

If you have been asked to revise the written English in your submission as a condition of publication, you must do so, and you are expected to provide evidence that you have received language editing support. The journal would prefer that you use a professional language editing service and provide a certificate of editing, but a signed letter from a colleague who is a proficient user of English is acceptable. Note the journal has arranged a number of discounts for authors

using professional language editing services
(<https://royalsociety.org/journals/authors/benefits/language-editing/>).

===PREPARING YOUR REVISION IN SCHOLARONE===

-- If you are requesting an article processing charge waiver, you must select the relevant waiver option (if requesting a discretionary waiver, the form should have been uploaded, see 'File upload' above).

-- If you have uploaded any electronic supplementary (ESM) files, please ensure you follow the guidance at <https://royalsociety.org/journals/authors/author-guidelines/#supplementary-material> to include a suitable title and informative caption. An example of appropriate titling and

captioning may be found at https://figshare.com/articles/Table_S2_from_Is_there_a_trade-off_between_peak_performance_and_performance_breadth_across_temperatures_for_aerobic_sc_ope_in_teleost_fishes_/3843624.

Author's Response to Decision Letter for (RSOS-211893.R0)

See Appendices G.

Decision letter (RSOS-211893.R1)

Dear Dr Prondvai,

I am pleased to inform you that your manuscript entitled "Radial Porosity Profiles: a new bone histological method for comparative developmental analysis of diametric limb bone growth" is now accepted for publication in Royal Society Open Science.

on behalf of Dr Jennifer Botha (Associate Editor) and Kevin Padian (Subject Editor)
openscience@royalsociety.org

Appendix A**ROYAL SOCIETY
OPEN SCIENCE****Radial Porosity Profiles: a new bone histological method for comparative developmental analysis of diametric limb bone growth**

Journal:	Royal Society Open Science
Manuscript ID	RSOS-211150
Article Type:	Research
Date Submitted by the Author:	08-Jul-2021
Complete List of Authors:	Prondvai, Edina; University of Birmingham, Earth & Environmental Sciences; MTM-ELTE Research Group for Paleontology Kocsis, Ádám; Friedrich-Alexander-Universität Erlangen-Nürnberg Department Geographie und Geowissenschaften, GeoZentrum Nordbayern, Fachgruppe Paläoumwelt Abourachid, Anick; UMR 7179 Muséum National d'Histoire Naturelle – CNRS, Département Adaptations du Vivant, Bâtiment d'Anatomie Comparée Adriaens, Dominique; Ghent University, Biology Godefroit, Pascal; Royal Belgian Institute of Natural Sciences Hu, Dongyu; Shenyang Normal University, Key Laboratory for Evolution of Past Life and Change of Past Environment Butler, Richard; University of Birmingham, School of Geography and Earth Sciences
Subject:	evolution < BIOLOGY, palaeontology < BIOLOGY
Keywords:	birds, growth and functional maturity, ontogeny, precocial–altricial development, quantitative bone histology, radial porosity profile
Subject Category:	Organismal and Evolutionary Biology

Author-supplied statements

Relevant information will appear here if provided.

Ethics

Does your article include research that required ethical approval or permits?:

This article does not present research with ethical considerations

Statement (if applicable):

CUST_IF_YES_ETHICS :No data available.

Data

It is a condition of publication that data, code and materials supporting your paper are made publicly available. Does your paper present new data?:

Yes

Statement (if applicable):

All data for editors and reviewers are available within the main MS and in supplementary material.

Conflict of interest

I/We declare we have no competing interests

Statement (if applicable):

CUST_STATE_CONFLICT :No data available.

Authors' contributions

This paper has multiple authors and our individual contributions were as below

Statement (if applicable):

EP conceived and designed the study and methodology, dissected the ducks, prepared the bone samples, photographed and evaluated the sections, took the measurements, ran the analyses, prepared the figures and wrote the manuscript; ÅTK helped to conceptualize and implement the numeric analyses, wrote the R codes, reviewed and edited final draft; AA acquired and currently curates the duck specimens used in this study, provided access to the hoatzin specimens, reviewed and edited final draft; DA provided necessary lab facilities, reviewed and edited final draft; PG and D-YH provided access to fossil paravian sections, reviewed and edited final draft; RJB reviewed and edited final draft. All authors gave final approval for publication.

**1 Radial Porosity Profiles: a new bone histological method for comparative developmental**
**2 analysis of diametric limb bone growth**

Edina Prondvai^{1,2*}, m T. Kocsis³, Anick Abourachid⁴, Dominique Adriaens⁵, Pascal
Godefroit⁶, Dong-Yu Hu^{7,8}, Richard J. Butler¹

¹School of Geography, Earth & Environmental Sciences, University of Birmingham,
Edgbaston, Birmingham, UK

²MTA-MTM-ELTE Research Group for Paleontology, Budapest, Hungary

³Department of Palaeobiology, Friedrich-Alexander-University of Erlangen-Nurnberg,
Erlangen, Germany

⁴Departement Adaptations du Vivant, UMR 7179 Museum National d’Histoire Naturelle –
CNRS, Paris, France

⁵Department of Biology, Evolutionary Morphology of Vertebrates, Ghent University, Ghent,
Belgium

⁶Directorate Earth & History of Life, Royal Belgian Institute of Natural Sciences, Brussels,
Belgium

⁷Paleontological Institute of Shenyang Normal University, Key Laboratory for Evolution of
Past Life in Northeast Asia, Ministry of Land and Resources, Shenyang, China

⁸Paleontological Museum of Liaoning, Shenyang, China

*Corresponding author. E-mail: edina.prondvai@gmail.com; E.Prondvai@bham.ac.uk

**Abstract**

Assessing ontogenetic patterns in fossil vertebrates is crucial in palaeobiology and many other
fields of vertebrate evolution. Limb bone histology is considered the most reliable tool not
only for skeletal maturity assessments but also for evaluating the growth record in the
ontogenetic window represented by the primary bone cortex. This is especially relevant in the
context of variable ontogenetic strategies along the precocial–altricial developmental
spectrum. Primary cortical vascularity is the most informative histological character for
reconstructing growth dynamics due to its complex relationship with bone growth and
functional maturation. Using this concept as our working hypothesis, we introduce here a new
quantitative osteohistological parameter, radial porosity profile (RPP), that captures relative
cortical porosity changes as trajectories in limb bones. We built a proof-of-concept RPP
dataset on extant birds to which we added fossil paravian dinosaurs. We performed a set of
trajectory-grouping analyses to identify potential RPP categories which were evaluated in the
context of our ontogeny – developmental strategy working hypothesis. Our results support the
analytical power of RPPs and reveal unexpected potential osteohistological correlates of
growth and functional development of limb bones. The diverse potential applications of RPPs
opens up new research directions in the evolution of locomotor ontogeny.

[revised manuscript text omitted]

strategies. We here propose a new quantitative histological method for long bones, referred to
as radial porosity profiles (RPP), that builds upon the general principles of diametric bone
growth and development in the primary cortex of terrestrial tetrapods. Our method abstracts
growth and developmental dynamics from the sampled element on the basis of relative
primary porosity measured along radial trajectories across the preserved posthatching cortex.
The course of these short trajectories, i.e. the RPPs, is not only informative of the specific
ontogenetic phase/period the studied specimen represents in its overall growth trajectory, but
may also give insight into the developmental strategy of the studied skeletal element.

[revised manuscript text omitted]

fine details of the individual's developmental phase and strategy which then can be compared
with that of other conspecifics or specimens of other taxa. Whether growth is overall slow or
fast, cyclical or continuous, isometric or allometric, precocial or altricial, porosity patterns are
influenced by the cumulative effects of all these developmental processes, and thus represent
an information-rich abstraction of the ontogenetic trajectories and potentially element-specific
functional aspects of tetrapods.

*2.2.2. Selection of the data-collecting technique*

Our ontogenetic and developmental approach is based on the osteohistological evaluation of
the RPPs in limb bone diaphyses. Limb bone shafts are the most frequently analysed elements
in palaeohistological studies because the mid-diaphyseal region usually has the highest
primary bone content providing insight into individual growth trajectories (Padian & Lamm,
2013). Long bone histology is also shaped by diverse locomotor demands (de Margerie, 2002;
Skedros *et al.* 2003; Currey 2003; de Margerie *et al.* 2005; Robling *et al.* 2006; Ponton *et al.*
2007; Simons & O'Connor. 2012; Smith & Clarke. 2014; Lee & Simons, 2015; Frongia *et al.*

[revised manuscript text omitted]
 (Fig. 2D). Thereafter, a mean RPP from the three sectors ($S = 3$) defined by a vector of four points for each bone (Fig. 2E) was calculated as:

$$RPP = \left(\sum_{n=1}^S P_{a_S}^{\frac{1}{S}} \sum_{n=1}^S P_{b_S}^{\frac{1}{S}} \sum_{n=1}^S P_{c_S}^{\frac{1}{S}} \sum_{n=1}^S P_{d_S}^{\frac{1}{S}} \right) (3).$$

The courses of these RPPs (Fig. S3) were numerically analysed by diverse trajectory modelling and clustering methods in various comparative contexts (see Section 2.2.6).

Vascular measurements were only taken in primary cortical areas. Wherever secondary remodelling was present, their areas were measured within the quadrants and subtracted from the sampled areas (Fig. S2B, D). In those bones where secondary remodelling affected extensive cortical regions obliterating most or all of the primary cortex in a given quadrant (most frequently Q_{a-b}), the porosity values of the affected quadrants were considered missing (NAs) (Fig. S2B, C). Only those RPPs were analysed in which a minimum of two of the four points defining the sector-averaged trajectory could be calculated at least in one sector. The extent of secondary remodelling was also considered in the interpretation of the results.

The relative thicknesses of the outer circumferential layer (OCL) and the endosteal lamellar layer (EL), rimming the medullary cavity in most bones of the paravian dinosaurs and some bones of the extant bird specimens (Fig. 1H), were quantified in relation to overall cortex thickness as the area percentage it occupies of the total cortex area in the measured sectors. Similar to secondarily remodelled areas, OCL and EL were not included in the numeric analyses in this study but were considered during the interpretation of the results.

2.2.6. Numeric analysis – a selection of methods

Three different main approaches with various settings were selected to demonstrate the diverse analytical potential in the RPPs of individual bones: 1) K-means clustering; 2) group-based trajectory modelling (GBTM; Nagin, 2014); and 3) three types of trajectory similarity measures (dynamic time warping [DTW], ‘edit distance’ [ED], and longest common subsequence [LCSS]; Toohey & Duckham, 2015). K-means clustering is a widely used method applied in a diverse range of fields for classifying multivariate observations into groups (e.g. MacKay, 2003). The similarity of trajectories included mean squared distances (MSD) that reflect the expected distance between corresponding points (homologous quadrants) of the RPPs. GBTM is a more specific statistical approach to analyse developmental trajectories – a progression of any phenomenon described by longitudinal data – and to identify distinct groups of individual trajectories within a population (Nagin, 1999, 2014). GBTM assigns group membership probabilities to individual trajectories and compares the analytical support for all models to find the optimal (best supported) number of groups (Nagin, 1999, 2014; Nielsen, 2018). The third approach, trajectory similarity measures, comes from the field of computational movement analysis in which trajectories are defined as a “sequence of time-stamped locations” (Gudmundsson et al. 2012) and similarity is calculated by various distance measures between trajectory pairs (Toohey & Duckham, 2015). In the DTW method, the sum of the distances at each point along the trajectories is calculated as the smallest warping path. In ED, the minimum number of edits are calculated for two trajectories to be considered equivalent. Finally, LCSS works with the translation of the trajectories in all

400 possible dimensions (in our case 2D) to find the maximum number of equivalent points
between two trajectories (Gudmundsson et al. 2012; Toohey & Duckham, 2015; Toohey,
2018).

In all analysis types, we first ran the tests separately on the RPPs of the duck ontogenetic
series (Fig. S3A, B), then on those of the extinct paravian dinosaurs (Fig. S3C, D), and finally
on the complete dataset including the ducks, the extinct paravians, the juvenile hoatzins and
the other adult birds (Fig. S3E, F). Using these separate analyses, we could adjust the
resolution of the results to broaden the context of our interpretations and elaborate on the finer
details. We also analyzed separately those datasets with and without trajectories containing
missing values to see their effect on the group compositions. Furthermore, we also ran tests on
centered RPPs calculated for each quadrant as the difference from the mean sector porosity
within a given trajectory (Fig. S3B, D, E). In these tests, only the shape of the trajectories
determines their group memberships. Finally, in the duck dataset, for which we had the most
complete background information, potential predictors of group memberships were also tested
by logistic regressions in the best supported RPP-clustering models.

All analyses were performed in RStudio (1.4.1) integrated development environment
(RStudio, PBC) for the free programming language and statistical computing software R (R
Core Team). K-means clustering was called in from the basic 'stats' function *kmeans()*,
GBTM was performed using the function *crimCV()* in package 'crimCV' (Nielsen et al. 2014;
Nielsen, 2018), and the three different trajectory similarity measures were run with the
functions *DTW()*, *EditDist()* and *LCSS()* in package 'SimilarityMeasures' (Toohey, 2015). For
the K-means and GBTM analyses, we used the available ontogenetic, age and developmental
strategy data of the specimens, and the graphical distribution patterns of individual RPPs, to
select the number of groups to be tested. K-means uses a priori defined number of clusters,
and we tested 3–5 clusters in the different analyses. Although treating the distance values as
coordinates might have a degrading effect on the signal present in the data, the
multidimensional structure of points remains very similar. GBTM, on the other hand, works
with a pre-set maximum number of groups, which we set to five in our analyses. Because
GBTM has a stochastic component, we ran 100 iterations in each analysis to find a globally
best supported model using three model selection methods (Bayesian Information Criterion
[BIC], Akaike Information Criterion [AIC], cross-validation error [CVE]) with the function
*crimCV()* in which CVE was developed for testing GBTM itself (Nielsen et al. 2014). The
similarity/distance measures of trajectory pairs resulting from the three types of trajectory
similarity analyses were organized into distance matrices and further analysed with K-means
clustering set to 3–5 clusters. General and ordinal logistic regressions for testing potential
predictor variables of group memberships in ducks were run with the functions *glm()*
available in {stats} (R Core Team, 2020) and *porl()* in package 'MASS' (Venables & Ripley,
2002). P-values for the regression coefficients in the ordinal logistic regression were
calculated as for t-values with the function *pt()*. Model selection was based on AIC values
using the function *step()* available in {stats} (R Core Team, 2020). See complete datasets,
detailed R scripts and results of all analyses in Supplementary Table 1 and Supplementary
Information 2-6.

Results were interpreted in various contexts, such as the developmental strategies of
different elements within an individual, individual variation within a cohort of a species,

ontogenetic patterns of development of homologous elements and the whole skeleton, and
broader within-clade and interspecific comparisons.

**3. Results**

**3.1. Qualitative evaluation of RPPs**

Inspection of the simple graphical representation of RPPs reveals interesting patterns
concerning the developmental trajectories of limb elements (Figs. 3, S4–S18). In this
qualitative description, we mostly focus on the ducks where the ontogenetic spectrum and the
presence of three individuals in each age group (hereafter cohort) up to 30 dph allowed more
detailed interpretations (Figs. S4–S12). Other taxa in our sample (Figs. S13–S18) are
considered in the context of the findings in ducks.

**3.1.1. Ducks**

Although sample size is obviously inadequate to properly assess intracohort variance in
RPPs, the three growth series of ducks seem to show certain ontogenetic trends in the course
of the trajectories and their variance that may be related to the disparate locomotor functions
of the fore- and hind limbs.

At 4 dph, neither the ulna, nor the carpometacarpus has developed enough posthatching
periosteal cortex to be evaluated. All the other analysed limb bones of the ducklings show
high overall porosity levels throughout the cortex (~20–70%) with erratic-looking RPPs. The
intrasectional variance of RPPs (i.e. among the three sectors) lies within the same range as the
intracohort variance among homologous bones, and no separation appears between altricial
forelimb and precocial hind limb elements either in absolute porosity levels or in the course of
RPPs (Figs. S4, S9, S10). By 8 dph, a trend appears in the hind limb elements with more
channelized RPP courses, that is, with lower intrasectional and intracohort variance in a well-
defined course and radially increasing porosity, especially in the femur and tibiotarsus. On the
other hand, wing elements show no such trend in their RPPs and continue to be largely
irregular (Figs. S5, S9, S10). At 15 dph, the general pattern stays the same as at 8 dph, except
that the humerus starts showing similar channelization, and porosity may increase in the inner
cortex of some elements due to the formation of large resorption cavities related to medullary
cavity expansion (Figs. 3, S6, S9, S10). Furthermore, overall porosity seems to be higher in
the altricial wing bones, although the precocial tarsometatarsus remains in the same porosity
range as the wing bones at this stage. By 30 dph, RPP channelization occurs in every element
with overall higher porosity levels in the altricial wing bones (~20–50%) than in the precocial
leg bones (~5–30%) (Figs. S7, S9, S10), although pooled bones of the forelimb
[revised manuscript text omitted]

(Prondvai et al. 2018). RPPs of all other specimens converge on <3% P_d levels (Fig. S17).

Interestingly, and in contrast to the general pattern seen in our extant bird sample, an EL is
present in all sampled elements of these fossil paravians, while a definite OCL is missing in
all *Eosinopteryx* bones, and in some bones of the other, more mature specimens. Since hind
limb elements were only represented by the femora in this sample of extinct paravians, no
meaningful comparison can be made between the RPPs of fore- and hind limbs. Nevertheless,
RPPs of femora seem to be neatly aligned with RPPs of the forelimb elements showing no
evident deviation at these ontogenetic stages (Figs. S17, S18).

**3.2. Numeric analyses of RPPs**

In essence, all analyses of the raw data by K-means, GBTM and Similarity Measures (DTW
and ED combined with K-means) gave similar results for each dataset, that is for the ducks,
extinct paravians, and the complete dataset (Figs. 4–7, S19–S21, Supplementary Information
4–6.). On the other hand, centred data and LCSS gave quite erratic memberships in each case
which apparently did not relate to any studied parameters (Fig. S22, S23). Therefore, only the
raw data-based results are shown here, whereas those of centred data and LCSS are
considered further only in the Discussion.

**3.2.1. Ducks**

When the duck growth series were analysed with 3, 4 and 5 groups for describing trajectory
types from 4 dph up to adulthood, the 4-group setup consistently represents the best model
across different analysis methods (Fig. S19). Numeric support is provided by GBTM
iterations in which the 4-group model shows the most frequently, as well as globally, the
lowest AIC, BIC and CVE values (Fig. 4). Group compositions are fairly consistent among
different analyses with 74% of the elements clustered the same way by all four methods
(Table S2). The most inconsistency in membership assignments across methods occurred in
the 30 dph cohort, followed by 15 dph. Group memberships of bones appear to largely reflect
the combination of the individual’s ontogenetic stage and the developmental strategy of the

element. Most importantly, 4-group models successfully show how altricial and precocial
bone development within an individual can assign an individual's elements to groups
containing specimens of less and more mature ontogenetic stages, respectively (Fig. 5). This
means that these RPP-based trajectory groups (Fig. 5A) tend to “dissociate” the animal's
skeleton along its differently developing limb modules. For instance, the precocially
developing hind limb bones of a subadult animal (50 dph) are likely grouped together with
bones of adult animals, while its altricially developing wing bones tend to be assigned to a
separate group composed mostly of bones representing earlier ontogenetic stages, such as
mid- and late juveniles (Fig. 5B; Table S2).

At this analytical resolution encompassing the entire ontogenetic series up to adulthood,
group memberships also reveal that the predictive power of precocial *vs* altricial development
in separation of element RPPs changes throughout ontogeny. Dynamics of RPP disparity can
be traced in group compositions along the age axis combined with the level of skeletal
‘dissociation’ of specimens, that is the number of different groups an individual's bones were
assigned to. In the analyses with four groups, skeletal dissociation is not evident at 4 dph, (i.e.
all bones within an individual tend to be clustered in a single group) but starts to appear at 8
dph where wing and leg bones of an individual tend to be separated into two groups. The level
of skeletal dissociation may even reach three groups in some specimens at 15 and 30 dph but
returns to a bipartite division at 50 dph with perfect separation of wing and leg bones by
GBTM and K-means (Fig. 5B, Table S2). Figure 5B further illustrates these dynamics in
clusters #3 and #4, which contain most elements of 15 and 30 dph, (Fig. 5B). The distribution
pattern in these clusters is a transition from the younger ages (left-hand side) being mostly
represented by precocial hind limb bones (white circles) to the older ages (right-hand side)
containing primarily altricial forelimb bones (black circles). The mid-age ranges show a more
balanced mixture of fore- and hind limb bones in the respective groups. Thus, modularity in
fore- and hind limb development results in groups spanning three ad hoc maturity degrees
(from early to late juveniles in #3 and from mid-juveniles to subadults in #4) and clustering
together elements that originate from very differently aged animals (Fig. 5B).

Even though these groupings clearly result in a mixture of possible ontogenetic staging of
the animal based on RPPs of its individual elements, there are consistent limits as to which
ontogenetic stages can be represented by the differently developing elements within the same
group. For example, no bones of early juveniles (up to 8 dph) were assigned the same group
membership as those of subadults (50 dph), and no bones of mid-juveniles (up to 15 dph)
were grouped together with any adult bones, irrespective of the bone's individual
developmental strategy.

To further unfold how this complex RPP pattern might be related to the disparate growth
and development of altricial *vs* precocial elements, we separately analyzed by GBTM and K-
means the subsets of those ontogenetic stages where we had at least 3 animals in each
category, that is for early (4–8 dph), mid- (15 dph) and late juveniles (30 dph), respectively
(Table S3). Here, we set the maximum number of groups to 2 to see whether and how
precocial *vs* altricial strategy is reflected in the RPP group compositions through ontogeny.
These higher resolution analyses showed a similar trend to that recovered in the analysis
spanning the entire ontogenetic series but provided a much clearer pattern and stronger signals
related to the developmental strategies of individual bones.

The major trend is still the progressively clearer separation of the RPPs of precocially vs
altricially developing hind vs forelimb elements as ontogeny progresses. GBTM of early
juveniles (4–8 dph) showed no separation of bones at 4 dph, whereas RPPs of differently
developing elements start to differ more evidently at 8 dph with all of the altricial wing bones
being grouped together with bones of 4 dph, while all femora and tibiotarsi of the precocial
leg bones were grouped separately (Table S3). Nevertheless, tarsometatarsi at 8 dph are
clustered with the wing bones, and the distinctness of the groups is still weak, which is also
reflected in the closeness of support for both number of groups (1 and 2) provided by all
GBTM model selection methods (Supplementary Information 4). K-means gave very similar
results, although the ulnae at 8 dph were consistently assigned to the same cluster as the
femora and tibiotarsi at 8 dph. On the other hand, in late juveniles (30 dph) both GBTM and
K-means were able to identify and separate precocially and altricially developing bones with
100% accuracy based on their RPPs. Finally, the strength of separation of RPPs of precocial
and altricial elements in mid-juveniles (15 dph) is intermediate with a less distinct group
composition than in late juveniles but higher support for the two groups than in early juveniles
by all GBTM model selection methods (Table S3, Supplementary Information 4).

Based on our working hypothesis and the observed patterns in group compositions in the
entire dataset and its cohort subsets, age and developmental strategy of bones were selected to
be tested as predictors of GBTM and K-means group memberships in the complete duck
dataset with both linear combination of and interaction between these predictor variables.
Model selection (AIC) supported the model with linear combination of predictors. Logistic
regressions in each case showed that age is an overall stronger predictor than developmental
strategy with both variables being significant predictors of group memberships ($p < 0.0001$
for age and $p < 0.001$ for developmental strategy). For further details see Supplementary
Information 4.

As for individual elements, group membership probabilities highlight the precocial
tarsometatarsus seems to be the most inconsistent concerning its development-related group
membership in all analyses. This bone usually shows a highly porous cortex as compared with
the other two precocial hind limb elements, the femur and the tibiotarsus, within the same
specimen, and hence is frequently grouped together with the altricial wing elements. Finally,
the altricial ulnae at 8 dph (the earliest age for measurable ulnar cortex) tend to group together
with their contemporary precocial femora and tibiae along with elements of mid- and late
juveniles. At 8 dph the ulnar posthatching cortex is still fairly underdeveloped which may
account for this unexpected pattern. However, the same trend in the ulnae is still observed at
15 dph, although the signal is less clear.

3.2.2. Fossil paravian dinosaurs

Different types of analyses of the five specimens of fossil paravian dinosaurs gave moderately
variable results. In GBTM, all model selection methods supported the model where RPPs of
all studied elements belong to a single group (Figs. 6, S20; Supplementary Information 5).
Nevertheless, in the 2-group setup, one group contains a single element, namely the humerus
of the late juvenile *Eosinopteryx*, while all other elements are clustered together in the other
group. The same is the case in the 2-cluster K-means and ED analyses, while DTW assigns
four of the six elements of *Eosinopteryx* in this separate group. The 3-group setting provided

more diverse group memberships with different analyses. GBTM resulted again in the
separation of the *Eosinopteryx* humerus in one group, and so did K-means and ED analyses
(Figs. 6, S20). This recurring separation of *Eosinopteryx* elements by different RPP analysis
methods is in accordance with the previously assigned juvenile status of this specimen
(Prondvai et al. 2018). However, the rest of the ‘dinobird’ elements were clustered in variable
ways. GBTM sorted three more *Eosinopteryx* elements into a separate group and all other
elements in the third group. On the other hand, K-means and ED consistently grouped
together most of the elements of *Eosinopteryx* and *Jeholornis* with half of those of *Serikornis*
across the 3-group analyses, while DTW gave a less interpretable element sorting
(Supplementary Information 5).

Group membership probabilities of RPPs are best explained when precocity ranks of
elements (a qualitative order of relative osteohistological maturity of elements within the
skeleton; Prondvai et al. 2018) are also considered in combination with the assigned
ontogenetic stage of the specimen (deduction based on overall osteohistological maturity;
Prondvai et al. 2018). This is shown by the clear separation of the humerus of *Eosinopteryx*
from all other elements in both; the 2-group and 3-group setups. Memberships of elements of
*Serikornis* also tend to reflect its immature ontogenetic status along with the assigned element
precocity ranks in its skeleton (Prondvai et al. 2018).

22 727 3.2.3. Complete dataset

When tested for 3–5 groups (Figs. 7, S21; Supplementary Information 6), model selection
methods performed in the GBTM analyses supported 5 groups as the best model to cluster
RPPs in the complete dataset. However, the four different analysis types gave much less
congruent results for the composition of the 5 groups (Fig. S21) than they did for the duck-
only dataset with 4 groups. Here, only 35% of the elements clustered identically by all four
methods, and 45% by three of the four methods. Nevertheless, this still gives a total of 80% of
the elements being grouped with a minimum of 75% consistency across different methods
(Table S4). Group memberships assigned by GBTM and K-means reflect most faithfully the
combination of the specimens’ ontogenetic stages and the elements’ developmental strategies
(Fig. 7B), while DTW and ED cluster some elements less clearly related to these attributes
(Supplementary Information 6). Overall, no phylogenetic coherence appears in any of the
RPP clustering patterns.

Within the duck subgroup in the complete dataset, only 56% of the bones were identically
clustered by all four methods, while 20% and 21% were clustered by three and two of the four
methods, respectively (Table S4). Furthermore, when cluster assignments of duck elements in
this pooled dataset with 5 groups was compared with that of the duck-only dataset with 4
groups, 73% of the elements were clustered the same way by all four methods as in the duck-
only analyses. Another 19% of the bones were grouped identically by three out of the four
methods altogether giving a $\geq 75\%$ clustering congruence for 92% of the duck elements across
these analyses.

The subgroup of juvenile hoatzin bones show less consistency revealing that only 27% of
their elements were clustered identically by all four methods, and another 61% by three of the
four (Table S4). This still means that a $\geq 75\%$ clustering congruency occurs in 88% of their
elements. Within the hoatzin subgroup, GBTM and K-means vs. DTW and ED scatter their

bones across three vs. four clusters, respectively. Hoatzin bone clusters resulting from GBTM
and K-means largely cover the same clusters as do duckling bones of 15 dph, while DTW and
ED group the hind limb elements of the oldest hoatzin chick together with duck elements of
30 dph and 50 dph. GBTM seems to best reflect the estimated ages / ontogenetic stages of
these juveniles clustering most bones of the ~6 dph and ~9 dph chicks together, and
separating them from all but the ulna and carpometacarpus of the ~16 dph chick. Even though
the bone distribution pattern in the other three analyses, spanning up to four clusters, cannot
be satisfactorily related to ontogenetic stage and/or developmental strategies, distinction of
the oldest individual from the two younger chicks is clearly detectable in these analyses as
well. Finally, all four methods place the tarsometatarsi of the two younger hoatzin chicks in
the group containing most of the elements of the youngest ducks (4 dph) emphasizing again
the highly porous nature of this limb element in hoatzins (Table S4).

With a few exceptions, bones of extant adult birds were largely grouped together with
those of extinct paravian dinosaurs by all four methods. Remarkably, only K-means separated
the elements of the not yet fully-grown kestrel from the rest of the adult birds, and grouped
them together with some hind limb elements of ≥ 30 dph ducks and a single ‘dinobird’
element, the *Eosinopteryx* humerus (Fig. 7B). DTW and ED uniformly clustered all extant
adult birds in the group containing the most mature specimens in the complete dataset,
including all bones of the extinct dinosaurs. On the other hand, GBTM picked out a single
paravian element – the carpometacarpus of *Aurornis* – for its fifth cluster, while filling up the
fourth cluster with all the other bones of adults and two hind limb elements of the 50 dph
duck (Table S4).

When the cluster memberships of the ‘dinobird’ subgroup in the complete dataset is
compared with those in the 2-group ‘dinobird’-only analyses, it appears that the recurring
separation of the *Eosinopteryx* humerus in the latter disappears from all but the K-means
analysis in the former. Due to GBTM’s strange clustering in the complete dataset, only a
single element (*Aurornis* carpometacarpus) was clustered identically by all four methods, and
81% of all elements by three of the four methods in the ‘dinobird’-only and ‘dinobird’
subgroup analyses. This gives a $\geq 75\%$ clustering congruence for 85% of the ‘dinobird’
elements across these analyses.

32 782 33 783 3.2.4. Performance of and congruency among different methods

Besides the comparisons of clustering congruency of elements presented above, performance
and consistency of each method across the analyses can also be compared. Missing values
affected each method differently. For instance, GBTM can only work with complete RPPs
that have all four trajectory-defining points, while the rest of the methods tolerated incomplete
RPPs with up to two missing values differently. K-means has proven to be the most sensitive
to these missing values with only 67% of the elements sorted the same way in the analyses of
the complete dataset with and without incomplete RPPs. DTW and ED performed better with
98% and 86%, respectively.

Congruency in element sorting was examined between method pairs as well. When we
removed all elements with incomplete RPPs, the duck and the complete dataset included 70
and 128 elements, respectively. In this reduced duck dataset, the highest congruency in
membership assignments was found between GBTM and K-means, and between DTW and

ED, with both method pairs resulting in 93% of the elements clustered the same way. The
lowest congruence was detected between the K-means and ED methods, although it was still
high (79%) (Table S2). However, in the complete dataset encompassing 128 elements
spanning the same ontogenetic range but broadening the included developmental strategies,
the congruency levels of GBTM with respect to all other methods dropped considerably.
GBTM still had the highest congruency with K-means but it only reached 59%, whereas the
other the methods showed 77%–86% corresponding membership assignments (Table S4).

15 805 **4. Discussion**

[revised manuscript text omitted]

**Author Contributions**

EP conceived and designed the study and methodology, dissected the ducks, prepared the
bone samples, photographed and evaluated the sections, took the measurements, ran the
analyses, prepared the figures and wrote the manuscript; ÁTK helped to conceptualize and
implement the numeric analyses, wrote the R codes, reviewed and edited final draft; AA
acquired and currently curates the duck specimens used in this study, provided access to the
hoatzin specimens, reviewed and edited final draft; DA provided necessary lab facilities,
reviewed and edited final draft; PG and D-YH provided access to fossil paravian sections,
reviewed and edited final draft; RJB reviewed and edited final draft. All authors gave final
approval for publication.

**References**

- Abourachid A., Herrel A., Decamps T., Pages F., Fabre A.-C., Van Hoorebeke L., Adriaens
D., Garcia Amado M. A., 2019. Hoatzin nestling locomotion: Acquisition of quadrupedal
limb coordination in birds. *Science Advances*, 5, eaat0787.
Bailleul AM, Scannella JB, Horner JR, Evans DC, 2016. Fusion patterns in the skulls of
modern archosaurs reveal that sutures are ambiguous maturity indicators for the
Dinosauria. *PLoS ONE* 11(2): e0147687.
Bennett C. S. 1993. Ontogeny of *Pteranodon*. *Paleobiology*, 19(1), 92-106.
Bennett, M. B. 2008. Post-hatching growth and development of the pectoral and pelvic limbs
in the black noddy, *Anous minutus*. *Comparative Biochemistry and Physiology*, Part A 150,
159–168.
Brinkman, D. 1988. Size-independent criteria for estimating relative age in *Ophiacodon* and
*Dimetrodon* (Reptilia, Pelycosauria) from the Admiral and lower Belle Plains Formations
of West-Central Texas. *Journal of Vertebrate Paleontology* 8:172–180.
Brochu, C. A. 1996. Closure of neurocentral sutures during crocodylian ontogeny:
implications for maturity assessment in fossil archosaurs. *Journal of Vertebrate*
*Paleontology* 16:49–62.
Bybee PJ, Lee AH, Lamm E-T. 2006. Sizing the Jurassic theropod dinosaur *Allosaurus*:
Assessing growth strategy and evolution of ontogenetic scaling of limbs. *Journal of*
*Morphology*, 267(3):347–359.
Carrier, D., Leon, L.R. 1990. Skeletal growth and function in the California gull (*Larus*
*californicus*). *J. Zool.* 222, 375–389.
Castanet, J., Grandin, A., Abourachid, A., de Ricqlès, A. 1996. Expression de la dynamique
de croissance dans la structure de l'os periostique chez *Anas platyrhynchos*. [Expression of
growth dynamics in the structure of the periosteal bone in the mallard, *Anas*
*platyrhynchos*.] *C. R. Acad. Sci.* 319, 301–308.
Chapelle K.E.J., Benson R.B.J., Stiegler J., Otero A., Zhao Q., Choiniere J.N. 2020. A
quantitative method for inferring locomotory shifts in amniotes during ontogeny, its
application to dinosaurs and its bearing on the evolution of posture. *Palaeontology* 63(2),
229-242.

- Chinsamy, A. 1995. Ontogenetic changes in the bone histology of the Late Jurassic
ornithopod *Dryosaurus lettowvorbecki*. *Journal of Vertebrate Palaeontology*, 15(3): 96-
104.
- Chinsamy A. 1997. Assessing the biology of fossil vertebrates through bone histology
*Palaeont. Afr.*, 33, 29-35.
- Chinsamy-Turan, A. 2005. *The Microstructure of Dinosaur Bone*. Johns Hopkins University
Press, Baltimore.
- Colbert MW, Rowe T. 2008. Ontogenetic Sequence Analysis: Using parsimony to
characterize developmental sequences and sequence polymorphism. *J Exp Zool B Mol*
*Dev Evol* 310(5):398–416.
- Cullen T.M., Evans D.C., Ryan M.J., Currie P.J., Kobayashi Y., 2014. Osteohistological
variation in growth marks and osteocyte lacunar density in a theropod dinosaur
(Coelurosauria: Ornithomimidae). *BMC Evolutionary Biology*, 14:231.
- Currey, J. D. 2003. The many adaptations of bone. *Journal of Biomechanics* 36, 1487–1495.
- De Margerie, E. 2002. Laminar bone as an adaptation to torsional loads in flapping flight.
*Journal of Anatomy* 201, 521–526.
- De Margerie, E., Sanchez, S., Cubo, J. & Castanet, J. 2005. Torsional resistance as a principal
component of the structural design of long bones: comparative multivariate evidence in
birds. *The Anatomical Record* 282, 49–66.
- De Ricqlès A. 1980. Tissue structures of dinosaur bone, functional significance and possible
relation to dinosaur physiology. In: Thomas RDK, Olson EC, editors. *A Cold Look at the*
*Warm-Blooded Dinosaurs*. New York: American Association for the Advancement of
Science; pp. 103–139.
- De Ricqlès A., Castanet, J. & Francillon-Vieillot, H. 2004. The ‘message’ of bone tissue in
paleoherpetology. *Italian Journal of Zoology* 71, 3–12.
- Dial, T.R., Carrier, D.R. 2012. Precocial hindlimbs and altricial forelimbs: partitioning
ontogenetic strategies in mallards (*Anas platyrhynchos*). *J. Exp. Biol.* 215, 3703–3710.
- Dial TR, Heers AM, Tobalske BW. 2012. Ontogeny of aerodynamics in mallards:
comparative performance and developmental implications. *J Exp Biol.* 215, 3693-3702.
- Dilkes D.W. 2001. An ontogenetic perspective on locomotion in the Late Cretaceous dinosaur
*Maiasaura peeblesorum* (Ornithischia: Hadrosauridae). *Canadian Journal of Earth*
*Sciences* 38(8), 1205-1227.
- Erickson, GM, 2014. On Dinosaur Growth. *Annual Review of Earth and Planetary*
*Sciences*, 42 (1), 675-697.
- Formann AK. 1984. Die Latent-Class-Analyse: Einführung in die Theorie und Anwendung.
Beltz, Weinheim.
- Frongia GN, Muzzeddu M, Mereu P, Leoni G, Berlinguer F, Zedda M, Farina V, Satta V, Di
Stefano M, Naitana S. 2018. Structural features of cross-sectional wing bones in the griffon
vulture (*Gyps fulvus*) as a prediction of flight style. *Journal of Morphology*, 279 (12),
1753–1763.
- Garn SM, Rohmann CG, Blumenthal T (1966) Ossification sequence polymorphism and
sexual dimorphism in skeletal development. *Am J Phys Anthropol* 24(1):101–115.

Griffin CT, Nesbitt SJ, 2016a. The histology and femoral ontogeny of the Middle Triassic
(?late Anisian) dinosauriform *Asilisaurus kongwe* and implications for the growth of early
dinosaurs. *J Vertebr Paleontol* 36 (3), e1111224.
- Griffin CT, Nesbitt SJ, 2016b. Anomalously high variation in growth is ancestral for
dinosaurs but lost in birds. *PNAS* 113, 14757–14762.
- Griffin, CT. 2018. Developmental patterns and variation among early theropods. *Journal of*
*Anatomy*, 232(4), 604-640.
- Griffin CT, Stocker MR, Colleary C, Stefanic CM, Lessner EJ, Riegler M, Formoso K,
Koeller K, Nesbitt SJ. 2021. Assessing ontogenetic maturity in extinct saurian reptiles. *Biol*
*Rev* 96, 470-525.
- Gudmundsson J, Laube P., Wolle T. 2012. Computational movement analysis. In W. Kresse
and D. Danko, editors, *Springer Handbook of Geographic Information*, pp. 725-741.
Dordrecht: Springer, 2012.
- Halliday TR, Verrell PA. 1988. Body size and age in amphibians and reptiles. *J. Herpetol.* 22,
253–265.
- Haug C., Haug J.T. 2016. Developmental Paleontology and Paleo-Evo-Devo. In Kliman R.M.
ed. *Encyclopedia of Evolutionary Biology*, pp: 420-429. Academic Press.
- Heers A.M., Dial K.P, 2012. From extant to extinct: locomotor ontogeny and the evolution of
avian flight. *Trends in Ecology & Evolution*, 27(5), 296-305.
- Hone DWE, Farke AA, Wedel MJ. 2016. Ontogeny and the fossil record: what, if anything, is
an adult dinosaur? *Biol. Lett.* 12: 20150947.
- Horner, J. R. & Padian, K. 2004. Age and growth dynamics of *Tyrannosaurus rex*.
*Proceedings of the Royal Society of London B* 271, 1875–1880.
- Horner, J. R., de Ricqlès, A. & Padian, K. 1999. Variation in dinosaur skeletochronology
indicators: implications for age assessment and physiology. *Paleobiology* 25, 295–304.
- Horner, J. R., de Ricqlès, A. & Padian, K. 2000. Long bone histology of the hadrosaurid
dinosaur *Maiasaura peeblesorum*: growth dynamics and physiology based on an
ontogenetic series of skeletal elements. *Journal of Vertebrate Paleontology* 20, 115–129.
- Horner, J. R., Padian, K. & De Ricqlès, A. 2001. Comparative osteohistology of some
embryonic and neonatal archosaurs: implications for variable life histories among
dinosaurs. *Paleobiology* 27, 39–58.
- Inkscape Project, 2020. *Inkscape*, Available at: <https://inkscape.org>.
- Johnson R, 1977. Size independent criteria for estimating relative age and the relationships
among growth parameters in a group of fossil reptiles (Reptilia: Ichthyosauria). *Canadian*
*Journal of Earth Sciences*, 14(8), 1916-1924.
- Khabazi A. 2017. Anatomie comparée du développement du système ostéo-musculaire des
oiseaux. Implications des contraintes fonctionnelles sur la croissance. Unpublished PhD
Thesis. (Available on request from Dr A. Abourachid).
- Klein, N., Sander, M. 2008. Ontogenetic stages in the long bone histology of sauropod
dinosaurs. *Paleobiology* 34, 248–264.
- Lee, A., Simons, E. 2015. Wing bone laminarity is not an adaptation for torsional loads in
bats. *PeerJ* 3, e823.
- Lee A. H., Huttenlocker A. K., Padian K, Woodward H. N. 2013. Analysis of growth rates.
In: Padian, K., Lamm, E-T. (eds). *Bone Histology of Fossil Tetrapods: Advancing*

*Methods, Analysis, and Interpretation*. University of California Press, California. Pp 217-
252.
MacKay D.J.C. 2003. *Information Theory, Inference, and Learning Algorithms*. Cambridge
*University Press*.
Maxwell E.E. 2008. Ossification sequence of the avian order Anseriformes, with comparison
to other precocial birds. *J Morphol*, 269:1095–1113.
Maxwell E.E., Harrison L.B., Larsson H.C.E. 2010. Assessing the phylogenetic utility of
sequence heterochrony: evolution of avian ossification sequences as a case study. *Zoology*
113, 57–66.
Mitgutsch C., Wimmer C., Sánchez-Villagra M.R., Hahnloser R., Schneider R.A. 2011.
Timing of ossification in duck, quail, and zebra finch: intraspecific variation,
heterochronies, and life history evolution. *Zool Sci*. 28(7): 491–500.
Morris JS, Carrier DR. 2016. Sexual selection on skeletal shape in Carnivora. *Evolution*,
70(4): 767-780.
Müllner, A. 2004. Breeding Ecology and Related Life-History Traits of the Hoatzin,
*Opisthocomus hoazin*, in a Primary Rainforest Habitat. PhD Thesis. Bayerischen Julius-
Maximilians-Universität, Würzburg.
Nagin, DS. 1999. Analyzing developmental trajectories: A semiparametric, group-based
approach. *Psychological Methods*, 4(2): 139-157.
Nagin, DS. 2014. Group-based trajectory modeling: An overview. *Annals of Nutrition &*
*Metabolism*, 65: 205-210.
Nielsen JD., Rosenthal JS., Sun Y., Day DM., Bevc I., Duchesne T. 2014. Group-based
criminal trajectory analysis using cross-validation criteria. *Communications in Statistics –*
*Theory and Methods*, 43(20), 4337-4356.
Nielsen JD. 2018. Package ‘crimCV’: Group-based modelling of longitudinal data.
<https://CRAN.R-project.org/package=crimCV>
Otero, A., Cuff, A.R., Allen, V. Sumner-Rooney, L., Pol, D., Hutchinson, J.R. 2019.
Ontogenetic changes in the body plan of the sauropodomorph dinosaur *Mussaurus*
*patagonicus* reveal shifts of locomotor stance during growth. *Sci Rep* **9**, 7614.
Padian, K., Lamm, E-T. 2013. *Bone Histology of Fossil Tetrapods: Advancing Methods,*
*Analysis, and Interpretation*. University of California Press, California.
Parsons KJ, McWhinnie K, Pilakouta N, Walker L. 2020. Does phenotypic plasticity initiate
developmental bias? *Evolution & Development*, 22:56–70.
Petermann H., Mongiardino Koch N., Gauthier J.A. 2017. Osteohistology and sequence of
suture fusion reveal complex environmentally influenced growth in the teiid lizard
*Aspidoscelis tigris* — Implications for fossil squamates. *Palaeogeography,*
*Palaeoclimatology, Palaeoecology*, 475, 12–22.
Ponton, F., Montes L., Castanet J., Cubo J. 2007. Bone histological correlates of high-
frequency flapping flight and body mass in the furculae of birds: a phylogenetic
approach. *Biological Journal of the Linnean Society*, 91(4), 729–38.
Prondvai E. 2014. Comparative bone histology of rhabdodontid dinosaurs. *Palaeovertebrata*
38(2)-el. pp. 31, Online ISSN: 2274-0333.

Prondvai E. 2017. Medullary bone in fossils: function, evolution and significance in growth
curve reconstructions of extinct vertebrates. *Journal of Evolutionary Biology*, 30(3): 440-
460.
- Prondvai, E., Stein, K.H.W., Ósi A., Sander P.M. 2012. Life history of *Rhamphorhynchus*
inferred from bone histology and the diversity of pterosaurian growth strategies. PLoS
ONE 7(2): e31392.
- Prondvai, E., Godefroit, P., Adriaens, D., Hu, D-Y. 2018. Intraskkeletal histovariability,
allometric growth patterns, and their functional implications in bird-like dinosaurs. *Sci.*
*Rep.* 8, 258.
- Prondvai E, Witten PE, Abourachid A, Huysseune A, Adrians D. 2020. Extensive chondroid
bone in juvenile duck limbs hints at accelerated growth mechanism in avian
skeletogenesis. *Journal of Anatomy*, 236(3): 463-473.
- R Core Team, 2020. R: A language and environment for statistical computing. R Foundation
for Statistical Computing, Vienna, Austria. URL <http://www.R-project.org/>.
- Reisz, R. R., Evans, D. C., Sues, H. D., Scott, D. 2010. Embryonic skeletal anatomy of the
sauropodomorph dinosaur *Massospondylus* from the Lower Jurassic of South Africa. *J.*
*Vert. Paleontol.* 30, 1653–1665.
- Riesenfeld, 1978. Sexual dimorphism of skeletal robusticity in several mammalian orders.
*Acta Anat.*, 102: 392–398.
- Robling AG, Castillo AB, Turner CH. 2006. Biomechanical and molecular regulation of bone
remodeling. *Annu Rev Biomed Eng.* 8:455-498.
- Roitberg, E.S., Smirina, E.M. 2006. Age, body size and growth of *Lacerta agilis boemica* and
*L. strigata*: a comparative study of two closely related lizard species based on
skeletochronology. *Herpetological Journal*, 16, 133-148.
- Roitberg E.S., Orlova V.F., Bulakhova N.A., Kuranova V.N., Eplanova G.V., Zinenko O.I.,
Arribas O., Kratochvíl L., Ljubisavljević K., Starikov V.P., Strijbosch H., Hofmann S.,
Leontyeva O.A., Böhme W. 2020. Variation in body size and sexual size dimorphism in
the most widely ranging lizard: testing the effects of reproductive mode and climate. *Ecol*
*Evol.* 10, 4531–4561.
- Sánchez-Villagra M.R. 2010. Developmental palaeontology in synapsids: the fossil record of
ontogeny in mammals and their closest relatives. *Proc. R. Soc. B.* 277, 1139–1147.
- Sander PM, Klein N. 2005. Developmental plasticity in the life history of a prosauropod
dinosaur. *Science* 310(5755): 1800–1802.
- Schneider, C. A., Rasband, W. S., Eliceiri, K. W. 2012. NIH Image to ImageJ: 25 years of
image analysis. *Nature Methods* 9(7): 671-675.
- Sharma, P., Clouse, R., & Wheeler, W. 2017. Hennig's semaphoront concept and the use of
ontogenetic stages in phylogenetic reconstruction. *Cladistics*, 33(1), 93-108.
- Sheil C, Greenbaum E. 2005. Reconsideration of skeletal development of *Chelydra*
*serpentina* (Reptilia: Testudinata: Chelydridae): Evidence for intraspecific variation. *J Zool*
265(3), 235–267.
- Simons, E., O'Connor, P. 2012. Bone laminarity in the avian forelimb skeleton and its
relationship to flight mode: Testing functional interpretations. *Anat. Rec.* 295, 386–396.

- Skedros, J. G., Hunt, K. J., Hughes, P. E. & Winet, H. 2003. Ontogenetic and regional morphologic variations in the turkey ulna diaphysis: implications for functional adaptation of cortical bone. *The Anatomical Record Part A* 273A, 609–629.
- Smith, A., Clarke, J. 2014. Osteological histology of the Pan-Alcidae (Aves, Charadriiformes): correlates of wing-propelled diving and flightlessness. *Anat. Rec.* 297, 188–199.
- Stamps, J. A., Mangel, M., Phillips, J. A. 1998. A new look at relationships between size at maturity and asymptotic size. *American Naturalist* 152, 470–479.
- Starck, M.J. 1993. Evolution of avian ontogenies. In *Current Ornithology*, Volume 10, (ed. Power D. M.) 275–366 (New York Plenum Press).
- Starck, M.J. 1994. Quantitative design of the skeleton in bird hatchlings: Does tissue compartmentalization limit posthatching growth rates? *J. Morph.* 222, 113–131.
- Starck, M.J., Ricklefs, R. E. 1998. Patterns of development: the altricial-precocial spectrum. In *Avian Growth and Development. Evolution within the Altricial Precocial Spectrum* (eds. Starck, M.J. & Ricklefs, R.E.), 3–30 (Oxford University Press).
- Starck, M.J., Sutter, E. 2000. Patterns of growth and heterochrony in moundbuilders (Megapodiidae) and fowl (Phasianidae). *J. Avian Biol.* 31, 527–547.
- Starck, J. M., Chinsamy, A. 2002. Bone microstructure and developmental plasticity in birds and other dinosaurs. *Journal of Morphology* 254, 232–246.
- Thewissen J.G.M., Cooper L.N., Behringer R.R. 2012. Developmental biology enriches paleontology. *Journal of Vertebrate Paleontology*, 32:6, 1223–1234.
- Toohey K. 2018. SimilarityMeasures: Trajectory Similarity Measures. <https://CRAN.R-project.org/package=SimilarityMeasures>
- Toohey K., Duckham M. 2015. Trajectory similarity measures. SIGSPATIAL Special Volume 7(1), 43–50. <https://doi.org/10.1145/2782759.2782767>
- Tumarkin-Deratzian AR, Vann DR, Dodson P 2006. Bone surface texture as an ontogenetic indicator in long bones of the Canada goose *Branta canadensis* (Anseriformes: Anatidae). *Zool J Linn Soc* 148(2):133–168.
- Unwin, D. 2005. *The Pterosaurs: From Deep Time*. New York: Pi Press. pp. 140–164.
- Varricchio, D.J. 2010. A distinct dinosaur life history? *Historical Biology*, 23:1, 91–107.
- Venables WN, Ripley BD. 2002. *Modern Applied Statistics with S*, Fourth edition. Springer, New York. ISBN 0-387-95457-0, <https://www.stats.ox.ac.uk/pub/MASS4/>.
- Wang, X., Kellner, A.W.A., Jiang, S., Cheng, X., Wang, Q., Ma, Y., Paidoula, Y., Rodrigues, T., Chen, H., Sayão, J.M., Li, N., Zhang, J., Bantim, R.A.M., Meng, X., Zhang, X., Qiu, R., Zhou, Z. 2017. Egg accumulation with 3D embryos provides insight into the life history of a pterosaur. *Science* 358, 1197–1201.
- Woodward, H., Horner, J. R. & Farlow, J. O. 2011. Osteohistological evidence for determinate growth in the American alligator. *Journal of Herpetology* 45, 339–342.
- Woodward H. N., Padian K, Lee A. H., 2013. Skeletochronology. In: Padian, K., Lamm, E-T. (eds). *Bone Histology Of Fossil Tetrapods: Advancing Methods, Analysis, and Interpretation*. University of California Press, California. pp 195–216.
- Woodward H.N., Horner J.R., Farlow J. O. 2014. Quantification of intraskeletal histovariability in *Alligator mississippiensis* and implications for vertebrate osteohistology. *PeerJ* 2, e422.

Woodward H.N., Freedman Fowler E.A., Farlow J.O., Horner J.R. 2015. *Maiasaura*, a model
organism for extinct vertebrate population biology: a large sample statistical assessment of
growth dynamics and survivorship. *Paleobiology*, 41(4), 503–527.
Woodward H.N., Tremaine K., Williams S.A., Zanno L.E., Horner J.R., Myhrvold N. 2020.
Growing up *Tyrannosaurus rex*: Osteohistology refutes the pygmy “*Nanotyrannus*” and
supports ontogenetic niche partitioning in juvenile *Tyrannosaurus*. *Sci. Adv.* 6, eaax6250.
Zhao, Q., Benton, M. J., Sullivan, C., Sander, P. M. & Xu, X. 2013. Histology and postural
change during the growth of the ceratopsian dinosaur *Psittacosaurus lujiatunensis*. *Nature*
*Communications* 4, 2079.

**Table 1. Sampled taxa, specimens and elements used in this study.** Asterisks indicate
estimated minimum ages that were based on the size and the incipient outer circumferential
layer in the elements of *Anas* sp, on the fresh body weight in hoatzin chicks, and on the
plumage maturity and flight capability in the context of known growth strategy and life style
in *Falco*, *Scolopax* and *Corvus*. Abbreviations: Cmc, carpometacarpus; Fe, femur; Hu,
humerus; Ti, tibiotarsus; Tmt, tarsometatarsus

Taxon	Specimen ID	Age (dph)	Sampled bones
Anas platyrhynchos domesticus	duck_215/4461	4	Hu; Fe; Ti; Tmt
Anas platyrhynchos domesticus	duck_230/4480	4	Hu; Fe; Ti; Tmt
Anas platyrhynchos domesticus	duck_290/4556	4	Hu; Fe; Ti; Tmt
Anas platyrhynchos domesticus	duck_227/4477	8	Hu; UL; Cmc; Fe; Ti; Tmt
Anas platyrhynchos domesticus	duck_218/4464	8	Hu; UL; Fe; Ti; Tmt
Anas platyrhynchos domesticus	duck_246/4496	8	Hu; UL; Fe; Ti; Tmt
Anas platyrhynchos domesticus	duck_241/4491	15	Hu; UL; Cmc; Fe; Ti; Tmt
Anas platyrhynchos domesticus	duck_254/4503	15	Hu; UL; Cmc; Fe; Ti; Tmt
Anas platyrhynchos domesticus	duck_255/4504	15	Hu; UL; Cmc; Fe; Ti; Tmt
Anas platyrhynchos domesticus	duck_208/4452	30	Hu; UL; Cmc; Fe; Ti; Tmt
Anas platyrhynchos domesticus	duck_221/4468	30	Hu; UL; Cmc; Fe; Ti; Tmt
Anas platyrhynchos domesticus	duck_298/4555	30	Hu; UL; Cmc; Fe; Ti; Tmt
Anas platyrhynchos domesticus	duck_236/4486	50	Hu; UL; Cmc; Fe; Ti; Tmt
Anas sp.	duck_6	≥ 90*	Fe; Ti
Opisthocomus hoatzin	hoatzin_P4	6*	Hu; UL; Cmc; Fe; Ti; Tmt
Opisthocomus hoatzin	hoatzin_P1	9*	Hu; UL; Cmc; Fe; Ti; Tmt

[revised manuscript text omitted]

to reflect the disparate locomotor development of the fore- and hind limbs by increasing
skeletal dissociation (i.e. assignment of an individual’s wing and leg bones into different
clusters) as ontogeny progresses.

**Figure 6. Summary of results of the GBTM analysis (A-D) and cluster composition by**
**K-means with three clusters (E) of the fossil paravian RPP dataset.** A, Model selection
output of CVE, AIC and BIC. B, Predicted trajectories with two groups of RPPs indicated by
different colours and the mean RPP averaged over the dataset shown with a dashed line. C
and D, RPP distribution pattern in each group overlain by their respective estimated
trajectories. E, Cluster memberships of elements assigned by K-means analysis with three
clusters. Colours indicate homologous elements as indicated in the legend. Element IDs as
given in Table S1.

**Figure 7. Clustering of the complete RPP dataset – excluding RPPs with missing points –**
**by K-means with five groups demonstrated by different graphical representations.** A,
Raw RPPs with their cluster memberships indicated by the five different colours. B, Cluster

compositions with each element identifiable within their respective groups. Elements are
sorted along the horizontal axis by the known and/or estimated ages of the animals. Question
mark on the age axis denotes the unknown status of fossil paravians.

[revised manuscript text omitted]

479x400mm (236 x 236 DPI)

Figure 5. Clustering of the duck RPP dataset by K-means with four groups demonstrated by different graphical representations. A, Raw RPPs with their cluster memberships indicated by the four different colours. B, Cluster compositions with each element identifiable within their respective groups. Elements are sorted along the horizontal axis by the age of the ducks. Note, how cluster memberships within cohorts increasingly tend to reflect the disparate locomotor development of the fore- and hind limbs by increasing skeletal dissociation (i.e. assignment of an individual's wing and leg bones into different clusters) as ontogeny progresses.

472x524mm (236 x 236 DPI)

Figure 6. Summary of results of the GBTM analysis (A-D) and cluster composition by K-means with three clusters (E) of the fossil paravian RPP dataset. A, Model selection output of CVE, AIC and BIC. B, Predicted trajectories with two groups of RPPs indicated by different colours and the mean RPP averaged over the dataset shown with a dashed line. C and D, RPP distribution pattern in each group overlain by their respective estimated trajectories. E, Cluster memberships of elements assigned by K-means analysis with three clusters. Colours indicate homologous elements as indicated in the legend. Element IDs as given in Table S1.

472x465mm (236 x 236 DPI)

Figure 7. Clustering of the complete RPP dataset – excluding RPPs with missing points – by K-means with five groups demonstrated by different graphical representations. A, Raw RPPs with their cluster memberships indicated by the five different colours. B, Cluster compositions with each element identifiable within their respective groups. Elements are sorted along the horizontal axis by the known and/or estimated ages of the animals. Question mark on the age axis denotes the unknown status of fossil paravians.

483x567mm (236 x 236 DPI)

Appendix B**ROYAL SOCIETY
OPEN SCIENCE****Radial Porosity Profiles: a new bone histological method for
comparative developmental analysis of diametric limb bone
growth**

Journal:	Royal Society Open Science
Manuscript ID	RSOS-211150
Article Type:	Research
Date Submitted by the Author:	08-Jul-2021
Complete List of Authors:	Prondvai, Edina; University of Birmingham, Earth & Environmental Sciences; MTM-ELTE Research Group for Paleontology Kocsis, Ádám; Friedrich-Alexander-Universität Erlangen-Nürnberg Department Geographie und Geowissenschaften, GeoZentrum Nordbayern, Fachgruppe Paläoumwelt Abourachid, Anick; UMR 7179 Muséum National d'Histoire Naturelle – CNRS, Département Adaptations du Vivant, Bâtiment d'Anatomie Comparée Adriaens, Dominique; Ghent University, Biology Godefroit, Pascal; Royal Belgian Institute of Natural Sciences Hu, Dongyu; Shenyang Normal University, Key Laboratory for Evolution of Past Life and Change of Past Environment Butler, Richard; University of Birmingham, School of Geography and Earth Sciences
Subject:	evolution < BIOLOGY, palaeontology < BIOLOGY
Keywords:	birds, growth and functional maturity, ontogeny, precocial–altricial development, quantitative bone histology, radial porosity profile
Subject Category:	Organismal and Evolutionary Biology

Author-supplied statements

Relevant information will appear here if provided.

Ethics

Does your article include research that required ethical approval or permits?:

This article does not present research with ethical considerations

Statement (if applicable):

CUST_IF_YES_ETHICS :No data available.

Data

It is a condition of publication that data, code and materials supporting your paper are made publicly available. Does your paper present new data?:

Yes

Statement (if applicable):

All data for editors and reviewers are available within the main MS and in supplementary material.

Conflict of interest

I/We declare we have no competing interests

Statement (if applicable):

CUST_STATE_CONFLICT :No data available.

Authors' contributions

This paper has multiple authors and our individual contributions were as below

Statement (if applicable):

EP conceived and designed the study and methodology, dissected the ducks, prepared the bone samples, photographed and evaluated the sections, took the measurements, ran the analyses, prepared the figures and wrote the manuscript; ÅTK helped to conceptualize and implement the numeric analyses, wrote the R codes, reviewed and edited final draft; AA acquired and currently curates the duck specimens used in this study, provided access to the hoatzin specimens, reviewed and edited final draft; DA provided necessary lab facilities, reviewed and edited final draft; PG and D-YH provided access to fossil paravian sections, reviewed and edited final draft; RJB reviewed and edited final draft. All authors gave final approval for publication.

**Radial Porosity Profiles: a new bone histological method for comparative developmental**
**analysis of diametric limb bone growth**

Edina Prondvai^{1,2*}, Ádám T. Kocsis³, Anick Abourachid⁴, Dominique Adriaens⁵, Pascal
Godefroit⁶, Dong-Yu Hu^{7,8}, Richard J. Butler¹

¹School of Geography, Earth & Environmental Sciences, University of Birmingham,
Edgbaston, Birmingham, UK

²MTA-MTM-ELTE Research Group for Paleontology, Budapest, Hungary

³Department of Palaeobiology, Friedrich-Alexander-University of Erlangen-Nürnberg,
Erlangen, Germany

⁴Département Adaptations du Vivant, UMR 7179 Muséum National d’Histoire Naturelle –
CNRS, Paris, France

⁵Department of Biology, Evolutionary Morphology of Vertebrates, Ghent University, Ghent,
Belgium

⁶Directorate Earth & History of Life, Royal Belgian Institute of Natural Sciences, Brussels,
Belgium

⁷Paleontological Institute of Shenyang Normal University, Key Laboratory for Evolution of
Past Life in Northeast Asia, Ministry of Land and Resources, Shenyang, China

⁸Paleontological Museum of Liaoning, Shenyang, China

*Corresponding author. E-mail: edina.prondvai@gmail.com; E.Prondvai@bham.ac.uk

Abstract

Assessing ontogenetic patterns in fossil vertebrates is crucial in palaeobiology and many other fields of vertebrate evolution. Limb bone histology is considered the most reliable tool not only for skeletal maturity assessments but also for evaluating the growth record in the ontogenetic window represented by the primary bone cortex. This is especially relevant in the context of variable ontogenetic strategies along the precocial–altricial developmental spectrum. Primary cortical vascularity is the most informative histological character for reconstructing growth dynamics due to its complex relationship with bone growth and functional maturation. Using this concept as our working hypothesis, we introduce here a new quantitative osteohistological parameter, radial porosity profile (RPP), that captures relative cortical porosity changes as trajectories in limb bones. We built a proof-of-concept RPP dataset on extant birds to which we added fossil paravian dinosaurs. We performed a set of trajectory-grouping analyses to identify potential RPP categories which were evaluated in the context of our ontogeny – developmental strategy working hypothesis. Our results support the analytical power of RPPs and reveal unexpected potential osteohistological correlates of growth and functional development of limb bones. The diverse potential applications of RPPs opens up new research directions in the evolution of locomotor ontogeny.

[revised manuscript text omitted]

strategies. We here propose a new quantitative histological method for long bones, referred to
as radial porosity profiles (RPP), that builds upon the general principles of diametric bone
growth and development in the primary cortex of terrestrial tetrapods. Our method abstracts
growth and developmental dynamics from the sampled element on the basis of relative
primary porosity measured along radial trajectories across the preserved posthatching cortex.
The course of these short trajectories, i.e. the RPPs, is not only informative of the specific
ontogenetic phase/period the studied specimen represents in its overall growth trajectory, but
may also give insight into the developmental strategy of the studied skeletal element.

[revised manuscript text omitted]

fine details of the individual's developmental phase and strategy which then can be compared
with that of other conspecifics or specimens of other taxa. Whether growth is overall slow or
fast, cyclical or continuous, isometric or allometric, precocial or altricial, porosity patterns are
influenced by the cumulative effects of all these developmental processes, and thus represent
an information-rich abstraction of the ontogenetic trajectories and potentially element-specific
functional aspects of tetrapods.

*2.2.2. Selection of the data-collecting technique*

Our ontogenetic and developmental approach is based on the osteohistological evaluation of
the RPPs in limb bone diaphyses. Limb bone shafts are the most frequently analysed elements
in palaeohistological studies because the mid-diaphyseal region usually has the highest
primary bone content providing insight into individual growth trajectories (Padian & Lamm,
2013). Long bone histology is also shaped by diverse locomotor demands (de Margerie, 2002;
Skedros *et al.* 2003; Currey 2003; de Margerie *et al.* 2005; Robling *et al.* 2006; Ponton *et al.*
2007; Simons & O'Connor. 2012; Smith & Clarke. 2014; Lee & Simons, 2015; Frongia *et al.*

[revised manuscript text omitted]
 (Fig. 2D). Thereafter, a mean RPP from the three sectors ($S = 3$) defined by a vector of four points for each bone (Fig. 2E) was calculated as:

$$RPP = \left(\sum_{n=1}^S P_{a_S}^1 \sum_{n=1}^S P_{b_S}^1 \sum_{n=1}^S P_{c_S}^1 \sum_{n=1}^S P_{d_S}^1 \right) (3).$$

The courses of these RPPs (Fig. S3) were numerically analysed by diverse trajectory modelling and clustering methods in various comparative contexts (see Section 2.2.6).

Vascular measurements were only taken in primary cortical areas. Wherever **secondary remodelling was present**, their areas were measured within the quadrants and subtracted from the sampled areas (Fig. S2B, D). In those bones where secondary remodelling affected extensive cortical regions obliterating most or all of the primary cortex in a given quadrant (most frequently Q_{a-b}), the porosity values of the affected quadrants were considered missing (NAs) (Fig. S2B, C). Only those RPPs were analysed in which a minimum of two of the four points defining the sector-averaged trajectory could be calculated at least in one sector. The extent of secondary remodelling was also considered in the interpretation of the results.

The relative thicknesses of the outer circumferential layer (OCL) and the endosteal lamellar layer (EL), rimming the medullary cavity in most bones of the paravian dinosaurs and some bones of the extant bird specimens (Fig. 1H), were quantified in relation to overall cortex thickness as the area percentage it occupies of the total cortex area in the measured sectors. Similar to secondarily remodelled areas, OCL and EL were not included in the numeric analyses in this study but were considered during the interpretation of the results.

2.2.6. Numeric analysis – a selection of methods

Three different main approaches with various settings were selected to demonstrate the diverse analytical potential in the RPPs of individual bones: 1) K-means clustering; 2) group-based trajectory modelling (GBTM; Nagin, 2014); and 3) three types of trajectory similarity measures (dynamic time warping [DTW], ‘edit distance’ [ED], and longest common subsequence [LCSS]; Toohey & Duckham, 2015). K-means clustering is a widely used method applied in a diverse range of fields for classifying multivariate observations into groups (e.g. MacKay, 2003). The similarity of trajectories included mean squared distances (MSD) that reflect the expected distance between corresponding points (**homologous** quadrants) of the RPPs. GBTM is a more specific statistical approach to analyse developmental trajectories – a progression of any phenomenon described by longitudinal data – and to identify distinct groups of individual trajectories within a population (Nagin, 1999, 2014). GBTM assigns group membership probabilities to individual trajectories and compares the analytical support for all models to find the optimal (best supported) number of groups (Nagin, 1999, 2014; Nielsen, 2018). The third approach, trajectory similarity measures, comes from the field of computational movement analysis in which trajectories are defined as a “sequence of time-stamped locations” (Gudmundsson et al. 2012) and similarity is calculated by various distance measures between trajectory pairs (Toohey & Duckham, 2015). In the DTW method, the sum of the distances at each point along the trajectories is calculated as the smallest warping path. In ED, the minimum number of edits are calculated for two trajectories to be considered equivalent. Finally, LCSS works with the translation of the trajectories in all

possible dimensions (in our case 2D) to find the maximum number of equivalent points between two trajectories (Gudmundsson et al. 2012; Toohey & Duckham, 2015; Toohey, 2018).

In all analysis types, we first ran the tests separately on the RPPs of the duck ontogenetic series (Fig. S3A, B), then on those of the extinct paravian dinosaurs (Fig. S3C, D), and finally on the complete dataset including the ducks, the extinct paravians, the juvenile hoatzins and the other adult birds (Fig. S3E, F). Using these separate analyses, we could adjust the resolution of the results to broaden the context of our interpretations and elaborate on the finer details. We also analyzed separately those datasets with and without trajectories containing missing values to see their effect on the group compositions. Furthermore, we also ran tests on centered RPPs calculated for each quadrant as the difference from the mean sector porosity within a given trajectory (Fig. S3B, D, E). In these tests, only the shape of the trajectories determines their group memberships. Finally, in the duck dataset, for which we had the most complete background information, potential predictors of group memberships were also tested by logistic regressions in the best supported RPP-clustering models.

All analyses were performed in RStudio (1.4.1) integrated development environment (RStudio, PBC) for the free programming language and statistical computing software R (R Core Team). K-means clustering was called in from the basic 'stats' function *kmeans()*, GBTM was performed using the function *crimCV()* in package 'crimCV' (Nielsen et al. 2014; Nielsen, 2018), and the three different trajectory similarity measures were run with the functions *DTW()*, *EditDist()* and *LCSS()* in package 'SimilarityMeasures' (Toohey, 2015). For the K-means and GBTM analyses, we used the available ontogenetic, age and developmental strategy data of the specimens, and the graphical distribution patterns of individual RPPs, to select the number of groups to be tested. K-means uses a priori defined number of clusters, and we tested 3–5 clusters in the different analyses. Although treating the distance values as coordinates might have a degrading effect on the signal present in the data, the multidimensional structure of points remains very similar. GBTM, on the other hand, works with a pre-set maximum number of groups, which we set to five in our analyses. Because GBTM has a stochastic component, we ran 100 iterations in each analysis to find a globally best supported model using three model selection methods (Bayesian Information Criterion [BIC], Akaike Information Criterion [AIC], cross-validation error [CVE]) with the function *crimCV()* in which CVE was developed for testing GBTM itself (Nielsen et al. 2014). The similarity/distance measures of trajectory pairs resulting from the three types of trajectory similarity analyses were organized into distance matrices and further analysed with K-means clustering set to 3–5 clusters. General and ordinal logistic regressions for testing potential predictor variables of group memberships in ducks were run with the functions *glm()* available in {stats} (R Core Team, 2020) and *porl()* in package 'MASS' (Venables & Ripley, 2002). P-values for the regression coefficients in the ordinal logistic regression were calculated as for t-values with the function *pt()*. Model selection was based on AIC values using the function *step()* available in {stats} (R Core Team, 2020). See complete datasets, detailed R scripts and results of all analyses in Supplementary Table 1 and Supplementary Information 2-6.

Results were interpreted in various contexts, such as the developmental strategies of different elements within an individual, individual variation within a cohort of a species,

ontogenetic patterns of development of homologous elements and the whole skeleton, and
broader within-clade and interspecific comparisons.

**3. Results**

**3.1. Qualitative evaluation of RPPs**

Inspection of the simple graphical representation of RPPs reveals interesting patterns
concerning the developmental trajectories of limb elements (Figs. 3, S4–S18). In this
qualitative description, we mostly focus on the ducks where the ontogenetic spectrum and the
presence of three individuals in each age group (hereafter cohort) up to 30 dph allowed more
detailed interpretations (Figs. S4–S12). Other taxa in our sample (Figs. S13–S18) are
considered in the context of the findings in ducks.

**3.1.1. Ducks**

Although sample size is obviously inadequate to properly assess intracohort variance in
RPPs, the three growth series of ducks seem to show certain ontogenetic trends in the course
of the trajectories and their variance that may be related to the disparate locomotor functions
of the fore- and hind limbs.

At 4 dph, neither the ulna, nor the carpometacarpus has developed enough posthatching
periosteal cortex to be evaluated. All the other analysed limb bones of the ducklings show
high overall porosity levels throughout the cortex (~20–70%) with erratic-looking RPPs. The
intrasectional variance of RPPs (i.e. among the three sectors) lies within the same range as the
intracohort variance among homologous bones, and no separation appears between altricial
forelimb and precocial hind limb elements either in absolute porosity levels or in the course of
RPPs (Figs. S4, S9, S10). By 8 dph, a trend appears in the hind limb elements with more
channelized RPP courses, that is, with lower intrasectional and intracohort variance in a well-
defined course and radially increasing porosity, especially in the femur and tibiotarsus. On the
other hand, wing elements show no such trend in their RPPs and continue to be largely
irregular (Figs. S5, S9, S10). At 15 dph, the general pattern stays the same as at 8 dph, except
that the humerus starts showing similar channelization, and porosity may increase in the inner
cortex of some elements due to the formation of large resorption cavities related to medullary
cavity expansion (Figs. 3, S6, S9, S10). Furthermore, overall porosity seems to be higher in
the altricial wing bones, although the precocial tarsometatarsus remains in the same porosity
range as the wing bones at this stage. By 30 dph, RPP channelization occurs in every element
with overall higher porosity levels in the altricial wing bones (~20–50%) than in the precocial
leg bones (~5–30%) (Figs. S7, S9, S10), although pooled bones of the forelimb
[revised manuscript text omitted]

(Prondvai et al. 2018). RPPs of all other specimens converge on <3% P_d levels (Fig. S17).

Interestingly, and in contrast to the general pattern seen in our extant bird sample, an EL is
present in all sampled elements of these fossil paravians, while a definite OCL is missing in
all *Eosinopteryx* bones, and in some bones of the other, more mature specimens. Since hind
limb elements were only represented by the femora in this sample of extinct paravians, no
meaningful comparison can be made between the RPPs of fore- and hind limbs. Nevertheless,
RPPs of femora seem to be neatly aligned with RPPs of the forelimb elements showing no
evident deviation at these ontogenetic stages (Figs. S17, S18).

**3.2. Numeric analyses of RPPs**

In essence, all analyses of the raw data by K-means, GBTM and Similarity Measures (DTW
and ED combined with K-means) gave similar results for each dataset, that is for the ducks,
extinct paravians, and the complete dataset (Figs. 4–7, S19–S21, Supplementary Information
4–6.). On the other hand, centred data and LCSS gave quite erratic memberships in each case
which apparently did not relate to any studied parameters (Fig. S22, S23). Therefore, only the
raw data-based results are shown here, whereas those of centred data and LCSS are
considered further only in the Discussion.

**3.2.1. Ducks**

When the duck growth series were analysed with 3, 4 and 5 groups for describing trajectory
types from 4 dph up to adulthood, the 4-group setup consistently represents the best model
across different analysis methods (Fig. S19). Numeric support is provided by GBTM
iterations in which the 4-group model shows the most frequently, as well as globally, the
lowest AIC, BIC and CVE values (Fig. 4). Group compositions are fairly consistent among
different analyses with 74% of the elements clustered the same way by all four methods
(Table S2). The most inconsistency in membership assignments across methods occurred in
the 30 dph cohort, followed by 15 dph. Group memberships of bones appear to largely reflect
the combination of the individual’s ontogenetic stage and the developmental strategy of the

element. Most importantly, 4-group models successfully show how altricial and precocial
bone development within an individual can assign an individual's elements to groups
containing specimens of less and more mature ontogenetic stages, respectively (Fig. 5). This
means that these RPP-based trajectory groups (Fig. 5A) tend to "dissociate" the animal's
skeleton along its differently developing limb modules. For instance, the precocially
developing hind limb bones of a subadult animal (50 dph) are likely grouped together with
bones of adult animals, while its altricially developing wing bones tend to be assigned to a
separate group composed mostly of bones representing earlier ontogenetic stages, such as
mid- and late juveniles (Fig. 5B; Table S2).

At this analytical resolution encompassing the entire ontogenetic series up to adulthood,
group memberships also reveal that the predictive power of precocial *vs* altricial development
in separation of element RPPs changes throughout ontogeny. Dynamics of RPP disparity can
be traced in group compositions along the age axis combined with the level of skeletal
'dissociation' of specimens, that is the number of different groups an individual's bones were
assigned to. In the analyses with four groups, skeletal dissociation is not evident at 4 dph, (i.e.
all bones within an individual tend to be clustered in a single group) but starts to appear at 8
dph where wing and leg bones of an individual tend to be separated into two groups. The level
of skeletal dissociation may even reach three groups in some specimens at 15 and 30 dph but
returns to a bipartite division at 50 dph with perfect separation of wing and leg bones by
GBTM and K-means (Fig. 5B, Table S2). Figure 5B further illustrates these dynamics in
clusters #3 and #4, which contain most elements of 15 and 30 dph, (Fig. 5B). The distribution
pattern in these clusters is a transition from the younger ages (left-hand side) being mostly
represented by precocial hind limb bones (white circles) to the older ages (right-hand side)
containing primarily altricial forelimb bones (black circles). The mid-age ranges show a more
balanced mixture of fore- and hind limb bones in the respective groups. Thus, modularity in
fore- and hind limb development results in groups spanning three ad hoc maturity degrees
(from early to late juveniles in #3 and from mid-juveniles to subadults in #4) and clustering
together elements that originate from very differently aged animals (Fig. 5B).

Even though these groupings clearly result in a mixture of possible ontogenetic staging of
the animal based on RPPs of its individual elements, there are consistent limits as to which
ontogenetic stages can be represented by the differently developing elements within the same
group. For example, no bones of early juveniles (up to 8 dph) were assigned the same group
membership as those of subadults (50 dph), and no bones of mid-juveniles (up to 15 dph)
were grouped together with any adult bones, irrespective of the bone's individual
developmental strategy.

To further unfold how this complex RPP pattern might be related to the disparate growth
and development of altricial *vs* precocial elements, we separately analyzed by GBTM and K-
means the subsets of those ontogenetic stages where we had at least 3 animals in each
category, that is for early (4–8 dph), mid- (15 dph) and late juveniles (30 dph), respectively
(Table S3). Here, we set the maximum number of groups to 2 to see whether and how
precocial *vs* altricial strategy is reflected in the RPP group compositions through ontogeny.
These higher resolution analyses showed a similar trend to that recovered in the analysis
spanning the entire ontogenetic series but provided a much clearer pattern and stronger signals
related to the developmental strategies of individual bones.

The major trend is still the progressively clearer separation of the RPPs of precocially vs
altricially developing hind vs forelimb elements as ontogeny progresses. GBTM of early
juveniles (4–8 dph) showed no separation of bones at 4 dph, whereas RPPs of differently
developing elements start to differ more evidently at 8 dph with all of the altricial wing bones
being grouped together with bones of 4 dph, while all femora and tibiotarsi of the precocial
leg bones were grouped separately (Table S3). Nevertheless, tarsometatarsi at 8 dph are
clustered with the wing bones, and the distinctness of the groups is still weak, which is also
reflected in the closeness of support for both number of groups (1 and 2) provided by all
GBTM model selection methods (Supplementary Information 4). K-means gave very similar
results, although the ulnae at 8 dph were consistently assigned to the same cluster as the
femora and tibiotarsi at 8 dph. On the other hand, in late juveniles (30 dph) both GBTM and
K-means were able to identify and separate precocially and altricially developing bones with
100% accuracy based on their RPPs. Finally, the strength of separation of RPPs of precocial
and altricial elements in mid-juveniles (15 dph) is intermediate with a less distinct group
composition than in late juveniles but higher support for the two groups than in early juveniles
by all GBTM model selection methods (Table S3, Supplementary Information 4).

Based on our working hypothesis and the observed patterns in group compositions in the
entire dataset and its cohort subsets, age and developmental strategy of bones were selected to
be tested as predictors of GBTM and K-means group memberships in the complete duck
dataset with both linear combination of and interaction between these predictor variables.
Model selection (AIC) supported the model with linear combination of predictors. Logistic
regressions in each case showed that age is an overall stronger predictor than developmental
strategy with both variables being significant predictors of group memberships ($p < 0.0001$
for age and $p < 0.001$ for developmental strategy). For further details see Supplementary
Information 4.

As for individual elements, group membership probabilities highlight the precocial
tarsometatarsus seems to be the most inconsistent concerning its development-related group
membership in all analyses. This bone usually shows a highly porous cortex as compared with
the other two precocial hind limb elements, the femur and the tibiotarsus, within the same
specimen, and hence is frequently grouped together with the altricial wing elements. Finally,
the altricial ulnae at 8 dph (the earliest age for measurable ulnar cortex) tend to group together
with their contemporary precocial femora and tibiae along with elements of mid- and late
juveniles. At 8 dph the ulnar posthatching cortex is still fairly underdeveloped which may
account for this unexpected pattern. However, the same trend in the ulnae is still observed at
15 dph, although the signal is less clear.

3.2.2. Fossil paravian dinosaurs

Different types of analyses of the five specimens of fossil paravian dinosaurs gave moderately
variable results. In GBTM, all model selection methods supported the model where RPPs of
all studied elements belong to a single group (Figs. 6, S20; Supplementary Information 5).
Nevertheless, in the 2-group setup, one group contains a single element, namely the humerus
of the late juvenile *Eosinopteryx*, while all other elements are clustered together in the other
group. The same is the case in the 2-cluster K-means and ED analyses, while DTW assigns
four of the six elements of *Eosinopteryx* in this separate group. The 3-group setting provided

more diverse group memberships with different analyses. GBTM resulted again in the
separation of the *Eosinopteryx* humerus in one group, and so did K-means and ED analyses
(Figs. 6, S20). This recurring separation of *Eosinopteryx* elements by different RPP analysis
methods is in accordance with the previously assigned juvenile status of this specimen
(Prondvai et al. 2018). However, the rest of the ‘dinobird’ elements were clustered in variable
ways. GBTM sorted three more *Eosinopteryx* elements into a separate group and all other
elements in the third group. On the other hand, K-means and ED consistently grouped
together most of the elements of *Eosinopteryx* and *Jeholornis* with half of those of *Serikornis*
across the 3-group analyses, while DTW gave a less interpretable element sorting
(Supplementary Information 5).

Group membership probabilities of RPPs are best explained when precocity ranks of
elements (a qualitative order of relative osteohistological maturity of elements within the
skeleton; Prondvai et al. 2018) are also considered in combination with the assigned
ontogenetic stage of the specimen (deduction based on overall osteohistological maturity;
Prondvai et al. 2018). This is shown by the clear separation of the humerus of *Eosinopteryx*
from all other elements in both, the 2-group and 3-group setups. Memberships of elements of
*Serikornis* also tend to reflect its immature ontogenetic status along with the assigned element
precocity ranks in its skeleton (Prondvai et al. 2018).

22 727 3.2.3. Complete dataset

When tested for 3–5 groups (Figs. 7, S21; Supplementary Information 6), model selection
methods performed in the GBTM analyses supported 5 groups as the best model to cluster
RPPs in the complete dataset. However, the four different analysis types gave much less
congruent results for the composition of the 5 groups (Fig. S21) than they did for the duck-
only dataset with 4 groups. Here, only 35% of the elements clustered identically by all four
methods, and 45% by three of the four methods. Nevertheless, this still gives a total of 80% of
the elements being grouped with a minimum of 75% consistency across different methods
(Table S4). Group memberships assigned by GBTM and K-means reflect most faithfully the
combination of the specimens’ ontogenetic stages and the elements’ developmental strategies
(Fig. 7B), while DTW and ED cluster some elements less clearly related to these attributes
(Supplementary Information 6). Overall, no phylogenetic coherence appears in any of the
RPP clustering patterns.

Within the duck subgroup in the complete dataset, only 56% of the bones were identically
clustered by all four methods, while 20% and 21% were clustered by three and two of the four
methods, respectively (Table S4). Furthermore, when cluster assignments of duck elements in
this pooled dataset with 5 groups was compared with that of the duck-only dataset with 4
groups, 73% of the elements were clustered the same way by all four methods as in the duck-
only analyses. Another 19% of the bones were grouped identically by three out of the four
methods altogether giving a $\geq 75\%$ clustering congruence for 92% of the duck elements across
these analyses.

The subgroup of juvenile hoatzin bones show less consistency revealing that only 27% of
their elements were clustered identically by all four methods, and another 61% by three of the
four (Table S4). This still means that a $\geq 75\%$ clustering congruency occurs in 88% of their
elements. Within the hoatzin subgroup, GBTM and K-means vs. DTW and ED scatter their

bones across three vs. four clusters, respectively. Hoatzin bone clusters resulting from GBTM
and K-means largely cover the same clusters as do duckling bones of 15 dph, while DTW and
ED group the hind limb elements of the oldest hoatzin chick together with duck elements of
30 dph and 50 dph. GBTM seems to best reflect the estimated ages / ontogenetic stages of
these juveniles clustering most bones of the ~6 dph and ~9 dph chicks together, and
separating them from all but the ulna and carpometacarpus of the ~16 dph chick. Even though
the bone distribution pattern in the other three analyses, spanning up to four clusters, cannot
be satisfactorily related to ontogenetic stage and/or developmental strategies, distinction of
the oldest individual from the two younger chicks is clearly detectable in these analyses as
well. Finally, all four methods place the tarsometatarsi of the two younger hoatzin chicks in
the group containing most of the elements of the youngest ducks (4 dph) emphasizing again
the highly porous nature of this limb element in hoatzins (Table S4).

With a few exceptions, bones of extant adult birds were largely grouped together with
those of extinct paravian dinosaurs by all four methods. Remarkably, only K-means separated
the elements of the not yet fully-grown kestrel from the rest of the adult birds, and grouped
them together with some hind limb elements of ≥ 30 dph ducks and a single ‘dinobird’
element, the *Eosinopteryx* humerus (Fig. 7B). DTW and ED uniformly clustered all extant
adult birds in the group containing the most mature specimens in the complete dataset,
including all bones of the extinct dinosaurs. On the other hand, GBTM picked out a single
paravian element – the carpometacarpus of *Aurornis* – for its fifth cluster, while filling up the
fourth cluster with all the other bones of adults and two hind limb elements of the 50 dph
duck (Table S4).

When the cluster memberships of the ‘dinobird’ subgroup in the complete dataset is
compared with those in the 2-group ‘dinobird’-only analyses, it appears that the recurring
separation of the *Eosinopteryx* humerus in the latter disappears from all but the K-means
analysis in the former. Due to GBTM’s strange clustering in the complete dataset, only a
single element (*Aurornis* carpometacarpus) was clustered identically by all four methods, and
81% of all elements by three of the four methods in the ‘dinobird’-only and ‘dinobird’
subgroup analyses. This gives a $\geq 75\%$ clustering congruence for 85% of the ‘dinobird’
elements across these analyses.

782 33 783 3.2.4. Performance of and congruency among different methods

Besides the comparisons of clustering congruency of elements presented above, performance
and consistency of each method across the analyses can also be compared. Missing values
affected each method differently. For instance, GBTM can only work with complete RPPs
that have all four trajectory-defining points, while the rest of the methods tolerated incomplete
RPPs with up to two missing values differently. K-means has proven to be the most sensitive
to these missing values with only 67% of the elements sorted the same way in the analyses of
the complete dataset with and without incomplete RPPs. DTW and ED performed better with
98% and 86%, respectively.

Congruency in element sorting was examined between method pairs as well. When we
removed all elements with incomplete RPPs, the duck and the complete dataset included 70
and 128 elements, respectively. In this reduced duck dataset, the highest congruency in
membership assignments was found between GBTM and K-means, and between DTW and

ED, with both method pairs resulting in 93% of the elements clustered the same way. The
lowest congruence was detected between the K-means and ED methods, although it was still
high (79%) (Table S2). However, in the complete dataset encompassing 128 elements
spanning the same ontogenetic range but broadening the included developmental strategies,
the congruency levels of GBTM with respect to all other methods dropped considerably.
GBTM still had the highest congruency with K-means but it only reached 59%, whereas the
other the methods showed 77%–86% corresponding membership assignments (Table S4).

15 805 **4. Discussion**

[revised manuscript text omitted]

**Author Contributions**

EP conceived and designed the study and methodology, dissected the ducks, prepared the
bone samples, photographed and evaluated the sections, took the measurements, ran the
analyses, prepared the figures and wrote the manuscript; ÁTK helped to conceptualize and
implement the numeric analyses, wrote the R codes, reviewed and edited final draft; AA
acquired and currently curates the duck specimens used in this study, provided access to the
hoatzin specimens, reviewed and edited final draft; DA provided necessary lab facilities,
reviewed and edited final draft; PG and D-YH provided access to fossil paravian sections,
reviewed and edited final draft; RJB reviewed and edited final draft. All authors gave final
approval for publication.

**References**

- Abourachid A., Herrel A., Decamps T., Pages F., Fabre A.-C., Van Hoorebeke L., Adriaens
D., Garcia Amado M. A., 2019. Hoatzin nestling locomotion: Acquisition of quadrupedal
limb coordination in birds. *Science Advances*, 5, eaat0787.
Bailleul AM, Scannella JB, Horner JR, Evans DC, 2016. Fusion patterns in the skulls of
modern archosaurs reveal that sutures are ambiguous maturity indicators for the
Dinosauria. *PLoS ONE* 11(2): e0147687.
Bennett C. S. 1993. Ontogeny of *Pteranodon*. *Paleobiology*, 19(1), 92-106.
Bennett, M. B. 2008. Post-hatching growth and development of the pectoral and pelvic limbs
in the black noddy, *Anous minutus*. *Comparative Biochemistry and Physiology*, Part A 150,
159–168.
Brinkman, D. 1988. Size-independent criteria for estimating relative age in *Ophiacodon* and
*Dimetrodon* (Reptilia, Pelycosauria) from the Admiral and lower Belle Plains Formations
of West-Central Texas. *Journal of Vertebrate Paleontology* 8:172–180.
Brochu, C. A. 1996. Closure of neurocentral sutures during crocodylian ontogeny:
implications for maturity assessment in fossil archosaurs. *Journal of Vertebrate*
*Paleontology* 16:49–62.
Bybee PJ, Lee AH, Lamm E-T. 2006. Sizing the Jurassic theropod dinosaur *Allosaurus*:
Assessing growth strategy and evolution of ontogenetic scaling of limbs. *Journal of*
*Morphology*, 267(3):347–359.
Carrier, D., Leon, L.R. 1990. Skeletal growth and function in the California gull (*Larus*
*californicus*). *J. Zool.* 222, 375–389.
Castanet, J., Grandin, A., Abourachid, A., de Ricqlès, A. 1996. Expression de la dynamique
de croissance dans la structure de l'os periostique chez *Anas platyrhynchos*. [Expression of
growth dynamics in the structure of the periosteal bone in the mallard, *Anas*
*platyrhynchos*.] *C. R. Acad. Sci.* 319, 301–308.
Chapelle K.E.J., Benson R.B.J., Stiegler J., Otero A., Zhao Q., Choiniere J.N. 2020. A
quantitative method for inferring locomotory shifts in amniotes during ontogeny, its
application to dinosaurs and its bearing on the evolution of posture. *Palaeontology* 63(2),
229-242.

- Chinsamy, A. 1995. Ontogenetic changes in the bone histology of the Late Jurassic
ornithopod *Dryosaurus lettowvorbecki*. *Journal of Vertebrate Palaeontology*, 15(3): 96-
104.
- Chinsamy A. 1997. Assessing the biology of fossil vertebrates through bone histology
*Palaeont. Afr.*, 33, 29-35.
- Chinsamy-Turan, A. 2005. *The Microstructure of Dinosaur Bone*. Johns Hopkins University
Press, Baltimore.
- Colbert MW, Rowe T. 2008. Ontogenetic Sequence Analysis: Using parsimony to
characterize developmental sequences and sequence polymorphism. *J Exp Zool B Mol*
*Dev Evol* 310(5):398–416.
- Cullen T.M., Evans D.C., Ryan M.J., Currie P.J., Kobayashi Y., 2014. Osteohistological
variation in growth marks and osteocyte lacunar density in a theropod dinosaur
(Coelurosauria: Ornithomimidae). *BMC Evolutionary Biology*, 14:231.
- Currey, J. D. 2003. The many adaptations of bone. *Journal of Biomechanics* 36, 1487–1495.
- De Margerie, E. 2002. Laminar bone as an adaptation to torsional loads in flapping flight.
*Journal of Anatomy* 201, 521–526.
- De Margerie, E., Sanchez, S., Cubo, J. & Castanet, J. 2005. Torsional resistance as a principal
component of the structural design of long bones: comparative multivariate evidence in
birds. *The Anatomical Record* 282, 49–66.
- De Ricqlès A. 1980. Tissue structures of dinosaur bone, functional significance and possible
relation to dinosaur physiology. In: Thomas RDK, Olson EC, editors. *A Cold Look at the*
*Warm-Blooded Dinosaurs*. New York: American Association for the Advancement of
Science; pp. 103–139.
- De Ricqlès A., Castanet, J. & Francillon-Vieillot, H. 2004. The ‘message’ of bone tissue in
paleoherpetology. *Italian Journal of Zoology* 71, 3–12.
- Dial, T.R., Carrier, D.R. 2012. Precocial hindlimbs and altricial forelimbs: partitioning
ontogenetic strategies in mallards (*Anas platyrhynchos*). *J. Exp. Biol.* 215, 3703–3710.
- Dial TR, Heers AM, Tobalske BW. 2012. Ontogeny of aerodynamics in mallards:
comparative performance and developmental implications. *J Exp Biol.* 215, 3693-3702.
- Dilkes D.W. 2001. An ontogenetic perspective on locomotion in the Late Cretaceous dinosaur
*Maiasaura peeblesorum* (Ornithischia: Hadrosauridae). *Canadian Journal of Earth*
*Sciences* 38(8), 1205-1227.
- Erickson, GM, 2014. On Dinosaur Growth. *Annual Review of Earth and Planetary*
*Sciences*, 42 (1), 675-697.
- Formann AK. 1984. Die Latent-Class-Analyse: Einführung in die Theorie und Anwendung.
Beltz, Weinheim.
- Frongia GN, Muzzeddu M, Mereu P, Leoni G, Berlinguer F, Zedda M, Farina V, Satta V, Di
Stefano M, Naitana S. 2018. Structural features of cross-sectional wing bones in the griffon
vulture (*Gyps fulvus*) as a prediction of flight style. *Journal of Morphology*, 279 (12),
1753–1763.
- Garn SM, Rohmann CG, Blumenthal T (1966) Ossification sequence polymorphism and
sexual dimorphism in skeletal development. *Am J Phys Anthropol* 24(1):101–115.

Griffin CT, Nesbitt SJ, 2016a. The histology and femoral ontogeny of the Middle Triassic
(?late Anisian) dinosauriform *Asilisaurus kongwe* and implications for the growth of early
dinosaurs. *J Vertebr Paleontol* 36 (3), e1111224.
- Griffin CT, Nesbitt SJ, 2016b. Anomalously high variation in growth is ancestral for
dinosaurs but lost in birds. *PNAS* 113, 14757–14762.
- Griffin, CT. 2018. Developmental patterns and variation among early theropods. *Journal of*
*Anatomy*, 232(4), 604-640.
- Griffin CT, Stocker MR, Colleary C, Stefanic CM, Lessner EJ, Riegler M, Formoso K,
Koeller K, Nesbitt SJ. 2021. Assessing ontogenetic maturity in extinct saurian reptiles. *Biol*
*Rev* 96, 470-525.
- Gudmundsson J, Laube P., Wolle T. 2012. Computational movement analysis. In W. Kresse
and D. Danko, editors, *Springer Handbook of Geographic Information*, pp. 725-741.
Dordrecht: Springer, 2012.
- Halliday TR, Verrell PA. 1988. Body size and age in amphibians and reptiles. *J. Herpetol.* 22,
253–265.
- Haug C., Haug J.T. 2016. Developmental Paleontology and Paleo-Evo-Devo. In Kliman R.M.
ed. *Encyclopedia of Evolutionary Biology*, pp: 420-429. Academic Press.
- Heers A.M., Dial K.P, 2012. From extant to extinct: locomotor ontogeny and the evolution of
avian flight. *Trends in Ecology & Evolution*, 27(5), 296-305.
- Hone DWE, Farke AA, Wedel MJ. 2016. Ontogeny and the fossil record: what, if anything, is
an adult dinosaur? *Biol. Lett.* 12: 20150947.
- Horner, J. R. & Padian, K. 2004. Age and growth dynamics of *Tyrannosaurus rex*.
*Proceedings of the Royal Society of London B* 271, 1875–1880.
- Horner, J. R., de Ricqlès, A. & Padian, K. 1999. Variation in dinosaur skeletochronology
indicators: implications for age assessment and physiology. *Paleobiology* 25, 295–304.
- Horner, J. R., de Ricqlès, A. & Padian, K. 2000. Long bone histology of the hadrosaurid
dinosaur *Maiasaura peeblesorum*: growth dynamics and physiology based on an
ontogenetic series of skeletal elements. *Journal of Vertebrate Paleontology* 20, 115–129.
- Horner, J. R., Padian, K. & De Ricqlès, A. 2001. Comparative osteohistology of some
embryonic and neonatal archosaurs: implications for variable life histories among
dinosaurs. *Paleobiology* 27, 39–58.
- Inkscape Project, 2020. *Inkscape*, Available at: <https://inkscape.org>.
- Johnson R, 1977. Size independent criteria for estimating relative age and the relationships
among growth parameters in a group of fossil reptiles (Reptilia: Ichthyosauria). *Canadian*
*Journal of Earth Sciences*, 14(8), 1916-1924.
- Khabazi A. 2017. Anatomie comparée du développement du système ostéo-musculaire des
oiseaux. Implications des contraintes fonctionnelles sur la croissance. Unpublished PhD
Thesis. (Available on request from Dr A. Abourachid).
- Klein, N., Sander, M. 2008. Ontogenetic stages in the long bone histology of sauropod
dinosaurs. *Paleobiology* 34, 248–264.
- Lee, A., Simons, E. 2015. Wing bone laminarity is not an adaptation for torsional loads in
bats. *PeerJ* 3, e823.
- Lee A. H., Huttenlocker A. K., Padian K, Woodward H. N. 2013. Analysis of growth rates.
In: Padian, K., Lamm, E-T. (eds). *Bone Histology of Fossil Tetrapods: Advancing*

*Methods, Analysis, and Interpretation*. University of California Press, California. Pp 217-
252.
MacKay D.J.C. 2003. *Information Theory, Inference, and Learning Algorithms*. Cambridge
*University Press*.
Maxwell E.E. 2008. Ossification sequence of the avian order Anseriformes, with comparison
to other precocial birds. *J Morphol*, 269:1095–1113.
- Maxwell E.E., Harrison L.B., Larsson H.C.E. 2010. Assessing the phylogenetic utility of
sequence heterochrony: evolution of avian ossification sequences as a case study. *Zoology*
113, 57–66.
Mitgutsch C., Wimmer C., Sánchez-Villagra M.R., Hahnloser R., Schneider R.A. 2011.
Timing of ossification in duck, quail, and zebra finch: intraspecific variation,
heterochronies, and life history evolution. *Zool Sci*. 28(7): 491–500.
- Morris JS, Carrier DR. 2016. Sexual selection on skeletal shape in Carnivora. *Evolution*,
70(4): 767-780.
Müllner, A. 2004. Breeding Ecology and Related Life-History Traits of the Hoatzin,
*Opisthocomus hoazin*, in a Primary Rainforest Habitat. PhD Thesis. Bayerischen Julius-
Maximilians-Universität, Würzburg.
Nagin, DS. 1999. Analyzing developmental trajectories: A semiparametric, group-based
approach. *Psychological Methods*, 4(2): 139-157.
- Nagin, DS. 2014. Group-based trajectory modeling: An overview. *Annals of Nutrition &*
*Metabolism*, 65: 205-210.
Nielsen JD., Rosenthal JS., Sun Y., Day DM., Bevc I., Duchesne T. 2014. Group-based
criminal trajectory analysis using cross-validation criteria. *Communications in Statistics –*
*Theory and Methods*, 43(20), 4337-4356.
Nielsen JD. 2018. Package ‘crimCV’: Group-based modelling of longitudinal data.
<https://CRAN.R-project.org/package=crimCV>
Otero, A., Cuff, A.R., Allen, V. Sumner-Rooney, L., Pol, D., Hutchinson, J.R. 2019.
Ontogenetic changes in the body plan of the sauropodomorph dinosaur *Mussaurus*
*patagonicus* reveal shifts of locomotor stance during growth. *Sci Rep* **9**, 7614.
Padian, K., Lamm, E-T. 2013. *Bone Histology of Fossil Tetrapods: Advancing Methods,*
*Analysis, and Interpretation*. University of California Press, California.
Parsons KJ, McWhinnie K, Pilakouta N, Walker L. 2020. Does phenotypic plasticity initiate
developmental bias? *Evolution & Development*, 22:56–70.
Petermann H., Mongiardino Koch N., Gauthier J.A. 2017. Osteohistology and sequence of
suture fusion reveal complex environmentally influenced growth in the teiid lizard
*Aspidoscelis tigris* — Implications for fossil squamates. *Palaeogeography,*
*Palaeoclimatology, Palaeoecology*, 475, 12–22.
Ponton, F., Montes L., Castanet J., Cubo J. 2007. Bone histological correlates of high-
frequency flapping flight and body mass in the furculae of birds: a phylogenetic
approach. *Biological Journal of the Linnean Society*, 91(4), 729–38.
Prondvai E. 2014. Comparative bone histology of rhabdodontid dinosaurs. *Palaeovertebrata*
38(2)-el. pp. 31, Online ISSN: 2274-0333.

Prondvai E. 2017. Medullary bone in fossils: function, evolution and significance in growth
curve reconstructions of extinct vertebrates. *Journal of Evolutionary Biology*, 30(3): 440-
460.
- Prondvai, E., Stein, K.H.W., Ósi A., Sander P.M. 2012. Life history of *Rhamphorhynchus*
inferred from bone histology and the diversity of pterosaurian growth strategies. PLoS
ONE 7(2): e31392.
- Prondvai, E., Godefroit, P., Adriaens, D., Hu, D-Y. 2018. Intraskkeletal histovariability,
allometric growth patterns, and their functional implications in bird-like dinosaurs. *Sci.*
*Rep.* 8, 258.
- Prondvai E, Witten PE, Abourachid A, Huysseune A, Adrians D. 2020. Extensive chondroid
bone in juvenile duck limbs hints at accelerated growth mechanism in avian
skeletogenesis. *Journal of Anatomy*, 236(3): 463-473.
- R Core Team, 2020. R: A language and environment for statistical computing. R Foundation
for Statistical Computing, Vienna, Austria. URL <http://www.R-project.org/>.
- Reisz, R. R., Evans, D. C., Sues, H. D., Scott, D. 2010. Embryonic skeletal anatomy of the
sauropodomorph dinosaur *Massospondylus* from the Lower Jurassic of South Africa. *J.*
*Vert. Paleontol.* 30, 1653–1665.
- Riesenfeld, 1978. Sexual dimorphism of skeletal robusticity in several mammalian orders.
*Acta Anat.* 102: 392–398.
- Robling AG, Castillo AB, Turner CH. 2006. Biomechanical and molecular regulation of bone
remodeling. *Annu Rev Biomed Eng.* 8:455-498.
- Roitberg, E.S., Smirina, E.M. 2006. Age, body size and growth of *Lacerta agilis boemica* and
*L. strigata*: a comparative study of two closely related lizard species based on
skeletochronology. *Herpetological Journal*, 16, 133-148.
- Roitberg E.S., Orlova V.F., Bulakhova N.A., Kuranova V.N., Eplanova G.V., Zinenko O.I.,
Arribas O., Kratochvíl L., Ljubisavljević K., Starikov V.P., Strijbosch H., Hofmann S.,
Leontyeva O.A., Böhme W. 2020. Variation in body size and sexual size dimorphism in
the most widely ranging lizard: testing the effects of reproductive mode and climate. *Ecol*
*Evol.* 10, 4531–4561.
- Sánchez-Villagra M.R. 2010. Developmental palaeontology in synapsids: the fossil record of
ontogeny in mammals and their closest relatives. *Proc. R. Soc. B.* 277, 1139–1147.
- Sander PM, Klein N. 2005. Developmental plasticity in the life history of a prosauropod
dinosaur. *Science* 310(5755): 1800–1802.
- Schneider, C. A., Rasband, W. S., Eliceiri, K. W. 2012. NIH Image to ImageJ: 25 years of
image analysis. *Nature Methods* 9(7): 671-675.
- Sharma, P., Clouse, R., & Wheeler, W. 2017. Hennig's semaphoront concept and the use of
ontogenetic stages in phylogenetic reconstruction. *Cladistics*, 33(1), 93-108.
- Sheil C, Greenbaum E. 2005. Reconsideration of skeletal development of *Chelydra*
*serpentina* (Reptilia: Testudinata: Chelydridae): Evidence for intraspecific variation. *J Zool*
265(3), 235–267.
- Simons, E., O'Connor, P. 2012. Bone laminarity in the avian forelimb skeleton and its
relationship to flight mode: Testing functional interpretations. *Anat. Rec.* 295, 386–396.

- Skedros, J. G., Hunt, K. J., Hughes, P. E. & Winet, H. 2003. Ontogenetic and regional
morphologic variations in the turkey ulna diaphysis: implications for functional adaptation
of cortical bone. *The Anatomical Record Part A* 273A, 609–629.
- Smith, A., Clarke, J. 2014. Osteological histology of the Pan-Alcidae (Aves,
Charadriiformes): correlates of wing-propelled diving and flightlessness. *Anat. Rec.* 297,
188–199.
- Stamps, J. A., Mangel, M., Phillips, J. A. 1998. A new look at relationships between size at
maturity and asymptotic size. *American Naturalist* 152, 470–479.
- Starck, M.J. 1993. Evolution of avian ontogenies. In *Current Ornithology*, Volume 10, (ed.
Power D. M.) 275–366 (New York Plenum Press).
- Starck, M.J. 1994. Quantitative design of the skeleton in bird hatchlings: Does tissue
compartmentalization limit posthatching growth rates? *J. Morph.* 222, 113–131.
- Starck, M.J., Ricklefs, R. E. 1998. Patterns of development: the altricial-precocial spectrum.
In *Avian Growth and Development. Evolution within the Altricial Precocial Spectrum* (eds.
Starck, M.J. & Ricklefs, R.E.), 3–30 (Oxford University Press).
- Starck, M.J., Sutter, E. 2000. Patterns of growth and heterochrony in moundbuilders
(Megapodiidae) and fowl (Phasianidae). *J. Avian Biol.* 31, 527–547.
- Starck, J. M., Chinsamy, A. 2002. Bone microstructure and developmental plasticity in birds
and other dinosaurs. *Journal of Morphology* 254, 232–246.
- Thewissen J.G.M., Cooper L.N., Behringer R.R. 2012. Developmental biology enriches
paleontology. *Journal of Vertebrate Paleontology*, 32:6, 1223–1234.
- Toohey K. 2018. SimilarityMeasures: Trajectory Similarity Measures. [https://CRAN.R-
project.org/package=SimilarityMeasures](https://CRAN.R-project.org/package=SimilarityMeasures)
- Toohey K., Duckham M. 2015. Trajectory similarity measures. SIGSPATIAL Special
Volume 7(1), 43–50. <https://doi.org/10.1145/2782759.2782767>
- Tumarkin-Deratzian AR, Vann DR, Dodson P 2006. Bone surface texture as an ontogenetic
indicator in long bones of the Canada goose *Branta canadensis* (Anseriformes: Anatidae).
*Zool J Linn Soc* 148(2):133–168.
- Unwin, D. 2005. *The Pterosaurs: From Deep Time*. New York: Pi Press. pp. 140–164.
- Varricchio, D.J. 2010. A distinct dinosaur life history? *Historical Biology*, 23:1, 91–107.
- Venables WN, Ripley BD. 2002. *Modern Applied Statistics with S*, Fourth edition. Springer,
New York. ISBN 0-387-95457-0, <https://www.stats.ox.ac.uk/pub/MASS4/>.
- Wang, X., Kellner, A.W.A., Jiang, S., Cheng, X., Wang, Q., Ma, Y., Paidoula, Y., Rodrigues,
1437 T., Chen, H., Sayão, J.M., Li, N., Zhang, J., Bantim, R.A.M., Meng, X., Zhang, X., Qiu,
R., Zhou, Z. 2017. Egg accumulation with 3D embryos provides insight into the life
history of a pterosaur. *Science* 358, 1197–1201.
- Woodward, H., Horner, J. R. & Farlow, J. O. 2011. Osteohistological evidence for
determinate growth in the American alligator. *Journal of Herpetology* 45, 339–342.
- Woodward H. N., Padian K, Lee A. H., 2013. Skeletochronology. In: Padian, K., Lamm, E-T.
(eds). *Bone Histology Of Fossil Tetrapods: Advancing Methods, Analysis, and
Interpretation*. University of California Press, California. pp 195–216.
- Woodward H.N., Horner J.R., Farlow J. O. 2014. Quantification of intraskeletal
histovariability in *Alligator mississippiensis* and implications for vertebrate osteohistology.
*PeerJ* 2, e422.

Woodward H.N., Freedman Fowler E.A., Farlow J.O., Horner J.R. 2015. *Maiasaura*, a model
organism for extinct vertebrate population biology: a large sample statistical assessment of
growth dynamics and survivorship. *Paleobiology*, 41(4), 503–527.
Woodward H.N., Tremaine K., Williams S.A., Zanno L.E., Horner J.R., Myhrvold N. 2020.
Growing up *Tyrannosaurus rex*: Osteohistology refutes the pygmy “*Nanotyrannus*” and
supports ontogenetic niche partitioning in juvenile *Tyrannosaurus*. *Sci. Adv.* 6, eaax6250.
Zhao, Q., Benton, M. J., Sullivan, C., Sander, P. M. & Xu, X. 2013. Histology and postural
change during the growth of the ceratopsian dinosaur *Psittacosaurus lujiatunensis*. *Nature*
*Communications* 4, 2079.

Table 1. Sampled taxa, specimens and elements used in this study. Asterisks indicate estimated minimum ages that were based on the size and the incipient outer circumferential layer in the elements of *Anas* sp, on the fresh body weight in hoatzin chicks, and on the plumage maturity and flight capability in the context of known growth strategy and life style in *Falco*, *Scolopax* and *Corvus*. Abbreviations: Cmc, carpometacarpus; Fe, femur; Hu, humerus; Ti, tibiotarsus; Tmt, tarsometatarsus

Taxon	Specimen ID	Age (dph)	Sampled bones
Anas platyrhynchos domesticus	duck_215/4461	4	Hu; Fe; Ti; Tmt
Anas platyrhynchos domesticus	duck_230/4480	4	Hu; Fe; Ti; Tmt
Anas platyrhynchos domesticus	duck_290/4556	4	Hu; Fe; Ti; Tmt
Anas platyrhynchos domesticus	duck_227/4477	8	Hu; UL; Cmc; Fe; Ti; Tmt
Anas platyrhynchos domesticus	duck_218/4464	8	Hu; UL; Fe; Ti; Tmt
Anas platyrhynchos domesticus	duck_246/4496	8	Hu; UL; Fe; Ti; Tmt
Anas platyrhynchos domesticus	duck_241/4491	15	Hu; UL; Cmc; Fe; Ti; Tmt
Anas platyrhynchos domesticus	duck_254/4503	15	Hu; UL; Cmc; Fe; Ti; Tmt
Anas platyrhynchos domesticus	duck_255/4504	15	Hu; UL; Cmc; Fe; Ti; Tmt
Anas platyrhynchos domesticus	duck_208/4452	30	Hu; UL; Cmc; Fe; Ti; Tmt
Anas platyrhynchos domesticus	duck_221/4468	30	Hu; UL; Cmc; Fe; Ti; Tmt
Anas platyrhynchos domesticus	duck_298/4555	30	Hu; UL; Cmc; Fe; Ti; Tmt
Anas platyrhynchos domesticus	duck_236/4486	50	Hu; UL; Cmc; Fe; Ti; Tmt
Anas sp.	duck_6	≥ 90*	Fe; Ti
Opisthocomus hoatzin	hoatzin_P4	6*	Hu; UL; Cmc; Fe; Ti; Tmt
Opisthocomus hoatzin	hoatzin_P1	9*	Hu; UL; Cmc; Fe; Ti; Tmt

[revised manuscript text omitted]

to reflect the disparate locomotor development of the fore- and hind limbs by increasing
skeletal dissociation (i.e. assignment of an individual’s wing and leg bones into different
clusters) as ontogeny progresses.

**Figure 6. Summary of results of the GBTM analysis (A-D) and cluster composition by**
**K-means with three clusters (E) of the fossil paravian RPP dataset.** A, Model selection
output of CVE, AIC and BIC. B, Predicted trajectories with two groups of RPPs indicated by
different colours and the mean RPP averaged over the dataset shown with a dashed line. C
and D, RPP distribution pattern in each group overlain by their respective estimated
trajectories. E, Cluster memberships of elements assigned by K-means analysis with three
clusters. Colours indicate homologous elements as indicated in the legend. Element IDs as
given in Table S1.

**Figure 7. Clustering of the complete RPP dataset – excluding RPPs with missing points –**
**by K-means with five groups demonstrated by different graphical representations.** A,
Raw RPPs with their cluster memberships indicated by the five different colours. B, Cluster

compositions with each element identifiable within their respective groups. Elements are
sorted along the horizontal axis by the known and/or estimated ages of the animals. Question
mark on the age axis denotes the unknown status of fossil paravians.

[revised manuscript text omitted]

479x400mm (236 x 236 DPI)

Figure 5. Clustering of the duck RPP dataset by K-means with four groups demonstrated by different graphical representations. A, Raw RPPs with their cluster memberships indicated by the four different colours. B, Cluster compositions with each element identifiable within their respective groups. Elements are sorted along the horizontal axis by the age of the ducks. Note, how cluster memberships within cohorts increasingly tend to reflect the disparate locomotor development of the fore- and hind limbs by increasing skeletal dissociation (i.e. assignment of an individual’s wing and leg bones into different clusters) as ontogeny progresses.

472x524mm (236 x 236 DPI)

Figure 6. Summary of results of the GBTM analysis (A-D) and cluster composition by K-means with three clusters (E) of the fossil paravian RPP dataset. A, Model selection output of CVE, AIC and BIC. B, Predicted trajectories with two groups of RPPs indicated by different colours and the mean RPP averaged over the dataset shown with a dashed line. C and D, RPP distribution pattern in each group overlain by their respective estimated trajectories. E, Cluster memberships of elements assigned by K-means analysis with three clusters. Colours indicate homologous elements as indicated in the legend. Element IDs as given in Table S1.

472x465mm (236 x 236 DPI)

Figure 7. Clustering of the complete RPP dataset – excluding RPPs with missing points – by K-means with five groups demonstrated by different graphical representations. A, Raw RPPs with their cluster memberships indicated by the five different colours. B, Cluster compositions with each element identifiable within their respective groups. Elements are sorted along the horizontal axis by the known and/or estimated ages of the animals. Question mark on the age axis denotes the unknown status of fossil paravians.

483x567mm (236 x 236 DPI)

Appendix C

1st December, 2021

Cover letter

to the revised manuscript titled '**Radial Porosity Profiles: a new bone histological method for comparative developmental analysis of diametric limb bone growth**'

by

Edina Prondvai, m T. Kocsis, Anick Abourachid, Dominique Adriaens, Pascal Godefroit,
Dong-Yu Hu & Richard J. Butler

Edina Prondvai*: edina.prontvai@gmail.com; E.Prontvai@bham.ac.uk

m T. Kocsis: adam.kocsis@fau.de

Anick Abourachid: anick.abourachid@mnhn.fr

Dominique Adriaens: dominique.adriaens@ugent.be

Pascal Godefroit: pgodefroit@naturalsciences.be

Dong-Yu Hu: hudongyu1@126.com

Richard J. Butler: r.butler.1@bham.ac.uk

*Corresponding author

To the Editors,

We would be most grateful if you would consider the revised version of the MS, '**Radial Porosity Profiles: a new bone histological method for comparative developmental analysis of diametric limb bone growth**' for publication in *Royal Society Open Science* as a Research Article.

We would like to thank the reviewers for their positive and appreciative comments on our manuscript.

In the "Response to Reviewers" file, we outlined our responses to the reviewers' major comments and we also replied to their minor comments in their respective annotated pdfs which we attached to this revision.

Following their advice, **we have changed some crucial aspects of our analyses (using raw RPPs in K-means; coupling hierarchical clustering with DTW and ED) and completely rerun all analyses (except GBTM) in all datasets.**

We also changed our predictor testing method to multinomial logistic regression which is a more realistic approach for our dataset.

In essence, these corrections have not changed the results or the conclusions, but we **redid all figures and tables and rewrote all related sections of the paper, so that it now precisely reflects these new results in every aspects and detail.**

However, using mixed-effects models for testing our predictors suggested by Reviewer #2 was not possible to our dataset and we summarized the reasons in our 'Response to Reviewers' letter.

We have added new sections into the Conclusion emphasizing the most important findings, and the advantages and conditions of using the RPP method, as suggested by Reviewer #1.

However, his suggestion of a compiling a summary figure was practically not possible. Instead, **in our revised MS, we summarized the most important findings in tables.**

Finally, **we also added some new and important sections on the meaning of skeletal dissociation and other identified ontogenetic patterns.**

We have uploaded our MS in Track Changes mode as well for the editors and reviewers to find the modifications easier in the revised version.

We hope that with the introduced corrections and amendments suggested by the Reviewers and our reply to their concerns, this work can be considered eligible for publication in *Royal Society Open Science*.

We declare that neither the current work, nor other, closely related study is being under consideration for publication in another journal or book.

Thank you very much for your time and effort in reassessing this manuscript.

Sincerely,
Edina Prondvai & co-authors

Appendix D**ROYAL SOCIETY
OPEN SCIENCE****Radial Porosity Profiles: a new bone histological method for comparative developmental analysis of diametric limb bone growth**

Journal:	Royal Society Open Science
Manuscript ID	RSOS-211150
Article Type:	Research
Date Submitted by the Author:	08-Jul-2021
Complete List of Authors:	Prondvai, Edina; University of Birmingham, Earth & Environmental Sciences; MTM-ELTE Research Group for Paleontology Kocsis, Ádám; Friedrich-Alexander-Universität Erlangen-Nürnberg Department Geographie und Geowissenschaften, GeoZentrum Nordbayern, Fachgruppe Paläoumwelt Abourachid, Anick; UMR 7179 Muséum National d'Histoire Naturelle – CNRS, Département Adaptations du Vivant, Bâtiment d'Anatomie Comparée Adriaens, Dominique; Ghent University, Biology Godefroit, Pascal; Royal Belgian Institute of Natural Sciences Hu, Dongyu; Shenyang Normal University, Key Laboratory for Evolution of Past Life and Change of Past Environment Butler, Richard; University of Birmingham, School of Geography and Earth Sciences
Subject:	evolution < BIOLOGY, palaeontology < BIOLOGY
Keywords:	birds, growth and functional maturity, ontogeny, precocial–altricial development, quantitative bone histology, radial porosity profile
Subject Category:	Organismal and Evolutionary Biology

Author-supplied statements

Relevant information will appear here if provided.

Ethics

Does your article include research that required ethical approval or permits?:

This article does not present research with ethical considerations

Statement (if applicable):

CUST_IF_YES_ETHICS :No data available.

Data

It is a condition of publication that data, code and materials supporting your paper are made publicly available. Does your paper present new data?:

Yes

Statement (if applicable):

All data for editors and reviewers are available within the main MS and in supplementary material.

Conflict of interest

I/We declare we have no competing interests

Statement (if applicable):

CUST_STATE_CONFLICT :No data available.

Authors' contributions

This paper has multiple authors and our individual contributions were as below

Statement (if applicable):

EP conceived and designed the study and methodology, dissected the ducks, prepared the bone samples, photographed and evaluated the sections, took the measurements, ran the analyses, prepared the figures and wrote the manuscript; ÅTK helped to conceptualize and implement the numeric analyses, wrote the R codes, reviewed and edited final draft; AA acquired and currently curates the duck specimens used in this study, provided access to the hoatzin specimens, reviewed and edited final draft; DA provided necessary lab facilities, reviewed and edited final draft; PG and D-YH provided access to fossil paravian sections, reviewed and edited final draft; RJB reviewed and edited final draft. All authors gave final approval for publication.

Abstract

Assessing ontogenetic patterns in fossil vertebrates is crucial in palaeobiology and many other fields of vertebrate evolution. Limb bone histology is considered the most reliable tool not only for skeletal maturity assessments but also for evaluating the growth record in the ontogenetic window represented by the primary bone cortex. This is especially relevant in the context of variable ontogenetic strategies along the precocial–altricial developmental spectrum. Primary cortical vascularity is the most informative histological character for reconstructing growth dynamics due to its complex relationship with bone growth and functional maturation. Using this concept as our working hypothesis, we introduce here a new quantitative osteohistological parameter, radial porosity profile (RPP), that captures relative cortical porosity changes as trajectories in limb bones. We built a proof-of-concept RPP dataset on extant birds to which we added fossil paravian dinosaurs. We performed a set of trajectory-grouping analyses to identify potential RPP categories which were evaluated in the context of our ontogeny – developmental strategy working hypothesis. Our results support the analytical power of RPPs and reveal unexpected potential osteohistological correlates of growth and functional development of limb bones. The diverse potential applications of RPPs opens up new research directions in the evolution of locomotor ontogeny.

[revised manuscript text omitted]

strategies. We here propose a new quantitative histological method for long bones, referred to
as radial porosity profiles (RPP), that builds upon the general principles of diametric bone
growth and development in the primary cortex of terrestrial tetrapods. Our method abstracts
growth and developmental dynamics from the sampled element on the basis of relative
primary porosity measured along radial trajectories across the preserved posthatching cortex.
The course of these short trajectories, i.e. the RPPs, is not only informative of the specific
ontogenetic phase/period the studied specimen represents in its overall growth trajectory, but
may also give insight into the developmental strategy of the studied skeletal element.

[revised manuscript text omitted]

fine details of the individual's developmental phase and strategy which then can be compared
with that of other conspecifics or specimens of other taxa. Whether growth is overall slow or
fast, cyclical or continuous, isometric or allometric, precocial or altricial, porosity patterns are
influenced by the cumulative effects of all these developmental processes, and thus represent
an information-rich abstraction of the ontogenetic trajectories and potentially element-specific
functional aspects of tetrapods.

*2.2.2. Selection of the data-collecting technique*

Our ontogenetic and developmental approach is based on the osteohistological evaluation of
the RPPs in limb bone diaphyses. Limb bone shafts are the most frequently analysed elements
in palaeohistological studies because the mid-diaphyseal region usually has the highest
primary bone content providing insight into individual growth trajectories (Padian & Lamm,
2013). Long bone histology is also shaped by diverse locomotor demands (de Margerie, 2002;
Skedros *et al.* 2003; Currey 2003; de Margerie *et al.* 2005; Robling *et al.* 2006; Ponton *et al.*
2007; Simons & O'Connor. 2012; Smith & Clarke. 2014; Lee & Simons, 2015; Frongia *et al.*
2018), and therefore is expected to reflect the functional development of the limbs, which is
intimately associated with precocial – altricial ontogenetic strategies.

The following forelimb and hind limb elements were selected for analysis in most extant
birds: humerus, ulna, carpometacarpus, femur, tibiotarsus and tarsometatarsus. The only
exceptions were some duckling specimens of the early ontogenetic cohorts (4 and 8 dph), in
which the ulna and carpometacarpus had not yet developed enough posthatching cortex yet
for a reasonable radial measurement, the adult *Anas* sp. for which only the femur and the
tibiotarsus were available for sampling, and the adult kestrel in which the tarsometatarsal was
lost during preparation. Since of the hind limb bones only the femur was sampled in the
paravian dinosaurs (Prondvai et al. 2018), we also evaluated sections of the radius and the
first phalanx of digit I. ('alula') to have enough between-element comparative data for each
fossil specimen. The non-homologous nature of elements was taken into account in the
interspecific comparisons.

Our input variable, the mid-diaphyseal post-hatching cortical porosity data, can be
collected by **transverse means**. For this demonstration, we used 2D data from undemineralized
ground sections, the most accessible and widely used technique for fossil osteohistological
investigations, to provide a common basis for direct comparisons between extant and extinct
taxa.

2.2.3. *Dissection and osteohistological ground section preparation*

[revised manuscript text omitted]

Each of the three sectors was divided up to four quadrants (Q) along the radial axis from
the inner to the outer cortex ($Q_a - Q_d$). Quadrants were delineated by dividing each radial side
of the sector into four sections of equal length and connecting the corresponding sections of
the two radial sides following the section circumference (Fig. 2B). These four quadrants
represented the sampling areas for the four points of measurement defining the RPP for each
sector as follows: 1) the area of each quadrant was measured, along with the area of each
primary vascular space (V_x) within the quadrant; 2) the porosity in each quadrant (P_x) was
calculated as

$$39 350 P_x = \text{area}(V_x) / \text{area}(Q_x) \times 100\% \quad (2);$$

3) RPP within a sector was given as a trajectory defined by the four P_x (i.e. $P_a - P_d$) values of
quadrants starting from the innermost (i.e. the oldest, Q_a) to the outermost (i.e. the youngest,
Q_d) cortical layer (Fig. 2C). Thus, this RPP trajectory, which is a vector defined here by four
points, reflects element-specific developmental dynamics within the ontogenetic window of
the specimen. First, the intrasectional variability was assessed based on the RPPs of the three

60

sectors (Fig. 2D). Thereafter, a mean RPP from the three sectors ($S = 3$) defined by a vector of four points for each bone (Fig. 2E) was calculated as:

$$RPP = \left(\sum_{n=1}^S P_{a_S} \frac{1}{S} \sum_{n=1}^S P_{b_S} \frac{1}{S} \sum_{n=1}^S P_{d_S} \right) (3).$$

The courses of these RPPs (Fig. S3) were numerically analysed by diverse trajectory modelling and clustering methods in various comparative contexts (see Section 2.2.6).

Vascular measurements were only taken in primary cortical areas. Wherever secondary remodelling was present, their areas were measured within the quadrants and subtracted from the sampled areas (Fig. S2B, D). In those bones where secondary remodelling affected extensive cortical regions obliterating most or all of the primary cortex in a given quadrant (most frequently Q_{a-b}), the porosity values of the affected quadrants were considered missing (NAs) (Fig. S2B, C). Only those RPPs were analysed in which a minimum of two of the four points defining the sector-averaged trajectory could be calculated at least in one sector. The extent of secondary remodelling was also considered in the interpretation of the results.

The relative thicknesses of the outer circumferential layer (OCL) and the endosteal lamellar layer (EL), rimming the medullary cavity in most bones of the paravian dinosaurs and some bones of the extant bird specimens (Fig. 1H), were quantified in relation to overall cortex thickness as the area percentage it occupies of the total cortex area in the measured sectors. Similar to secondarily remodelled areas, OCL and EL were not included in the numeric analyses in this study but were considered during the interpretation of the results.

2.2.6. Numeric analysis – a selection of methods

Three different main approaches with various settings were selected to demonstrate the diverse analytical potential in the RPPs of individual bones: 1) K-means clustering; 2) group-based trajectory modelling (GBTM; Nagin, 2014); and 3) three types of trajectory similarity measures (dynamic time warping [DTW], ‘edit distance’ [ED], and longest common subsequence [LCSS]; Toohey & Duckham, 2015). K-means clustering is a widely used method applied in a diverse range of fields for classifying multivariate observations into groups (e.g. MacKay, 2003). The similarity of trajectories included mean squared distances (MSD) that reflect the expected distance between corresponding points (homologous quadrants) of the RPPs. GBTM is a more specific statistical approach to analyse developmental trajectories – a progression of any phenomenon described by longitudinal data – and to identify distinct groups of individual trajectories within a population (Nagin, 1999, 2014). GBTM assigns group membership probabilities to individual trajectories and compares the analytical support for all models to find the optimal (best supported) number of groups (Nagin, 1999, 2014; Nielsen, 2018). The third approach, trajectory similarity measures, comes from the field of computational movement analysis in which trajectories are defined as a “sequence of time-stamped locations” (Gudmundsson et al. 2012) and similarity is calculated by various distance measures between trajectory pairs (Toohey & Duckham, 2015). In the DTW method, the sum of the distances at each point along the trajectories is calculated as the smallest warping path. In ED, the minimum number of edits are calculated for two trajectories to be considered equivalent. Finally, LCSS works with the translation of the trajectories in all

400 possible dimensions (in our case 2D) to find the maximum number of equivalent points
between two trajectories (Gudmundsson et al. 2012; Toohey & Duckham, 2015; Toohey,
2018).

In all analysis types, we first ran the tests separately on the RPPs of the duck ontogenetic
series (Fig. S3A, B), then on those of the extinct paravian dinosaurs (Fig. S3C, D), and finally
on the complete dataset including the ducks, the extinct paravians, the juvenile hoatzins and
the other adult birds (Fig. S3E, F). Using these separate analyses, we could adjust the
resolution of the results to broaden the context of our interpretations and elaborate on the finer
details. We also analyzed separately those datasets with and without trajectories containing
missing values to see their effect on the group compositions. Furthermore, we also ran tests on
centered RPPs calculated for each quadrant as the difference from the mean sector porosity
within a given trajectory (Fig. S3B, D, E). In these tests, only the shape of the trajectories
determines their group memberships. Finally, in the duck dataset, for which we had the most
complete background information, potential predictors of group memberships were also tested
by logistic regressions in the best supported RPP-clustering models.

All analyses were performed in RStudio (1.4.1) integrated development environment
(RStudio, PBC) for the free programming language and statistical computing software R (R
Core Team). K-means clustering was called in from the basic 'stats' function *kmeans()*,
GBTM was performed using the function *crimCV()* in package 'crimCV' (Nielsen et al. 2014;
Nielsen, 2018), and the three different trajectory similarity measures were run with the
functions *DTW()*, *EditDist()* and *LCSS()* in package 'SimilarityMeasures' (Toohey, 2015). For
the K-means and GBTM analyses, we used the available ontogenetic, age and developmental
strategy data of the specimens, and the graphical distribution patterns of individual RPPs, to
select the number of groups to be tested. K-means uses a priori defined number of clusters,
and we tested 3–5 clusters in the different analyses. Although treating the distance values as
coordinates might have a degrading effect on the signal present in the data, the
multidimensional structure of points remains very similar. GBTM, on the other hand, works
with a pre-set maximum number of groups, which we set to five in our analyses. Because
GBTM has a stochastic component, we ran 100 iterations in each analysis to find a globally
best supported model using three model selection methods (Bayesian Information Criterion
[BIC], Akaike Information Criterion [AIC], cross-validation error [CVE]) with the function
*crimCV()* in which CVE was developed for testing GBTM itself (Nielsen et al. 2014). The
similarity/distance measures of trajectory pairs resulting from the three types of trajectory
similarity analyses were organized into distance matrices and further analysed with K-means
clustering set to 3–5 clusters. General and ordinal logistic regressions for testing potential
predictor variables of group memberships in ducks were run with the functions *glm()*
available in {stats} (R Core Team, 2020) and *porl()* in package 'MASS' (Venables & Ripley,
2002). P-values for the regression coefficients in the ordinal logistic regression were
calculated as for t-values with the function *pt()*. Model selection was based on AIC values
using the function *step()* available in {stats} (R Core Team, 2020). See complete datasets,
detailed R scripts and results of all analyses in Supplementary Table 1 and Supplementary
Information 2-6.

Results were interpreted in various contexts, such as the developmental strategies of
different elements within an individual, individual variation within a cohort of a species,

ontogenetic patterns of development of homologous elements and the whole skeleton, and
broader within-clade and interspecific comparisons.

**3. Results**

**3.1. Qualitative evaluation of RPPs**

Inspection of the simple graphical representation of RPPs reveals interesting patterns
concerning the developmental trajectories of limb elements (Figs. 3, S4–S18). In this
qualitative description, we mostly focus on the ducks where the ontogenetic spectrum and the
presence of three individuals in each age group (hereafter cohort) up to 30 dph allowed more
detailed interpretations (Figs. S4–S12). Other taxa in our sample (Figs. S13–S18) are
considered in the context of the findings in ducks.

**3.1.1. Ducks**

Although sample size is obviously inadequate to properly assess intracohort variance in
RPPs, the three growth series of ducks seem to show certain ontogenetic trends in the course
of the trajectories and their variance that may be related to the disparate locomotor functions
of the fore- and hind limbs.

At 4 dph, neither the ulna, nor the carpometacarpus has developed enough posthatching
periosteal cortex to be evaluated. All the other analysed limb bones of the ducklings show
high overall porosity levels throughout the cortex (~20–70%) with erratic-looking RPPs. The
intrasectional variance of RPPs (i.e. among the three sectors) lies within the same range as the
intracohort variance among homologous bones, and no separation appears between altricial
forelimb and precocial hind limb elements either in absolute porosity levels or in the course of
RPPs (Figs. S4, S9, S10). By 8 dph, a trend appears in the hind limb elements with more
channelized RPP courses, that is, with lower intrasectional and intracohort variance in a well-
defined course and radially increasing porosity, especially in the femur and tibiotarsus. On the
other hand, wing elements show no such trend in their RPPs and continue to be largely
irregular (Figs. S5, S9, S10). At 15 dph, the general pattern stays the same as at 8 dph, except
that the humerus starts showing similar channelization, and porosity may increase in the inner
cortex of some elements due to the formation of large resorption cavities related to medullary
cavity expansion (Figs. 3, S6, S9, S10). Furthermore, overall porosity seems to be higher in
the altricial wing bones, although the precocial tarsometatarsus remains in the same porosity
range as the wing bones at this stage. By 30 dph, RPP channelization occurs in every element
with overall higher porosity levels in the altricial wing bones (~20–50%) than in the precocial
leg bones (~5–30%) (Figs. S7, S9, S10), although pooled bones of the forelimb
[revised manuscript text omitted]

(Prondvai et al. 2018). RPPs of all other specimens converge on <3% P_d levels (Fig. S17).

Interestingly, and in contrast to the general pattern seen in our extant bird sample, an EL is
present in all sampled elements of these fossil paravians, while a definite OCL is missing in
all *Eosinopteryx* bones, and in some bones of the other, more mature specimens. Since hind
limb elements were only represented by the femora in this sample of extinct paravians, no
meaningful comparison can be made between the RPPs of fore- and hind limbs. Nevertheless,
RPPs of femora seem to be neatly aligned with RPPs of the forelimb elements showing no
evident deviation at these ontogenetic stages (Figs. S17, S18).

**3.2. Numeric analyses of RPPs**

In essence, all analyses of the raw data by K-means, GBTM and Similarity Measures (DTW
and ED combined with K-means) gave similar results for each dataset, that is for the ducks,
extinct paravians, and the complete dataset (Figs. 4–7, S19–S21, Supplementary Information
4–6.). On the other hand, centred data and LCSS gave quite erratic memberships in each case
which apparently did not relate to any studied parameters (Fig. S22, S23). Therefore, only the
raw data-based results are shown here, whereas those of centred data and LCSS are
considered further only in the Discussion.

**3.2.1. Ducks**

When the duck growth series were analysed with 3, 4 and 5 groups for describing trajectory
types from 4 dph up to adulthood, the 4-group setup consistently represents the best model
across different analysis methods (Fig. S19). Numeric support is provided by GBTM
iterations in which the 4-group model shows the most frequently, as well as globally, the
lowest AIC, BIC and CVE values (Fig. 4). Group compositions are fairly consistent among
different analyses with 74% of the elements clustered the same way by all four methods
(Table S2). The most inconsistency in membership assignments across methods occurred in
the 30 dph cohort, followed by 15 dph. Group memberships of bones appear to largely reflect
the combination of the individual’s ontogenetic stage and the developmental strategy of the

element. Most importantly, 4-group models successfully show how altricial and precocial
bone development within an individual can assign an individual's elements to groups
containing specimens of less and more mature ontogenetic stages, respectively (Fig. 5). This
means that these RPP-based trajectory groups (Fig. 5A) tend to “dissociate” the animal's
skeleton along its differently developing limb modules. For instance, the precocially
developing hind limb bones of a subadult animal (50 dph) are likely grouped together with
bones of adult animals, while its altricially developing wing bones tend to be assigned to a
separate group composed mostly of bones representing earlier ontogenetic stages, such as
mid- and late juveniles (Fig. 5B; Table S2).

At this analytical resolution encompassing the entire ontogenetic series up to adulthood,
group memberships also reveal that the predictive power of precocial *vs* altricial development
in separation of element RPPs changes throughout ontogeny. Dynamics of RPP disparity can
be traced in group compositions along the age axis combined with the level of skeletal
‘dissociation’ of specimens, that is the number of different groups an individual's bones were
assigned to. In the analyses with four groups, skeletal dissociation is not evident at 4 dph, (i.e.
all bones within an individual tend to be clustered in a single group) but starts to appear at 8
dph where wing and leg bones of an individual tend to be separated into two groups. The level
of skeletal dissociation may even reach three groups in some specimens at 15 and 30 dph but
returns to a bipartite division at 50 dph with perfect separation of wing and leg bones by
GBTM and K-means (Fig. 5B, Table S2). Figure 5B further illustrates these dynamics in
clusters #3 and #4, which contain most elements of 15 and 30 dph, (Fig. 5B). The distribution
pattern in these clusters is a transition from the younger ages (left-hand side) being mostly
represented by precocial hind limb bones (white circles) to the older ages (right-hand side)
containing primarily altricial forelimb bones (black circles). The mid-age ranges show a more
balanced mixture of fore- and hind limb bones in the respective groups. Thus, modularity in
fore- and hind limb development results in groups spanning three ad hoc maturity degrees
(from early to late juveniles in #3 and from mid-juveniles to subadults in #4) and clustering
together elements that originate from very differently aged animals (Fig. 5B).

Even though these groupings clearly result in a mixture of possible ontogenetic staging of
the animal based on RPPs of its individual elements, there are consistent limits as to which
ontogenetic stages can be represented by the differently developing elements within the same
group. For example, no bones of early juveniles (up to 8 dph) were assigned the same group
membership as those of subadults (50 dph), and no bones of mid-juveniles (up to 15 dph)
were grouped together with any adult bones, irrespective of the bone's individual
developmental strategy.

To further unfold how this complex RPP pattern might be related to the disparate growth
and development of altricial *vs* precocial elements, we separately analyzed by GBTM and K-
means the subsets of those ontogenetic stages where we had at least 3 animals in each
category, that is for early (4–8 dph), mid- (15 dph) and late juveniles (30 dph), respectively
(Table S3). Here, we set the maximum number of groups to 2 to see whether and how
precocial *vs* altricial strategy is reflected in the RPP group compositions through ontogeny.
These higher resolution analyses showed a similar trend to that recovered in the analysis
spanning the entire ontogenetic series but provided a much clearer pattern and stronger signals
related to the developmental strategies of individual bones.

The major trend is still the progressively clearer separation of the RPPs of precocially vs
altricially developing hind vs forelimb elements as ontogeny progresses. GBTM of early
juveniles (4–8 dph) showed no separation of bones at 4 dph, whereas RPPs of differently
developing elements start to differ more evidently at 8 dph with all of the altricial wing bones
being grouped together with bones of 4 dph, while all femora and tibiotarsi of the precocial
leg bones were grouped separately (Table S3). Nevertheless, tarsometatarsi at 8 dph are
clustered with the wing bones, and the distinctness of the groups is still weak, which is also
reflected in the closeness of support for both number of groups (1 and 2) provided by all
GBTM model selection methods (Supplementary Information 4). K-means gave very similar
results, although the ulnae at 8 dph were consistently assigned to the same cluster as the
femora and tibiotarsi at 8 dph. On the other hand, in late juveniles (30 dph) both GBTM and
K-means were able to identify and separate precocially and altricially developing bones with
100% accuracy based on their RPPs. Finally, the strength of separation of RPPs of precocial
and altricial elements in mid-juveniles (15 dph) is intermediate with a less distinct group
composition than in late juveniles but higher support for the two groups than in early juveniles
by all GBTM model selection methods (Table S3, Supplementary Information 4).

Based on our working hypothesis and the observed patterns in group compositions in the
entire dataset and its cohort subsets, age and developmental strategy of bones were selected to
be tested as predictors of GBTM and K-means group memberships in the complete duck
dataset with both linear combination of and interaction between these predictor variables.
Model selection (AIC) supported the model with linear combination of predictors. Logistic
regressions in each case showed that age is an overall stronger predictor than developmental
strategy with both variables being significant predictors of group memberships ($p < 0.0001$
for age and $p < 0.001$ for developmental strategy). For further details see Supplementary
Information 4.

As for individual elements, group membership probabilities highlight the precocial
tarsometatarsus seems to be the most inconsistent concerning its development-related group
membership in all analyses. This bone usually shows a highly porous cortex as compared with
the other two precocial hind limb elements, the femur and the tibiotarsus, within the same
specimen, and hence is frequently grouped together with the altricial wing elements. Finally,
the altricial ulnae at 8 dph (the earliest age for measurable ulnar cortex) tend to group together
with their contemporary precocial femora and tibiae along with elements of mid- and late
juveniles. At 8 dph the ulnar posthatching cortex is still fairly underdeveloped which may
account for this unexpected pattern. However, the same trend in the ulnae is still observed at
15 dph, although the signal is less clear.

3.2.2. *Fossil paravian dinosaurs*

Different types of analyses of the five specimens of fossil paravian dinosaurs gave moderately
variable results. In GBTM, all model selection methods supported the model where RPPs of
all studied elements belong to a single group (Figs. 6, S20; Supplementary Information 5).
Nevertheless, in the 2-group setup, one group contains a single element, namely the humerus
of the late juvenile *Eosinopteryx*, while all other elements are clustered together in the other
group. The same is the case in the 2-cluster K-means and ED analyses, while DTW assigns
four of the six elements of *Eosinopteryx* in this separate group. The 3-group setting provided

more diverse group memberships with different analyses. GBTM resulted again in the
separation of the *Eosinopteryx* humerus in one group, and so did K-means and ED analyses
(Figs. 6, S20). This recurring separation of *Eosinopteryx* elements by different RPP analysis
methods is in accordance with the previously assigned juvenile status of this specimen
(Prondvai et al. 2018). However, the rest of the ‘dinobird’ elements were clustered in variable
ways. GBTM sorted three more *Eosinopteryx* elements into a separate group and all other
elements in the third group. On the other hand, K-means and ED consistently grouped
together most of the elements of *Eosinopteryx* and *Jeholornis* with half of those of *Serikornis*
across the 3-group analyses, while DTW gave a less interpretable element sorting
(Supplementary Information 5).

Group membership probabilities of RPPs are best explained when precocity ranks of
elements (a qualitative order of relative osteohistological maturity of elements within the
skeleton; Prondvai et al. 2018) are also considered in combination with the assigned
ontogenetic stage of the specimen (deduction based on overall osteohistological maturity;
Prondvai et al. 2018). This is shown by the clear separation of the humerus of *Eosinopteryx*
from all other elements in both; the 2-group and 3-group setups. Memberships of elements of
*Serikornis* also tend to reflect its immature ontogenetic status along with the assigned element
precocity ranks in its skeleton (Prondvai et al. 2018).

3.2.3. Complete dataset

When tested for 3–5 groups (Figs. 7, S21; Supplementary Information 6), model selection
methods performed in the GBTM analyses supported 5 groups as the best model to cluster
RPPs in the complete dataset. However, the four different analysis types gave much less
congruent results for the composition of the 5 groups (Fig. S21) than they did for the duck-
only dataset with 4 groups. Here, only 35% of the elements clustered identically by all four
methods, and 45% by three of the four methods. Nevertheless, this still gives a total of 80% of
the elements being grouped with a minimum of 75% consistency across different methods
(Table S4). Group memberships assigned by GBTM and K-means reflect most faithfully the
combination of the specimens’ ontogenetic stages and the elements’ developmental strategies
(Fig. 7B), while DTW and ED cluster some elements less clearly related to these attributes
(Supplementary Information 6). Overall, no phylogenetic coherence appears in any of the
RPP clustering patterns.

Within the duck subgroup in the complete dataset, only 56% of the bones were identically
clustered by all four methods, while 20% and 21% were clustered by three and two of the four
methods, respectively (Table S4). Furthermore, when cluster assignments of duck elements in
this pooled dataset with 5 groups was compared with that of the duck-only dataset with 4
groups, 73% of the elements were clustered the same way by all four methods as in the duck-
only analyses. Another 19% of the bones were grouped identically by three out of the four
methods altogether giving a $\geq 75\%$ clustering congruence for 92% of the duck elements across
these analyses.

The subgroup of juvenile hoatzin bones show less consistency revealing that only 27% of
their elements were clustered identically by all four methods, and another 61% by three of the
four (Table S4). This still means that a $\geq 75\%$ clustering congruency occurs in 88% of their
elements. Within the hoatzin subgroup, GBTM and K-means vs. DTW and ED scatter their

bones across three vs. four clusters, respectively. Hoatzin bone clusters resulting from GBTM
and K-means largely cover the same clusters as do duckling bones of 15 dph, while DTW and
ED group the hind limb elements of the oldest hoatzin chick together with duck elements of
30 dph and 50 dph. GBTM seems to best reflect the estimated ages / ontogenetic stages of
these juveniles clustering most bones of the ~6 dph and ~9 dph chicks together, and
separating them from all but the ulna and carpometacarpus of the ~16 dph chick. Even though
the bone distribution pattern in the other three analyses, spanning up to four clusters, cannot
be satisfactorily related to ontogenetic stage and/or developmental strategies, distinction of
the oldest individual from the two younger chicks is clearly detectable in these analyses as
well. Finally, all four methods place the tarsometatarsi of the two younger hoatzin chicks in
the group containing most of the elements of the youngest ducks (4 dph) emphasizing again
the highly porous nature of this limb element in hoatzins (Table S4).

With a few exceptions, bones of extant adult birds were largely grouped together with
those of extinct paravian dinosaurs by all four methods. Remarkably, only K-means separated
the elements of the not yet fully-grown kestrel from the rest of the adult birds, and grouped
them together with some hind limb elements of ≥ 30 dph ducks and a single ‘dinobird’
element, the *Eosinopteryx* humerus (Fig. 7B). DTW and ED uniformly clustered all extant
adult birds in the group containing the most mature specimens in the complete dataset,
including all bones of the extinct dinosaurs. On the other hand, GBTM picked out a single
paravian element – the carpometacarpus of *Aurornis* – for its fifth cluster, while filling up the
fourth cluster with all the other bones of adults and two hind limb elements of the 50 dph
duck (Table S4).

When the cluster memberships of the ‘dinobird’ subgroup in the complete dataset is
compared with those in the 2-group ‘dinobird’-only analyses, it appears that the recurring
separation of the *Eosinopteryx* humerus in the latter disappears from all but the K-means
analysis in the former. Due to GBTM’s strange clustering in the complete dataset, only a
single element (*Aurornis* carpometacarpus) was clustered identically by all four methods, and
81% of all elements by three of the four methods in the ‘dinobird’-only and ‘dinobird’
subgroup analyses. This gives a $\geq 75\%$ clustering congruence for 85% of the ‘dinobird’
elements across these analyses.

32 782 33 783 3.2.4. Performance of and congruency among different methods

Besides the comparisons of clustering congruency of elements presented above, performance
and consistency of each method across the analyses can also be compared. Missing values
affected each method differently. For instance, GBTM can only work with complete RPPs
that have all four trajectory-defining points, while the rest of the methods tolerated incomplete
RPPs with up to two missing values differently. K-means has proven to be the most sensitive
to these missing values with only 67% of the elements sorted the same way in the analyses of
the complete dataset with and without incomplete RPPs. DTW and ED performed better with
98% and 86%, respectively.

Congruency in element sorting was examined between method pairs as well. When we
removed all elements with incomplete RPPs, the duck and the complete dataset included 70
and 128 elements, respectively. In this reduced duck dataset, the highest congruency in
membership assignments was found between GBTM and K-means, and between DTW and

ED, with both method pairs resulting in 93% of the elements clustered the same way. The
lowest congruence was detected between the K-means and ED methods, although it was still
high (79%) (Table S2). However, in the complete dataset encompassing 128 elements
spanning the same ontogenetic range but broadening the included developmental strategies,
the congruency levels of GBTM with respect to all other methods dropped considerably.
GBTM still had the highest congruency with K-means but it only reached 59%, whereas the
other the methods showed 77%–86% corresponding membership assignments (Table S4).

15 805 **4. Discussion**

[revised manuscript text omitted]

**Author Contributions**

EP conceived and designed the study and methodology, dissected the ducks, prepared the
bone samples, photographed and evaluated the sections, took the measurements, ran the
analyses, prepared the figures and wrote the manuscript; ÁTK helped to conceptualize and
implement the numeric analyses, wrote the R codes, reviewed and edited final draft; AA
acquired and currently curates the duck specimens used in this study, provided access to the
hoatzin specimens, reviewed and edited final draft; DA provided necessary lab facilities,
reviewed and edited final draft; PG and D-YH provided access to fossil paravian sections,
reviewed and edited final draft; RJB reviewed and edited final draft. All authors gave final
approval for publication.

**References**

- Abourachid A., Herrel A., Decamps T., Pages F., Fabre A.-C., Van Hoorebeke L., Adriaens
D., Garcia Amado M. A., 2019. Hoatzin nestling locomotion: Acquisition of quadrupedal
limb coordination in birds. *Science Advances*, 5, eaat0787.
Bailleul AM, Scannella JB, Horner JR, Evans DC, 2016. Fusion patterns in the skulls of
modern archosaurs reveal that sutures are ambiguous maturity indicators for the
Dinosauria. *PLoS ONE* 11(2): e0147687.
Bennett C. S. 1993. Ontogeny of *Pteranodon*. *Paleobiology*, 19(1), 92-106.
Bennett, M. B. 2008. Post-hatching growth and development of the pectoral and pelvic limbs
in the black noddy, *Anous minutus*. *Comparative Biochemistry and Physiology*, Part A 150,
159–168.
Brinkman, D. 1988. Size-independent criteria for estimating relative age in *Ophiacodon* and
*Dimetrodon* (Reptilia, Pelycosauria) from the Admiral and lower Belle Plains Formations
of West-Central Texas. *Journal of Vertebrate Paleontology* 8:172–180.
Brochu, C. A. 1996. Closure of neurocentral sutures during crocodylian ontogeny:
implications for maturity assessment in fossil archosaurs. *Journal of Vertebrate*
*Paleontology* 16:49–62.
Bybee PJ, Lee AH, Lamm E-T. 2006. Sizing the Jurassic theropod dinosaur *Allosaurus*:
Assessing growth strategy and evolution of ontogenetic scaling of limbs. *Journal of*
*Morphology*, 267(3):347–359.
Carrier, D., Leon, L.R. 1990. Skeletal growth and function in the California gull (*Larus*
*californicus*). *J. Zool.* 222, 375–389.
Castanet, J., Grandin, A., Abourachid, A., de Ricqlès, A. 1996. Expression de la dynamique
de croissance dans la structure de l'os periostique chez *Anas platyrhynchos*. [Expression of
growth dynamics in the structure of the periosteal bone in the mallard, *Anas*
*platyrhynchos*.] *C. R. Acad. Sci.* 319, 301–308.
Chapelle K.E.J., Benson R.B.J., Stiegler J., Otero A., Zhao Q., Choiniere J.N. 2020. A
quantitative method for inferring locomotory shifts in amniotes during ontogeny, its
application to dinosaurs and its bearing on the evolution of posture. *Palaeontology* 63(2),
229-242.

- Chinsamy, A. 1995. Ontogenetic changes in the bone histology of the Late Jurassic
ornithopod *Dryosaurus lettowvorbecki*. *Journal of Vertebrate Palaeontology*, 15(3): 96-
104.
- Chinsamy A. 1997. Assessing the biology of fossil vertebrates through bone histology
*Palaeont. Afr.*, 33, 29-35.
- Chinsamy-Turan, A. 2005. *The Microstructure of Dinosaur Bone*. Johns Hopkins University
Press, Baltimore.
- Colbert MW, Rowe T. 2008. Ontogenetic Sequence Analysis: Using parsimony to
characterize developmental sequences and sequence polymorphism. *J Exp Zool B Mol*
*Dev Evol* 310(5):398–416.
- Cullen T.M., Evans D.C., Ryan M.J., Currie P.J., Kobayashi Y., 2014. Osteohistological
variation in growth marks and osteocyte lacunar density in a theropod dinosaur
(Coelurosauria: Ornithomimidae). *BMC Evolutionary Biology*, 14:231.
- Currey, J. D. 2003. The many adaptations of bone. *Journal of Biomechanics* 36, 1487–1495.
- De Margerie, E. 2002. Laminar bone as an adaptation to torsional loads in flapping flight.
*Journal of Anatomy* 201, 521–526.
- De Margerie, E., Sanchez, S., Cubo, J. & Castanet, J. 2005. Torsional resistance as a principal
component of the structural design of long bones: comparative multivariate evidence in
birds. *The Anatomical Record* 282, 49–66.
- De Ricqlès A. 1980. Tissue structures of dinosaur bone, functional significance and possible
relation to dinosaur physiology. In: Thomas RDK, Olson EC, editors. *A Cold Look at the*
*Warm-Blooded Dinosaurs*. New York: American Association for the Advancement of
Science; pp. 103–139.
- De Ricqlès A., Castanet, J. & Francillon-Vieillot, H. 2004. The ‘message’ of bone tissue in
paleoherpetology. *Italian Journal of Zoology* 71, 3–12.
- Dial, T.R., Carrier, D.R. 2012. Precocial hindlimbs and altricial forelimbs: partitioning
ontogenetic strategies in mallards (*Anas platyrhynchos*). *J. Exp. Biol.* 215, 3703–3710.
- Dial TR, Heers AM, Tobalske BW. 2012. Ontogeny of aerodynamics in mallards:
comparative performance and developmental implications. *J Exp Biol.* 215, 3693-3702.
- Dilkes D.W. 2001. An ontogenetic perspective on locomotion in the Late Cretaceous dinosaur
*Maiasaura peeblesorum* (Ornithischia: Hadrosauridae). *Canadian Journal of Earth*
*Sciences* 38(8), 1205-1227.
- Erickson, GM, 2014. On Dinosaur Growth. *Annual Review of Earth and Planetary*
*Sciences*, 42 (1), 675-697.
- Formann AK. 1984. Die Latent-Class-Analyse: Einführung in die Theorie und Anwendung.
Beltz, Weinheim.
- Frongia GN, Muzzeddu M, Mereu P, Leoni G, Berlinguer F, Zedda M, Farina V, Satta V, Di
Stefano M, Naitana S. 2018. Structural features of cross-sectional wing bones in the griffon
vulture (*Gyps fulvus*) as a prediction of flight style. *Journal of Morphology*, 279 (12),
1753–1763.
- Garn SM, Rohmann CG, Blumenthal T (1966) Ossification sequence polymorphism and
sexual dimorphism in skeletal development. *Am J Phys Anthropol* 24(1):101–115.

Griffin CT, Nesbitt SJ, 2016a. The histology and femoral ontogeny of the Middle Triassic
(?late Anisian) dinosauriform *Asilisaurus kongwe* and implications for the growth of early
dinosaurs. *J Vertebr Paleontol* 36 (3), e1111224.
- Griffin CT, Nesbitt SJ, 2016b. Anomalously high variation in growth is ancestral for
dinosaurs but lost in birds. *PNAS* 113, 14757–14762.
- Griffin, CT. 2018. Developmental patterns and variation among early theropods. *Journal of*
*Anatomy*, 232(4), 604-640.
- Griffin CT, Stocker MR, Colleary C, Stefanic CM, Lessner EJ, Riegler M, Formoso K,
Koeller K, Nesbitt SJ. 2021. Assessing ontogenetic maturity in extinct saurian reptiles. *Biol*
*Rev* 96, 470-525.
- Gudmundsson J, Laube P., Wolle T. 2012. Computational movement analysis. In W. Kresse
and D. Danko, editors, *Springer Handbook of Geographic Information*, pp. 725-741.
Dordrecht: Springer, 2012.
- Halliday TR, Verrell PA. 1988. Body size and age in amphibians and reptiles. *J. Herpetol.* 22,
253–265.
- Haug C., Haug J.T. 2016. Developmental Paleontology and Paleo-Evo-Devo. In Kliman R.M.
ed. *Encyclopedia of Evolutionary Biology*, pp: 420-429. Academic Press.
- Heers A.M., Dial K.P, 2012. From extant to extinct: locomotor ontogeny and the evolution of
avian flight. *Trends in Ecology & Evolution*, 27(5), 296-305.
- Hone DWE, Farke AA, Wedel MJ. 2016. Ontogeny and the fossil record: what, if anything, is
an adult dinosaur? *Biol. Lett.* 12: 20150947.
- Horner, J. R. & Padian, K. 2004. Age and growth dynamics of *Tyrannosaurus rex*.
*Proceedings of the Royal Society of London B* 271, 1875–1880.
- Horner, J. R., de Ricqlès, A. & Padian, K. 1999. Variation in dinosaur skeletochronology
indicators: implications for age assessment and physiology. *Paleobiology* 25, 295–304.
- Horner, J. R., de Ricqlès, A. & Padian, K. 2000. Long bone histology of the hadrosaurid
dinosaur *Maiasaura peeblesorum*: growth dynamics and physiology based on an
ontogenetic series of skeletal elements. *Journal of Vertebrate Paleontology* 20, 115–129.
- Horner, J. R., Padian, K. & De Ricqlès, A. 2001. Comparative osteohistology of some
embryonic and neonatal archosaurs: implications for variable life histories among
dinosaurs. *Paleobiology* 27, 39–58.
- Inkscape Project, 2020. *Inkscape*, Available at: <https://inkscape.org>.
- Johnson R, 1977. Size independent criteria for estimating relative age and the relationships
among growth parameters in a group of fossil reptiles (Reptilia: Ichthyosauria). *Canadian*
*Journal of Earth Sciences*, 14(8), 1916-1924.
- Khabazi A. 2017. Anatomie comparée du développement du système ostéo-musculaire des
oiseaux. Implications des contraintes fonctionnelles sur la croissance. Unpublished PhD
Thesis. (Available on request from Dr A. Abourachid).
- Klein, N., Sander, M. 2008. Ontogenetic stages in the long bone histology of sauropod
dinosaurs. *Paleobiology* 34, 248–264.
- Lee, A., Simons, E. 2015. Wing bone laminarity is not an adaptation for torsional loads in
bats. *PeerJ* 3, e823.
- Lee A. H., Huttenlocker A. K., Padian K, Woodward H. N. 2013. Analysis of growth rates.
In: Padian, K., Lamm, E-T. (eds). *Bone Histology of Fossil Tetrapods: Advancing*

*Methods, Analysis, and Interpretation*. University of California Press, California. Pp 217-
252.
MacKay D.J.C. 2003. Information Theory, Inference, and Learning Algorithms. *Cambridge*
*University Press*.
Maxwell E.E. 2008. Ossification sequence of the avian order Anseriformes, with comparison
to other precocial birds. *J Morphol*, 269:1095–1113.
Maxwell E.E., Harrison L.B., Larsson H.C.E. 2010. Assessing the phylogenetic utility of
sequence heterochrony: evolution of avian ossification sequences as a case study. *Zoology*
113, 57–66.
Mitgutsch C., Wimmer C., Sánchez-Villagra M.R., Hahnloser R., Schneider R.A. 2011.
Timing of ossification in duck, quail, and zebra finch: intraspecific variation,
heterochronies, and life history evolution. *Zool Sci*. 28(7): 491–500.
Morris JS, Carrier DR. 2016. Sexual selection on skeletal shape in Carnivora. *Evolution*,
70(4): 767-780.
Müllner, A. 2004. Breeding Ecology and Related Life-History Traits of the Hoatzin,
*Opisthocomus hoazin*, in a Primary Rainforest Habitat. PhD Thesis. Bayerischen Julius-
Maximilians-Universität, Würzburg.
Nagin, DS. 1999. Analyzing developmental trajectories: A semiparametric, group-based
approach. *Psychological Methods*, 4(2): 139-157.
Nagin, DS. 2014. Group-based trajectory modeling: An overview. *Annals of Nutrition &*
*Metabolism*, 65: 205-210.
Nielsen JD., Rosenthal JS., Sun Y., Day DM., Bevc I., Duchesne T. 2014. Group-based
criminal trajectory analysis using cross-validation criteria. *Communications in Statistics –*
*Theory and Methods*, 43(20), 4337-4356.
Nielsen JD. 2018. Package ‘crimCV’: Group-based modelling of longitudinal data.
<https://CRAN.R-project.org/package=crimCV>
Otero, A., Cuff, A.R., Allen, V. Sumner-Rooney, L., Pol, D., Hutchinson, J.R. 2019.
Ontogenetic changes in the body plan of the sauropodomorph dinosaur *Mussaurus*
*patagonicus* reveal shifts of locomotor stance during growth. *Sci Rep* **9**, 7614.
Padian, K., Lamm, E-T. 2013. *Bone Histology of Fossil Tetrapods: Advancing Methods,*
*Analysis, and Interpretation*. University of California Press, California.
Parsons KJ, McWhinnie K, Pilakouta N, Walker L. 2020. Does phenotypic plasticity initiate
developmental bias? *Evolution & Development*, 22:56–70.
Petermann H., Mongiardino Koch N., Gauthier J.A. 2017. Osteohistology and sequence of
suture fusion reveal complex environmentally influenced growth in the teiid lizard
*Aspidoscelis tigris* — Implications for fossil squamates. *Palaeogeography,*
*Palaeoclimatology, Palaeoecology*, 475, 12–22.
Ponton, F., Montes L., Castanet J., Cubo J. 2007. Bone histological correlates of high-
frequency flapping flight and body mass in the furculae of birds: a phylogenetic
approach. *Biological Journal of the Linnean Society*, 91(4), 729–38.
Prondvai E. 2014. Comparative bone histology of rhabdodontid dinosaurs. *Palaeovertebrata*
38(2)-el. pp. 31, Online ISSN: 2274-0333.

Prondvai E. 2017. Medullary bone in fossils: function, evolution and significance in growth
curve reconstructions of extinct vertebrates. *Journal of Evolutionary Biology*, 30(3): 440-
460.
- Prondvai, E., Stein, K.H.W., Ósi A., Sander P.M. 2012. Life history of *Rhamphorhynchus*
inferred from bone histology and the diversity of pterosaurian growth strategies. PLoS
ONE 7(2): e31392.
- Prondvai, E., Godefroit, P., Adriaens, D., Hu, D-Y. 2018. Intraskkeletal histovariability,
allometric growth patterns, and their functional implications in bird-like dinosaurs. *Sci.*
*Rep.* 8, 258.
- Prondvai E, Witten PE, Abourachid A, Huysseune A, Adrians D. 2020. Extensive chondroid
bone in juvenile duck limbs hints at accelerated growth mechanism in avian
skeletogenesis. *Journal of Anatomy*, 236(3): 463-473.
- R Core Team, 2020. R: A language and environment for statistical computing. R Foundation
for Statistical Computing, Vienna, Austria. URL <http://www.R-project.org/>.
- Reisz, R. R., Evans, D. C., Sues, H. D., Scott, D. 2010. Embryonic skeletal anatomy of the
sauropodomorph dinosaur *Massospondylus* from the Lower Jurassic of South Africa. *J.*
*Vert. Paleontol.* 30, 1653–1665.
- Riesenfeld, 1978. Sexual dimorphism of skeletal robusticity in several mammalian orders.
*Acta Anat.* 102: 392–398.
- Robling AG, Castillo AB, Turner CH. 2006. Biomechanical and molecular regulation of bone
remodeling. *Annu Rev Biomed Eng.* 8:455-498.
- Roitberg, E.S., Smirina, E.M. 2006. Age, body size and growth of *Lacerta agilis boemica* and
*L. strigata*: a comparative study of two closely related lizard species based on
skeletochronology. *Herpetological Journal*, 16, 133-148.
- Roitberg E.S., Orlova V.F., Bulakhova N.A., Kuranova V.N., Eplanova G.V., Zinenko O.I.,
Arribas O., Kratochvíl L., Ljubisavljević K., Starikov V.P., Strijbosch H., Hofmann S.,
Leontyeva O.A., Böhme W. 2020. Variation in body size and sexual size dimorphism in
the most widely ranging lizard: testing the effects of reproductive mode and climate. *Ecol*
*Evol.* 10, 4531–4561.
- Sánchez-Villagra M.R. 2010. Developmental palaeontology in synapsids: the fossil record of
ontogeny in mammals and their closest relatives. *Proc. R. Soc. B.* 277, 1139–1147.
- Sander PM, Klein N. 2005. Developmental plasticity in the life history of a prosauropod
dinosaur. *Science* 310(5755): 1800–1802.
- Schneider, C. A., Rasband, W. S., Eliceiri, K. W. 2012. NIH Image to ImageJ: 25 years of
image analysis. *Nature Methods* 9(7): 671-675.
- Sharma, P., Clouse, R., & Wheeler, W. 2017. Hennig's semaphoront concept and the use of
ontogenetic stages in phylogenetic reconstruction. *Cladistics*, 33(1), 93-108.
- Sheil C, Greenbaum E. 2005. Reconsideration of skeletal development of *Chelydra*
*serpentina* (Reptilia: Testudinata: Chelydridae): Evidence for intraspecific variation. *J Zool*
265(3), 235–267.
- Simons, E., O'Connor, P. 2012. Bone laminarity in the avian forelimb skeleton and its
relationship to flight mode: Testing functional interpretations. *Anat. Rec.* 295, 386–396.

- Skedros, J. G., Hunt, K. J., Hughes, P. E. & Winet, H. 2003. Ontogenetic and regional morphologic variations in the turkey ulna diaphysis: implications for functional adaptation of cortical bone. *The Anatomical Record Part A* 273A, 609–629.
- Smith, A., Clarke, J. 2014. Osteological histology of the Pan-Alcidae (Aves, Charadriiformes): correlates of wing-propelled diving and flightlessness. *Anat. Rec.* 297, 188–199.
- Stamps, J. A., Mangel, M., Phillips, J. A. 1998. A new look at relationships between size at maturity and asymptotic size. *American Naturalist* 152, 470–479.
- Starck, M.J. 1993. Evolution of avian ontogenies. In *Current Ornithology*, Volume 10, (ed. Power D. M.) 275–366 (New York Plenum Press).
- Starck, M.J. 1994. Quantitative design of the skeleton in bird hatchlings: Does tissue compartmentalization limit posthatching growth rates? *J. Morph.* 222, 113–131.
- Starck, M.J., Ricklefs, R. E. 1998. Patterns of development: the altricial-precocial spectrum. In *Avian Growth and Development. Evolution within the Altricial Precocial Spectrum* (eds. Starck, M.J. & Ricklefs, R.E.), 3–30 (Oxford University Press).
- Starck, M.J., Sutter, E. 2000. Patterns of growth and heterochrony in moundbuilders (Megapodiidae) and fowl (Phasianidae). *J. Avian Biol.* 31, 527–547.
- Starck, J. M., Chinsamy, A. 2002. Bone microstructure and developmental plasticity in birds and other dinosaurs. *Journal of Morphology* 254, 232–246.
- Thewissen J.G.M., Cooper L.N., Behringer R.R. 2012. Developmental biology enriches paleontology. *Journal of Vertebrate Paleontology*, 32:6, 1223–1234.
- Toohey K. 2018. SimilarityMeasures: Trajectory Similarity Measures. <https://CRAN.R-project.org/package=SimilarityMeasures>
- Toohey K., Duckham M. 2015. Trajectory similarity measures. SIGSPATIAL Special Volume 7(1), 43–50. <https://doi.org/10.1145/2782759.2782767>
- Tumarkin-Deratzian AR, Vann DR, Dodson P 2006. Bone surface texture as an ontogenetic indicator in long bones of the Canada goose *Branta canadensis* (Anseriformes: Anatidae). *Zool J Linn Soc* 148(2):133–168.
- Unwin, D. 2005. *The Pterosaurs: From Deep Time*. New York: Pi Press. pp. 140–164.
- Varricchio, D.J. 2010. A distinct dinosaur life history? *Historical Biology*, 23:1, 91–107.
- Venables WN, Ripley BD. 2002. *Modern Applied Statistics with S*, Fourth edition. Springer, New York. ISBN 0-387-95457-0, <https://www.stats.ox.ac.uk/pub/MASS4/>.
- Wang, X., Kellner, A.W.A., Jiang, S., Cheng, X., Wang, Q., Ma, Y., Paidoula, Y., Rodrigues, T., Chen, H., Sayão, J.M., Li, N., Zhang, J., Bantim, R.A.M., Meng, X., Zhang, X., Qiu, R., Zhou, Z. 2017. Egg accumulation with 3D embryos provides insight into the life history of a pterosaur. *Science* 358, 1197–1201.
- Woodward, H., Horner, J. R. & Farlow, J. O. 2011. Osteohistological evidence for determinate growth in the American alligator. *Journal of Herpetology* 45, 339–342.
- Woodward H. N., Padian K, Lee A. H., 2013. Skeletochronology. In: Padian, K., Lamm, E-T. (eds). *Bone Histology Of Fossil Tetrapods: Advancing Methods, Analysis, and Interpretation*. University of California Press, California. pp 195–216.
- Woodward H.N., Horner J.R., Farlow J. O. 2014. Quantification of intraskeletal histovariability in *Alligator mississippiensis* and implications for vertebrate osteohistology. *PeerJ* 2, e422.

Woodward H.N., Freedman Fowler E.A., Farlow J.O., Horner J.R. 2015. *Maiasaura*, a model
organism for extinct vertebrate population biology: a large sample statistical assessment of
growth dynamics and survivorship. *Paleobiology*, 41(4), 503–527.
Woodward H.N., Tremaine K., Williams S.A., Zanno L.E., Horner J.R., Myhrvold N. 2020.
Growing up *Tyrannosaurus rex*: Osteohistology refutes the pygmy “*Nanotyrannus*” and
supports ontogenetic niche partitioning in juvenile *Tyrannosaurus*. *Sci. Adv.* 6, eaax6250.
Zhao, Q., Benton, M. J., Sullivan, C., Sander, P. M. & Xu, X. 2013. Histology and postural
change during the growth of the ceratopsian dinosaur *Psittacosaurus lujiatunensis*. *Nature*
*Communications* 4, 2079.

**Table 1. Sampled taxa, specimens and elements used in this study.** Asterisks indicate
estimated minimum ages that were based on the size and the incipient outer circumferential
layer in the elements of *Anas* sp, on the fresh body weight in hoatzin chicks, and on the
plumage maturity and flight capability in the context of known growth strategy and life style
in *Falco*, *Scolopax* and *Corvus*. Abbreviations: Cmc, carpometacarpus; Fe, femur; Hu,
humerus; Ti, tibiotarsus; Tmt, tarsometatarsus

Taxon	Specimen ID	Age (dph)	Sampled bones
Anas platyrhynchos domesticus	duck_215/4461	4	Hu; Fe; Ti; Tmt
Anas platyrhynchos domesticus	duck_230/4480	4	Hu; Fe; Ti; Tmt
Anas platyrhynchos domesticus	duck_290/4556	4	Hu; Fe; Ti; Tmt
Anas platyrhynchos domesticus	duck_227/4477	8	Hu; UL; Cmc; Fe; Ti; Tmt
Anas platyrhynchos domesticus	duck_218/4464	8	Hu; UL; Fe; Ti; Tmt
Anas platyrhynchos domesticus	duck_246/4496	8	Hu; UL; Fe; Ti; Tmt
Anas platyrhynchos domesticus	duck_241/4491	15	Hu; UL; Cmc; Fe; Ti; Tmt
Anas platyrhynchos domesticus	duck_254/4503	15	Hu; UL; Cmc; Fe; Ti; Tmt
Anas platyrhynchos domesticus	duck_255/4504	15	Hu; UL; Cmc; Fe; Ti; Tmt
Anas platyrhynchos domesticus	duck_208/4452	30	Hu; UL; Cmc; Fe; Ti; Tmt
Anas platyrhynchos domesticus	duck_221/4468	30	Hu; UL; Cmc; Fe; Ti; Tmt
Anas platyrhynchos domesticus	duck_298/4555	30	Hu; UL; Cmc; Fe; Ti; Tmt
Anas platyrhynchos domesticus	duck_236/4486	50	Hu; UL; Cmc; Fe; Ti; Tmt
Anas sp.	duck_6	≥ 90*	Fe; Ti
Opisthocomus hoatzin	hoatzin_P4	6*	Hu; UL; Cmc; Fe; Ti; Tmt
Opisthocomus hoatzin	hoatzin_P1	9*	Hu; UL; Cmc; Fe; Ti; Tmt

[revised manuscript text omitted]

to reflect the disparate locomotor development of the fore- and hind limbs by increasing
skeletal dissociation (i.e. assignment of an individual’s wing and leg bones into different
clusters) as ontogeny progresses.

**Figure 6. Summary of results of the GBTM analysis (A-D) and cluster composition by**
**K-means with three clusters (E) of the fossil paravian RPP dataset.** A, Model selection
output of CVE, AIC and BIC. B, Predicted trajectories with two groups of RPPs indicated by
different colours and the mean RPP averaged over the dataset shown with a dashed line. C
and D, RPP distribution pattern in each group overlain by their respective estimated
trajectories. E, Cluster memberships of elements assigned by K-means analysis with three
clusters. Colours indicate homologous elements as indicated in the legend. Element IDs as
given in Table S1.

**Figure 7. Clustering of the complete RPP dataset – excluding RPPs with missing points –**
**by K-means with five groups demonstrated by different graphical representations.** A,
Raw RPPs with their cluster memberships indicated by the five different colours. B, Cluster

compositions with each element identifiable within their respective groups. Elements are
sorted along the horizontal axis by the known and/or estimated ages of the animals. Question
mark on the age axis denotes the unknown status of fossil paravians.

[revised manuscript text omitted]

479x400mm (236 x 236 DPI)

Figure 5. Clustering of the duck RPP dataset by K-means with four groups demonstrated by different graphical representations. A, Raw RPPs with their cluster memberships indicated by the four different colours. B, Cluster compositions with each element identifiable within their respective groups. Elements are sorted along the horizontal axis by the age of the ducks. Note, how cluster memberships within cohorts increasingly tend to reflect the disparate locomotor development of the fore- and hind limbs by increasing skeletal dissociation (i.e. assignment of an individual's wing and leg bones into different clusters) as ontogeny progresses.

472x524mm (236 x 236 DPI)

Figure 6. Summary of results of the GBTM analysis (A-D) and cluster composition by K-means with three clusters (E) of the fossil paravian RPP dataset. A, Model selection output of CVE, AIC and BIC. B, Predicted trajectories with two groups of RPPs indicated by different colours and the mean RPP averaged over the dataset shown with a dashed line. C and D, RPP distribution pattern in each group overlain by their respective estimated trajectories. E, Cluster memberships of elements assigned by K-means analysis with three clusters. Colours indicate homologous elements as indicated in the legend. Element IDs as given in Table S1.

472x465mm (236 x 236 DPI)

Figure 7. Clustering of the complete RPP dataset – excluding RPPs with missing points – by K-means with five groups demonstrated by different graphical representations. A, Raw RPPs with their cluster memberships indicated by the five different colours. B, Cluster compositions with each element identifiable within their respective groups. Elements are sorted along the horizontal axis by the known and/or estimated ages of the animals. Question mark on the age axis denotes the unknown status of fossil paravians.

483x567mm (236 x 236 DPI)

Appendix E**ROYAL SOCIETY
OPEN SCIENCE****Radial Porosity Profiles: a new bone histological method for comparative developmental analysis of diametric limb bone growth**

Journal:	Royal Society Open Science
Manuscript ID	RSOS-211150
Article Type:	Research
Date Submitted by the Author:	08-Jul-2021
Complete List of Authors:	Prondvai, Edina; University of Birmingham, Earth & Environmental Sciences; MTM-ELTE Research Group for Paleontology Kocsis, Ádám; Friedrich-Alexander-Universität Erlangen-Nürnberg Department Geographie und Geowissenschaften, GeoZentrum Nordbayern, Fachgruppe Paläoumwelt Abourachid, Anick; UMR 7179 Muséum National d'Histoire Naturelle – CNRS, Département Adaptations du Vivant, Bâtiment d'Anatomie Comparée Adriaens, Dominique; Ghent University, Biology Godefroit, Pascal; Royal Belgian Institute of Natural Sciences Hu, Dongyu; Shenyang Normal University, Key Laboratory for Evolution of Past Life and Change of Past Environment Butler, Richard; University of Birmingham, School of Geography and Earth Sciences
Subject:	evolution < BIOLOGY, palaeontology < BIOLOGY
Keywords:	birds, growth and functional maturity, ontogeny, precocial–altricial development, quantitative bone histology, radial porosity profile
Subject Category:	Organismal and Evolutionary Biology

Author-supplied statements

Relevant information will appear here if provided.

Ethics

Does your article include research that required ethical approval or permits?:

This article does not present research with ethical considerations

Statement (if applicable):

CUST_IF_YES_ETHICS :No data available.

Data

It is a condition of publication that data, code and materials supporting your paper are made publicly available. Does your paper present new data?:

Yes

Statement (if applicable):

All data for editors and reviewers are available within the main MS and in supplementary material.

Conflict of interest

I/We declare we have no competing interests

Statement (if applicable):

CUST_STATE_CONFLICT :No data available.

Authors' contributions

This paper has multiple authors and our individual contributions were as below

Statement (if applicable):

EP conceived and designed the study and methodology, dissected the ducks, prepared the bone samples, photographed and evaluated the sections, took the measurements, ran the analyses, prepared the figures and wrote the manuscript; ÅTK helped to conceptualize and implement the numeric analyses, wrote the R codes, reviewed and edited final draft; AA acquired and currently curates the duck specimens used in this study, provided access to the hoatzin specimens, reviewed and edited final draft; DA provided necessary lab facilities, reviewed and edited final draft; PG and D-YH provided access to fossil paravian sections, reviewed and edited final draft; RJB reviewed and edited final draft. All authors gave final approval for publication.

**1 Radial Porosity Profiles: a new bone histological method for comparative developmental**
**2 analysis of diametric limb bone growth**

4 Edina Prondvai^{1,2*}, mm T. Kocsis³, Anick Abourachid⁴, Dominique Adriaens⁵, Pascal
5 Godefroit⁶, Dong-Yu Hu^{7,8}, Richard J. Butler¹

¹School of Geography, Earth & Environmental Sciences, University of Birmingham,
Edgbaston, Birmingham, UK

²MTA-MTM-ELTE Research Group for Paleontology, Budapest, Hungary

³Department of Palaeobiology, Friedrich-Alexander-University of Erlangen-Nurnberg,
Erlangen, Germany

⁴Departement Adaptations du Vivant, UMR 7179 Museum National d’Histoire Naturelle –
CNRS, Paris, France

⁵Department of Biology, Evolutionary Morphology of Vertebrates, Ghent University, Ghent,
Belgium

⁶Directorate Earth & History of Life, Royal Belgian Institute of Natural Sciences, Brussels,
Belgium

⁷Paleontological Institute of Shenyang Normal University, Key Laboratory for Evolution of
Past Life in Northeast Asia, Ministry of Land and Resources, Shenyang, China

⁸Paleontological Museum of Liaoning, Shenyang, China

23 *Corresponding author. E-mail: edina.prondvai@gmail.com; E.Prondvai@bham.ac.uk

Abstract

Assessing ontogenetic patterns in fossil vertebrates is crucial in palaeobiology and many other fields of vertebrate evolution. Limb bone histology is considered the most reliable tool not only for skeletal maturity assessments but also for evaluating the growth record in the ontogenetic window represented by the primary bone cortex. This is especially relevant in the context of variable ontogenetic strategies along the precocial–altricial developmental spectrum. Primary cortical vascularity is the most informative histological character for reconstructing growth dynamics due to its complex relationship with bone growth and functional maturation. Using this concept as our working hypothesis, we introduce here a new quantitative osteohistological parameter, radial porosity profile (RPP), that captures relative cortical porosity changes as trajectories in limb bones. We built a proof-of-concept RPP dataset on extant birds to which we added fossil paravian dinosaurs. We performed a set of trajectory-grouping analyses to identify potential RPP categories which were evaluated in the context of our ontogeny – developmental strategy working hypothesis. Our results support the analytical power of RPPs and reveal unexpected potential osteohistological correlates of growth and functional development of limb bones. The diverse potential applications of RPPs opens up new research directions in the evolution of locomotor ontogeny.

[revised manuscript text omitted]

139 strategies. We here propose a new quantitative histological method for long bones, referred to
140 as radial porosity profiles (RPP), that builds upon the general principles of diametric bone
growth and development in the primary cortex of terrestrial tetrapods. Our method abstracts
growth and developmental dynamics from the sampled element on the basis of relative
primary porosity measured along radial trajectories across the preserved posthatching cortex.
The course of these short trajectories, i.e. the RPPs, is not only informative of the specific
ontogenetic phase/period the studied specimen represents in its overall growth trajectory, but
may also give insight into the developmental strategy of the studied skeletal element.

[revised manuscript text omitted]

fine details of the individual's developmental phase and strategy which then can be compared
with that of other conspecifics or specimens of other taxa. Whether growth is overall slow or
fast, cyclical or continuous, isometric or allometric, precocial or altricial, porosity patterns are
influenced by the cumulative effects of all these developmental processes, and thus represent
an information-rich abstraction of the ontogenetic trajectories and potentially element-specific
functional aspects of tetrapods.

*2.2.2. Selection of the data-collecting technique*

Our ontogenetic and developmental approach is based on the osteohistological evaluation of
the RPPs in limb bone diaphyses. Limb bone shafts are the most frequently analysed elements
in palaeohistological studies because the mid-diaphyseal region usually has the highest
primary bone content providing insight into individual growth trajectories (Padian & Lamm,
2013). Long bone histology is also shaped by diverse locomotor demands (de Margerie, 2002;
Skedros *et al.* 2003; Currey 2003; de Margerie *et al.* 2005; Robling *et al.* 2006; Ponton *et al.*
2007; Simons & O'Connor. 2012; Smith & Clarke. 2014; Lee & Simons, 2015; Frongia *et al.*

[revised manuscript text omitted]
 (Fig. 2D). Thereafter, a mean RPP from the three sectors ($S = 3$) defined by a vector of four points for each bone (Fig. 2E) was calculated as:

$$RPP = \left(\sum_{n=1}^S P_{a_S}^1 \sum_{n=1}^S P_{b_S}^1 \sum_{n=1}^S P_{c_S}^1 \sum_{n=1}^S P_{d_S}^1 \right) (3).$$

The courses of these RPPs (Fig. S3) were numerically analysed by diverse trajectory modelling and clustering methods in various comparative contexts (see Section 2.2.6).

Vascular measurements were only taken in primary cortical areas. Wherever secondary remodelling was present, their areas were measured within the quadrants and subtracted from the sampled areas (Fig. S2B, D). In those bones where secondary remodelling affected extensive cortical regions obliterating most or all of the primary cortex in a given quadrant (most frequently Q_{a-b}), the porosity values of the affected quadrants were considered missing (NAs) (Fig. S2B, C). Only those RPPs were analysed in which a minimum of two of the four points defining the sector-averaged trajectory could be calculated at least in one sector. The extent of secondary remodelling was also considered in the interpretation of the results.

The relative thicknesses of the outer circumferential layer (OCL) and the endosteal lamellar layer (EL), rimming the medullary cavity in most bones of the paravian dinosaurs and some bones of the extant bird specimens (Fig. 1H), were quantified in relation to overall cortex thickness as the area percentage it occupies of the total cortex area in the measured sectors. Similar to secondarily remodelled areas, OCL and EL were not included in the numeric analyses in this study but were considered during the interpretation of the results.

2.2.6. Numeric analysis – a selection of methods

Three different main approaches with various settings were selected to demonstrate the diverse analytical potential in the RPPs of individual bones: 1) K-means clustering; 2) group-based trajectory modelling (GBTM; Nagin, 2014); and 3) three types of trajectory similarity measures (dynamic time warping [DTW], ‘edit distance’ [ED], and longest common subsequence [LCSS]; Toohey & Duckham, 2015). K-means clustering is a widely used method applied in a diverse range of fields for classifying multivariate observations into groups (e.g. MacKay, 2003). The similarity of trajectories included mean squared distances (MSD) that reflect the expected distance between corresponding points (homologous quadrants) of the RPPs. GBTM is a more specific statistical approach to analyse developmental trajectories – a progression of any phenomenon described by longitudinal data – and to identify distinct groups of individual trajectories within a population (Nagin, 1999, 2014). GBTM assigns group membership probabilities to individual trajectories and compares the analytical support for all models to find the optimal (best supported) number of groups (Nagin, 1999, 2014; Nielsen, 2018). The third approach, trajectory similarity measures, comes from the field of computational movement analysis in which trajectories are defined as a “sequence of time-stamped locations” (Gudmundsson et al. 2012) and similarity is calculated by various distance measures between trajectory pairs (Toohey & Duckham, 2015). In the DTW method, the sum of the distances at each point along the trajectories is calculated as the smallest warping path. In ED, the minimum number of edits are calculated for two trajectories to be considered equivalent. Finally, LCSS works with the translation of the trajectories in all

possible dimensions (in our case 2D) to find the maximum number of equivalent points between two trajectories (Gudmundsson et al. 2012; Toohey & Duckham, 2015; Toohey, 2018).

In all analysis types, we first ran the tests separately on the RPPs of the duck ontogenetic series (Fig. S3A, B), then on those of the extinct paravian dinosaurs (Fig. S3C, D), and finally on the complete dataset including the ducks, the extinct paravians, the juvenile hoatzins and the other adult birds (Fig. S3E, F). Using these separate analyses, we could adjust the resolution of the results to broaden the context of our interpretations and elaborate on the finer details. We also analyzed separately those datasets with and without trajectories containing missing values to see their effect on the group compositions. Furthermore, we also ran tests on centered RPPs calculated for each quadrant as the difference from the mean sector porosity within a given trajectory (Fig. S3B, D, E). In these tests, only the shape of the trajectories determines their group memberships. Finally, in the duck dataset, for which we had the most complete background information, potential predictors of group memberships were also tested by logistic regressions in the best supported RPP-clustering models.

All analyses were performed in RStudio (1.4.1) integrated development environment (RStudio, PBC) for the free programming language and statistical computing software R (R Core Team). K-means clustering was called in from the basic 'stats' function *kmeans()*, GBTM was performed using the function *crimCV()* in package 'crimCV' (Nielsen et al. 2014; Nielsen, 2018), and the three different trajectory similarity measures were run with the functions *DTW()*, *EditDist()* and *LCSS()* in package 'SimilarityMeasures' (Toohey, 2015). For the K-means and GBTM analyses, we used the available ontogenetic, age and developmental strategy data of the specimens, and the graphical distribution patterns of individual RPPs, to select the number of groups to be tested. K-means uses a priori defined number of clusters, and we tested 3–5 clusters in the different analyses. Although treating the distance values as coordinates might have a degrading effect on the signal present in the data, the multidimensional structure of points remains very similar. GBTM, on the other hand, works with a pre-set maximum number of groups, which we set to five in our analyses. Because GBTM has a stochastic component, we ran 100 iterations in each analysis to find a globally best supported model using three model selection methods (Bayesian Information Criterion [BIC], Akaike Information Criterion [AIC], cross-validation error [CVE]) with the function *crimCV()* in which CVE was developed for testing GBTM itself (Nielsen et al. 2014). The similarity/distance measures of trajectory pairs resulting from the three types of trajectory similarity analyses were organized into distance matrices and further analysed with K-means clustering set to 3–5 clusters. General and ordinal logistic regressions for testing potential predictor variables of group memberships in ducks were run with the functions *glm()* available in {stats} (R Core Team, 2020) and *porl()* in package 'MASS' (Venables & Ripley, 2002). P-values for the regression coefficients in the ordinal logistic regression were calculated as for t-values with the function *pt()*. Model selection was based on AIC values using the function *step()* available in {stats} (R Core Team, 2020). See complete datasets, detailed R scripts and results of all analyses in Supplementary Table 1 and Supplementary Information 2-6.

Results were interpreted in various contexts, such as the developmental strategies of different elements within an individual, individual variation within a cohort of a species,

ontogenetic patterns of development of homologous elements and the whole skeleton, and
broader within-clade and interspecific comparisons.

3. Results

3.1. Qualitative evaluation of RPPs

Inspection of the simple graphical representation of RPPs reveals interesting patterns
concerning the developmental trajectories of limb elements (Figs. 3, S4–S18). In this
qualitative description, we mostly focus on the ducks where the ontogenetic spectrum and the
presence of three individuals in each age group (hereafter cohort) up to 30 dph allowed more
detailed interpretations (Figs. S4–S12). Other taxa in our sample (Figs. S13–S18) are
considered in the context of the findings in ducks.

3.1.1. Ducks

Although sample size is obviously inadequate to properly assess intracohort variance in
RPPs, the three growth series of ducks seem to show certain ontogenetic trends in the course
of the trajectories and their variance that may be related to the disparate locomotor functions
of the fore- and hind limbs.

At 4 dph, neither the ulna, nor the carpometacarpus has developed enough posthatching
periosteal cortex to be evaluated. All the other analysed limb bones of the ducklings show
high overall porosity levels throughout the cortex (~20–70%) with erratic-looking RPPs. The
intrasectional variance of RPPs (i.e. among the three sectors) lies within the same range as the
intracohort variance among homologous bones, and no separation appears between altricial
forelimb and precocial hind limb elements either in absolute porosity levels or in the course of
RPPs (Figs. S4, S9, S10). By 8 dph, a trend appears in the hind limb elements with more
channelized RPP courses, that is, with lower intrasectional and intracohort variance in a well-
defined course and radially increasing porosity, especially in the femur and tibiotarsus. On the
other hand, wing elements show no such trend in their RPPs and continue to be largely
irregular (Figs. S5, S9, S10). At 15 dph, the general pattern stays the same as at 8 dph, except
that the humerus starts showing similar channelization, and porosity may increase in the inner
cortex of some elements due to the formation of large resorption cavities related to medullary
cavity expansion (Figs. 3, S6, S9, S10). Furthermore, overall porosity seems to be higher in
the altricial wing bones, although the precocial tarsometatarsus remains in the same porosity
range as the wing bones at this stage. By 30 dph, RPP channelization occurs in every element
with overall higher porosity levels in the altricial wing bones (~20–50%) than in the precocial
leg bones (~5–30%) (Figs. S7, S9, S10), although pooled bones of the forelimb
[revised manuscript text omitted]

(Prondvai et al. 2018). RPPs of all other specimens converge on <3% P_d levels (Fig. S17).

Interestingly, and in contrast to the general pattern seen in our extant bird sample, an EL is
present in all sampled elements of these fossil paravians, while a definite OCL is missing in
all *Eosinopteryx* bones, and in some bones of the other, more mature specimens. Since hind
limb elements were only represented by the femora in this sample of extinct paravians, no
meaningful comparison can be made between the RPPs of fore- and hind limbs. Nevertheless,
RPPs of femora seem to be neatly aligned with RPPs of the forelimb elements showing no
evident deviation at these ontogenetic stages (Figs. S17, S18).

**3.2. Numeric analyses of RPPs**

In essence, all analyses of the raw data by K-means, GBTM and Similarity Measures (DTW
and ED combined with K-means) gave similar results for each dataset, that is for the ducks,
extinct paravians, and the complete dataset (Figs. 4–7, S19–S21, Supplementary Information
4–6.). On the other hand, centred data and LCSS gave quite erratic memberships in each case
which apparently did not relate to any studied parameters (Fig. S22, S23). Therefore, only the
raw data-based results are shown here, whereas those of centred data and LCSS are
considered further only in the Discussion.

**3.2.1. Ducks**

When the duck growth series were analysed with 3, 4 and 5 groups for describing trajectory
types from 4 dph up to adulthood, the 4-group setup consistently represents the best model
across different analysis methods (Fig. S19). Numeric support is provided by GBTM
iterations in which the 4-group model shows the most frequently, as well as globally, the
lowest AIC, BIC and CVE values (Fig. 4). Group compositions are fairly consistent among
different analyses with 74% of the elements clustered the same way by all four methods
(Table S2). The most inconsistency in membership assignments across methods occurred in
the 30 dph cohort, followed by 15 dph. Group memberships of bones appear to largely reflect
the combination of the individual’s ontogenetic stage and the developmental strategy of the

element. Most importantly, 4-group models successfully show how altricial and precocial
bone development within an individual can assign an individual's elements to groups
containing specimens of less and more mature ontogenetic stages, respectively (Fig. 5). This
means that these RPP-based trajectory groups (Fig. 5A) tend to “dissociate” the animal's
skeleton along its differently developing limb modules. For instance, the precocially
developing hind limb bones of a subadult animal (50 dph) are likely grouped together with
bones of adult animals, while its altricially developing wing bones tend to be assigned to a
separate group composed mostly of bones representing earlier ontogenetic stages, such as
mid- and late juveniles (Fig. 5B; Table S2).

At this analytical resolution encompassing the entire ontogenetic series up to adulthood,
group memberships also reveal that the predictive power of precocial *vs* altricial development
in separation of element RPPs changes throughout ontogeny. Dynamics of RPP disparity can
be traced in group compositions along the age axis combined with the level of skeletal
‘dissociation’ of specimens, that is the number of different groups an individual's bones were
assigned to. In the analyses with four groups, skeletal dissociation is not evident at 4 dph, (i.e.
all bones within an individual tend to be clustered in a single group) but starts to appear at 8
dph where wing and leg bones of an individual tend to be separated into two groups. The level
of skeletal dissociation may even reach three groups in some specimens at 15 and 30 dph but
returns to a bipartite division at 50 dph with perfect separation of wing and leg bones by
GBTM and K-means (Fig. 5B, Table S2). Figure 5B further illustrates these dynamics in
clusters #3 and #4, which contain most elements of 15 and 30 dph, (Fig. 5B). The distribution
pattern in these clusters is a transition from the younger ages (left-hand side) being mostly
represented by precocial hind limb bones (white circles) to the older ages (right-hand side)
containing primarily altricial forelimb bones (black circles). The mid-age ranges show a more
balanced mixture of fore- and hind limb bones in the respective groups. Thus, modularity in
fore- and hind limb development results in groups spanning three ad hoc maturity degrees
(from early to late juveniles in #3 and from mid-juveniles to subadults in #4) and clustering
together elements that originate from very differently aged animals (Fig. 5B).

Even though these groupings clearly result in a mixture of possible ontogenetic staging of
the animal based on RPPs of its individual elements, there are consistent limits as to which
ontogenetic stages can be represented by the differently developing elements within the same
group. For example, no bones of early juveniles (up to 8 dph) were assigned the same group
membership as those of subadults (50 dph), and no bones of mid-juveniles (up to 15 dph)
were grouped together with any adult bones, irrespective of the bone's individual
developmental strategy.

To further unfold how this complex RPP pattern might be related to the disparate growth
and development of altricial *vs* precocial elements, we separately analyzed by GBTM and K-
means the subsets of those ontogenetic stages where we had at least 3 animals in each
category, that is for early (4–8 dph), mid- (15 dph) and late juveniles (30 dph), respectively
(Table S3). Here, we set the maximum number of groups to 2 to see whether and how
precocial *vs* altricial strategy is reflected in the RPP group compositions through ontogeny.
These higher resolution analyses showed a similar trend to that recovered in the analysis
spanning the entire ontogenetic series but provided a much clearer pattern and stronger signals
related to the developmental strategies of individual bones.

The major trend is still the progressively clearer separation of the RPPs of precocially vs
altricially developing hind vs forelimb elements as ontogeny progresses. GBTM of early
juveniles (4–8 dph) showed no separation of bones at 4 dph, whereas RPPs of differently
developing elements start to differ more evidently at 8 dph with all of the altricial wing bones
being grouped together with bones of 4 dph, while all femora and tibiotarsi of the precocial
leg bones were grouped separately (Table S3). Nevertheless, tarsometatarsi at 8 dph are
clustered with the wing bones, and the distinctness of the groups is still weak, which is also
reflected in the closeness of support for both number of groups (1 and 2) provided by all
GBTM model selection methods (Supplementary Information 4). K-means gave very similar
results, although the ulnae at 8 dph were consistently assigned to the same cluster as the
femora and tibiotarsi at 8 dph. On the other hand, in late juveniles (30 dph) both GBTM and
K-means were able to identify and separate precocially and altricially developing bones with
100% accuracy based on their RPPs. Finally, the strength of separation of RPPs of precocial
and altricial elements in mid-juveniles (15 dph) is intermediate with a less distinct group
composition than in late juveniles but higher support for the two groups than in early juveniles
by all GBTM model selection methods (Table S3, Supplementary Information 4).

Based on our working hypothesis and the observed patterns in group compositions in the
entire dataset and its cohort subsets, age and developmental strategy of bones were selected to
be tested as predictors of GBTM and K-means group memberships in the complete duck
dataset with both linear combination of and interaction between these predictor variables.
Model selection (AIC) supported the model with linear combination of predictors. Logistic
regressions in each case showed that age is an overall stronger predictor than developmental
strategy with both variables being significant predictors of group memberships ($p < 0.0001$
for age and $p < 0.001$ for developmental strategy). For further details see Supplementary
Information 4.

As for individual elements, group membership probabilities highlight the precocial
tarsometatarsus seems to be the most inconsistent concerning its development-related group
membership in all analyses. This bone usually shows a highly porous cortex as compared with
the other two precocial hind limb elements, the femur and the tibiotarsus, within the same
specimen, and hence is frequently grouped together with the altricial wing elements. Finally,
the altricial ulnae at 8 dph (the earliest age for measurable ulnar cortex) tend to group together
with their contemporary precocial femora and tibiae along with elements of mid- and late
juveniles. At 8 dph the ulnar posthatching cortex is still fairly underdeveloped which may
account for this unexpected pattern. However, the same trend in the ulnae is still observed at
15 dph, although the signal is less clear.

*3.2.2. Fossil paravian dinosaurs*
Different types of analyses of the five specimens of fossil paravian dinosaurs gave moderately
variable results. In GBTM, all model selection methods supported the model where RPPs of
all studied elements belong to a single group (Figs. 6, S20; Supplementary Information 5).
Nevertheless, in the 2-group setup, one group contains a single element, namely the humerus
of the late juvenile *Eosinopteryx*, while all other elements are clustered together in the other
group. The same is the case in the 2-cluster K-means and ED analyses, while DTW assigns
four of the six elements of *Eosinopteryx* in this separate group. The 3-group setting provided

more diverse group memberships with different analyses. GBTM resulted again in the
separation of the *Eosinopteryx* humerus in one group, and so did K-means and ED analyses
(Figs. 6, S20). This recurring separation of *Eosinopteryx* elements by different RPP analysis
methods is in accordance with the previously assigned juvenile status of this specimen
(Prondvai et al. 2018). However, the rest of the ‘dinobird’ elements were clustered in variable
ways. GBTM sorted three more *Eosinopteryx* elements into a separate group and all other
elements in the third group. On the other hand, K-means and ED consistently grouped
together most of the elements of *Eosinopteryx* and *Jeholornis* with half of those of *Serikornis*
across the 3-group analyses, while DTW gave a less interpretable element sorting
(Supplementary Information 5).

Group membership probabilities of RPPs are best explained when precocity ranks of
elements (a qualitative order of relative osteohistological maturity of elements within the
skeleton; Prondvai et al. 2018) are also considered in combination with the assigned
ontogenetic stage of the specimen (deduction based on overall osteohistological maturity;
Prondvai et al. 2018). This is shown by the clear separation of the humerus of *Eosinopteryx*
from all other elements in both, the 2-group and 3-group setups. Memberships of elements of
*Serikornis* also tend to reflect its immature ontogenetic status along with the assigned element
precocity ranks in its skeleton (Prondvai et al. 2018).

3.2.3. Complete dataset

When tested for 3–5 groups (Figs. 7, S21; Supplementary Information 6), model selection
methods performed in the GBTM analyses supported 5 groups as the best model to cluster
RPPs in the complete dataset. However, the four different analysis types gave much less
congruent results for the composition of the 5 groups (Fig. S21) than they did for the duck-
only dataset with 4 groups. Here, only 35% of the elements clustered identically by all four
methods, and 45% by three of the four methods. Nevertheless, this still gives a total of 80% of
the elements being grouped with a minimum of 75% consistency across different methods
(Table S4). Group memberships assigned by GBTM and K-means reflect most faithfully the
combination of the specimens’ ontogenetic stages and the elements’ developmental strategies
(Fig. 7B), while DTW and ED cluster some elements less clearly related to these attributes
(Supplementary Information 6). Overall, no phylogenetic coherence appears in any of the
RPP clustering patterns.

Within the duck subgroup in the complete dataset, only 56% of the bones were identically
clustered by all four methods, while 20% and 21% were clustered by three and two of the four
methods, respectively (Table S4). Furthermore, when cluster assignments of duck elements in
this pooled dataset with 5 groups was compared with that of the duck-only dataset with 4
groups, 73% of the elements were clustered the same way by all four methods as in the duck-
only analyses. Another 19% of the bones were grouped identically by three out of the four
methods altogether giving a $\geq 75\%$ clustering congruence for 92% of the duck elements across
these analyses.

The subgroup of juvenile hoatzin bones show less consistency revealing that only 27% of
their elements were clustered identically by all four methods, and another 61% by three of the
four (Table S4). This still means that a $\geq 75\%$ clustering congruency occurs in 88% of their
elements. Within the hoatzin subgroup, GBTM and K-means vs. DTW and ED scatter their

bones across three vs. four clusters, respectively. Hoatzin bone clusters resulting from GBTM
and K-means largely cover the same clusters as do duckling bones of 15 dph, while DTW and
ED group the hind limb elements of the oldest hoatzin chick together with duck elements of
30 dph and 50 dph. GBTM seems to best reflect the estimated ages / ontogenetic stages of
these juveniles clustering most bones of the ~6 dph and ~9 dph chicks together, and
separating them from all but the ulna and carpometacarpus of the ~16 dph chick. Even though
the bone distribution pattern in the other three analyses, spanning up to four clusters, cannot
be satisfactorily related to ontogenetic stage and/or developmental strategies, distinction of
the oldest individual from the two younger chicks is clearly detectable in these analyses as
well. Finally, all four methods place the tarsometatarsi of the two younger hoatzin chicks in
the group containing most of the elements of the youngest ducks (4 dph) emphasizing again
the highly porous nature of this limb element in hoatzins (Table S4).

With a few exceptions, bones of extant adult birds were largely grouped together with
those of extinct paravian dinosaurs by all four methods. Remarkably, only K-means separated
the elements of the not yet fully-grown kestrel from the rest of the adult birds, and grouped
them together with some hind limb elements of ≥ 30 dph ducks and a single ‘dinobird’
element, the *Eosinopteryx* humerus (Fig. 7B). DTW and ED uniformly clustered all extant
adult birds in the group containing the most mature specimens in the complete dataset,
including all bones of the extinct dinosaurs. On the other hand, GBTM picked out a single
paravian element – the carpometacarpus of *Aurornis* – for its fifth cluster, while filling up the
fourth cluster with all the other bones of adults and two hind limb elements of the 50 dph
duck (Table S4).

When the cluster memberships of the ‘dinobird’ subgroup in the complete dataset is
compared with those in the 2-group ‘dinobird’-only analyses, it appears that the recurring
separation of the *Eosinopteryx* humerus in the latter disappears from all but the K-means
analysis in the former. Due to GBTM’s strange clustering in the complete dataset, only a
single element (*Aurornis* carpometacarpus) was clustered identically by all four methods, and
81% of all elements by three of the four methods in the ‘dinobird’-only and ‘dinobird’
subgroup analyses. This gives a $\geq 75\%$ clustering congruence for 85% of the ‘dinobird’
elements across these analyses.

32 782 33 783 3.2.4. Performance of and congruency among different methods

Besides the comparisons of clustering congruency of elements presented above, performance
and consistency of each method across the analyses can also be compared. Missing values
affected each method differently. For instance, GBTM can only work with complete RPPs
that have all four trajectory-defining points, while the rest of the methods tolerated incomplete
RPPs with up to two missing values differently. K-means has proven to be the most sensitive
to these missing values with only 67% of the elements sorted the same way in the analyses of
the complete dataset with and without incomplete RPPs. DTW and ED performed better with
98% and 86%, respectively.

Congruency in element sorting was examined between method pairs as well. When we
removed all elements with incomplete RPPs, the duck and the complete dataset included 70
and 128 elements, respectively. In this reduced duck dataset, the highest congruency in
membership assignments was found between GBTM and K-means, and between DTW and

ED, with both method pairs resulting in 93% of the elements clustered the same way. The
lowest congruence was detected between the K-means and ED methods, although it was still
high (79%) (Table S2). However, in the complete dataset encompassing 128 elements
spanning the same ontogenetic range but broadening the included developmental strategies,
the congruency levels of GBTM with respect to all other methods dropped considerably.
GBTM still had the highest congruency with K-means but it only reached 59%, whereas the
other the methods showed 77%–86% corresponding membership assignments (Table S4).

15 805 **4. Discussion**

[revised manuscript text omitted]

**Author Contributions**

EP conceived and designed the study and methodology, dissected the ducks, prepared the
bone samples, photographed and evaluated the sections, took the measurements, ran the
analyses, prepared the figures and wrote the manuscript; ÁTK helped to conceptualize and
implement the numeric analyses, wrote the R codes, reviewed and edited final draft; AA
acquired and currently curates the duck specimens used in this study, provided access to the
hoatzin specimens, reviewed and edited final draft; DA provided necessary lab facilities,
reviewed and edited final draft; PG and D-YH provided access to fossil paravian sections,
reviewed and edited final draft; RJB reviewed and edited final draft. All authors gave final
approval for publication.

**References**

- Abourachid A., Herrel A., Decamps T., Pages F., Fabre A.-C., Van Hoorebeke L., Adriaens
D., Garcia Amado M. A., 2019. Hoatzin nestling locomotion: Acquisition of quadrupedal
limb coordination in birds. *Science Advances*, 5, eaat0787.
Bailleul AM, Scannella JB, Horner JR, Evans DC, 2016. Fusion patterns in the skulls of
modern archosaurs reveal that sutures are ambiguous maturity indicators for the
Dinosauria. *PLoS ONE* 11(2): e0147687.
Bennett C. S. 1993. Ontogeny of *Pteranodon*. *Paleobiology*, 19(1), 92-106.
Bennett, M. B. 2008. Post-hatching growth and development of the pectoral and pelvic limbs
in the black noddy, *Anous minutus*. *Comparative Biochemistry and Physiology*, Part A 150,
159–168.
Brinkman, D. 1988. Size-independent criteria for estimating relative age in *Ophiacodon* and
*Dimetrodon* (Reptilia, Pelycosauria) from the Admiral and lower Belle Plains Formations
of West-Central Texas. *Journal of Vertebrate Paleontology* 8:172–180.
Brochu, C. A. 1996. Closure of neurocentral sutures during crocodylian ontogeny:
implications for maturity assessment in fossil archosaurs. *Journal of Vertebrate*
*Paleontology* 16:49–62.
Bybee PJ, Lee AH, Lamm E-T. 2006. Sizing the Jurassic theropod dinosaur *Allosaurus*:
Assessing growth strategy and evolution of ontogenetic scaling of limbs. *Journal of*
*Morphology*, 267(3):347–359.
Carrier, D., Leon, L.R. 1990. Skeletal growth and function in the California gull (*Larus*
*californicus*). *J. Zool.* 222, 375–389.
Castanet, J., Grandin, A., Abourachid, A., de Ricqlès, A. 1996. Expression de la dynamique
de croissance dans la structure de l'os periostique chez *Anas platyrhynchos*. [Expression of
growth dynamics in the structure of the periosteal bone in the mallard, *Anas*
*platyrhynchos*.] *C. R. Acad. Sci.* 319, 301–308.
Chapelle K.E.J., Benson R.B.J., Stiegler J., Otero A., Zhao Q., Choiniere J.N. 2020. A
quantitative method for inferring locomotory shifts in amniotes during ontogeny, its
application to dinosaurs and its bearing on the evolution of posture. *Palaeontology* 63(2),
229-242.

- Chinsamy, A. 1995. Ontogenetic changes in the bone histology of the Late Jurassic
ornithopod *Dryosaurus lettowvorbecki*. *Journal of Vertebrate Palaeontology*, 15(3): 96-
104.
- Chinsamy A. 1997. Assessing the biology of fossil vertebrates through bone histology
*Palaeont. Afr.*, 33, 29-35.
- Chinsamy-Turan, A. 2005. *The Microstructure of Dinosaur Bone*. Johns Hopkins University
Press, Baltimore.
- Colbert MW, Rowe T. 2008. Ontogenetic Sequence Analysis: Using parsimony to
characterize developmental sequences and sequence polymorphism. *J Exp Zool B Mol*
*Dev Evol* 310(5):398–416.
- Cullen T.M., Evans D.C., Ryan M.J., Currie P.J., Kobayashi Y., 2014. Osteohistological
variation in growth marks and osteocyte lacunar density in a theropod dinosaur
(Coelurosauria: Ornithomimidae). *BMC Evolutionary Biology*, 14:231.
- Currey, J. D. 2003. The many adaptations of bone. *Journal of Biomechanics* 36, 1487–1495.
- De Margerie, E. 2002. Laminar bone as an adaptation to torsional loads in flapping flight.
*Journal of Anatomy* 201, 521–526.
- De Margerie, E., Sanchez, S., Cubo, J. & Castanet, J. 2005. Torsional resistance as a principal
component of the structural design of long bones: comparative multivariate evidence in
birds. *The Anatomical Record* 282, 49–66.
- De Ricqlès A. 1980. Tissue structures of dinosaur bone, functional significance and possible
relation to dinosaur physiology. In: Thomas RDK, Olson EC, editors. *A Cold Look at the*
*Warm-Blooded Dinosaurs*. New York: American Association for the Advancement of
Science; pp. 103–139.
- De Ricqlès A., Castanet, J. & Francillon-Vieillot, H. 2004. The ‘message’ of bone tissue in
paleoherpetology. *Italian Journal of Zoology* 71, 3–12.
- Dial, T.R., Carrier, D.R. 2012. Precocial hindlimbs and altricial forelimbs: partitioning
ontogenetic strategies in mallards (*Anas platyrhynchos*). *J. Exp. Biol.* 215, 3703–3710.
- Dial TR, Heers AM, Tobalske BW. 2012. Ontogeny of aerodynamics in mallards:
comparative performance and developmental implications. *J Exp Biol.* 215, 3693-3702.
- Dilkes D.W. 2001. An ontogenetic perspective on locomotion in the Late Cretaceous dinosaur
*Maiasaura peeblesorum* (Ornithischia: Hadrosauridae). *Canadian Journal of Earth*
*Sciences* 38(8), 1205-1227.
- Erickson, GM, 2014. On Dinosaur Growth. *Annual Review of Earth and Planetary*
*Sciences*, 42 (1), 675-697.
- Formann AK. 1984. Die Latent-Class-Analyse: Einführung in die Theorie und Anwendung.
Beltz, Weinheim.
- Frongia GN, Muzzeddu M, Mereu P, Leoni G, Berlinguer F, Zedda M, Farina V, Satta V, Di
Stefano M, Naitana S. 2018. Structural features of cross-sectional wing bones in the griffon
vulture (*Gyps fulvus*) as a prediction of flight style. *Journal of Morphology*, 279 (12),
1753–1763.
- Garn SM, Rohmann CG, Blumenthal T (1966) Ossification sequence polymorphism and
sexual dimorphism in skeletal development. *Am J Phys Anthropol* 24(1):101–115.

Griffin CT, Nesbitt SJ, 2016a. The histology and femoral ontogeny of the Middle Triassic
(?late Anisian) dinosauriform *Asilisaurus kongwe* and implications for the growth of early
dinosaurs. *J Vertebr Paleontol* 36 (3), e1111224.
- Griffin CT, Nesbitt SJ, 2016b. Anomalously high variation in growth is ancestral for
dinosaurs but lost in birds. *PNAS* 113, 14757–14762.
- Griffin, CT. 2018. Developmental patterns and variation among early theropods. *Journal of*
*Anatomy*, 232(4), 604-640.
- Griffin CT, Stocker MR, Colleary C, Stefanic CM, Lessner EJ, Riegler M, Formoso K,
Koeller K, Nesbitt SJ. 2021. Assessing ontogenetic maturity in extinct saurian reptiles. *Biol*
*Rev* 96, 470-525.
- Gudmundsson J, Laube P., Wolle T. 2012. Computational movement analysis. In W. Kresse
and D. Danko, editors, *Springer Handbook of Geographic Information*, pp. 725-741.
Dordrecht: Springer, 2012.
- Halliday TR, Verrell PA. 1988. Body size and age in amphibians and reptiles. *J. Herpetol.* 22,
253–265.
- Haug C., Haug J.T. 2016. Developmental Paleontology and Paleo-Evo-Devo. In Kliman R.M.
ed. *Encyclopedia of Evolutionary Biology*, pp: 420-429. Academic Press.
- Heers A.M., Dial K.P, 2012. From extant to extinct: locomotor ontogeny and the evolution of
avian flight. *Trends in Ecology & Evolution*, 27(5), 296-305.
- Hone DWE, Farke AA, Wedel MJ. 2016. Ontogeny and the fossil record: what, if anything, is
an adult dinosaur? *Biol. Lett.* 12: 20150947.
- Horner, J. R. & Padian, K. 2004. Age and growth dynamics of *Tyrannosaurus rex*.
*Proceedings of the Royal Society of London B* 271, 1875–1880.
- Horner, J. R., de Ricqlès, A. & Padian, K. 1999. Variation in dinosaur skeletochronology
indicators: implications for age assessment and physiology. *Paleobiology* 25, 295–304.
- Horner, J. R., de Ricqlès, A. & Padian, K. 2000. Long bone histology of the hadrosaurid
dinosaur *Maiasaura peeblesorum*: growth dynamics and physiology based on an
ontogenetic series of skeletal elements. *Journal of Vertebrate Paleontology* 20, 115–129.
- Horner, J. R., Padian, K. & De Ricqlès, A. 2001. Comparative osteohistology of some
embryonic and neonatal archosaurs: implications for variable life histories among
dinosaurs. *Paleobiology* 27, 39–58.
- Inkscape Project, 2020. *Inkscape*, Available at: <https://inkscape.org>.
- Johnson R, 1977. Size independent criteria for estimating relative age and the relationships
among growth parameters in a group of fossil reptiles (Reptilia: Ichthyosauria). *Canadian*
*Journal of Earth Sciences*, 14(8), 1916-1924.
- Khabazi A. 2017. Anatomie comparée du développement du système ostéo-musculaire des
oiseaux. Implications des contraintes fonctionnelles sur la croissance. Unpublished PhD
Thesis. (Available on request from Dr A. Abourachid).
- Klein, N., Sander, M. 2008. Ontogenetic stages in the long bone histology of sauropod
dinosaurs. *Paleobiology* 34, 248–264.
- Lee, A., Simons, E. 2015. Wing bone laminarity is not an adaptation for torsional loads in
bats. *PeerJ* 3, e823.
- Lee A. H., Huttenlocker A. K., Padian K, Woodward H. N. 2013. Analysis of growth rates.
In: Padian, K., Lamm, E-T. (eds). *Bone Histology of Fossil Tetrapods: Advancing*

*Methods, Analysis, and Interpretation*. University of California Press, California. Pp 217-
252.
MacKay D.J.C. 2003. Information Theory, Inference, and Learning Algorithms. *Cambridge*
*University Press*.
Maxwell E.E. 2008. Ossification sequence of the avian order Anseriformes, with comparison
to other precocial birds. *J Morphol*, 269:1095–1113.
- Maxwell E.E., Harrison L.B., Larsson H.C.E. 2010. Assessing the phylogenetic utility of
sequence heterochrony: evolution of avian ossification sequences as a case study. *Zoology*
113, 57–66.
Mitgutsch C., Wimmer C., Sánchez-Villagra M.R., Hahnloser R., Schneider R.A. 2011.
Timing of ossification in duck, quail, and zebra finch: intraspecific variation,
heterochronies, and life history evolution. *Zool Sci*. 28(7): 491–500.
- Morris JS, Carrier DR. 2016. Sexual selection on skeletal shape in Carnivora. *Evolution*,
70(4): 767-780.
Müllner, A. 2004. Breeding Ecology and Related Life-History Traits of the Hoatzin,
*Opisthocomus hoazin*, in a Primary Rainforest Habitat. PhD Thesis. Bayerischen Julius-
Maximilians-Universität, Würzburg.
Nagin, DS. 1999. Analyzing developmental trajectories: A semiparametric, group-based
approach. *Psychological Methods*, 4(2): 139-157.
- Nagin, DS. 2014. Group-based trajectory modeling: An overview. *Annals of Nutrition &*
*Metabolism*, 65: 205-210.
Nielsen JD., Rosenthal JS., Sun Y., Day DM., Bevc I., Duchesne T. 2014. Group-based
criminal trajectory analysis using cross-validation criteria. *Communications in Statistics –*
*Theory and Methods*, 43(20), 4337-4356.
Nielsen JD. 2018. Package ‘crimCV’: Group-based modelling of longitudinal data.
<https://CRAN.R-project.org/package=crimCV>
Otero, A., Cuff, A.R., Allen, V. Sumner-Rooney, L., Pol, D., Hutchinson, J.R. 2019.
Ontogenetic changes in the body plan of the sauropodomorph dinosaur *Mussaurus*
*patagonicus* reveal shifts of locomotor stance during growth. *Sci Rep* **9**, 7614.
Padian, K., Lamm, E-T. 2013. *Bone Histology of Fossil Tetrapods: Advancing Methods,*
*Analysis, and Interpretation*. University of California Press, California.
Parsons KJ, McWhinnie K, Pilakouta N, Walker L. 2020. Does phenotypic plasticity initiate
developmental bias? *Evolution & Development*, 22:56–70.
Petermann H., Mongiardino Koch N., Gauthier J.A. 2017. Osteohistology and sequence of
suture fusion reveal complex environmentally influenced growth in the teiid lizard
*Aspidoscelis tigris* — Implications for fossil squamates. *Palaeogeography,*
*Palaeoclimatology, Palaeoecology*, 475, 12–22.
Ponton, F., Montes L., Castanet J., Cubo J. 2007. Bone histological correlates of high-
frequency flapping flight and body mass in the furculae of birds: a phylogenetic
approach. *Biological Journal of the Linnean Society*, 91(4), 729–38.
Prondvai E. 2014. Comparative bone histology of rhabdodontid dinosaurs. *Palaeovertebrata*
38(2)-el. pp. 31, Online ISSN: 2274-0333.

Prondvai E. 2017. Medullary bone in fossils: function, evolution and significance in growth
curve reconstructions of extinct vertebrates. *Journal of Evolutionary Biology*, 30(3): 440-
460.
- Prondvai, E., Stein, K.H.W., Ósi A., Sander P.M. 2012. Life history of *Rhamphorhynchus*
inferred from bone histology and the diversity of pterosaurian growth strategies. PLoS
ONE 7(2): e31392.
- Prondvai, E., Godefroit, P., Adriaens, D., Hu, D-Y. 2018. Intraskkeletal histovariability,
allometric growth patterns, and their functional implications in bird-like dinosaurs. *Sci.*
*Rep.* 8, 258.
- Prondvai E, Witten PE, Abourachid A, Huysseune A, Adrians D. 2020. Extensive chondroid
bone in juvenile duck limbs hints at accelerated growth mechanism in avian
skeletogenesis. *Journal of Anatomy*, 236(3): 463-473.
- R Core Team, 2020. R: A language and environment for statistical computing. R Foundation
for Statistical Computing, Vienna, Austria. URL <http://www.R-project.org/>.
- Reisz, R. R., Evans, D. C., Sues, H. D., Scott, D. 2010. Embryonic skeletal anatomy of the
sauropodomorph dinosaur *Massospondylus* from the Lower Jurassic of South Africa. *J.*
*Vert. Paleontol.* 30, 1653–1665.
- Riesenfeld, 1978. Sexual dimorphism of skeletal robusticity in several mammalian orders.
*Acta Anat.* 102: 392–398.
- Robling AG, Castillo AB, Turner CH. 2006. Biomechanical and molecular regulation of bone
remodeling. *Annu Rev Biomed Eng.* 8:455-498.
- Roitberg, E.S., Smirina, E.M. 2006. Age, body size and growth of *Lacerta agilis boemica* and
*L. strigata*: a comparative study of two closely related lizard species based on
skeletochronology. *Herpetological Journal*, 16, 133-148.
- Roitberg E.S., Orlova V.F., Bulakhova N.A., Kuranova V.N., Eplanova G.V., Zinenko O.I.,
Arribas O., Kratochvíl L., Ljubisavljević K., Starikov V.P., Strijbosch H., Hofmann S.,
Leontyeva O.A., Böhme W. 2020. Variation in body size and sexual size dimorphism in
the most widely ranging lizard: testing the effects of reproductive mode and climate. *Ecol*
*Evol.* 10, 4531–4561.
- Sánchez-Villagra M.R. 2010. Developmental palaeontology in synapsids: the fossil record of
ontogeny in mammals and their closest relatives. *Proc. R. Soc. B.* 277, 1139–1147.
- Sander PM, Klein N. 2005. Developmental plasticity in the life history of a prosauropod
dinosaur. *Science* 310(5755): 1800–1802.
- Schneider, C. A., Rasband, W. S., Eliceiri, K. W. 2012. NIH Image to ImageJ: 25 years of
image analysis. *Nature Methods* 9(7): 671-675.
- Sharma, P., Clouse, R., & Wheeler, W. 2017. Hennig's semaphoront concept and the use of
ontogenetic stages in phylogenetic reconstruction. *Cladistics*, 33(1), 93-108.
- Sheil C, Greenbaum E. 2005. Reconsideration of skeletal development of *Chelydra*
*serpentina* (Reptilia: Testudinata: Chelydridae): Evidence for intraspecific variation. *J Zool*
265(3), 235–267.
- Simons, E., O'Connor, P. 2012. Bone laminarity in the avian forelimb skeleton and its
relationship to flight mode: Testing functional interpretations. *Anat. Rec.* 295, 386–396.

- Skedros, J. G., Hunt, K. J., Hughes, P. E. & Winet, H. 2003. Ontogenetic and regional
morphologic variations in the turkey ulna diaphysis: implications for functional adaptation
of cortical bone. *The Anatomical Record Part A* 273A, 609–629.
- Smith, A., Clarke, J. 2014. Osteological histology of the Pan-Alcidae (Aves,
Charadriiformes): correlates of wing-propelled diving and flightlessness. *Anat. Rec.* 297,
188–199.
- Stamps, J. A., Mangel, M., Phillips, J. A. 1998. A new look at relationships between size at
maturity and asymptotic size. *American Naturalist* 152, 470–479.
- Starck, M.J. 1993. Evolution of avian ontogenies. In *Current Ornithology*, Volume 10, (ed.
Power D. M.) 275–366 (New York Plenum Press).
- Starck, M.J. 1994. Quantitative design of the skeleton in bird hatchlings: Does tissue
compartmentalization limit posthatching growth rates? *J. Morph.* 222, 113–131.
- Starck, M.J., Ricklefs, R. E. 1998. Patterns of development: the altricial-precocial spectrum.
In *Avian Growth and Development. Evolution within the Altricial Precocial Spectrum* (eds.
Starck, M.J. & Ricklefs, R.E.), 3–30 (Oxford University Press).
- Starck, M.J., Sutter, E. 2000. Patterns of growth and heterochrony in moundbuilders
(Megapodiidae) and fowl (Phasianidae). *J. Avian Biol.* 31, 527–547.
- Starck, J. M., Chinsamy, A. 2002. Bone microstructure and developmental plasticity in birds
and other dinosaurs. *Journal of Morphology* 254, 232–246.
- Thewissen J.G.M., Cooper L.N., Behringer R.R. 2012. Developmental biology enriches
paleontology. *Journal of Vertebrate Paleontology*, 32:6, 1223–1234.
- Toohey K. 2018. SimilarityMeasures: Trajectory Similarity Measures. [https://CRAN.R-
project.org/package=SimilarityMeasures](https://CRAN.R-project.org/package=SimilarityMeasures)
- Toohey K., Duckham M. 2015. Trajectory similarity measures. SIGSPATIAL Special
Volume 7(1), 43–50. <https://doi.org/10.1145/2782759.2782767>
- Tumarkin-Deratzian AR, Vann DR, Dodson P 2006. Bone surface texture as an ontogenetic
indicator in long bones of the Canada goose *Branta canadensis* (Anseriformes: Anatidae).
*Zool J Linn Soc* 148(2):133–168.
- Unwin, D. 2005. *The Pterosaurs: From Deep Time*. New York: Pi Press. pp. 140–164.
- Varricchio, D.J. 2010. A distinct dinosaur life history? *Historical Biology*, 23:1, 91–107.
- Venables WN, Ripley BD. 2002. *Modern Applied Statistics with S*, Fourth edition. Springer,
New York. ISBN 0-387-95457-0, <https://www.stats.ox.ac.uk/pub/MASS4/>.
- Wang, X., Kellner, A.W.A., Jiang, S., Cheng, X., Wang, Q., Ma, Y., Paidoula, Y., Rodrigues,
1437 T., Chen, H., Sayão, J.M., Li, N., Zhang, J., Bantim, R.A.M., Meng, X., Zhang, X., Qiu,
R., Zhou, Z. 2017. Egg accumulation with 3D embryos provides insight into the life
history of a pterosaur. *Science* 358, 1197–1201.
- Woodward, H., Horner, J. R. & Farlow, J. O. 2011. Osteohistological evidence for
determinate growth in the American alligator. *Journal of Herpetology* 45, 339–342.
- Woodward H. N., Padian K, Lee A. H., 2013. Skeletochronology. In: Padian, K., Lamm, E-T.
(eds). *Bone Histology Of Fossil Tetrapods: Advancing Methods, Analysis, and
Interpretation*. University of California Press, California. pp 195–216.
- Woodward H.N., Horner J.R., Farlow J. O. 2014. Quantification of intraskeletal
histovariability in *Alligator mississippiensis* and implications for vertebrate osteohistology.
*PeerJ* 2, e422.

Woodward H.N., Freedman Fowler E.A., Farlow J.O., Horner J.R. 2015. *Maiasaura*, a model
organism for extinct vertebrate population biology: a large sample statistical assessment of
growth dynamics and survivorship. *Paleobiology*, 41(4), 503–527.
Woodward H.N., Tremaine K., Williams S.A., Zanno L.E., Horner J.R., Myhrvold N. 2020.
Growing up *Tyrannosaurus rex*: Osteohistology refutes the pygmy “*Nanotyrannus*” and
supports ontogenetic niche partitioning in juvenile *Tyrannosaurus*. *Sci. Adv.* 6, eaax6250.
Zhao, Q., Benton, M. J., Sullivan, C., Sander, P. M. & Xu, X. 2013. Histology and postural
change during the growth of the ceratopsian dinosaur *Psittacosaurus lujiatunensis*. *Nature*
*Communications* 4, 2079.

Table 1. Sampled taxa, specimens and elements used in this study. Asterisks indicate estimated minimum ages that were based on the size and the incipient outer circumferential layer in the elements of *Anas* sp, on the fresh body weight in hoatzin chicks, and on the plumage maturity and flight capability in the context of known growth strategy and life style in *Falco*, *Scolopax* and *Corvus*. Abbreviations: Cmc, carpometacarpus; Fe, femur; Hu, humerus; Ti, tibiotarsus; Tmt, tarsometatarsus

Taxon	Specimen ID	Age (dph)	Sampled bones
Anas platyrhynchos domesticus	duck_215/4461	4	Hu; Fe; Ti; Tmt
Anas platyrhynchos domesticus	duck_230/4480	4	Hu; Fe; Ti; Tmt
Anas platyrhynchos domesticus	duck_290/4556	4	Hu; Fe; Ti; Tmt
Anas platyrhynchos domesticus	duck_227/4477	8	Hu; UL; Cmc; Fe; Ti; Tmt
Anas platyrhynchos domesticus	duck_218/4464	8	Hu; UL; Fe; Ti; Tmt
Anas platyrhynchos domesticus	duck_246/4496	8	Hu; UL; Fe; Ti; Tmt
Anas platyrhynchos domesticus	duck_241/4491	15	Hu; UL; Cmc; Fe; Ti; Tmt
Anas platyrhynchos domesticus	duck_254/4503	15	Hu; UL; Cmc; Fe; Ti; Tmt
Anas platyrhynchos domesticus	duck_255/4504	15	Hu; UL; Cmc; Fe; Ti; Tmt
Anas platyrhynchos domesticus	duck_208/4452	30	Hu; UL; Cmc; Fe; Ti; Tmt
Anas platyrhynchos domesticus	duck_221/4468	30	Hu; UL; Cmc; Fe; Ti; Tmt
Anas platyrhynchos domesticus	duck_298/4555	30	Hu; UL; Cmc; Fe; Ti; Tmt
Anas platyrhynchos domesticus	duck_236/4486	50	Hu; UL; Cmc; Fe; Ti; Tmt
Anas sp.	duck_6	≥ 90*	Fe; Ti
Opisthocomus hoatzin	hoatzin_P4	6*	Hu; UL; Cmc; Fe; Ti; Tmt
Opisthocomus hoatzin	hoatzin_P1	9*	Hu; UL; Cmc; Fe; Ti; Tmt

[revised manuscript text omitted]

to reflect the disparate locomotor development of the fore- and hind limbs by increasing
skeletal dissociation (i.e. assignment of an individual’s wing and leg bones into different
clusters) as ontogeny progresses.

**Figure 6. Summary of results of the GBTM analysis (A-D) and cluster composition by**
**K-means with three clusters (E) of the fossil paravian RPP dataset.** A, Model selection
output of CVE, AIC and BIC. B, Predicted trajectories with two groups of RPPs indicated by
different colours and the mean RPP averaged over the dataset shown with a dashed line. C
and D, RPP distribution pattern in each group overlain by their respective estimated
trajectories. E, Cluster memberships of elements assigned by K-means analysis with three
clusters. Colours indicate homologous elements as indicated in the legend. Element IDs as
given in Table S1.

**Figure 7. Clustering of the complete RPP dataset – excluding RPPs with missing points –**
**by K-means with five groups demonstrated by different graphical representations.** A,
Raw RPPs with their cluster memberships indicated by the five different colours. B, Cluster

compositions with each element identifiable within their respective groups. Elements are
sorted along the horizontal axis by the known and/or estimated ages of the animals. Question
mark on the age axis denotes the unknown status of fossil paravians.

[revised manuscript text omitted]

479x400mm (236 x 236 DPI)

Figure 5. Clustering of the duck RPP dataset by K-means with four groups demonstrated by different graphical representations. A, Raw RPPs with their cluster memberships indicated by the four different colours. B, Cluster compositions with each element identifiable within their respective groups. Elements are sorted along the horizontal axis by the age of the ducks. Note, how cluster memberships within cohorts increasingly tend to reflect the disparate locomotor development of the fore- and hind limbs by increasing skeletal dissociation (i.e. assignment of an individual's wing and leg bones into different clusters) as ontogeny progresses.

472x524mm (236 x 236 DPI)

Figure 6. Summary of results of the GBTM analysis (A-D) and cluster composition by K-means with three clusters (E) of the fossil paravian RPP dataset. A, Model selection output of CVE, AIC and BIC. B, Predicted trajectories with two groups of RPPs indicated by different colours and the mean RPP averaged over the dataset shown with a dashed line. C and D, RPP distribution pattern in each group overlain by their respective estimated trajectories. E, Cluster memberships of elements assigned by K-means analysis with three clusters. Colours indicate homologous elements as indicated in the legend. Element IDs as given in Table S1.

472x465mm (236 x 236 DPI)

Figure 7. Clustering of the complete RPP dataset – excluding RPPs with missing points – by K-means with five groups demonstrated by different graphical representations. A, Raw RPPs with their cluster memberships indicated by the five different colours. B, Cluster compositions with each element identifiable within their respective groups. Elements are sorted along the horizontal axis by the known and/or estimated ages of the animals. Question mark on the age axis denotes the unknown status of fossil paravians.

483x567mm (236 x 236 DPI)

Appendix F

Dear RSOS editors, dear Reviewers,

Please, find our responses (in black) to the referees major comments (in red) one by one below. We replied to their minor comments in their respective annotated pdfs which we also attached to this revision.

We hope that the introduced changes in the revised MS and our replies to the questions and comments will be satisfactory for the Editor(s) and Reviewers as well.

Sincerely,
Edina Prondvai & co-authors

Response to the comments of Reviewer #1

Please find enclosed my review for the manuscript " Radial Porosity Profiles: a new bone histological method for comparative developmental analysis of diametric limb bone growth" (RSOS-211150), submitted to Royal Society Open Science by Prondvai and colleagues, along with an annotated PDF with additional comments on specific sections of the paper.

Thank you, we have replied to each of those comments in the annotated pdf which we've uploaded with our revision.

There are a few issues that should be resolved before acceptance. The main one is a potential issue in the way some parts of the clustering procedure are performed: outputs of the trajectory similarity measure analyses are treated as distance matrices for k-means clustering, which is inadequate to treat such data format (see annotated PDF); the use of e.g. k-medoids would be more appropriate in this context, and likely will not change the results much (especially since only a fraction of the results and discussion focuses on these analyses).

Thank you for drawing our attention to this. Indeed, we have made a mistake by feeding k-means distance matrices instead of the raw RPP data in all but the GBTM analyses. This has now been corrected by completely redoing all analyses (except GBTM) in all datasets the following way:

- 1) K-means clustering of RPPs was performed on raw RPP data
- 2) DTW and ED-derived trajectory similarities were further analysed by hierarchical clustering which works with distance matrices (so trajectory similarity pairs were organized into distance matrices and used as input for hclust).

In essence, these corrections have not changed the results, but we redid all figures and tables and rewrote all related sections of the paper, so that it now precisely reflects these new results in every aspects and detail.

The paper could also be improved in the way it sells its main point – the novelty and efficiency of the method – as something that would be easily replicable by other bone histologists. As we all know, there are too many papers and too little time, and readers will inevitably skim through this study looking for one main way to implement RPPs, without necessarily meandering through its caveats as much as they should. In this context, it is important to make that information available in a clear and accessible format, so that even the hastiest reader understands the main pros and cons of the approach.

One way to do this would be further illustrate the results and discussion sections, which are well written but very lengthy and technical, and currently only cite supplementary figures without referring to data visualizations actually present in the main paper (or barely). These long sections would greatly benefit from summary figures, with compilations of e.g. 1 RPP for each specimen of duck or species in the sample (I realize that Figs 3 and 4 have that function already, but they are barely mentioned in the discussion). Similarly, a comparative figure with a panel for each clustering procedure for a given sample, showing the difference in output for each method and displaying the percentage of congruence between them, would really help the reader understand the value of the use of all these different procedures to assess congruence, and show that such a visual check can be used to interpret results efficiently (this is in part what Table S2–4 do). Finally, as also stated in the annotated PDF, I think the conclusion would be much clearer if it listed a few numbered bullet points (3 to 5) as the main takeaways from the paper, both in terms of what this method brings to the field and what more should be done/accounted for in the near future to make it fully operational for any sample of birds (and other amniotes too).

Thank you for highlighting these points. Indeed, there was not enough in-text reference to crucial figures and SI, which we now added.

However, there are some problems with summary figures as suggested by the reviewer. First, a single RPP on its own cannot be represented in any sensible way in this study. It either needs to be put in the context of RPPs of other sectors or bones for visual comparison (as in Fig. 3) or in the group membership context (as in Fig. 5A). As for the different visual representations, it is practically impossible to put them all in a single summary figure. Taking the examples of the visual output of GBTM (Fig. 4), K-means (Fig 5), and the two dendrograms derived from DTW-hlclust and ED-hlclust (new Fig. 6) makes this issue quite clear. We could put two different visual representations in Fig 5, two dendrograms in Fig. 6 and they are already full-page figures. GBTM on its own is almost a full-page figure as well. There is no way we can summarize the diverse visual outputs of all four methods in a single figure with still interpretable labels.

Given this practical constraint as well as realizing that some of the supplementary tables actually summarize the most important results across these methods, we have now moved four supplementary tables into the main text as tables (and two of them will still exceed one page in length). Finally, we have added new sections into the Conclusion emphasizing the most important findings, and the advantages and conditions of using the RPP method, as suggested by the reviewer.

Response to the comments of Reviewer: #2

Comments to the Author(s)

...I do want to raise a few points that I think can be addressed with revisions.

First, I think that the focus on radial porosity profiles needs more explanation. For instance, the introduction is not clear why porosity needs to be measured regular intervals along radial tracts instead of a single averaged value (either the average of the anatomical side of bone or the whole bone). If a single average value of porosity is sufficient to infer relative ontogenetic stage, then why go through the complex RPP methodology?

The core importance of the RPP methodology lies in documenting the growth dynamics in the entire ontogenetic window represented by the preserved primary cortex. Thus, it is not used only to make a statement about the ontogenetic status at the time of death but to extract as much information of the earlier recorded growth as possible. If we averaged porosity over the entire cortex thickness, we would lose exactly the information and fine resolution of individual growth dynamics that is expressed in the relative radial porosity changes throughout the preserved primary cortex. We have now explained all this in the revised Introduction, as suggested by the Reviewer.

As to why applying porosity measures at regular radial intervals, the concept is that growth dynamics information is coded in the radial porosity changes that are captured by the shape and quadrant porosity values of these trajectories. This is made quite clear in the theoretical background of RPPs in the Methods section combined with the lengthy SI 1. In addition, three paragraphs in section 4.5 in the Discussion are devoted to point out the significance of the number of quadrants taken and how and why to optimize it under certain circumstances. Thus, we do feel that the purpose, meaning and significance of RPP trajectories, the datapoints of which are defined by the radial porosity measurements at regular intervals, are discussed in much detail in the manuscript already.

Second, many readers may be unfamiliar with the clustering techniques, and I think that the current text would benefit from additional details regarding the assumptions and setup for each clustering method.

In the previous version of this MS, K-means was the only clustering method that we used and the rest were different trajectory analysing methods. In the revised version we also use hierarchical clustering. In our revision, we extended our description to each method used in our analyses and referred to relevant literature for further details. The referred papers give detailed account of their use, assumptions (in fact they have very few restrictions on the datasets, as we now also stated this in the revised version) and implementation. As this manuscript is lengthy and very technical already – as it was pointed out by Referee #1 as well – we do not feel that further background details on the methods should be given in this work.

Here are few questions that I had after reading the manuscript a few times: Do the clusters represent relative ontogenetic stages – so a 3-cluster grouping could represent young, intermediate, and old clusters?

As this study was largely a proof-of-concept study, what the RPP clusters would represent was exactly the important question we aimed to find out. The RPP grouping methods applied here do not include pre-defined categories, except for the number of groups to test. These groups were then tested against possible predictors – in this case, predictors included our expected candidates, that is age and the bone's developmental strategy – by multinomial logistic regression on the duck RPPs. Only the duck dataset was tested like this because only ducks had a complete ontogenetic series and proper background information for testing predictors.

Does a cluster method assume that the data points are independently sampled (the data points in each RPP and RPPs from bones of a single individual are not independent because they represent repeated measures)? Do you need to adjust the degrees of freedom to reflect the number of individuals rather

than the number of bones? If the clustering method involves model selection (3 vs. 5 groups), what exactly is the model specification?

These grouping methods do not require independence of data points or trajectories. The assumptions of K-means and hierarchical clustering (hclust), the latter of which we applied now in the revised version for analysing distance matrices derived from DTW and ED, do not consider independence of data points. DTW and ED in themselves are developed to analyse animal movement trajectories where it can be applied on herds or repeated tracking of the same individual, which are in a lot of ways not independent, either. GBTM was initially used in tracking individual behavioural changes in time, so for the same reason, observations are not independent. In our study, K-means and hclust are both only applied as exploratory methods rather than numeric analyses that classically require such assumptions to be met. Thus, to our understanding, specimen-dependence should not be an issue in any of these analyses.

As for model selection of best group numbers, only GBTM had an in-built numeric model selection method based on CVE, BIC and AIC values of different models where the number of groups and the polynomial level used in describing trajectories were used to define and evaluate these models. Therefore, when deciding about the best supported group numbers in the duck-only, extinct paravian, and complete datasets, we mostly relied on the results of the GBTM model selection, but also on the visual inspection of the distinctness of trajectories within groups and the cluster compositions.

Third, the analysis involving GLM and logistic regression needs adjustment. The regression models assume that data points are independent of each other. However, multiple bones are sampled from a single individual. This repeated sampling, if not properly accounted for, will artificially inflate the degrees of freedom and make p-values too small. I suggest adding a random-effect to the model with specimen as a factor. In addition, you need to include a null model to test whether group membership is predicted by age and/or developmental strategy. If the dataset and relationship is strong, then model selection will exclude the null model. But that test still needs to be performed even if the additive and interaction models have significant coefficients.

First of all, it is important to emphasize here that we did not do any regression analyses on the RPPs themselves, which are, indeed, as the Reviewer pointed out, not all independent (bones belonging to the same animal are certainly not).

The regressions were done on the cluster memberships that were derived from the RPPs by the different trajectory similarity and clustering methods which do not have independence assumptions, as explained above.

Also, if there was a strong specimen-dependence in the RPPs' clustering, these methods would show that by placing those bones in the same clusters which come from a single individual, which, however, is clearly not the case here. What we see instead is an age-dependent degree of skeletal dissociation of the specimens.

In our study, one of the most important things is exactly to show how this skeletal dissociation across clusters reflects the ontogenetic changes in development of different elements. By your reflections it has become clear that this message was not conveyed properly, so we have rewritten those parts of the MS and put more emphasis on this finding in our revision by also including two tables summarizing the levels of skeletal dissociation in each specimen across the different methods. In addition, we included specimens as predictors in our tests to see whether their effect is indeed a considerable constraint on the group memberships (which in each case turned out not to be the case); see our model building approach below.

The second issue is the right selection of the method to detect important predictors of cluster/group memberships. In the first version of this MS, we opted for glm and ordinal regression because of the mere observation that our clusters have a natural ordering.

However, while revising our analyses we realized that the actual ordering that we saw in the clusters/groups was due to the age of the individual + developmental strategy of the element, so exactly the parameters we wanted to test as predictors. Thus, while building the model, we should not have assumed any natural ordering in the clusters that is due to the actual factors that we were going to test because then it becomes a circular reasoning.

Thus, in the revised version we used multinomial regression models instead, where the response variable – cluster memberships – is nominal with no ‘assumed’ natural ordering, and we tested these nominal clusters with the assumed predictors.

Concerning modelling specimen-dependence, we did follow the Reviewer’s suggestion about inclusion of specimens in the revised analyses. However, we could not use any mixed-models which would take ‘specimen’ as random effect because the packages and functions we found for mixed-effect multinomial logistic regressions in current R libraries wouldn’t work for our dataset (the package `mlogit` for multinomial regressions cannot include random effects, while `mclogit` (function `mblogit()`), which works with mixed effect models, keeps giving us error and warning messages about NaNs created, algorithms not converging, fitted probabilities being numerically zero and half of the models wouldn’t run at all with messages such as system is computationally singular, etc.).

Instead, we used the function ‘`multinom()`’ in package `nnet` without mixed effects inclusion as follows:

- 1) First we tested a null model with only specimens being predictors of cluster memberships;
- 2) then we constructed a model that included all probable predictors, including specimens;
- 3) thereafter we did a model selection by the function `step()` which drops unimportant predictors one by one and leaves only the most important predictors in the best supported model shown by the lowest AIC values;
- 4) we converted all AIC values into AICc values (better for smaller datasets), following the Reviewer’s suggestion;
- 5) we modelled memberships predicted by the different models and checked their correspondence % with the actual memberships;
- 6) finally, we selected the best models based on the combination of AIC, AICc and prediction matches.

Fourth, I’m not so sure about the identity of the endosteal lamellae mentioned on lines 516-518 and shown in Figure 1. Could that tissue be remnants of outermost periosteal bone from an early stage? For example, the 15-d radius in Figure 1 has what is called endosteal lamellae. But does that tissue match the outer periosteal bone of an 8-d radius? I’ve attached an image from my collection of mallards that shows how the outer periosteal bone of a 8-d radius overlaps the tissue that you called endosteal lamellae. In the absence of data from in vivo bone labeling (i.e., label 8-d chicks and see if those labels have an endosteal-most position in 15-d chicks), I suggest that you temper the statement that the “observations clearly disprove the general and widespread assumption that EL is a hallmark of cessation of medullary cavity expansion and radial bone growth.” You should at least present the alternative hypothesis and acknowledge that in vivo testing is needed.

We are pretty positive about the identification of that structure shown in Fig 1E (15d radius) as endosteal lamellae (EL), and here we demonstrate why in more detail. I prepared two figures for this purpose: one showing an ontogenetic series of radii from day 1 to day 15 posthatching age in our ducklings (1d cohort was not included in this MS because there is no evaluable posthatching cortex for RPPs but it is relevant here for clarifications); and one with additional labelling to the Reviewer’s image showing our interpretation of the structures visible in the Reviewer’s mallard slides.

The first figure includes two specimens of 1d, 4d and 8d to represent the considerable individual diversity in the relative level of development of homologous bones in each cohort (e.g. specimen #2

radius at 1d is still exclusively cartilaginous, whereas specimen #1 has a well-defined prehatching bony cortex; specimen #3 at 4d has asymmetrical posthatching cortex development, while specimen #4 has no posthatching cortex yet whatsoever), and a single 15d (specimen #7) with a well-defined EL:

Mid-diaphyseal cross sections of radius at different posthatching age (days) in ducklings
preh:prehatching cortex; **hl**: hatching line; **posth**: posthatching cortex; **po**: periosteum; **EL**: endosteal lamellae
 0.12 mm

What you can see here is that there is a fairly compact prehatching cortex (in red) in the radii that is preserved up to 15 days age. This is separated from the posthatching cortex (in blue) by a hatching line (yellow arrowhead) of variable distinctness around the cortex. From 4-8 days on, regional endosteal deposition starts from within the medullary cavity, again, with various distinctness and

thickness, but it is clearly there and distinguishable from the original prehatching cortex. This deposition is variable also because of the counteracting process of medullary cavity expansion and resorption of the original prehatching cortex. At 15 days this endosteally deposited structure in specimen #7 clearly shows the histocharacteristics of endosteal lamellae (in green), as shown in the magnified image of the 15d radius. (I am preparing a separate, more descriptive paper on these duckling ontogenetic series in which I am planning on going into such fine details as well that are not included in the current study.)

Based on these, we interpret the structures on your mallard slides as follows:

In my view, most of the cortex in your 8d mallard radius is composed of the more compact prehatching cortex, and only the highly porous outer region is posthatching. Indeed, in your 15d radius, the inner compact ring of bone is prehatching cortex (so in this sense the earlier periosteal cortex).

However, regionally, I can see a thin line of remodelled region rimming the medullary cavity in both specimens, although lamellar characteristics are not evident from these slides (and may not be there at all in these specimens). But there is certainly endosteal remodelling and deposition going on from the medullary cavity in your slides as well.

Based on all the sections I have investigated in an ontogenetic context, endosteal deposition seems to be a very dynamic process that is ongoing from very early on and is balanced (and/or temporarily eliminated) by endosteal resorption due to medullary cavity expansion. But EL can be present from a very early ontogenetic stage (even if only regionally and temporarily), just like other kind of secondary remodelling also occurs even in prehatching cortex very frequently before it gets resorbed entirely later on in ontogeny.

In sum, I think that the presented histocharacteristics in our referred 15d radius satisfy the criteria of endosteal lamellae – be it a primary or secondary endosteal deposition – and this can be seen without in vivo bone labelling as well. If we couldn't use these characteristics for tissue identification, then no fossil bone tissues could be identified on a similar basis. Thus, we wish to keep this statement in an

attempt to weaken the almost dogmatic concept in paleohistology that EL only appears when medullary cavity expansion (and growth) ceases.

Finally, Figures 5-7 could be clearer. In particular, the overlapping labels within each cluster are illegible. I suggest placing labels in boxes off to the sides. That way readers can clearly see group membership.

We have revised our related figures and in these there are no overlapping labels anymore. However, due to the number of specimens (132) to show in the new dendrogram in Fig 6, the size of the labels had to be minimized and one will need to zoom in to read the individual labels at the tip of the branches. We couldn't find any other way to visualize the position of the individual bones in the dendrograms in a more identifiable and informative manner.

I've prepared an annotated version of the manuscript with the complete list of my comments and suggestions.

Thank you, we have replied to each of those comments in the annotated pdf which we've uploaded with our revision.

Appendix G

Dear RSOS editors, dear Reviewers,

Please, find our responses (in black) to the referees comments (in red) one by one below. We replied to their minor comments in their respective annotated pdfs which we also attached to this revision.

We hope that the introduced changes in the revised MS and our replies to the questions and comments will be satisfactory for the Editor(s) and Reviewers as well.

Sincerely,
Edina Prondvai & co-authors

Reviewer: 1

Comments to the Author(s)

I only have two minor comments on the manuscript as it is:

A) l. 485–486: "The level of skeletal dissociation of specimens was assessed in each analysis."

I assume you mean how many clusters were bones separated into for a given specimen (e.g. depending on whether the RPP comes from a wing or leg bone), but is there a specific metric you used to estimate skeletal dissociation? Please clarify.

We did not estimate skeletal dissociation, as it was simply the output of the different types of grouping analyses for each specimen in each case. The 'metric' of skeletal dissociation is thus the number of different groups the skeletal elements of a specimen were assigned to. This definition was given in the MS but inappropriately placed in the Results section, so now we've moved it to the Methods section.

B) What agglomeration method did you use in 'hclust' (e.g. Ward, complete)? This is important to ensure full replication of these results.

We used the complete linkage method which is the default method for hclust in R. We've now added this information in the MS. Thank you for drawing our attention to this.

The edits performed in this new version are overall highly satisfactory, and I recommend acceptance of the paper after the two above minor remarks have been answered.

We hope that we managed to answer your questions and modify our MS satisfactorily, and that it will now be acceptable for publication.

Reviewer: 2

Comments to the Author(s)

Other than a couple of minor typos in the Conclusions (e.g., L 1278: “information-rick”; L 1333: “tertapods”), I think that this revision is now publication worthy.

Thank you for finding these typos, they have now been corrected in the revised MS.